# Position Coupling: Improving Length Generalization of Arithmetic Transformers Using Task Structure

**Hanseul Cho**[*]    **Jaeyoung Cha**[*]
Graduate School of AI, KAIST
{jhs4015,chajaeyoung}@kaist.ac.kr

**Pranjal Awasthi**
Google Research
pranjalawasthi@google.com

**Srinadh Bhojanapalli**
Google Research
bsrinadh@google.com

**Anupam Gupta**
NYU & Google Research
anupam.g@nyu.edu

**Chulhee Yun**
Graduate School of AI, KAIST
chulhee.yun@kaist.ac.kr

## Abstract

Even for simple arithmetic tasks like integer addition, it is challenging for Transformers to generalize to longer sequences than those encountered during training. To tackle this problem, we propose *position coupling*, a simple yet effective method that directly embeds the structure of the tasks into the positional encoding of a (decoder-only) Transformer. Taking a departure from the vanilla absolute position mechanism assigning unique position IDs to each of the tokens, we assign the same position IDs to two or more "relevant" tokens; for integer addition tasks, we regard digits of the same significance as in the same position. On the empirical side, we show that with the proposed position coupling, a small (1-layer) Transformer trained on 1 to 30-digit additions can generalize up to *200-digit* additions ($6.67\times$ of the trained length). On the theoretical side, we prove that a 1-layer Transformer with coupled positions can solve the addition task involving exponentially many digits, whereas any 1-layer Transformer without positional information cannot entirely solve it. We also demonstrate that position coupling can be applied to other algorithmic tasks such as $N \times 2$ multiplication and a two-dimensional task. Our codebase is available at `github.com/HanseulJo/position-coupling`.

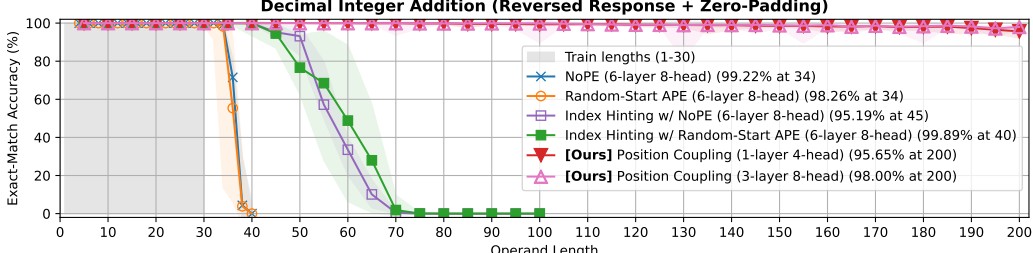

Figure 1: **Methods for Length Generalization in the Integer Addition Task.** We report exact-match (EM) accuracies (markers: *medians* over experiments; light area: 95% confidence intervals). We employ the reversed format and zero-paddings (Lee et al., 2024) into the input sequence. With our proposed **position coupling**, we achieve more than 95% exact-match accuracy for up to 200-digit additions with decoder-only Transformers trained on up to 30-digit additions. For index hinting (Zhou et al., 2024a), we separately test absolute positional embedding (APE) with a random starting position ID (mimicking the original implementation by Zhou et al. (2024a)) and without positional encoding (NoPE) (Kazemnejad et al., 2023) (as tested by Zhou et al. (2024b)).

---

[*]Authors contributed equally to this paper.

38th Conference on Neural Information Processing Systems (NeurIPS 2024).

# 1  Introduction

Since the appearance of a sequence-to-sequence deep neural architecture called Transformer (Vaswani et al., 2017), it has brought tremendous success in various fields including natural language process (NLP) (Chowdhery et al., 2023; Gemini et al., 2023; OpenAI, 2023; Thoppilan et al., 2022) and many applications such as mathematical reasoning and theorem proving (Lewkowycz et al., 2022; Trinh et al., 2024; Wu et al., 2022). Despite its triumph, it has recently been illuminated that Transformers often lack the ability of *length generalization* (Anil et al., 2022; Deletang et al., 2023; Press et al., 2022; Zhang et al., 2023). It refers to a special kind of out-of-distribution generalization capability to extrapolate the model's performance to longer sequences than those encountered during training. Understanding length generalization is of great importance because a lack of it provides evidence that language models do not genuinely understand the structure of a given task. Improving Transformer's length generalization has received much attention, particularly because the time/memory complexities for training Transformers grow up to quadratically in the sequence length.

Even for simple arithmetic tasks such as integer addition, length generalization is still difficult for Transformers (Kazemnejad et al., 2023; Kim et al., 2021; Lee et al., 2024; Nogueira et al., 2021; Zhou et al., 2024a,b). Humans can length-generalize in integer addition because they understand the essential principle of the task. Nevertheless, it is observed that Transformers typically learn to solve addition only up to the training sequence length (Lee et al., 2024), which is different from the true arithmetic algorithm that humans "implement". This raises an important question: *can we make a Transformer truly understand the structure of a task so that it can generalize to the longer sequences without training on them? In other words, can we inject the known structure of a task into a Transformer so that it can automatically length-generalize?*

In this paper, we propose **position coupling**, a simple yet effective method for length generalization that directly embeds the structure of the tasks into a Transformer. In contrast to the vanilla absolute position mechanism assigning unique and consecutive position IDs to each token, we assign the *same* position IDs to certain input tokens that are semantically *relevant*. Coupling such tokens together helps the model learn to solve the task regardless of the length of the given input sequence. For example, in the addition task, it is important to consider the significance of digits, so we couple the positions at the same significance (unique in each operand and the answer).

## 1.1  Summary of Contributions

- We propose **position coupling** to tackle the length generalization problem of decoder-only Transformers. Our approach injects the structure of the task into the absolute position encoding by assigning the same position IDs to relevant tokens (Section 3).

- With position coupling, we achieve a robust and near-perfect generalization up to 200-digit additions by training Transformers on up to 30-digit additions, which is a $6.67\times$ extrapolation of the operand lengths (Figure 1, Section 4). It is promising since it was unclear whether the length generalization on the addition task can be solved reliably with Transformers (Zhou et al., 2024b).

- We theoretically prove by concrete construction that a small (1-layer, 2-head) Transformer equipped with coupled position IDs can add two decimal integers whose lengths are exponential in the embedding dimension (Theorem 5.1). Interestingly, we observe a striking similarity between the attention patterns from our theoretical construction and those extracted from a Transformer trained with a standard optimizer (Section 5.1.1). As a complementary result, we also prove that any 1-layer Transformer without positional information cannot fully solve any permutation-sensitive tasks such as addition (Section 5.2).

- We empirically demonstrate that position coupling can effectively address various tasks beyond addition, including multiplication between $N$-digit and 2-digit integers (Section 6.1, in which we also provide a theoretical construction of a 2-layer Transformer that solves this task for exponentially large $N$). We also verify that position coupling can aid Transformers in learning tasks with multi-dimensional structures (Section 6.2). Moreover, we evaluate position coupling on some other tasks (addition with multiple operands, copy/reverse) in Appendix B.

# 2  Preliminaries

We focus on decoder-only Transformers that solve the tasks using next-token prediction (See Appendix A for a brief background on it). Since we study deterministic tasks with a unique answer, we consider greedy decoding throughout the paper.

## 2.1 Data Formats

Each task in this work is represented as a collection of sequences of the form '(query)=(response)': given a *query*, our task is to infer the *response* correctly. Thus, we only care about the result of the next-token prediction for the '=' token and the tokens in the response (except its last token). That is, we only compute the losses and accuracies for those output tokens.

Previous works commonly observe that data formats play an important role in solving downstream tasks with Transformers because a proper data format enables the model to learn a simple function to solve a task. Here we overview some well-known methods we apply, focusing on the addition task.

**Reversed Format.** Lee et al. (2024) observe that reversing the response leads to improvement in both performance and sample efficiency. For example, '$653 + 49 = 702$' becomes '$653 + 49 = 207$' in a reversed format. This enables a decoder-only Transformer to infer the response from the least significant digit to the most significant digit, similar to how humans add two numbers.

**Zero-padding.** Zero-paddings ensure that the length of both operands in a query is the same and the length of a response is fixed when the length of the operand is given. That is, by padding the query and the response of an $M$-digit + $N$-digit addition with 0's, the input sequence becomes a $\max\{M, N\}$-digit addition with $(\max\{M, N\} + 1)$-digit response. For example, '$653 + 49 = 702$' becomes '$653 + 049 = 0702$'.

**Wrapping with BOS/EOS token(s).** It is conventional in NLP to put BOS/EOS (beginning-/end-of-sequence) tokens at the beginning/end of the sequence. Lee et al. (2024) use the same token '$' for BOS and EOS tokens and observe that it is beneficial to wrap each sequence with the $ token when solving the addition task. We do not observe any significant difference in the performance between sequences with the same and different BOS and EOS tokens.

## 2.2 Positional Embeddings/Encodings (PE)

Vaswani et al. (2017) introduce the absolute positional embedding (APE) to Transformers to inject the positional information into the model. The usual APE works as follows: given an input sequence of tokens, we assign a sequence of consecutive position IDs (integers). Each position ID is mapped to a unique PE vector, and the vector is either added or concatenated to the corresponding token embedding vector. We focus on the learned APE initially proposed by Gehring et al. (2017).

**Length Generalization and PE.** It is actively studied whether PE is a crucial factor in solving the length generalization problem of Transformers. Kazemnejad et al. (2023) argue that decoder-only Transformers with no positional encoding (NoPE) can achieve length generalization of downstream tasks since a Transformer decoder can implicitly capture the generalizable positional information due to its causal nature. However, there is a line of works proposing new PE methods to improve the length generalization performance of Transformers (Li et al., 2024; Ruoss et al., 2023).

# 3 Position Coupling: A Method for Length Generalization

We propose *position coupling*, which assigns position IDs that directly encode the structure of given tasks. Here, we explain the general position ID assignment rule of position coupling in two steps.

First, we partition the tokens of the input sequence. The detailed principles for grouping the tokens differ by task, but the common desiderata are the following: there are two or more groups of consecutive tokens, and each token in a group must have a unique semantic meaning so that a one-to-one correspondence between tokens in different groups can be made.

Next, for each group of tokens, we assign a sequence of consecutive numbers (usually, positive integers) as position IDs, starting from a random number (at training time) or a fixed predetermined number (at evaluation time). We use random position IDs at training time for inducing length generalization by enabling all position embedding vectors to be trained, up to a pre-defined hyperparameter of maximum position ID (`max_pos`).[1] Very importantly, we assign the same position IDs to the tokens in all groups that are relevant to each other for solving the given task: we refer to this as "coupling the positions". Lastly, we set 0 as the position IDs of special tokens like BOS/EOS tokens and the PAD token (padding for minibatch training and evaluation).

---

[1]The hyperparameter `max_pos` determines the maximum testable sequence length that a Transformer can handle. Note that the idea of random assignment of position IDs is similar to *randomized PE* (Ruoss et al., 2023) although it is different since it assigns a sequence of increasing integers which are generally not consecutive.

### 3.1 Position Coupling for Decimal Integer Addition Task

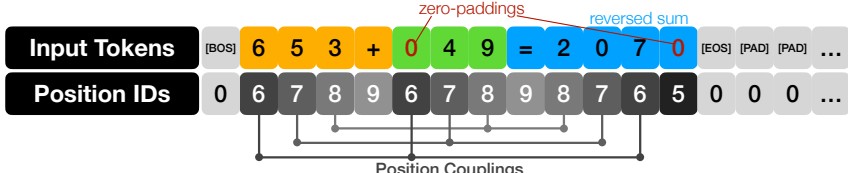

Figure 2: Position coupling for decimal integer addition task, displaying $653 + 49 = 702$ with appropriate input formats. The starting position ID '6' is an arbitrarily chosen number.

We illustrate position coupling for the decimal integer addition task (or addition task for short). To study the length generalization of the addition task, we regard each digit (0–9) as a single token. We will use an explicit example of the addition '$653 + 49 = 702$' for illustration.

Before applying the position coupling, we adopt an input format similar to Lee et al. (2024) so that we reverse the response, but we use zero-padding and wrapping with BOS/EOS token '$' at the same time. For example, '$653 + 49 = 702$' becomes '$653 + 049 = 2070$'.

We partition the tokens in the sequence into three groups: (1) first operand & '+', (2) second operand, and (3) '=' & response (which we call 'sum'). Then each number token is "unique" in the corresponding group in terms of significance, which naturally induces a one-to-one correspondence between (most of) the tokens across different groups. We group '=' and the sum together because these tokens are where we perform next-token prediction.

Now we assign the coupled position IDs to the tokens. Most importantly, we assign the same position ID to the digits of the same significance. Let us say that the random starting number is 6. In our example, we assign 6, 7, and 8 to the tokens in the operands, and assign 5, 6, 7, and 8 to the tokens in the sum in a reversed order: see Figure 2. We remark that, first, we assign 9 as position IDs of '+' and '=' tokens because they are adjacent to the number token with position ID 8, even if there are no 'significances' for those tokens. Second, we assign 5 as a position ID of the most significant digit of the sum (which may be '0' due to the zero-padding) just because it is next to the number token with position ID 6, even though there are no other corresponding tokens in the other groups (operands). We also note that the '+' token is not grouped with the second operand and is not given the ID 5; this is to prevent unnecessary coupling between '+' and the most significant digit of the sum.

**Remark.** A concurrent work by McLeish et al. (2024) proposes an analogous approach for solving arithmetic tasks, while they employ a different input format. We provide a detailed comparison with our work in Appendix A.

**Comparison with Index Hinting.** Even though the idea of implanting the structure of a task into the positional encoding is novel, there is an existing approach named *index hinting* (Zhou et al., 2024a) that applies a similar idea but to the input sequence. Index hinting is an input augmentation technique that places position markers in front of the tokens to couple the semantically relevant tokens. For example, Zhou et al. (2024a) transform '$653 + 49 = 702$' into 'a0b6c5d3 + a0b0c4d9 = a0b7c0d2' with some zero-paddings, where a, b, c, and d are consecutive index hints. Here, the starting hint character a is randomly selected during training, similar to our method of choosing the starting position ID. The reversed format and BOS/EOS tokens can be applied as well.

One way in which index hinting differs from position coupling is that it doubles the input sequence length. This is because the position information and the token information do not merge: the index hints and the normal tokens are mapped to separate token embedding vectors which are alternately placed in the input embedding matrix. As a result, a Transformer must figure out the correspondence between each adjacent pair of an index hint and a normal token. Moreover, the doubled input length requires up to $4\times$ the training time and memory consumption. In contrast, position coupling explicitly combines token and position information: every token embedding and corresponding position embedding are mixed into a single vector. Hence, a Transformer can effortlessly utilize the positional structure of the task, without hurting the training time. We highlight that, as will be mentioned in Section 4.1, position coupling exhibits better length generalization than index hinting.

Another difference is that the index hints should be inferred by Transformers in addition to the normal tokens in the response, which might be an additional burden. Our position coupling circumvents this difficulty, eliminating the need to estimate anything other than the tokens in the original response.

# 4 Experiments on the Addition Task

In this section, we empirically demonstrate that position coupling allows extensive length generalization of Transformers on the addition task. We delve into the impact of training length and architecture on the length generalization performance and provide comparisons with NoPE, APE with a random starting position ID (we call random-start APE), and index hinting (Zhou et al., 2024a).

**Data Sampling.** We opt for the balanced sampling in terms of the number of digits (Nogueira et al., 2021). Given the maximum number of digits $D_{\max}$, we do balanced sampling for each operand in two steps. First, we sample the number of digits $D \in [1, D_{\max}]$ uniformly at random. Next, we sample an operand from $[10^{D-1}, 10^D - 1]$ uniformly at random, except for $D = 1$ where we sample from $[0, 9]$. This procedure addresses the imbalance problem in the number of digits of operands.

**Model and Training.** We train decoder-only Transformer models from scratch. We properly choose `max_pos` so that the maximum testable length of summands is 200. We do not use packing or shifting for simplicity of implementation. Since we manually put coupled position IDs with a random starting index during training, we can train all the positions without packing and shifting. We run each experiment 8 times with 2 different random seeds for data generation and 4 different random seeds for model initialization & stochastic optimization unless mentioned otherwise. We summarize all hyperparameters in Appendix C.

## 4.1 Results

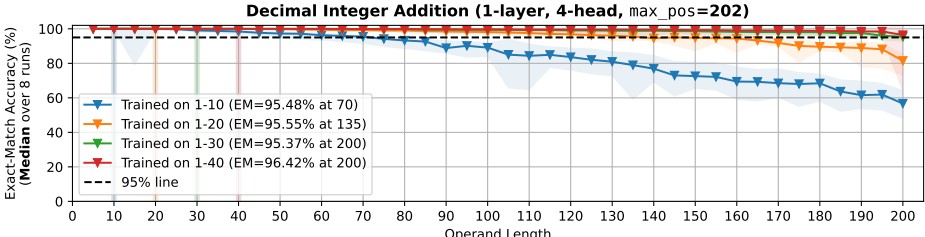

Figure 3: Ablation on the trained operand lengths (1-layer 4-head models).

**Longer Trained Sequences Lead to Longer Generalizable Lengths.** We train 1-layer 4-head models with $D_{\max} \in \{10, 20, 30, 40\}$ and evaluate them on up to 200-digit additions. For each run of training, we choose and evaluate the best model in terms of the validation loss for 200-digit additions. The result is showcased in Figure 3. We decide that a model successfully generalizes to a certain length of operands (referred to as "generalizable length") if the median EM accuracy exceeds 95%.

We observe that the generalizable length becomes longer as we train on longer training sequences. The generalizable length is 70 for the models trained on additions involving 1–10 digits, 135 for models trained on 1–20, and 200 for 1–30 and 1–40. We believe that we could achieve even longer generalizable length for the models trained on 1–40 if we use a larger `max_pos`. We note that we could scale up the generalizable length to 500 by training with lengths 1–160: refer to Appendix B.1. Although each test sample contains the operands of the same length, we also provide an extended evaluation on test samples with operands of different lengths: see Appendix B.2.

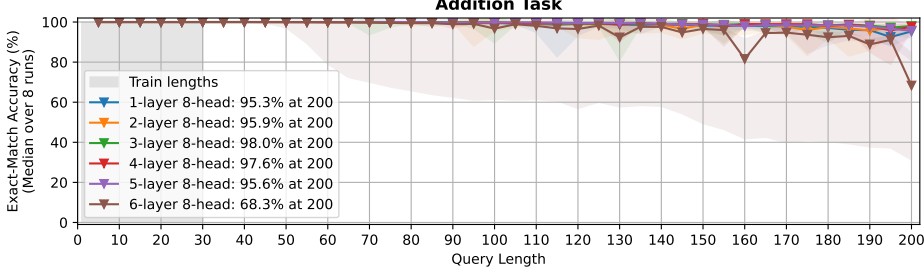

Figure 4: Ablation on the number of layers (trained with position coupling).

**Ablation on Number of Layers.** Since Zhou et al. (2024a) and Zhou et al. (2024b) choose 6-layer 8-head models as their base models, we also test our method to deeper models. The results evaluated with models trained on 1–30 digits are displayed in Figure 4, whose experimental details are listed in Table 2. Overall, the performances are well extrapolated to test lengths (longer than trained lengths)

and are similar for the models with 1–5 layers. For the 6-layer model, however, the performance slightly deteriorates. We hypothesize that the performance degradation is due to the bad implicit bias of deeper models (learning shortcuts only to achieve in-distribution generalization) when learning a simple algorithm to solve the task. Since the theoretical construction of a 1-layer addition Transformer (that will appear in Section 5.1) naturally extends to larger architectures, deeper models have at least as much generalization capability as shallower models. We believe exploring a theoretical explanation for the bad implicit bias of large models on low-complexity tasks is a promising research direction. We also highlight that we present median accuracies over multiple runs, while Zhou et al. (2024b) report maximum accuracies. To better the comparison, we also report the maximum accuracies (for the experiments in Figures 3 and 4) in Appendix B.3, showing that our 6-layer models can achieve near-perfect generalization for up to 200-digit additions as well.

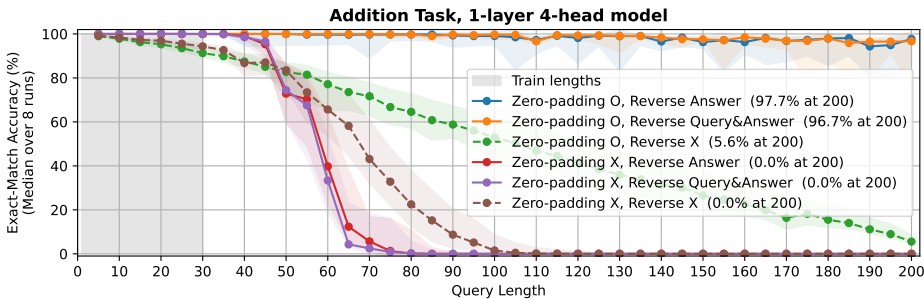

Figure 5: Ablation on the data formats (1-layer 4-head models trained with position coupling).

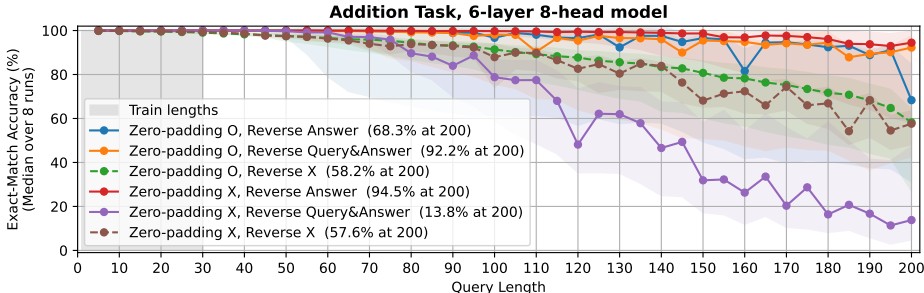

Figure 6: Ablation on the data formats (6-layer 8-head models trained with position coupling).

**Ablation on Data Formats.**  Our input format is primarily selected to simplify the algorithm of solving the addition task, not through extensive ablation studies. Thus, we are not arguing that our choice of input format is empirically optimal for training Transformers. However, since the data format is one of the crucial factors in general machine learning, we test various formats for 1-layer 4-head (Figure 5) and 6-layer 8-head models (Figure 6). The results clearly show that the performance varies with input formats, as they affect the complexity of the algorithm that the model should learn.

Small models (1-layer 4-head) achieve near-perfect generalization when the numbers are zero-padded and the response or all the numbers are reversed. We believe this is because the combination of zero-padding and reversing enabled a small Transformer to learn a simple length-generalizing algorithm. Zero-padding seems crucial since it aids length generalization to some extent even without reversing. Without reversing any numbers, however, even the in-distribution performance slightly decays.

Larger models (6-layer 8-head) perform better than small models when the numbers are no longer zero-padded or reversed. We believe this is because the task-solving algorithm without reversing or zero-padding that the model should learn is more sophisticated, which larger models can learn more easily. Contrarily, we observe a slight degradation in performance when we add zero-padding and reversing in the larger model, which suggests that the model may have learned a "shortcut" due to its (overly) strong expressive power relative to the problem's complexity.

**Comparison with NoPE and APE (with random starting position ID).**  Our experiments demonstrate that simple PE methods like NoPE and random-start APE cannot length-generalize well on the addition task. In particular, we implement random-start APE to mimic the training process with the usual APE combined with packing and shifting. The results showcased in Figure 1 imply that naively

training all position embeddings does not necessarily help produce a strictly better model in terms of length generalization than that does not use position embeddings at all. We also remark that even training itself is difficult for shallower models (e.g., 1-layer) with NoPE and random-start APE.

**Comparison with Index Hinting.** We test index hinting by running the code we implemented ourselves since the original code is unavailable. From Figure 1, we observe that index hinting indeed helps the model to length-generalize more than the baselines (NoPE & random-start APE). However, the generalizable lengths of the models trained with index hinting do not extend further than 50; the models completely fail starting from the length 70. We also observe that Transformers with index hinting require enough depth to achieve high enough training and in-distribution validation accuracies. Particularly, the training accuracies of 1-layer models do not deviate from near zero. Thus, we only present the results for the 6-layer 8-head model as done by Zhou et al. (2024a).

**Comparison & Combination with RoPE.** We also examine the possibility of combining position coupling and RoPE (Su et al., 2024): See Appendix B.4 for the experimental results and details.

## 5 Theoretical Analyses on 1-layer Transformers

In the previous section, we provided empirical results exhibiting the outstanding performance of position coupling. One might ask *why* and *how* position coupling works so effectively. In Section 5.1, we provide a theoretical explanation by carefully constructing a 1-layer Transformer model that is capable of solving the addition task involving exponentially long operands when the input is encoded with position coupling. We also present the necessity of proper positional information for a 1-layer Transformer to solve the addition task in Section 5.2.

### 5.1 1-layer Transformer with Coupled Positions can Perform Long Additions

For the sake of simplicity of presentation, we consider a Transformer without any normalization layers, as conventionally done in theoretical constructions by previous works (Awasthi and Gupta, 2023; Yun et al., 2020a,b). For the sake of completeness, readers can find a mathematical formulation of the decoder-only Transformer architecture in Appendix D.

**Theorem 5.1.** *With the input format described in Section 3.1, there exists a depth-1 two-head decoder-only Transformer with coupled positions that solves the addition task with next-token prediction. Here, the operand length is at most $2^{\lfloor (d-17)/2 \rfloor} - 2$, where the embedding dimension is $d \geq 21$.*

We provide our proof in Appendix E. We highlight that our proof is constructive and does not rely on any universal approximation result of neural networks.

Theorem 5.1 shows that a 1-layer 2-head Transformer is *sufficient* for implementing addition between two *exponentially long* integers. We emphasize that this result can be naturally extended to larger architectures with more layers/heads, with the help of residual connections.

#### 5.1.1 Probing the Attention Patterns in Trained Transformers with Position Coupling

We discover a striking similarity between the attention patterns in our theoretical construction (Theorem 5.1) and those extracted from a Transformer trained with position coupling and a standard optimizer. In particular, the manually constructed attention patterns described in Tables 11 and 17 in Appendix E closely resemble the actual attention patterns in Figure 7.[2] Drawn from this discovery, we claim that a Transformer trained with position coupling spontaneously learns two separate components of the addition task: (1) adding two numbers without carries, and (2) predicting the carries.

Let us revisit the example in Figure 2 and consider predicting '7' (position ID 6) as the next token of '0' (position ID 7). Note that the token '7' is the result of combining the digit-wise sum 6+0=6 and a propagated carry 1. To find out the sum without carry, it is enough for the model to attend to the *two* previous positions with ID 6: tokens '6' and '0'. On the other hand, to predict the carry, the model may attend to the *three* positions with ID 7: tokens '5', '4', and '0'. The reason why we should care about '0' is that considering the sum 5+4 (=9) of the two digits in the operands is not sufficient to determine the existence of the carry. By looking at the token '0' in the response (with position ID 7), we can detect that the actual sum in this position is $\underline{1}0$ (=5+4+**1**, where **1** is another carry propagated from the previous position) and hence we need to propagate a carry $\underline{1}$ to the next position (with ID 6).

---

[2]Note that they match up to matrix transpose, which is due to the difference in the formulations.

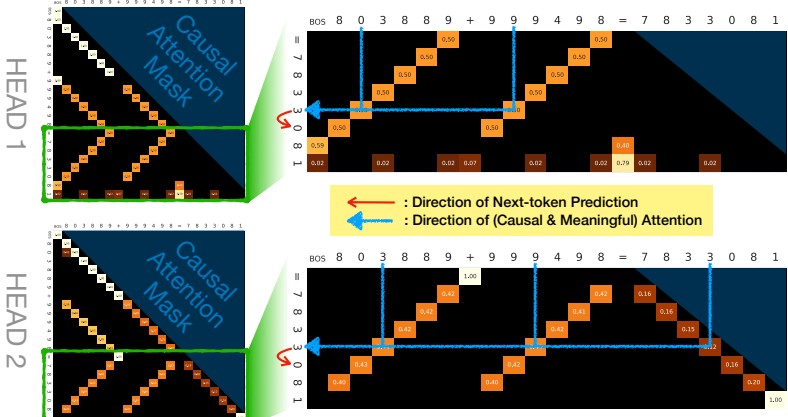

Figure 7: Probing attention matrices of a 1-layer 2-head Transformer with position coupling, trained on up to 5-digit additions. **(Left)** There are two heatmaps (clipped to zero below 0.01) corresponding to the (transposed) attention matrices observed from the attention heads. Averaged over 10K sequences of 6-digit additions. **(Right)** We magnify parts of the attention matrices that are involved in inferring the response (sum). The arrows explain the process of inferring the next token '0' from '3'.

Now we inspect the aforementioned claim by examining the attention matrices of an actual trained Transformer. In the model, we discover two different patterns of attention matrices,[3] playing distinct roles. The first attention pattern (top of the figure) seems to correspond to the addition without carries: each token in the response (including '=') attends to two positions needed to find out the sum without carry. Conversely, the second attention pattern (bottom of the figure) seems to correspond to the carry prediction: again, each token in the response attends to three positions required to find out the carry.

**Remark.** Similarly to our analysis, Quirke and Barez (2024) study the attention patterns of a 1-layer 3-head decoder-only Transformer model trained solely on 5-digit addition. They also observe that each head handles different subtasks of addition, such as digit-wise summation and carry detection.

## 5.2 1-layer Transformers Require Positional Information

In Section 4.1, we observed that 1-layer Transformers fail to perform the addition task without position coupling. Here, we provide a partial result that theoretically explains why this happens inevitably, particularly in the case of NoPE. We start with a general proposition: a 1-layer Transformer without positional encoding cannot distinguish queries that are identical up to permutation when inferring the first token of the response using greedy next-token prediction.

**Proposition 5.2.** *Consider any depth-1 finite-head decoder-only Transformer model $\mathcal{T}$ without positional encoding (NoPE). Given an input sequence $\mathcal{I}$ and its arbitrary permutation $\mathcal{I}'$, if the last tokens of $\mathcal{I}$ and $\mathcal{I}'$ are identical, then the next tokens predicted by $\mathcal{T}$ will also be identical for both sequences when applying a greedy decoding scheme.*

The proof is deferred to Appendix F. According to the proposition above, the 1-layer Transformer without positional encoding will always output the same values starting from the '=' token, provided that the combination of query tokens is identical, even if their order varies. However, the addition task is permutation-sensitive, meaning that the permuted queries may result in different responses. Therefore, the 1-layer Transformer cannot completely solve the task without positional encoding. It is important to note that this result remains unchanged regardless of the input format: neither reversed format nor index hinting provides any benefit. We also highlight that this impossibility result can be extended to any other permutation-sensitive tasks, such as arithmetic tasks and copy/reverse tasks.

Based on this, we write code to directly calculate the maximum EM accuracy on the $m$-digit addition task that a 1-layer decoder-only Transformer can achieve (see Appendix F for the code). The accuracies rapidly decrease to zero: 6.2% for 3-digit addition, 1% for 4-digit integers, and 0.13% for 5-digit integers. We leave it for future work to investigate the necessary conditions of the architecture for implementing addition when other positional encoding schemes are employed.

---

[3]The attention matrices depicted in Figure 7 are square, lower-triangular (due to causal attention pattern), and row-stochastic (all entries are nonnegative and the sum of each row equals 1).

# 6 Applying Position Coupling Beyond Addition Task

To demonstrate the versatility of position coupling, we consider two other tasks in this section: $N \times 2$ multiplication and a two-dimensional (2D) task. Other example tasks (e.g., addition with multiple summands, copy/reverse allowing duplicates) can be found in Appendix B.

## 6.1 Position Coupling for $N \times 2$ Multiplication Tasks

Here, we study length generalization on the $N$-digit $\times$ 2-digit multiplication task in terms of the length $N$ of the first operand, while fixing the length of the second operand by 2. Similar tasks have been studied before (Duan and Shi, 2023; Jelassi et al., 2023); we discuss further in Appendix A.

We reverse and zero-pad the response, setting the length of it as $N + 2$. We couple the position starting from the least significant digits of both operands and response, decrementing the ID as we move to their most significant digits: see Figure 17 in Appendix B.5. The experimental results showcased in Figure 8 verify the efficacy of position coupling compared to NoPE and random-start APE. We observe that a 1-layer model fails even with position coupling, even for training. However, as the depth increases to 2 or more, it immediately becomes capable of length generalization.

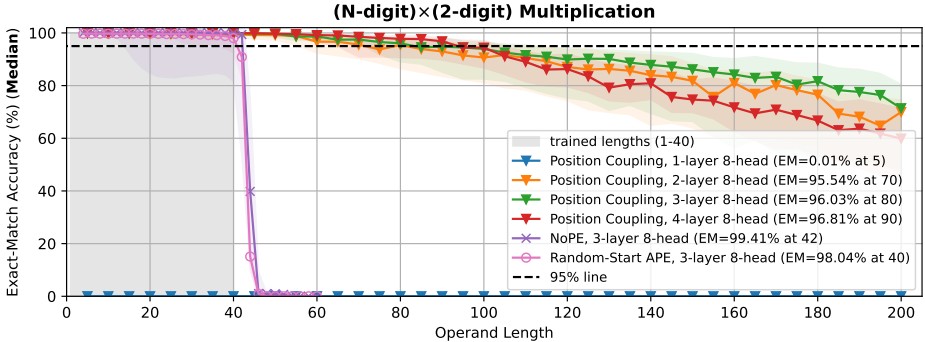

Figure 8: $N \times 2$ multiplication task, trained on sequences of length 1–40.

Unlike addition, position coupling for $N \times 2$ multiplication is less intuitive, as predicting the token in the middle of the response requires multiple digits from both operands while each token in the response is linked with at most 2 tokens in the query. Perhaps surprisingly, we can still construct a Transformer that provably solves this task for exponentially long sequences.

**Theorem 6.1.** *Given an appropriate format of the input sequence, there exists a depth-2 decoder-only Transformer model with coupled positions that can perform the $N \times 2$ multiplication task with next-token prediction. Here, the number of total heads is 10 and the length of the first operand is at most $2^{\lfloor (d-34)/6 \rfloor} - 3$, where we denote the token embedding dimension by $d \geq 46$.*

We defer the proof to Appendix G. This result suggests that the proposed position coupling scheme for the $N \times 2$ multiplication task sufficiently captures the inherent structure of the task, and thus provides the potential for the trained model to generalize across unseen lengths. Also, we believe that Theorem 6.1 is optimal in terms of the number of attention layers, as the depth-1 model exhibits total failure even for in-distribution samples in our experiment.

## 6.2 Two-dimensional Position Coupling for Minesweeper Generator Task

Now, we investigate the extension of position coupling for handling a 2D task, where the query and the response are originally 2D objects. In particular, we define and investigate a task we call *minesweeper generator*. Given a rectangular board where each cell is filled with either 'M' (mine) or '∗' (an empty cell), the task is to generate a new board of the same size, having each cell filled with:

- 'M', if the corresponding cell in the original board contains 'M';
- The count of mines in 8 adjacent cells, if the corresponding cell in the original board contains '∗'.

**Data Format & Position Coupling.** We introduce two position coupling modules: one for the row direction and another for the column direction. Following this, we flatten the board to feed it into a Transformer: see Figure 9. Within the model, an embedding vector for each token (cell) is generated by adding the token embedding vector and corresponding two PE vectors.

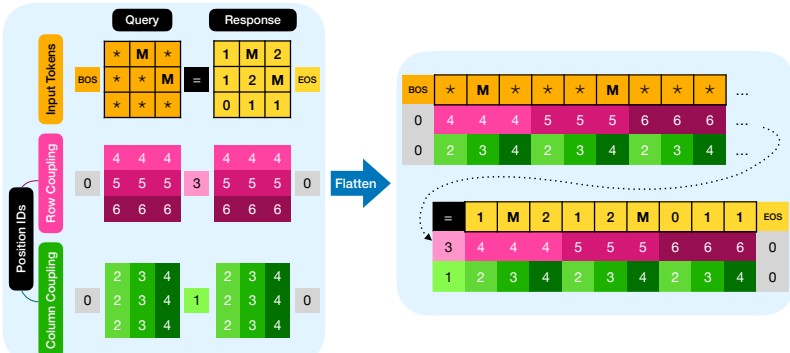

Figure 9: Position coupling for the two-dimensional 'minesweeper generator' task. **(Left)** The idea of assigning coupled position IDs. **(Right)** The model receives a flattened sequence of input tokens and two-dimensional position IDs.

**Experiments.** To assess the efficacy of position coupling, we contrast its performance with NoPE. The training samples are designed with the width and height of the board between 5 and 9 inclusively. We allow the width and height to be different for training samples. We evaluate the test performance on a square board with a width between 5 and 14 inclusively. We also employ a 4-layer 8-head model for position coupling and a 6-layer 8-head model for NoPE. In particular, for position coupling, we use the same embedding layer for both position coupling modules, as this approach empirically performs better than using distinct embedding layers for each module (see Appendix B.8).

The experimental results are described in Figure 10. Position coupling maintains over 98% accuracy until a width of 12 and near 90% accuracy even at a width of 14. In contrast, NoPE fails even for in-distribution samples. One might be concerned that the generalizable length of 12 seems only slightly higher than the trained length of 9. However, we stress that our query is a 2D board, therefore the actual length generalization is from 81 to 144.

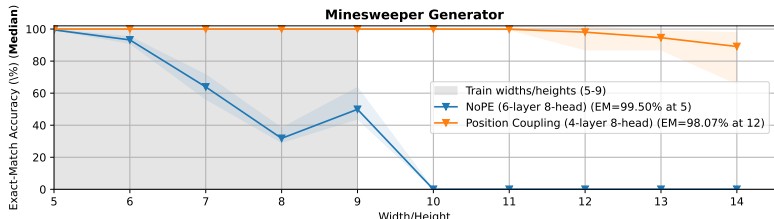

Figure 10: Minesweeper generator task, trained on sequences of length $(5\text{–}9)\times(5\text{–}9)$.

# 7 Conclusion

Achieving length generalization of Transformers even in the simple case of the addition task has been a challenge that received a lot of attention. We propose position coupling, a variant of learned APE, which enables capturing task structure to improve the length generalization performance of Transformers for addition. We show that a Transformer trained on 1–30 digit addition can generalize up to 200-digit addition. We also provide the construction of a 1-layer Transformer model capable of adding two exponentially long integers when position coupling is applied. Furthermore, we verify the efficacy of position coupling for length generalization in other arithmetic and algorithmic tasks.

**Limitations & Future Directions.** We intentionally limited ourselves to the tasks with an explicit structure between the tokens in each sequence. This is because we are proposing a method to instill the *known* structure of the task into a Transformer by training on short sequences. Designing the coupling of positions for tasks whose structure is implicit or black-box (e.g., for general NLP tasks) remains a fascinating next step: we leave the methodology for uncovering hidden structures and autonomously creating appropriate couplings (without manually designing them) for future work.

We also leave two challenging arithmetic tasks to length-generalize for future work. One is the addition with a varying number of summands, i.e., determining if the model can generalize to summing multiple integers when trained on samples with fewer summands. The second task is multiplication, where the lengths of both operands can vary. Note that our method is further extended to solve these two challenging length generalization problems in a recent work (Cho et al., 2024).

## Acknowledgments and Disclosure of Funding

This work was partly supported by a National Research Foundation of Korea (NRF) grant (No. RS-2024-00421203) funded by the Korean government (MSIT), and an Institute for Information & communications Technology Planning & Evaluation (IITP) grant (No.RS-2019-II190075, Artificial Intelligence Graduate School Program (KAIST)) funded by the Korean government (MSIT). HC, JC, and CY acknowledge support from a Google Gift on the research related to Long Context Transformers. The experiments contained in this work were supported in part through a Google Cloud Platform Credit Award.

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

# Contents

# A Omitted Backgrounds

## A.1 Next-token Prediction with Decoder-only Transformers

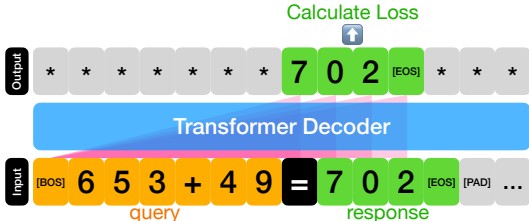

Figure 11: Schematic of solving an integer addition task instance using next-token prediction with a decoder-only Transformers. BOS/EOS mean beginning-/end-of-sequence tokens, respectively. PAD means a padding token, used for matching the sequence lengths in a single minibatch of sequences. Here we assume a basic input format (plain, no zero-padding), which is different from that we used in our experiment.

A decoder-only Transformer returns an output sequence of the same length as the input sequence. One difference from a Transformer encoder is that the attention mechanism in a Transformer decoder occurs only in a single forward direction due to the causal attention mask. Due to this causal nature, the Transformer decoder is mostly used for inferring the next token of each token, just based on the information of the current and the previous tokens.

## A.2 Related Works

**Length Generalization in the Addition Tasks.** Lee et al. (2024) observe that reversing the output in the addition task enables the model to learn a simple function. Shen et al. (2023) propose "Random Spacing" and "Recursive Scratchpad", achieving near-perfect generalization from 10-digits to 12-digits addition. Zhou et al. (2024a) introduce "index hints", position markers placed in front of each token, in both the input and output of addition tasks. Most recently, Zhou et al. (2024b) demonstrate a possibility of extrapolation to the length 100 with training length 1–40 in the addition task by combining appropriate input format and advanced PE, yet they also observe that the performances are not robust and highly depend on the random seeds.

**Length Generalization in the $N \times M$ Multiplication Task ($M$ is fixed).** Jelassi et al. (2023) investigate $N \times 3$ using an encoder-only model and Duan and Shi (2023) study $N \times 1$ with an encoder-decoder Transformer architecture. Besides the architectural difference, Jelassi et al. (2023) fail to observe length generalization with RPE and only achieve it by supplementing a small number of long samples to the training set. Furthermore, although Duan and Shi (2023) provide perfect length generalization results even for test samples $10\times$ longer than those observed during training, their approach requires a retraining step with hand-crafted bias correction on attention score matrices.

**Analyzing Length Generalization in Theoretical Perspectives.** An emerging line of research seeks to theoretically address why length generalization is difficult and under what conditions it can be achieved. In Abbe et al. (2023), the authors demonstrate that various neural network models have an implicit bias towards min-degree interpolators, which may not be ideal for various reasoning tasks. Xiao and Liu (2023, 2024) investigate problems whose reasoning processes can be formulated as directed acyclic graph (DAG) structures, introducing the concept of *maximal input element distance* to identify a sufficient condition for length generalization. Recently, Ahuja and Mansouri (2024) formulate the conditions of function classes required to guarantee the length generalization of the empirical risk minimizer function.

**Comparison with McLeish et al. (2024)** A very recent concurrent work by McLeish et al. (2024) proposes a new position embedding method called "Abacus". From a methodological perspective, Abacus is almost identical to our position coupling except for two main differences: Abacus reverses both the query and the response and does not use padding. From now on, we outline the differences between their work and ours beyond the methodology.

In terms of the model architecture, they use a depth-16 decoder-only Transformer model. They combine their method with looped Transformers and input injection and report an improved performance. In contrast, our main results are obtained with shallower models (up to 6 layers) with standard Transformer architecture of stacked decoder layers.

Besides the addition task, they study multiplication, sorting, and Bitwise OR. On the other hand, we study multiplication, triple addition, copy/reverse, and a 2D task. Specifically, for the multiplication task, their study mainly considers the case where the length of both operands could vary up to 15. In contrast, we focus solely on the $N \times 2$ task, fixing the length of the second operand by 2. While we achieve length generalization up to 90-digit multiplication by training the model on up to 40-digit multiplication, they report near-perfect in-distribution performance but poor length generalization.

Finally and notably, we provide novel theoretical analyses, including (1) the constructive proof that a depth-1 Transformer equipped with position coupling can completely solve the addition task for exponentially long digits and (2) the impossibility of the same model being capable of the addition task. We also present theoretical results for the $N \times 2$ multiplication task.

# B More Applications & Experiments of Position Couping

## B.1 Decimal Integer Addition Task: Scale-up to Length of 500

Here, we demonstrate the scalability of our proposed position coupling approach for large lengths of up to 500. Specifically, we again train a depth-1 decoder-only model for the addition task and evaluate the performance for instances with up to 500 digits. The results are shown in Figure 12. We notice that at a train length of 160, we achieve excellent length generalization for 500-digit addition. On the other hand, training on sequences of length up to 40 or 80 is insufficient for extreme length generalization. The results demonstrate that position coupling, as an approach, is highly scalable.

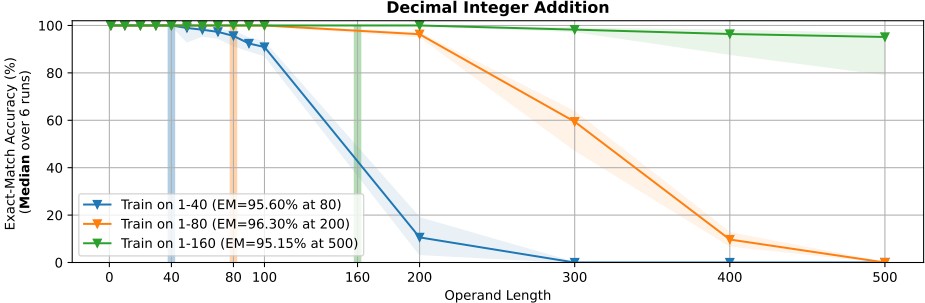

Figure 12: The exact-match accuracies obtained by training a depth-1 transformer on the addition task. We see that while training with sequences of length up to 40 and 80 is insufficient for generalization to large lengths, at training length 160 we achieve strong performance for lengths up to 500. The experimental details can be found in Table 3.

## B.2 Decimal Integer Addition Task: Operands of Different Lengths

In the main text, we mainly focus on the test examples of additions where the lengths of both operands are the same. For the sake of completeness, we also report the evaluation results for the cases where the operand lengths can be different (although the zero-padding is applied to ensure the consistency of the data format). See Figure 13.

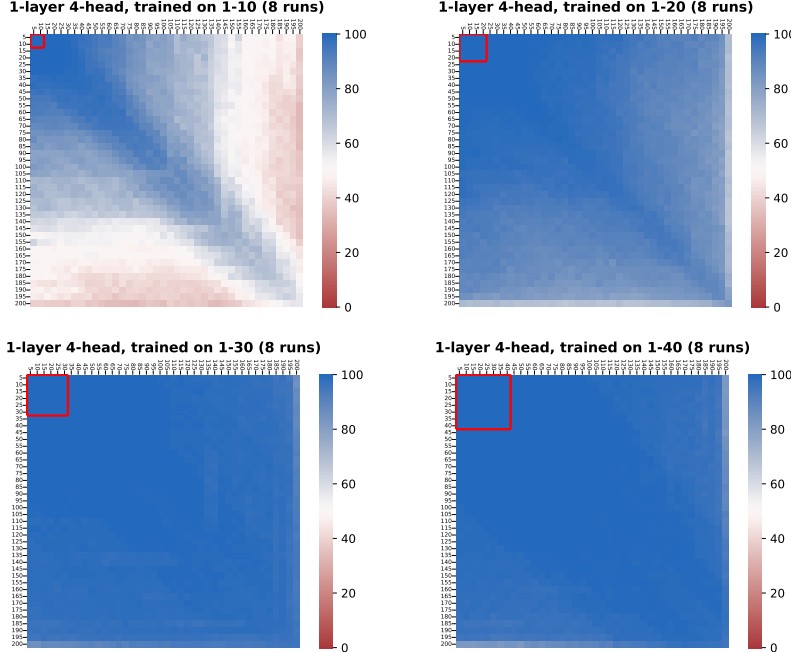

Figure 13: Exact-match accuracies (%) on additions with operands of different lengths. Different heatmap corresponds to different trained length of operands (1–10, 1–20, 1–30, and 1–40, expressed with a red box for each). For each heatmap, the $x$-axis and the $y$-axis are for the length of the first and the second operand, respectively.

### B.3 Decimal Integer Addition Task: Maximum Exact-Match Accuracies

A prior work by Zhou et al. (2024b) provides a similar analysis on the addition tasks as ours. Combining appropriate input format and advanced PE, they achieve ≥98% EM accuracy for 100-digit additions with a 6-layer 8-head model trained on 1–40. Moreover, they achieve a generalizable length of 45 for a model trained on 1–30, 25 for 1–20, and 10 for 1–10 (no length generalization). One big difference between their analysis and ours is they report the *maximum* accuracy for each testing length over trials, while we report the *medians*. Thus, we choose a bit lower threshold (95%) for generalizability than theirs. For a better comparison with Zhou et al. (2024b), we report the maximum exact-match (EM) accuracies. See Figures 14 and 15.

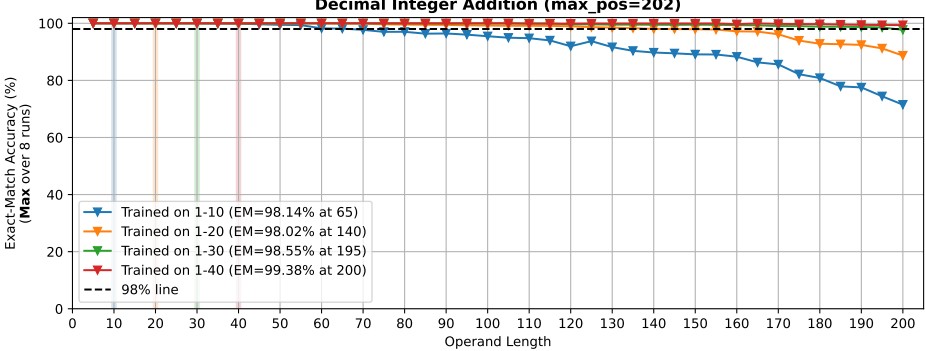

Figure 14: Ablation on the trained lengths (1-layer 4-head model trained with position coupling). Here, we report maximum EM accuracies over 8 runs for each tested length.

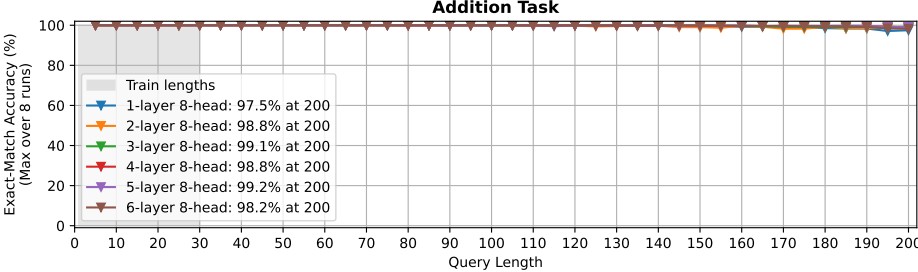

Figure 15: Ablation on the number of layers (trained with position coupling). Here, we report maximum EM accuracies over 8 runs for each tested length.

### B.4 Decimal Integer Addition Task: Comparison & Combination with RoPE

We examine further the possibility of combining our position coupling and RoPE (Su et al., 2024).

RoPE incorporates the positional information into the key and the query vectors by rotating them. Suppose a position ID $m$ is assigned to a key vector $k$, for every two consecutive entries of $k$ (i.e., $(k_{2i-1}, k_{2i})$), we rotate by a predefined angle $\theta_i$ multiplied by $m$:

$$\begin{pmatrix} k_{2i-1} \\ k_{2i} \end{pmatrix} \mapsto \begin{pmatrix} k_{2i-1} \cos m\theta_i - k_{2i} \sin m\theta_i \\ k_{2i-1} \sin m\theta_i + k_{2i} \cos m\theta_i \end{pmatrix}.$$

We also apply a similar rotation to the query vectors (say, $q$ with position ID $n$). As a result, the attention score for this key-query pair becomes a function of $k$, $q$, and the relative distance $n - m$.

Unlike the original implementation of RoPE, we apply rotations based on the re-assigned position IDs according to our position coupling method. We incorporate this RoPE variant into every attention head of every layer. One more difference in implementation is that, during training, we randomly sampled an integer scaler $\ell \in \{1, 2, 3, 4, 5\}$ and multiplied it by the rotation angle. By such random re-scaling of rotation angles, we expect the model could handle unseen large rotation angles at test time.

The result on 12-layer models is showcased in Figure 16 (the orange line). Unlike vanilla RoPE (the blue line), which fails immediately outside the trained lengths (1–40), our combination of RoPE and position coupling achieves a much better generalization up to operand lengths 100.

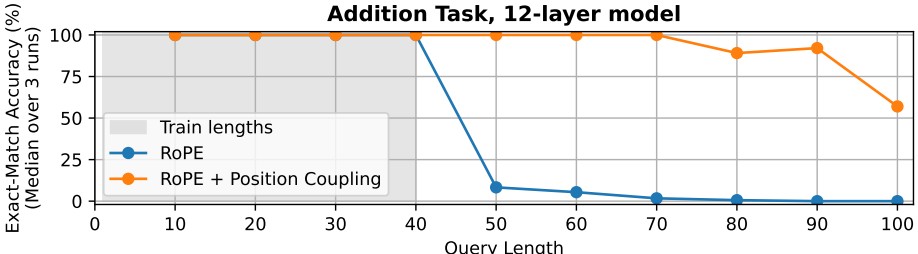

Figure 16: RoPE-based position coupling, 12-layer model.

## B.5 Position Coupling for $N \times 2$ Multiplication Tasks

We present an example of the position coupling method for $N \times 2$ ($N$-digit by 2-digit) multiplication, which is omitted from the main text. See Section 6.1 for the experimental results.

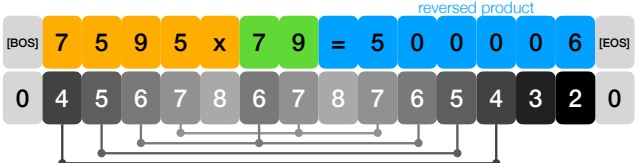

Figure 17: Illustration of position coupling for $N \times 2$ multiplication task.

## B.6 Addition Task with Multiple Summands

The position coupling scheme for the vanilla addition task (with two operands) can naturally extend to the addition task with multiple summands: assign position IDs in ascending order from most significant digits to least significant digits for every operand and the response. Here, we focus on the addition of three summands.

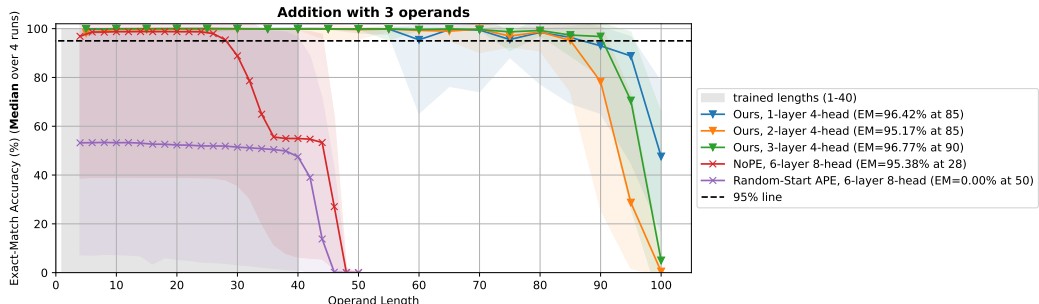

Figure 18: Exact-match accuracy (median over 4 runs) for triple addition task, trained on sequences of length 1-40 with position coupling, NoPE, and random-start APE. For further experiment details, refer to Table 6.

**Experiments.** We train on sequences with operands of 1–40 digits. Our choice of `max_pos` is 102, so we test the operands of up to 100 digits. We investigate the performance of 3 different architectures, each with a different depth. The experimental results are described in Figure 18. 1-layer models keep their generalization capability until 100 digits, whereas the 3-layer models exhibit great stability across random seeds and achieve the highest generalizable length of 90.

Lastly, we note that the result of Theorem 5.1 can be extended to addition tasks with multiple summands with slight adjustments to the feed-forward layer in the construction.

### B.7  Position Coupling for Copy/Reverse Tasks

**Data Format & Position Coupling.**  Each token of the query sequence is a digit (10 distinct characters). We couple the positions in the query and the response by their correspondence. Note that the position ID assigned to the equal token is different for the two tasks because as per our design principle (Sec 3.1), the equal token is grouped to the response tokens and position IDs have to be consecutive numbers within each group.

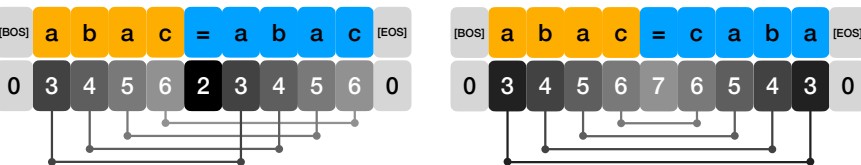

Figure 19: Illustration of position coupling for copy/reverse tasks.

**Experiments.**  We compare position coupling with NoPE and random-start APE. We train a model on lengths 1–40 and evaluate its performance on lengths from 5 to 300, at intervals of 5. While a 1-layer 4-head model is used for the position coupling, we observe that the same architecture fails to memorize training samples for both NoPE and random-start APE. Therefore, we use a 6-layer 8-head model for the latter cases as it is commonly used in the literature (Zhou et al., 2024a).

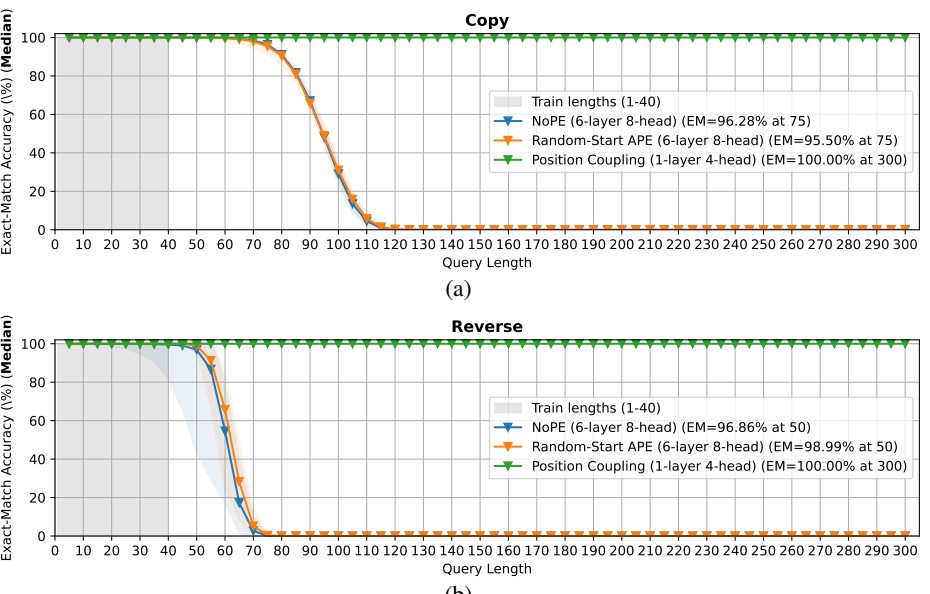

Figure 20: Exact-match accuracy (median over 4 runs) for (a) copying task and (b) reversing task, trained on sequences of length 1–40 with position coupling, NoPE, and random-start APE. For further experiment details, refer to the Table 7.

The experimental results are described in Figure 20. For both copy and reverse tasks, position coupling exhibits near-perfect accuracy across the entire test length (7.5× for the trained length). In contrast, NoPE and random-start APE immediately fail to length-generalize.

### B.8  Position Coupling for Minesweeper Generator Tasks

Here, we present the extra experimental results for training the minesweeper generator task with position coupling. Specifically, we compare the performance of two configurations: one where the model shares the same positional embedding layer for both position coupling modules, and another where the model uses separate positional embedding layers for each position coupling module.

The results are described in Figure 21. When sharing the same positional embedding layer, position coupling achieves over 98% accuracy on a 12×12 board, and maintains near 90% accuracy on a

14×14 board. However, with distinct positional embedding layers, position coupling only successfully generalizes to a 10×10 board. We currently do not have a clear explanation for why the former method exhibits significantly better performance than the latter one. We leave the investigation and explanation of this phenomenon for future work.

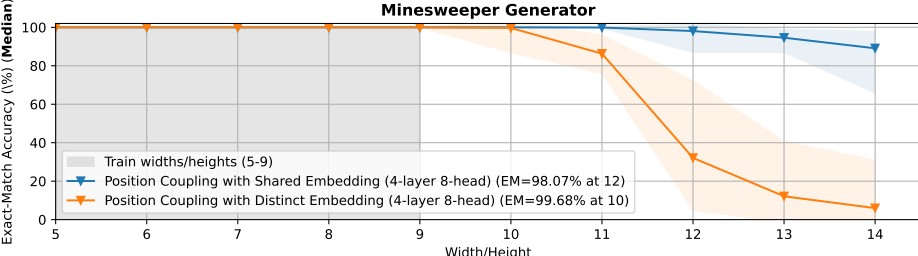

Figure 21: Exact-match accuracy (median over 4 runs) for minesweeper generator task, trained on sequences of length (5–9)×(5–9) with position coupling. For further experiment details, see Table 8.

# C  Experiment Details and Hyperparameters

Position coupling can be easily implemented on top of usual libraries of training transformer models like HuggingFace (Wolf et al., 2019) and Flaxformer[4] since these libraries support an arbitrary array of position IDs (in the case of using APE). All we need is to build up a short routine implementing the assigning rule of position IDs when establishing the dataset and data loaders. To compare with NoPE, we use the code base provided by Kazemnejad et al. (2023) for most of the experiments.[5] It contains a custom implementation of decoder-only T5 (Raffel et al., 2020) established on top of PyTorch (Paszke et al., 2019) and Huggingface, including several PE methods. We additionally implement a custom RMSNorm module (Zhang and Sennrich, 2019) and various positioning schemes of normalization layers (e.g., PreNorm (Xiong et al., 2020), PostNorm (Vaswani et al., 2017), and their combination), to follow the implementation details of Zhou et al. (2024b).

Table 1: Hyperparameter summary for decimal integer addition task: comparison between trained lengths (Figures 3 and 14).

| Hyperparameter | Value |
| --- | --- |
| Architecture | Decoder-only Transformer |
| Number of Layers | 1 |
| Number of Attention Heads | 4 |
| Embedding Dimension | 512 |
| Dimension per Head | 128 |
| Hidden Width of Feed-forward Layer | 2048 |
| Activation Function of Feed-forward Layer | GEGLU (Shazeer, 2020) |
| Normalization Layer | RMSNorm (Zhang and Sennrich, 2019) |
| Normalization Layer Position | PreNorm and PostNorm |
| Training Steps | 50,000 |
| Batch Size | 1,000 |
| Optimizer | Adam (Kingma and Ba, 2015) |
| Learning Rate (LR) | 0.0001 |
| LR Warm-up | Linear (From 0 to LR), 1% of total steps |
| LR Cool-down | Cosine Decay (From LR to 0.1LR) |
| Maximum Position ID (`max_pos`) | 202 |
| Training Dataset Size | 1,000,000 |
| Evaluation Dataset Size | 100,000 |
| Device | NVIDIA RTX A6000 48GB |
| Training Time | $\leq$ 10 hours |

---

[4] github.com/google/flaxformer
[5] github.com/McGill-NLP/length-generalization

Table 2: Hyperparameter summary for decimal integer addition task: comparison between the number of layers (Figures 4 and 15).

| Hyperparameter | Value |
|---|---|
| Architecture | Decoder-only Transformer |
| Number of Layers | 1-6 |
| Number of Attention Heads | 8 |
| Embedding Dimension | 1024 |
| Dimension per Head | 128 |
| Hidden Width of Feed-forward Layer | 2048 |
| Activation Function of Feed-forward Layer | GEGLU |
| Normalization Layer | RMSNorm |
| Normalization Layer Position | PreNorm and PostNorm |
| Trained Lengths of Operands | 1–30 |
| Training Steps | 50,000 |
| Batch Size | 1000 |
| Optimizer | Adam |
| Learning Rate (LR) | 0.00003 |
| LR Warm-up | Linear (From 0 to LR), 1% of total steps |
| LR Cool-down | Cosine Decay (From LR to 0.1LR) |
| Maximum Position ID (`max_pos`) | 202 |
| Training Dataset Size | 1,000,000 |
| Evaluation Dataset Size | 100,000 |
| Device | NVIDIA RTX A6000 48GB |
| Training Time | $\leq 10$ hours |

Table 3: Hyperparameter summary for decimal integer addition task: generalization up to length 500 (Figure 12).

| Hyperparameter | Value |
|---|---|
| Architecture | Decoder-only Transformer |
| Number of Layers | 1 |
| Number of Attention Heads | 2 |
| Embedding Dimension | 512 |
| Dimension per Head | 256 |
| Hidden Width of Feed-forward Layer | 2048 |
| Activation Function of Feed-forward Layer | GEGLU |
| Normalization Layer | LayerNorm (Ba et al., 2016) |
| Normalization Layer Position | PostNorm |
| Training Steps | 1,000,000 |
| Batch Size | 128 |
| Optimizer | Adam |
| Learning Rate (LR) | 0.0001 |
| LR Warm-up | Linear (From 0 to LR), 500 steps |
| LR Cool-down | Cosine Decay (From LR to 0.0) |
| Maximum Position ID (`max_pos`) | 1003 |
| Training Dataset Size | 1,000,000 |
| Evaluation Dataset Size | 100,000 |
| Device | 64 TPU V4 Chips |
| Training Time | $\leq 4$ hours |

Table 4: Hyperparameter summary for decimal integer addition task: extracting attention patterns (Figure 7).

| Hyperparameter | Value |
|---|---|
| Architecture | Decoder-only Transformer |
| Number of Layers | 1 |
| Number of Attention Heads | 2 |
| Embedding Dimension | 512 |
| Dimension per Head | 256 |
| Hidden Width of Feed-forward Layer | 2048 |
| Activation Function of Feed-forward Layer | GEGLU |
| Normalization Layer | RMSNorm |
| Normalization Layer Position | PreNorm and PostNorm |
| Trained Lengths of Operands | 1–5 |
| Training Steps | 50,000 |
| Batch Size | 100 |
| Optimizer | Adam |
| Learning Rate (LR) | 0.00005 |
| LR Warm-up | Linear (From 0 to LR), 1% of total steps |
| LR Cool-down | Cosine Decay (From LR to 0.1LR) |
| Maximum Position ID (`max_pos`) | 17 |
| Training Dataset Size | 100,000 |
| Device | NVIDIA RTX A6000 48GB |
| Training Time | $\leq$ 6 hours |

Table 5: Hyperparameter summary for $N \times 2$ multiplication task (Figure 8).

| Hyperparameter | Value |
|---|---|
| Architecture | Decoder-only Transformer |
| Number of Layers | 1-4 (Ours), 3 (NoPE & Random-start APE) |
| Number of Attention Heads | 8 |
| Embedding Dimension | 512 |
| Dimension per Head | 64 |
| Hidden Width of Feed-forward Layer | 2048 |
| Activation Function of Feed-forward Layer | GEGLU |
| Normalization Layer | RMSNorm |
| Normalization Layer Position | PreNorm and PostNorm |
| Trained Lengths of Operands | 1–40 |
| Training Steps | 50,000 |
| Batch Size | 200 (Ours), 800 (NoPE & Random-start APE) |
| Optimizer | Adam |
| Learning Rate (LR) | 0.0001 |
| LR Warm-up | Linear (From 0 to LR), 1% of total steps |
| LR Cool-down | Cosine Decay (From LR to 0.1LR) |
| Maximum Position ID (`max_pos`) | 203 (Ours), 1023 (Random-start APE) |
| Training Dataset Size | 50,000 (Ours), 500,000 (Others) |
| Evaluation Dataset Size | 100,000 |
| Device | NVIDIA RTX A6000 48GB |
| Training Time | $\leq$ 8 hours |

Table 6: Hyperparameter summary for addition task with three summands (Figure 18).

| Hyperparameter | Value |
|---|---|
| Architecture | Decoder-only Transformer |
| Number of Layers | 1-3 (Ours), 6 (NoPE & Random-start APE) |
| Number of Attention Heads | 4 |
| Embedding Dimension | 512 |
| Dimension per Head | 128 |
| Hidden Width of Feed-forward Layer | 2048 |
| Activation Function of Feed-forward Layer | GEGLU |
| Normalization Layer | RMSNorm |
| Normalization Layer Position | PreNorm and PostNorm |
| Trained Lengths of Operands | 1–40 |
| Training Steps | 50,000 |
| Batch Size | 1000 (Ours), 800 (Others) |
| Optimizer | Adam |
| Learning Rate (LR) | 0.0001 |
| LR Warm-up | Linear (From 0 to LR), 1% of total steps |
| LR Cool-down | Cosine Decay (From LR to 0.1LR) |
| Maximum Position ID (`max_pos`) | 102 (Ours), 1023 (Random-start APE) |
| Training Dataset Size | 1,000,000 |
| Evaluation Dataset Size | 100,000 |
| Device | NVIDIA RTX A6000 48GB |
| Training Time | $\leq$ 12 hours |

Table 7: Hyperparameter summary for copy/reverse task (Figure 20).

| Hyperparameter | Value |
|---|---|
| Architecture | Decoder-only Transformer |
| Number of Layers | 1 (Ours), 6 (NoPE & Random-start APE) |
| Number of Attention Heads | 4 (Ours), 8 (NoPE & Random-start APE) |
| Embedding Dimension | 512 |
| Dimension per Head | 128 |
| Hidden Width of Feed-forward Layer | 2048 |
| Activation Function of Feed-forward Layer | GEGLU |
| Normalization Layer | RMSNorm |
| Normalization Layer Position | PreNorm and PostNorm |
| Trained Lengths of Query | 1–40 |
| Training Steps | 50,000 |
| Batch Size | 1000 (Ours), 500 (Others) |
| Optimizer | Adam |
| Learning Rate (LR) | 0.0001 |
| LR Warm-up | Linear (From 0 to LR), 1% of total steps |
| LR Cool-down | Cosine Decay (From LR to 0.1LR) |
| Maximum Position ID (`max_pos`) | 301 (Ours), 601 (Random-start APE) |
| Training Dataset Size | 1,000,000 |
| Evaluation Dataset Size | 100,000 |
| Device | NVIDIA RTX A6000 48GB |
| Training Time | $\leq$ 8 hours |

Table 8: Hyperparameter summary for minesweeper generator task (Figures 10 and 21).

| Hyperparameter | Value |
| --- | --- |
| Architecture | Decoder-only Transformer |
| Number of Layers | 4 (Ours), 6 (NoPE) |
| Number of Attention Heads | 8 |
| Embedding Dimension | 512 |
| Dimension per Head | 64 |
| Hidden Width of Feed-forward Layer | 2048 |
| Activation Function of Feed-forward Layer | GEGLU |
| Normalization Layer | RMSNorm |
| Normalization Layer Position | PreNorm and PostNorm |
| Trained Lengths of Query | $(5\text{–}9) \times (5\text{–}9)$ |
| Training Steps | 100,000 |
| Batch Size | 200 |
| Optimizer | Adam |
| Learning Rate (LR) | 0.0001 (Ours), 0.0002 (NoPE) |
| LR Warm-up | Linear (From 0 to LR), 1% of total steps |
| LR Cool-down | Cosine Decay (From LR to 0.1LR) |
| Maximum Position ID (`max_pos`) | 15 |
| Training Dataset Size | 100,000 |
| Evaluation Dataset Size | 100,000 |
| Device | NVIDIA RTX A6000 48GB |
| Training Time | $\leq 30$ hours |

# D  Decoder-only Transformer Architecture

Here we detail the architecture of a depth-$L$, $H$-head decoder-only Transformer (Vaswani et al., 2017). For a simple presentation, we ignore the normalization layers, as in Yun et al. (2020a).

Let $\mathcal{V}$ be the (ordered) vocabulary, a set of all tokens. Given an input sequence $\mathcal{I} \in \mathcal{V}^N$ and its length $N$, the *encoding function* $\texttt{Enc} : \mathcal{V}^N \to \mathbb{R}^{d \times N}$ maps it to

$$\boldsymbol{X}^{(0)} := \texttt{Enc}(\mathcal{I}). \tag{1}$$

It is a sum of the token embedding and the position embedding.

Next, there are $L$ Transformer blocks that sequentially transform this input. We denote by $\texttt{Tf}_l : \mathbb{R}^{d \times N} \to \mathbb{R}^{d \times N}$ the operation of the $l$-th block ($l \in [L]$), so that

$$\boldsymbol{X}^{(l)} := \texttt{Tf}_l \left( \boldsymbol{X}^{(l-1)} \right). \tag{2}$$

The block $\texttt{Tf}_l$ consists of a (causal) attention layer $\texttt{Att}_l : \mathbb{R}^{d \times N} \to \mathbb{R}^{d \times N}$ and a (token-wise) feed-forward layer $\texttt{FF}_l : \mathbb{R}^{d \times N} \to \mathbb{R}^{d \times N}$, each of which contains a residual connection:

$$\texttt{Tf}_l := (\texttt{id} + \texttt{FF}_l) \circ (\texttt{id} + \texttt{Att}_l), \tag{3}$$

where we denote by $\texttt{id} : \mathbb{R}^{d \times N} \to \mathbb{R}^{d \times N}$ an identity map.

Each attention layer $\texttt{Att}_l$ consists of $H$ attention heads. Its $h$-th head ($h \in [H]$) has matrices $\boldsymbol{Q}_h^{(l)}, \boldsymbol{K}_h^{(l)} \in \mathbb{R}^{d_{QK,h}^{(l)} \times d}$, $\boldsymbol{V}_h^{(l)} \in \mathbb{R}^{d_{V,h}^{(l)} \times d}$ and $\boldsymbol{U}_h^{(l)} \in \mathbb{R}^{d \times d_{V,h}^{(l)}}$ as its parameters.[6] With these matrices, borrowing the notation from Yun et al. (2020a), the attention layer with an input $\boldsymbol{X} \in \mathbb{R}^{d \times N}$ can be written as

$$\texttt{Att}_l(\boldsymbol{X}) := \sum_{h=1}^{H} \boldsymbol{U}_h^{(l)} \boldsymbol{V}_h^{(l)} \boldsymbol{X} \cdot \texttt{softmax}\left( (\boldsymbol{K}_h^{(l)} \boldsymbol{X})^\top \boldsymbol{Q}_h^{(l)} \boldsymbol{X} \right). \tag{4}$$

Here the $\texttt{softmax}$ operator takes a square matrix $\boldsymbol{M} \in \mathbb{R}^{N \times N}$ and outputs an $N \times N$ upper-triangular column-stochastic[7] matrix

$$[\texttt{softmax}(\boldsymbol{M})]_{ij} = \frac{e^{\boldsymbol{M}_{ij}}}{\sum_{1 \leq i' \leq j} e^{\boldsymbol{M}_{i'j}}} \mathbb{1}_{\{i \leq j\}}, \tag{5}$$

where $\mathbb{1}_{\{\mathcal{E}\}}$ is an indicator function for a predicate $\mathcal{E}$: it equals 1 if $\mathcal{E}$ is true and 0 otherwise. Note that the upper triangularity captures the auto-regressive behavior of the causal attention. For the sake of convenience, we denote by $\boldsymbol{Y}^{(l)} := \boldsymbol{X}^{(l-1)} + \texttt{Att}_l(\boldsymbol{X}^{(l-1)}) \in \mathbb{R}^{d \times N}$ which is a consequence of residual connection right after the attention layer.

Each feed-forward layer $\texttt{FF}_l$ is a two-layer perceptron having $\boldsymbol{W}_1^{(l)} \in \mathbb{R}^{d_F \times d}$, $\boldsymbol{b}_1^{(l)} \in \mathbb{R}^{d_F}$, $\boldsymbol{W}_2^{(l)} \in \mathbb{R}^{d \times d_F}$, $\boldsymbol{b}_2^{(l)} \in \mathbb{R}^d$ as its parameters. It applies the following map to each column $\boldsymbol{y}$ of an input $\boldsymbol{Y}$:

$$\boldsymbol{y} \mapsto \boldsymbol{W}_2^{(l)} \phi(\boldsymbol{W}_1^{(l)} \boldsymbol{y} + \boldsymbol{b}_1^{(l)}) + \boldsymbol{b}_2^{(l)}, \tag{6}$$

where $\phi$ is a component-wise activation function. That is, the feed-forward layer is defined as

$$\texttt{FF}_l(\boldsymbol{Y}) := \boldsymbol{W}_2^{(l)} \phi(\boldsymbol{W}_1^{(l)} \boldsymbol{Y} + \boldsymbol{b}_1^{(l)} \boldsymbol{1}_{d_F}^\top) + \boldsymbol{b}_2^{(l)} \boldsymbol{1}_d^\top, \tag{7}$$

where $\boldsymbol{1}_d$ is the $d$-dimensional vectors filled with 1's. Here we mainly use the ReLU operation $\phi(\cdot) = \max\{\cdot, 0\}$ (Jarrett et al., 2009; Nair and Hinton, 2010), but there are many other popular choices such as GeLU (Hendrycks and Gimpel, 2016), GLU (Dauphin et al., 2017), ReGLU, and GEGLU (Shazeer, 2020).

The final component of the Transformer model is the decoding function $\texttt{Dec} : \mathbb{R}^{d \times N} \to \mathcal{V}^N$, which is composed of a linear readout and a (token-wise) arg-max operation. Here, the linear readout is simply a linear layer having $\boldsymbol{W}_{\text{out}} \in \mathbb{R}^{|\mathcal{V}| \times d}$ as its parameter. The decoding function produces the output sequence

$$\mathcal{O} := \texttt{Dec}(\boldsymbol{X}^{(L)}) \in \mathcal{V}^N. \tag{8}$$

---

[6] One can let $d_H = \max_{l,h} \max\{d_{QK,h}^{(l)}, d_{V,h}^{(l)}\}$ as an inner dimension of each head. This makes our formal constructions a bit messier with redundant entries 0.

[7] Every entry is non-negative and the sum of entries in each column is 1.

# E Formal Construction of Addition Transformer with Position Coupling

Here we show how to implement the addition by employing a single-layer two-head decoder-only Transformer equipped with position coupling. We restate the theorem for the sake of readability.

**Theorem 5.1.** *With the input format described in Section 3.1, there exists a depth-1 two-head decoder-only Transformer with coupled positions that solves the addition task with next-token prediction. Here, the operand length is at most $2^{\lfloor (d-17)/2 \rfloor} - 2$, where the embedding dimension is $d \geq 21$.*

**Organization of the Proof.** A whole section is dedicated to prove Theorem 5.1.

- We start with the notation (Appendix E.1).

- We review and formalize the format of the input sequence (zero-padding, reversed format, and wrapping with BOS/EOS) (Appendix E.2).

- We define the encoding function `Enc` with a table of a concrete example (Appendix E.3), where `Enc` maps an input sequence of length $N$ to a $d \times N$ encoding matrix $\boldsymbol{X}^{(0)}$.

- We devote a lot of pages to the detailed construction of the parameters of a causal attention layer $\mathtt{Att}_1$ to generate desired attention patterns (Appendix E.4). The attention layer has two attention heads playing distinct roles: (1) preparing for a sum without considering carries; and (2) preparing for the carry prediction & EOS detection.

- We provide a construction of a token-wise feed-forward neural network $\mathtt{FF}_1$ which is a two-layer ReLU network (Appendix E.5). It consists of two subnetworks playing different roles: (1) producing one-hot vectors, each of which indicates a digit of the sum (response); and (2) binary values indicating whether the position is the end of the sequence.

- We conclude the proof by defining the decoding function `Dec` which performs the linear readout and the arg-max operation to generate the output sequence (Appendix E.6).

We illustrate the roadmap of the proof in Figure 22.

## E.1 Notation

For the architecture of the decoder-only Transformer, we follow the notation introduced in Appendix D.

Let $\boldsymbol{e}_i^d$ denote the $i$-th standard basis vector of $\mathbb{R}^d$. For example, $\boldsymbol{e}_1^3 = \begin{bmatrix} 1 & 0 & 0 \end{bmatrix}^\top$. Let $\boldsymbol{I}_m$ be the $m \times m$ identity matrix. Let $\boldsymbol{0}_p$ and $\boldsymbol{1}_p$ denote the $p$-dimensional vectors filled with 0's and 1's, respectively. Similarly, let $\boldsymbol{0}_{m \times n}$ denote the $m \times n$ zero matrix. For a positive integer $n$, we frequently use the set $[n] := \{1, ..., n\}$. For any matrix $\boldsymbol{A}$, denote the $i$-th row and $j$-th column of $\boldsymbol{A}$ by $\boldsymbol{A}_{i\bullet}$ and $\boldsymbol{A}_{\bullet j}$, respectively. Given two non-negative integers $a$ and $b$, let $\ell(a,b)$ be the length of a longer one between $a$ and $b$. For example, $\ell(12, 3456) = 4$.

Consider an ordered vocabulary $\mathcal{V} = (0, 1, 2, 3, 4, 5, 6, 7, 8, 9, +, =, \$)$. We include a special token '\$' that plays the role of both the beginning-of-sequence (BOS) token and the end-of-sequence (EOS) token.[8] We denote $\mathcal{V}_k$ as $k$-th element of $\mathcal{V}$. For instance, $\mathcal{V}_4 = 3$ and $\mathcal{V}_{13} = \$$. Lastly, since we employ only one Transformer block, we omit the superscripts $(l)$ in the parameter matrices/vectors and the size of dimensions $d_{QK,h}^{(l)}$ and $d_{V,h}^{(l)}$.

## E.2 Input Sequence

We seek to perform an addition $a + b = c$ using next-token prediction. To this end, we want to transform it into an input sequence $\mathcal{I} = \overline{\$A + B = C}$ of an appropriate format. Note that the EOS token is the last token that needs to be predicted, so we exclude EOS in the input sequence. Let $\ell := \ell(a,b)$.

We first zero-pad the shorter one between $a$ and $b$ to match the length of the part $A$ and part $B$ as $\ell$. Sometimes, the sum $c$ might be longer than $a$ or $b$ due to a carry. To make the length of the part $C$

---

[8]BOS and EOS tokens do not need to be identical. We regard them as the same token just for the simplicity of the presentation.

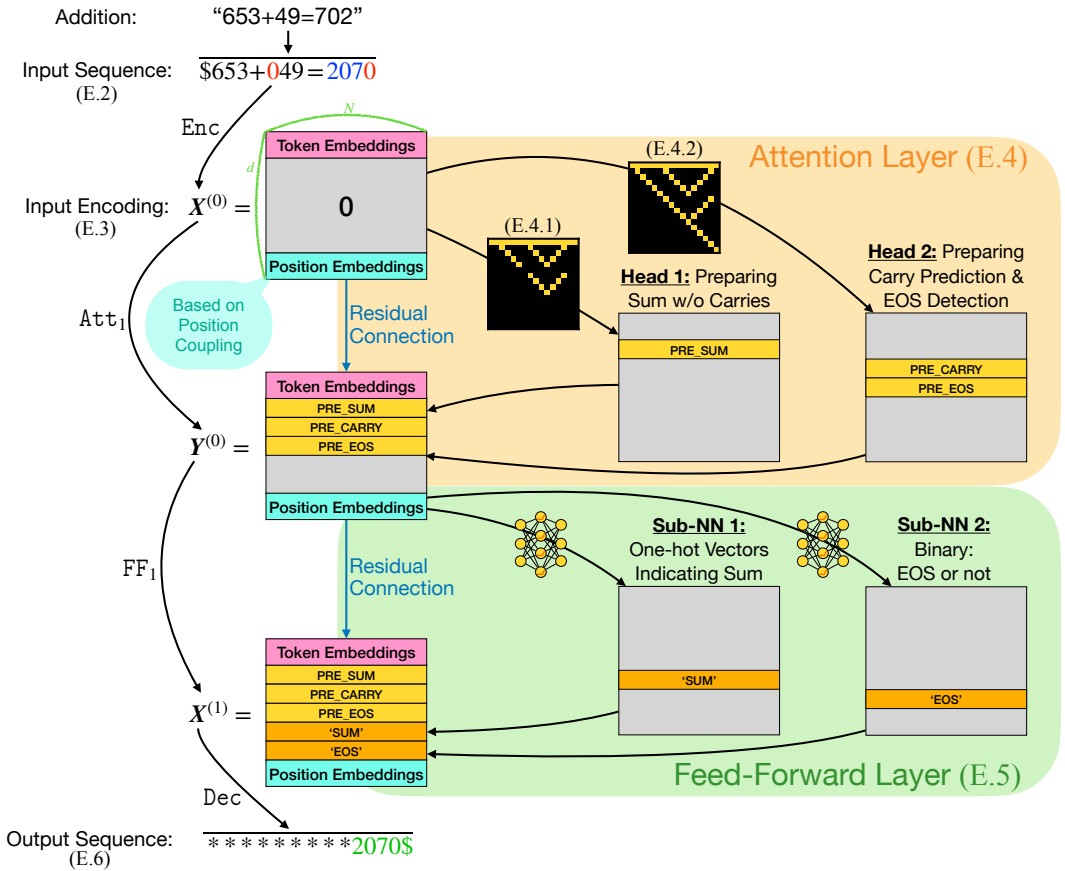

Figure 22: Roadmap to the formal construction of addition Transformer with position coupling.

consistent, we also put a zero-pad in front of $c$ to set its length as $\ell + 1$. Also, to ease calculating the addition with next-token prediction, we reverse the sum $c$ to make the part $C$. For example, if we have a sum $3812 + 98 = 3910$, we use $\overline{\$3812 + 0098 = 01930}$ as an input sequence; if a sum $98 + 9907 = 10005$ is given, we use $\overline{\$0098 + 9907 = 50001}$ as an input sequence. The red digits are zero-paddings, and the blue digits are the reversed sum.

To recap, the input sequence $\mathcal{I} = \overline{\sigma_1 \sigma_2 \ldots \sigma_N} \in \mathcal{V}^N$ of length $N = 3\ell + 4$ consists of six parts:

1. the BOS token $\sigma_1 = $ '$\$$'

2. the first operand $A = \overline{\sigma_2 \ldots \sigma_{\ell+1}}$ where $\sigma_i \in \{0, \ldots, 9\}$;

3. the addition symbol $\sigma_{\ell+2} = $ '+';

4. the second operand $B = \overline{\sigma_{\ell+3} \ldots \sigma_{2\ell+2}}$ where $\sigma_i \in \{0, \ldots, 9\}$;

5. the equality symbol $\sigma_{2\ell+3} = $ '=';

6. the (reversed) sum $C = \overline{\sigma_{2\ell+4} \ldots \sigma_{3\ell+4}}$ where $\sigma_i \in \{0, \ldots, 9\}$.

Note that the part $C$ might be incomplete (i.e., $N < 3\ell + 4$) at the inference time; we infer the digits of the part $C$ one by one using next-token prediction. Throughout this section on a formal construction, however, we only consider the train time setup in which we infer all the digits of the part $C$ at once using *simultaneous* next-token prediction in a single forward pass. Precisely, we want to use an input sequence $\mathcal{I} = \overline{\sigma_1 \ldots \sigma_N}$ to produce an output sequence $\mathcal{O} = \overline{\sigma'_1 \ldots \sigma'_N}$ where $\overline{\sigma'_{2\ell+3} \ldots \sigma'_{N-1}} = C = \overline{\sigma_{2\ell+4} \ldots \sigma_N}$ and $\sigma'_N = $ '$\$$' (EOS).

## E.3 Encoding Function

We plan to produce an input encoding, given an input sequence $\mathcal{I}$ designed as above. The encoding matrix $\boldsymbol{X}^{(0)}$ is of size $d \times N$: each column represents an embedding vector for a token, while each row represents a particular *named* dimension. What we mean by *named* dimension is that we give a name to each dimension for a clear description of our formal construction.

We construct an input encoding by *concatenating* the token embedding and the position embedding, which can be viewed as a *sum* of two different embedding matrices of the same size.

Table 9: Example initial encoding. Here we consider the input sequence $\overline{\$653 + 049 = 2070}$ and the starting position ID is chosen as $s = 2$. The vectors $\boldsymbol{v}_\square^P$ are defined in Equation (11). The gray rows will be filled in later.

| $\mathcal{I}$ | | $ | 6 | 5 | 3 | + | 0 | 4 | 9 | = | 2 | 0 | 7 | 0 |
|---|---|---|---|---|---|---|---|---|---|---|---|---|---|---|
| 1: NUM | | 0 | 6 | 5 | 3 | 0 | 0 | 4 | 9 | 0 | 2 | 0 | 7 | 0 |
| 2: IS_BOS | | 1 | 0 | 0 | 0 | 0 | 0 | 0 | 0 | 0 | 0 | 0 | 0 | 0 |
| 3: FULL_ONES | | 1 | 1 | 1 | 1 | 1 | 1 | 1 | 1 | 1 | 1 | 1 | 1 | 1 |
| 4: PRE_SUM | | 0 | 0 | 0 | 0 | 0 | 0 | 0 | 0 | 0 | 0 | 0 | 0 | 0 |
| 5: PRE_CARRY | | 0 | 0 | 0 | 0 | 0 | 0 | 0 | 0 | 0 | 0 | 0 | 0 | 0 |
| 6: PRE_EOS | | 0 | 0 | 0 | 0 | 0 | 0 | 0 | 0 | 0 | 0 | 0 | 0 | 0 |
| 7–16: SUM | | $\boldsymbol{0}_{10}$ | $\boldsymbol{0}_{10}$ | $\boldsymbol{0}_{10}$ | $\boldsymbol{0}_{10}$ | $\boldsymbol{0}_{10}$ | $\boldsymbol{0}_{10}$ | $\boldsymbol{0}_{10}$ | $\boldsymbol{0}_{10}$ | $\boldsymbol{0}_{10}$ | $\boldsymbol{0}_{10}$ | $\boldsymbol{0}_{10}$ | $\boldsymbol{0}_{10}$ | $\boldsymbol{0}_{10}$ |
| 17: IS_EOS | | 0 | 0 | 0 | 0 | 0 | 0 | 0 | 0 | 0 | 0 | 0 | 0 | 0 |
| 18–$(P+17)$: POS_1 | | $\boldsymbol{0}_P$ | $\boldsymbol{v}_3^P$ | $\boldsymbol{v}_4^P$ | $\boldsymbol{v}_5^P$ | $\boldsymbol{v}_6^P$ | $\boldsymbol{v}_3^P$ | $\boldsymbol{v}_4^P$ | $\boldsymbol{v}_5^P$ | $\boldsymbol{v}_6^P$ | $\boldsymbol{v}_5^P$ | $\boldsymbol{v}_4^P$ | $\boldsymbol{v}_3^P$ | $\boldsymbol{v}_2^P$ |
| $(P+18)$-$(2P+17)$: POS_2 | | $\boldsymbol{0}_P$ | $\boldsymbol{v}_4^P$ | $\boldsymbol{v}_5^P$ | $\boldsymbol{v}_6^P$ | $\boldsymbol{v}_7^P$ | $\boldsymbol{v}_4^P$ | $\boldsymbol{v}_5^P$ | $\boldsymbol{v}_6^P$ | $\boldsymbol{v}_7^P$ | $\boldsymbol{v}_6^P$ | $\boldsymbol{v}_5^P$ | $\boldsymbol{v}_4^P$ | $\boldsymbol{v}_3^P$ |

### E.3.1 Token Embedding

The token embedding consists of 17 dimensions: we call them

$$1=\text{NUM}, 2=\text{IS\_BOS}, 3=\text{FULL\_ONES},$$

$$4=\text{PRE\_SUM}, 5=\text{PRE\_CARRY}, 6=\text{PRE\_EOS},$$

$$\{7,...,16\}=\text{SUM}, \text{ and } 17=\text{IS\_EOS}.$$

Initially, we let the last 14 dimensions be empty (i.e., all zeros). Thus, we explain the first three dimensions, NUM, IS_BOS, and FULL_ONES.

**Dimension 1 (NUM).** For a number token $(0, \ldots, 9)$, we put itself in the dimension NUM. For the other tokens $(+, =, \$)$, we put $0$.

**Dimension 2 (IS_BOS).** For a special token '$\$$', we put $1$ in the dimension IS_BOS. Otherwise, we put $0$.

**Dimension 3 (FULL_ONES).** We put $1$ everywhere in this dimension.

### E.3.2 Coupled Position IDs and Position Embedding

Before constructing a position embedding, we specify the coupled position IDs for the addition task. Let max_pos be a hyperparameter of the maximum position IDs, where position IDs are non-negative integers. Basically, we match the significance of the digits: e.g., a least significant digit is always coupled to the other least significant digits. To this end, we first randomly choose a *starting position ID* $s \in [\text{max\_pos} - \ell - 1]$. (For that, $\text{max\_pos} \geq \ell + 2$ must hold.) Then we allocate the position IDs of token $\sigma_i$ in the input sequence $\mathcal{I} = \overline{\sigma_1 \ldots \sigma_N}$ as

$$p(i) = \begin{cases} 0, & i = 1, \\ s + i - 1, & i = 2, \ldots, \ell + 2, \\ s + i - (\ell + 2), & i = \ell + 3, \ldots, 2\ell + 3, \\ s + (3\ell + 4) - i & i = 2\ell + 4, \ldots, 3\ell + 4. \end{cases} \tag{9}$$

Recall that $N = 3\ell + 4$. Also, observe that for $i \in \{2, \ldots, \ell + 1\}$,
$$p(i) = p(i + \ell + 1) = p(3\ell + 5 - i) = s + i, \tag{10}$$
which couples the position of $(\ell - i + 2)$-th significant digit in the first operand $(A)$, the second operand $(B)$, and the sum $(C)$. Also, the position of tokens '+' and '=' are coupled. Lastly, the only token that has the position ID 0 is the special token '\$'.

Before moving on to the positional embedding, we define $\boldsymbol{v}_k^D$ $(k \in [2^D])$ as
$$\boldsymbol{v}_k^D = \left[ (-1)^{b_i^{(D,k)}} \right]_{i=1}^D \in \mathbb{R}^D \tag{11}$$
where $b_i^{(D,k)}$ is defined as the $i$-th (from left) digit of $D$-digit binary representation of $k - 1$. For example, if $D = 2$,
$$\boldsymbol{v}_1^2 = [1 \quad 1]^\top, \ \boldsymbol{v}_2^2 = [-1 \quad 1]^\top, \ \boldsymbol{v}_3^2 = [1 \quad -1]^\top, \ \boldsymbol{v}_4^2 = [-1 \quad -1]^\top. \tag{12}$$
We remark that the points $\boldsymbol{v}_k^D$ are the vertices of $D$-dimensional hypercube with side length 2, centered at the origin.[9] Note that for $k \neq l$,
$$\left\| \boldsymbol{v}_k^D \right\|^2 = D, \quad \left\langle \boldsymbol{v}_k^D, \boldsymbol{v}_l^D \right\rangle \leq D - 2. \tag{13}$$

Now we explain the position embedding. It consists of $2P$ dimensions, which eventually become from 18-th to $(2P + 17)$-th dimension after concatenation. If $p(i) = 0$, we let $\boldsymbol{0}_{2P}$ as a position embedding vector. For the positive position IDs $p(i) \geq 1$, we let a concatenation
$$\begin{bmatrix} \boldsymbol{v}_{p(i)}^P \\ \boldsymbol{v}_{p(i)+1}^P \end{bmatrix} \tag{14}$$
as a position embedding vector of a token $\sigma_i$. (In case of $p(i) = 2^P$, we use $\boldsymbol{v}_1^P$ instead of $\boldsymbol{v}_{p(i)+1}^P$.) We call the former $P$ dimensions for the position embedding as POS_1 and the latter $P$ dimensions as POS_2.

Concatenating the token embedding and the position embedding, we get the input embedding $\boldsymbol{X}^{(0)}$. See Table 9 for an example. As a result, the total embedding dimension is $d = 2P + 17$. Note the maximum possible position ID that can be represented with $\boldsymbol{v}_k^P$'s is $\texttt{max\_pos} = 2^P = 2^{\lfloor (d-17)/2 \rfloor}$. Therefore, the length of an operand must be $\ell \leq \texttt{max\_pos} - 2 = 2^{\lfloor (d-17)/2 \rfloor} - 2$.

### E.4 Transformer Block — Causal Attention Layer

The goal of the causal attention layer is to fill in the zero-blanks[10] of the encoding matrix at dimensions PRE_SUM, PRE_CARRY, and PRE_EOS. We divide the roles into two different heads.

#### E.4.1 Attention Head 1: Digit-wise Addition without Carries

The goal of the first head is to perform a *digit-wise addition* and to fill in the blanks of the encoding matrix at dimension PRE_SUM. Later, using this dimension, combined with the dimension PRE_CARRY, we will be able to perform the next-token prediction for addition. For now, we do not care about the carries, which will be dealt with in a later section. Formally, we aim to perform $\sigma_i + \sigma_{i+\ell+1}$ for each $i \in \{2, \cdots, \ell + 1\}$ and put its result at the $(3\ell + 4 - i)$-th position (column) of the dimension PRE_SUM (row). To this end, we utilize our position embedding.

Recall that $d = 2P + 17$ and let $d_{QK,1} = P + 1$. Let $M > 0$ be a number determined later. Let
$$\boldsymbol{Q}_1 = \begin{pmatrix} \boldsymbol{0}_{P \times 17} & \sqrt{M} \boldsymbol{I}_P & \boldsymbol{0}_{P \times P} \\ \sqrt{MP}(\boldsymbol{e}_{\texttt{FULL\_ONES}}^{17})^\top & \boldsymbol{0}_{1 \times P} & \boldsymbol{0}_{1 \times P} \end{pmatrix} \in \mathbb{R}^{d_{QK,1} \times d}, \tag{15}$$
$$\boldsymbol{K}_1 = \begin{pmatrix} \boldsymbol{0}_{P \times 17} & \boldsymbol{0}_{P \times P} & \sqrt{M} \boldsymbol{I}_P \\ \sqrt{MP}(\boldsymbol{e}_{\texttt{IS\_BOS}}^{17})^\top & \boldsymbol{0}_{1 \times P} & \boldsymbol{0}_{1 \times P} \end{pmatrix} \in \mathbb{R}^{d_{QK,1} \times d}. \tag{16}$$

---

[9]The choice of the vectors $\boldsymbol{v}_k^D$ is not strict. They only need to have the same length and be distinguishable (for at least a constant order) in terms of inner products. That is, there should be a noticeable difference between $\left\| \boldsymbol{v}_k^D \right\|^2$ and $\left\langle \boldsymbol{v}_k^D, \boldsymbol{v}_l^D \right\rangle$ for $k \neq l$.

[10]Such an idea of filling in the blacks of the encoding matrix is borrowed from the literature of RASP language(s) (Friedman et al., 2023; Lindner et al., 2023; Weiss et al., 2021; Zhou et al., 2024a). This can be done with the help of residual connections.

The linear transformations with matrices $\boldsymbol{Q}_1$ and $\boldsymbol{K}_1$ do two different jobs at once. (1) $\boldsymbol{Q}_1$ ($\boldsymbol{K}_1$, resp.) takes the dimensions POS_1 (POS_2, resp.) from the input encoding matrix and scale them up by $\sqrt{M}$; (2) $\boldsymbol{Q}_1$ ($\boldsymbol{K}_1$, resp.) takes the dimension FULL_ONES (IS_BOS, resp.) and scale it up by $\sqrt{MP}$. For concrete examples, please refer to Tables 12 and 13. By these, the attention *score* matrix $\boldsymbol{C}_1 := (\boldsymbol{K}_1 \boldsymbol{X}^{(0)})^\top \boldsymbol{Q}_1 \boldsymbol{X}^{(0)}$ becomes as in Table 10. The blanks in Table 10 are the numbers smaller than $M(P-2)$; the asterisks ('*') are the entries (or lower triangular submatrices) ignored by the causal softmax operator; the dots represents the hidden $MP$'s.

Table 10: Exact attention score matrix $\boldsymbol{C}_1$ (with explicit row/column indices) of Head 1.

| row \ col | $j=1$ | 2 | 3 | $\cdots$ | $\ell+2$ | $\ell+3$ | $\ell+4$ | $\cdots$ | $2\ell+3$ | $\cdots$ | $3\ell+2$ | $3\ell+3$ | $3\ell+4$ |
|---|---|---|---|---|---|---|---|---|---|---|---|---|---|
| $i=1$ | $MP$ | $MP$ | $MP$ | $\cdots$ | $MP$ | $MP$ | $MP$ | $\cdots$ | $MP$ | $\cdots$ | $MP$ | $MP$ | $MP$ |
| 2 | * | | $MP$ | | | $MP$ | | | | | $MP$ | | |
| $\vdots$ | * | * | | $\ddots$ | | | | $\ddots$ | | $\cdot^{\cdot^{\cdot}}$ | | | |
| $\ell+1$ | * | * | * | | $MP$ | | | | $MP$ | | | | |
| $\ell+2$ | * | * | * | * | | | | | | | | | |
| $\ell+3$ | * | * | * | * | * | | $MP$ | | | | $MP$ | | |
| $\vdots$ | * | * | * | * | * | * | | $\ddots$ | | $\cdot^{\cdot^{\cdot}}$ | | | |
| $2\ell+2$ | * | * | * | * | * | * | * | | $MP$ | | | | |
| $2\ell+3$ | * | * | * | * | * | * | * | * | | | | | |
| $\vdots$ | * | * | * | * | * | * | * | * | * | | | | |
| $3\ell+2$ | * | * | * | * | * | * | * | * | * | * | | | |
| $3\ell+3$ | * | * | * | * | * | * | * | * | * | * | * | | |
| $3\ell+4$ | * | * | * | * | * | * | * | * | * | * | * | * | |

Now consider the *attention matrix* $\boldsymbol{A}_1 := \texttt{softmax}(\boldsymbol{C}_1) \in \mathbb{R}^{N \times N}$. Its exact form is a bit messy due to the softmax operation of finite numbers. However, one can observe that, if the number $M$ is large enough, it gets close to the column-stochastic matrix $\boldsymbol{T}_1 \in \mathbb{R}^{N \times N}$ described in Table 11. The blanks in Table 11 are zeros; the dots represent the omitted nonzero entries.

Table 11: Limiting attention matrix $\boldsymbol{T}_1$ (with explicit row/column indices) of Head 1, as $M$ gets large.

| row \ col | $j=1$ | 2 | 3 | $\cdots$ | $\ell+2$ | $\ell+3$ | $\ell+4$ | $\cdots$ | $2\ell+3$ | $\cdots$ | $3\ell+2$ | $3\ell+3$ | $3\ell+4$ |
|---|---|---|---|---|---|---|---|---|---|---|---|---|---|
| $i=1$ | 1 | 1 | 1/2 | $\cdots$ | 1/2 | 1 | 1/3 | $\cdots$ | 1/3 | $\cdots$ | 1/3 | 1 | 1 |
| 2 | 0 | 0 | 1/2 | | 0 | 0 | 1/3 | | 0 | | 1/3 | 0 | 0 |
| $\vdots$ | | | | $\ddots$ | | | | $\ddots$ | | $\cdot^{\cdot^{\cdot}}$ | | | |
| $\ell+1$ | 0 | 0 | 0 | | 1/2 | 0 | 0 | | 1/3 | | 0 | 0 | 0 |
| $\ell+2$ | 0 | 0 | 0 | | 0 | 0 | 0 | | 0 | | 0 | 0 | 0 |
| $\ell+3$ | 0 | 0 | 0 | | 0 | 0 | 1/3 | | 0 | | 1/3 | 0 | 0 |
| $\vdots$ | | | | | | | | $\ddots$ | | $\cdot^{\cdot^{\cdot}}$ | | | |
| $2\ell+2$ | 0 | 0 | 0 | | 0 | 0 | 0 | | 1/3 | | 0 | 0 | 0 |
| $2\ell+3$ | 0 | 0 | 0 | | 0 | 0 | 0 | | 0 | | 0 | 0 | 0 |
| $\vdots$ | | | | | | | | | | | | | |
| $3\ell+4$ | 0 | 0 | 0 | | 0 | 0 | 0 | | 0 | | 0 | 0 | 0 |

Let $\boldsymbol{R}_1 = \boldsymbol{A}_1 - \boldsymbol{T}_1 \in \mathbb{R}^{N \times N}$ be the error matrix, which is upper triangular. Its exact form is messy as well, but we can obtain the bounds of their entries. Consider a pair of indices $(i,j) \in [N]^2$ such that $i \le j$. Let $x_j = 1/[\boldsymbol{T}_1]_{1j} \in \{1,2,3\}$. If $[\boldsymbol{T}_1]_{ij} = \frac{1}{x_j}$, $[\boldsymbol{R}_1]_{ij} < 0$ and

$$-[\boldsymbol{R}_1]_{ij} \le \frac{1}{x_j} - \frac{e^{MP}}{x_j e^{MP} + (j - x_j)e^{M(P-2)}} = \frac{j - x_j}{x_j(x_j e^{2M} + (j - x_j))}. \tag{17}$$

On the other hand, if $[\boldsymbol{T}_1]_{ij} = 0$, $[\boldsymbol{R}_1]_{ij} > 0$ and

$$[\boldsymbol{R}_1]_{ij} \le \frac{e^{M(P-2)}}{x_j e^{MP} + (j - x_j)e^{M(P-2)}} = \frac{1}{x_j e^{2M} + (j - x_j)}. \tag{18}$$

Now let $d_{V,1} = 1$ and

$$\boldsymbol{V}_1 = 3(\boldsymbol{e}_{\text{NUM}}^d)^\top \in \mathbb{R}^{d_{V,1} \times d}, \tag{19}$$

$$\boldsymbol{U}_1 = \boldsymbol{e}_{\text{PRE\_SUM}}^d \in \mathbb{R}^{d \times d_{V,1}}. \tag{20}$$

The linear transformation with matrix $\boldsymbol{U}_1\boldsymbol{V}_1$ takes the dimension NUM from the input encoding matrix, scales it up by 3, and puts it to the dimension PRE_SUM. A concrete example is provided in Table 14.

Obtaining $\boldsymbol{U}_1\boldsymbol{V}_1\boldsymbol{X}^{(0)}\boldsymbol{A}_1$, its every entry is zero except at the dimension PRE_SUM. Observe that $[\boldsymbol{U}_1\boldsymbol{V}_1\boldsymbol{X}^{(0)}]_{(\text{PRE\_SUM})1} = 0$, because in the input encoding matrix, the dimension NUM starts with 0. Also, note that it is enough to focus on the columns $j \in \{2\ell+3,\dots,3\ell+4\}$ since we only care about the next-token prediction of the tokens after $\sigma_{2\ell+3} =$ '='. Specifying the dimension (i.e., the particular row) for these columns, we have

$$[\boldsymbol{U}_1\boldsymbol{V}_1\boldsymbol{X}^{(0)}\boldsymbol{T}_1]_{(\text{PRE\_SUM})j} = \begin{cases} \boldsymbol{X}^{(0)}_{(\text{NUM})(3\ell+4-j)} + \boldsymbol{X}^{(0)}_{(\text{NUM})(4\ell+5-j)} & \text{if } j \in \{2\ell+3,\dots,3\ell+2\}, \\ 0 & \text{if } j \in \{3\ell+3, 3\ell+4\}, \end{cases} \tag{21}$$

$$= \begin{cases} \sigma_{(3\ell+4)-j} + \sigma_{(4\ell+5)-j} & \text{if } j \in \{2\ell+3,\dots,3\ell+2\}, \\ 0 & \text{if } j \in \{3\ell+3, 3\ell+4\}. \end{cases} \tag{22}$$

Refer to Table 15 for a concrete example of computing $\boldsymbol{U}_1\boldsymbol{V}_1\boldsymbol{X}^{(0)}\boldsymbol{T}_1$. Also, for the softmax errors,

$$[\boldsymbol{U}_1\boldsymbol{V}_1\boldsymbol{X}^{(0)}\boldsymbol{R}_1]_{(\text{PRE\_SUM})j} = \sum_{2 \leq i \leq j} 3\boldsymbol{X}^{(0)}_{(\text{NUM})i}[\boldsymbol{R}_1]_{ij}. \tag{23}$$

Specifically, if $j \in \{2\ell+3,\dots,3\ell+2\}$ (thus $x_j = 3$),

$$[\boldsymbol{U}_1\boldsymbol{V}_1\boldsymbol{X}^{(0)}\boldsymbol{R}_1]_{(\text{PRE\_SUM})j} = \underbrace{\sum_{i \in \{(3\ell+4)-j,(4\ell+5)-j\}} 3\boldsymbol{X}^{(0)}_{(\text{NUM})i}[\boldsymbol{R}_1]_{ij}}_{\text{negative}} + \underbrace{\sum_{\substack{2 \leq i \leq j \\ i \neq (3\ell+4)-j \\ i \neq (4\ell+5)-j}} 3\boldsymbol{X}^{(0)}_{(\text{NUM})i}[\boldsymbol{R}_1]_{ij}}_{\text{positive}}, \tag{24}$$

where

$$0 \leq - \sum_{i \in \{(3\ell+4)-j,(4\ell+5)-j\}} 3\boldsymbol{X}^{(0)}_{(\text{NUM})i}[\boldsymbol{R}_1]_{ij} \leq \frac{2 \cdot 9(j-3)}{3e^{2M} + (j-3)} \tag{25}$$

holds by Equation (17), and

$$0 \leq \sum_{\substack{2 \leq i \leq j \\ i \neq (3\ell+4)-j \\ i \neq (4\ell+5)-j}} 3\boldsymbol{X}^{(0)}_{(\text{NUM})i}[\boldsymbol{R}_1]_{ij} \leq \frac{27(j-3)}{3e^{2M} + (j-3)} \tag{26}$$

holds by Equation (18). On the other hand, if $j \in \{3\ell+3, 3\ell+4\}$,

$$0 \leq [\boldsymbol{U}_1\boldsymbol{V}_1\boldsymbol{X}^{(0)}\boldsymbol{R}_1]_{(\text{PRE\_SUM})j} = \sum_{2 \leq i \leq j} 3\boldsymbol{X}^{(0)}_{(\text{NUM})i}[\boldsymbol{R}_1]_{ij} \leq \frac{27(j-1)}{e^{2M} + (j-1)}. \tag{27}$$

One can easily prove these inequalities by using the bounds of $[\boldsymbol{R}_1]_{ij}$'s and the fact that the entries in $\boldsymbol{X}^{(0)}_{(\text{NUM})\bullet}$ lie in the interval $[0, 9]$.

If we let $M \geq \frac{1}{2}\log(N-1) + 3$, we can ensure that $\left|[\boldsymbol{U}_1\boldsymbol{V}_1\boldsymbol{X}^{(0)}\boldsymbol{R}_1]_{(\text{PRE\_SUM})j}\right|$ smaller than 0.1 for each $j \in \{2\ell+3,\dots,3\ell+4\}$. The proof is simple: it is enough to check

$$\frac{27(N-3)}{3e^{2M} + (N-3)} < \frac{1}{10}, \quad \frac{27(N-1)}{e^{2M} + (N-1)} < \frac{1}{10}. \tag{28}$$

Table 12: Example of $\boldsymbol{Q}_1\boldsymbol{X}^{(0)}$, continuing from Table 9.

| $\mathcal{I}$ | \$ | 6 | 5 | 3 | + | 0 | 4 | 9 | = | 2 | 0 | 7 | 0 |
|---|---|---|---|---|---|---|---|---|---|---|---|---|---|
| 1–$P$: | $\boldsymbol{0}_P$ | $\sqrt{M}\boldsymbol{v}_3^P$ | $\sqrt{M}\boldsymbol{v}_4^P$ | $\sqrt{M}\boldsymbol{v}_5^P$ | $\sqrt{M}\boldsymbol{v}_6^P$ | $\sqrt{M}\boldsymbol{v}_3^P$ | $\sqrt{M}\boldsymbol{v}_4^P$ | $\sqrt{M}\boldsymbol{v}_5^P$ | $\sqrt{M}\boldsymbol{v}_6^P$ | $\sqrt{M}\boldsymbol{v}_5^P$ | $\sqrt{M}\boldsymbol{v}_4^P$ | $\sqrt{M}\boldsymbol{v}_3^P$ | $\sqrt{M}\boldsymbol{v}_2^P$ |
| $P+1$: | $\sqrt{MP}$ | $\sqrt{MP}$ | $\sqrt{MP}$ | $\sqrt{MP}$ | $\sqrt{MP}$ | $\sqrt{MP}$ | $\sqrt{MP}$ | $\sqrt{MP}$ | $\sqrt{MP}$ | $\sqrt{MP}$ | $\sqrt{MP}$ | $\sqrt{MP}$ | $\sqrt{MP}$ |

Table 13: Example of $\boldsymbol{K}_1\boldsymbol{X}^{(0)}$, continuing from Table 9.

| $\mathcal{I}$ | \$ | 6 | 5 | 3 | + | 0 | 4 | 9 | = | 2 | 0 | 7 | 0 |
|---|---|---|---|---|---|---|---|---|---|---|---|---|---|
| 1–$P$: | $\boldsymbol{0}_P$ | $\sqrt{M}\boldsymbol{v}_4^P$ | $\sqrt{M}\boldsymbol{v}_5^P$ | $\sqrt{M}\boldsymbol{v}_6^P$ | $\sqrt{M}\boldsymbol{v}_7^P$ | $\sqrt{M}\boldsymbol{v}_4^P$ | $\sqrt{M}\boldsymbol{v}_5^P$ | $\sqrt{M}\boldsymbol{v}_6^P$ | $\sqrt{M}\boldsymbol{v}_7^P$ | $\sqrt{M}\boldsymbol{v}_6^P$ | $\sqrt{M}\boldsymbol{v}_5^P$ | $\sqrt{M}\boldsymbol{v}_4^P$ | $\sqrt{M}\boldsymbol{v}_3^P$ |
| $P+1$: | $\sqrt{MP}$ | 0 | 0 | 0 | 0 | 0 | 0 | 0 | 0 | 0 | 0 | 0 | 0 |

Table 14: Example of $\boldsymbol{U}_1\boldsymbol{V}_1\boldsymbol{X}^{(0)}$, continuing from Table 9.

| $\mathcal{I}$ | \$ | 6 | 5 | 3 | + | 0 | 4 | 9 | = | 2 | 0 | 7 | 0 |
|---|---|---|---|---|---|---|---|---|---|---|---|---|---|
| 1: NUM | 0 | 0 | 0 | 0 | 0 | 0 | 0 | 0 | 0 | 0 | 0 | 0 | 0 |
| 2: IS_BOS | 0 | 0 | 0 | 0 | 0 | 0 | 0 | 0 | 0 | 0 | 0 | 0 | 0 |
| 3: FULL_ONES | 0 | 0 | 0 | 0 | 0 | 0 | 0 | 0 | 0 | 0 | 0 | 0 | 0 |
| 4: PRE_SUM | 0 | 18 | 15 | 9 | 0 | 0 | 12 | 27 | 0 | 6 | 0 | 21 | 0 |
| 5: PRE_CARRY | 0 | 0 | 0 | 0 | 0 | 0 | 0 | 0 | 0 | 0 | 0 | 0 | 0 |
| 6: PRE_EOS | 0 | 0 | 0 | 0 | 0 | 0 | 0 | 0 | 0 | 0 | 0 | 0 | 0 |
| 7–16: SUM | $\boldsymbol{0}_{10}$ | $\boldsymbol{0}_{10}$ | $\boldsymbol{0}_{10}$ | $\boldsymbol{0}_{10}$ | $\boldsymbol{0}_{10}$ | $\boldsymbol{0}_{10}$ | $\boldsymbol{0}_{10}$ | $\boldsymbol{0}_{10}$ | $\boldsymbol{0}_{10}$ | $\boldsymbol{0}_{10}$ | $\boldsymbol{0}_{10}$ | $\boldsymbol{0}_{10}$ | $\boldsymbol{0}_{10}$ |
| 17: IS_EOS | 0 | 0 | 0 | 0 | 0 | 0 | 0 | 0 | 0 | 0 | 0 | 0 | 0 |
| 18–end: POS_1,POS_2 | $\boldsymbol{0}_{2P}$ | $\boldsymbol{0}_{2P}$ | $\boldsymbol{0}_{2P}$ | $\boldsymbol{0}_{2P}$ | $\boldsymbol{0}_{2P}$ | $\boldsymbol{0}_{2P}$ | $\boldsymbol{0}_{2P}$ | $\boldsymbol{0}_{2P}$ | $\boldsymbol{0}_{2P}$ | $\boldsymbol{0}_{2P}$ | $\boldsymbol{0}_{2P}$ | $\boldsymbol{0}_{2P}$ | $\boldsymbol{0}_{2P}$ |

Table 15: Example of $\boldsymbol{U}_1\boldsymbol{V}_1\boldsymbol{X}^{(0)}\boldsymbol{T}_1$, continuing from Table 14. See Table 11 for the definition of $\boldsymbol{T}_1$.

| $\mathcal{I}$ | \$ | 6 | 5 | 3 | + | 0 | 4 | 9 | = | 2 | 0 | 7 | 0 |
|---|---|---|---|---|---|---|---|---|---|---|---|---|---|
| 1: NUM | 0 | 0 | 0 | 0 | 0 | 0 | 0 | 0 | 0 | 0 | 0 | 0 | 0 |
| 2: IS_BOS | 0 | 0 | 0 | 0 | 0 | 0 | 0 | 0 | 0 | 0 | 0 | 0 | 0 |
| 3: FULL_ONES | 0 | 0 | 0 | 0 | 0 | 0 | 0 | 0 | 0 | 0 | 0 | 0 | 0 |
| 4: PRE_SUM | 0 | 0 | 9 | 7.5 | 4.5 | 0 | 6 | 9 | 12 | 9 | 6 | 0 | 0 |
| 5: PRE_CARRY | 0 | 0 | 0 | 0 | 0 | 0 | 0 | 0 | 0 | 0 | 0 | 0 | 0 |
| 6: PRE_EOS | 0 | 0 | 0 | 0 | 0 | 0 | 0 | 0 | 0 | 0 | 0 | 0 | 0 |
| 7–16: SUM | $\boldsymbol{0}_{10}$ | $\boldsymbol{0}_{10}$ | $\boldsymbol{0}_{10}$ | $\boldsymbol{0}_{10}$ | $\boldsymbol{0}_{10}$ | $\boldsymbol{0}_{10}$ | $\boldsymbol{0}_{10}$ | $\boldsymbol{0}_{10}$ | $\boldsymbol{0}_{10}$ | $\boldsymbol{0}_{10}$ | $\boldsymbol{0}_{10}$ | $\boldsymbol{0}_{10}$ | $\boldsymbol{0}_{10}$ |
| 17: IS_EOS | 0 | 0 | 0 | 0 | 0 | 0 | 0 | 0 | 0 | 0 | 0 | 0 | 0 |
| 18–end: POS_1,POS_2 | $\boldsymbol{0}_{2P}$ | $\boldsymbol{0}_{2P}$ | $\boldsymbol{0}_{2P}$ | $\boldsymbol{0}_{2P}$ | $\boldsymbol{0}_{2P}$ | $\boldsymbol{0}_{2P}$ | $\boldsymbol{0}_{2P}$ | $\boldsymbol{0}_{2P}$ | $\boldsymbol{0}_{2P}$ | $\boldsymbol{0}_{2P}$ | $\boldsymbol{0}_{2P}$ | $\boldsymbol{0}_{2P}$ | $\boldsymbol{0}_{2P}$ |

#### E.4.2    Attention Head 2: Carry & EOS Detection

The goal of the second head is to fill in the blanks of the encoding matrix at dimensions PRE_CARRY and PRE_EOS. At dimension PRE_EOS, we will put (approximately) 1 if the next token would be the EOS token ('\$'), otherwise, we will put strictly smaller numbers like (approximately) 2/3 and 1/2.

What we will put at dimension PRE_CARRY is the evidence of the presence of an additional carry, which is not quite straightforward to understand. Let us take a look at some examples. Consider an addition $3 + 9 = 12$. Since it is greater than or equal to 10, the least significant digits in the operands generate a carry 1. But in some cases, a pair of digits with a sum less than 10 can make a carry. Next, consider an addition $53 + 49 = 102$. In the second least significant digits, An addition of 5 and 4 occurs. However, a carry is already produced in the least significant digits ($3 + 9 = 12$), so the total sum including the carry is 10, not 9. Thus, it also produces a carry. But how can we know the presence of a carry while only looking at the second least significant digits? The answer is to observe

the second least significant digit in the sum, 0 of 102. Somehow, the consequence of adding 5 and 4 is 0, (or 10, implicitly) so it makes a carry.

To generalize this explanation, let $a$ and $b$ be digits of the operands in the same significance, and $c$ be a digit of the sum in the same significance as $a$ and $b$. We find that the rule of recognizing that the addition of $a$ and $b$ generates a carry is that

$$\begin{cases} \text{If } a + b - c \in \{9, 10\}, & \text{then a carry is generated}, \\ \text{Otherwise}, & \text{then the carry is not generated}. \end{cases} \tag{29}$$

Thus, it is crucial to store the information of $a + b - c$ or any related one somewhere. In fact, we can store $a + b + c$ at dimension PRE_CARRY of the encoding matrix, and it can be transformed into $a+b-c$ and used later in the feed-forward layer. Formally, we aim to perform $\sigma_i + \sigma_{i+\ell+1} + \sigma_{3\ell+5-i}$ for each $i \in \{2, ..., \ell+1\}$ and put its result at the $(3\ell + 5 - i)$-th position (column) of the dimension PRE_CARRY (row). To this end, we again utilize our position embedding.

Recall that $d = 2P + 17$ and let $d_{QK,2} = P + 1$. Let

$$Q_2 = \begin{pmatrix} \mathbf{0}_{P \times 17} & \sqrt{M}I_P & \mathbf{0}_{P \times P} \\ \sqrt{MP}(e_{\text{FULL\_ONES}}^{17})^\top & \mathbf{0}_{1 \times P} & \mathbf{0}_{1 \times P} \end{pmatrix} \in \mathbb{R}^{d_{QK,2} \times d}, \tag{30}$$

$$K_2 = \begin{pmatrix} \mathbf{0}_{P \times 17} & \sqrt{M}I_P & \mathbf{0}_{P \times P} \\ \sqrt{MP}(e_{\text{IS\_BOS}}^{17})^\top & \mathbf{0}_{1 \times P} & \mathbf{0}_{1 \times P} \end{pmatrix} \in \mathbb{R}^{d_{QK,2} \times d}. \tag{31}$$

The linear transformations with matrices $Q_2$ and $K_2$ do two different jobs at once. (1) they take the dimensions POS_1 from the input encoding matrix and scale them up by $\sqrt{M}$; (2) $Q_2$ ($K_2$, resp.) takes the dimension FULL_ONES (IS_BOS, resp.) and scale it up by $\sqrt{MP}$. For concrete examples, refer to Tables 18 and 19. By these, the attention score matrix $C_2 := (K_2 X^{(0)})^\top Q_2 X^{(0)}$ becomes as in Table 16. The blanks in Table 16 are the numbers less than equal to $M(P-2)$; the asterisks ('*') are the entries (or lower triangular submatrices) ignored by the causal softmax operator; the dots represent the hidden $MP$'s.

Table 16: Exact attention score matrix $C_2$ (with explicit row/column indices) of Head 2.

| row \ col | $j = 1$ | $2$ | $\cdots$ | $\ell+1$ | $\ell+2$ | $\ell+3$ | $\cdots$ | $2\ell+2$ | $2\ell+3$ | $2\ell+4$ | $\cdots$ | $3\ell+3$ | $3\ell+4$ |
|---|---|---|---|---|---|---|---|---|---|---|---|---|---|
| $i = 1$ | $MP$ | $MP$ | $\cdots$ | $MP$ | $MP$ | $MP$ | $\cdots$ | $MP$ | $MP$ | $MP$ | $\cdots$ | $MP$ | $MP$ |
| $2$ | $*$ | $MP$ | | | | $MP$ | | | | | | | $MP$ |
| $\vdots$ | $*$ | $*$ | $\ddots$ | | | | $\ddots$ | | | | | $\cdot^{\cdot^{\cdot}}$ | |
| $\ell+1$ | $*$ | $*$ | $*$ | $MP$ | | | | $MP$ | | $MP$ | | | |
| $\ell+2$ | $*$ | $*$ | $*$ | $*$ | $MP$ | | | | $MP$ | | | | |
| $\ell+3$ | $*$ | $*$ | $*$ | $*$ | $*$ | $MP$ | | | | | | | $MP$ |
| $\vdots$ | $*$ | $*$ | $*$ | $*$ | $*$ | $*$ | $\ddots$ | | | | | $\cdot^{\cdot^{\cdot}}$ | |
| $2\ell+2$ | $*$ | $*$ | $*$ | $*$ | $*$ | $*$ | $*$ | $MP$ | | $MP$ | | | |
| $2\ell+3$ | $*$ | $*$ | $*$ | $*$ | $*$ | $*$ | $*$ | $*$ | $MP$ | | | | |
| $2\ell+4$ | $*$ | $*$ | $*$ | $*$ | $*$ | $*$ | $*$ | $*$ | $*$ | $MP$ | | | |
| $\vdots$ | $*$ | $*$ | $*$ | $*$ | $*$ | $*$ | $*$ | $*$ | $*$ | $*$ | $\ddots$ | | |
| $3\ell+3$ | $*$ | $*$ | $*$ | $*$ | $*$ | $*$ | $*$ | $*$ | $*$ | $*$ | $*$ | $MP$ | |
| $3\ell+4$ | $*$ | $*$ | $*$ | $*$ | $*$ | $*$ | $*$ | $*$ | $*$ | $*$ | $*$ | $*$ | $MP$ |

Now consider the attention matrix $A_2 := \text{softmax}(C_2) \in \mathbb{R}^{N \times N}$. Similarly to the previous head, if the number $M$ is large enough, it gets close to the column-stochastic matrix $T_2 \in \mathbb{R}^{N \times N}$ described in Table 17. The blanks in Table 17 are zeros; the dots represent the omitted nonzero entries.

Let $R_2 = A_2 - T_2 \in \mathbb{R}^{N \times N}$ be the error matrix, which is upper triangular as well. Its exact form is messy as well, but we can obtain the bounds of their entries. Consider a pair of indices $(i, j) \in [N]^2$ such that $i \le j$. Let $x_j = 1/[T_2]_{1j} \in \{1, 2, 3, 4\}$. If $[T_2]_{ij} = \frac{1}{x_j}$, $[R_2]_{ij} < 0$ and

$$-[R_2]_{ij} \le \frac{1}{x_j} - \frac{e^{MP}}{x_j e^{MP} + (j - x_j)e^{M(P-2)}} = \frac{j - x_j}{x_j(x_j e^{2M} + (j - x_j))}. \tag{32}$$

Table 17: Limiting attention matrix $\boldsymbol{T}_2$ (with explicit row/column indices) of Head 2, as $M$ gets large.

| row \ col | $j=1$ | 2 | $\cdots$ | $\ell+1$ | $\ell+2$ | $\ell+3$ | $\cdots$ | $2\ell+2$ | $2\ell+3$ | $2\ell+4$ | $\cdots$ | $3\ell+3$ | $3\ell+4$ |
|---|---|---|---|---|---|---|---|---|---|---|---|---|---|
| $i=1$ | 1 | 1/2 | $\cdots$ | 1/2 | 1/2 | 1/3 | $\cdots$ | 1/3 | 1/3 | 1/4 | $\cdots$ | 1/4 | 1/2 |
| 2 | * | 1/2 | | 0 | 0 | 1/3 | | 0 | 0 | 0 | | 1/4 | 0 |
| $\vdots$ | * | * | $\ddots$ | | | | $\ddots$ | | | | $\ddots^{\cdot}$ | | |
| $\ell+1$ | * | * | * | 1/2 | 0 | 0 | | 1/3 | 0 | 1/4 | | 0 | 0 |
| $\ell+2$ | * | * | * | * | 1/2 | 0 | | 0 | 1/3 | 0 | | 0 | 0 |
| $\ell+3$ | * | * | * | * | * | 1/3 | | 0 | 0 | 0 | | 1/4 | 0 |
| $\vdots$ | * | * | * | * | * | * | $\ddots$ | | | | $\ddots^{\cdot}$ | | |
| $2\ell+2$ | * | * | * | * | * | * | * | 1/3 | 0 | 1/4 | | 0 | 0 |
| $2\ell+3$ | * | * | * | * | * | * | * | * | 1/3 | 0 | | 0 | 0 |
| $2\ell+4$ | * | * | * | * | * | * | * | * | * | 1/4 | | 0 | 0 |
| $\vdots$ | * | * | * | * | * | * | * | * | * | * | $\ddots$ | | |
| $3\ell+3$ | * | * | * | * | * | * | * | * | * | * | * | 1/4 | 0 |
| $3\ell+4$ | * | * | * | * | * | * | * | * | * | * | * | * | 1/2 |

On the other hand, if $[\boldsymbol{T}_2]_{ij} = 0$, $[\boldsymbol{R}_2]_{ij} > 0$ and

$$[\boldsymbol{R}_2]_{ij} \leq \frac{e^{M(P-2)}}{x_j e^{MP} + (j - x_j)e^{M(P-2)}} = \frac{1}{x_j e^{2M} + (j - x_j)}. \tag{33}$$

Now let $d_{V,2} = 2$ and

$$\boldsymbol{V}_2 = \begin{pmatrix} 4(\boldsymbol{e}_{\text{NUM}}^d)^\top \\ 2(\boldsymbol{e}_{\text{IS\_BOS}}^d)^\top \end{pmatrix} \in \mathbb{R}^{d_{V,2} \times d}, \tag{34}$$

$$\boldsymbol{U}_2 = \begin{pmatrix} \boldsymbol{e}_{\text{PRE\_CARRY}}^d & \boldsymbol{e}_{\text{PRE\_EOS}}^d \end{pmatrix} \in \mathbb{R}^{d \times d_{V,2}}. \tag{35}$$

The linear combination with matrix $\boldsymbol{U}_2\boldsymbol{V}_2$ does two jobs at once. First, it takes the dimension NUM from the encoding matrix, scales it up by 4, and puts it to the dimension PRE_CARRY. Second, it takes the dimension IS_BOS from the encoding matrix, scales it up by 2, and puts it to the dimension PRE_EOS. A concrete example is provided in Table 20.

Obtaining $\boldsymbol{U}_2\boldsymbol{V}_2\boldsymbol{X}^{(0)}\boldsymbol{A}_2$, its every entry is zero except at the dimensions PRE_CARRY and PRE_EOS. Observe that $[\boldsymbol{U}_2\boldsymbol{V}_2\boldsymbol{X}^{(0)}]_{(\text{PRE\_CARRY})1} = 0$, because in the input encoding matrix, the dimension NUM starts with 0. Also, note again that it is enough to focus on the columns $j \in \{2\ell+3, \ldots, 3\ell+4\}$, since we only care about the next-token prediction of the tokens after $\sigma_{2\ell+3} = \text{`='}$. Specifying the dimensions (i.e., the particular rows) for these columns, we have

$$[\boldsymbol{U}_2\boldsymbol{V}_2\boldsymbol{X}^{(0)}\boldsymbol{T}_2]_{(\text{PRE\_CARRY})j} = \begin{cases} \frac{4}{3}\left(\boldsymbol{X}_{(\text{NUM})(\ell+2)}^{(0)} + \boldsymbol{X}_{(\text{NUM})j}^{(0)}\right) & \text{if } (2\ell+3) = 2\ell+3, \\ \boldsymbol{X}_{(\text{NUM})(3\ell+5-j)}^{(0)} + \boldsymbol{X}_{(\text{NUM})(4\ell+6-j)}^{(0)} + \boldsymbol{X}_{(\text{NUM})j}^{(0)} & \text{if } j \in \{2\ell+4, \ldots, 3\ell+3\}, \\ 0 & \text{if } j = 3\ell+4, \end{cases} \tag{36}$$

$$= \begin{cases} 0 & \text{if } j \in \{2\ell+3, 3\ell+4\}, \\ \sigma_{(3\ell+5)-j} + \sigma_{(4\ell+6)-j} + \sigma_j & \text{if } j \in \{2\ell+4, \ldots, 3\ell+3\}, \end{cases} \tag{37}$$

$$[\boldsymbol{U}_2\boldsymbol{V}_2\boldsymbol{X}^{(0)}\boldsymbol{T}_2]_{(\text{PRE\_EOS})j} = \begin{cases} 2/3 & \text{if } j = 2\ell+3, \\ 1/2 & \text{if } j \in \{2\ell+4, \ldots, 3\ell+3\}, \\ 1 & \text{if } j = 3\ell+4. \end{cases} \tag{38}$$

Refer to Table 21 for a concrete example of computing $\boldsymbol{U}_2\boldsymbol{V}_2\boldsymbol{X}^{(0)}\boldsymbol{T}_2$. Also, for the softmax errors,

$$[\boldsymbol{U}_2\boldsymbol{V}_2\boldsymbol{X}^{(0)}\boldsymbol{R}_2]_{(\text{PRE\_CARRY})j} = \sum_{2 \leq i \leq j} 4\boldsymbol{X}_{(\text{NUM})i}^{(0)}[\boldsymbol{R}_1]_{ij}, \tag{39}$$

$$[\boldsymbol{U}_2\boldsymbol{V}_2\boldsymbol{X}^{(0)}\boldsymbol{R}_2]_{(\text{PRE\_EOS})j} = \sum_{1 \leq i \leq j} 2\boldsymbol{X}_{(\text{IS\_BOS})i}^{(0)}[\boldsymbol{R}_1]_{ij}. \tag{40}$$

Let us first obtain a bound of the softmax error term at dimension PRE_CARRY. If $j = 2\ell + 3$, since $\boldsymbol{X}^{(0)}_{(\text{NUM})(\ell+2)} = \boldsymbol{X}^{(0)}_{(\text{NUM})(2\ell+3)} = 0$,

$$[\boldsymbol{U}_2\boldsymbol{V}_2\boldsymbol{X}^{(0)}\boldsymbol{R}_2]_{(\text{PRE\_CARRY})(2\ell+3)} = \sum_{\substack{2 \le i \le 2\ell+2 \\ i \ne \ell+2}} 4\boldsymbol{X}^{(0)}_{(\text{NUM})i}[\boldsymbol{R}_1]_{ij} \tag{41}$$

and

$$0 \le \sum_{\substack{2 \le i \le 2\ell+2 \\ i \ne \ell+2}} 4\boldsymbol{X}^{(0)}_{(\text{NUM})i}[\boldsymbol{R}_1]_{ij} \le \frac{36(2\ell)}{3e^{2M} + 2\ell}. \tag{42}$$

If $j \in \{2\ell+4, \ldots, 3\ell+3\}$,

$$[\boldsymbol{U}_2\boldsymbol{V}_2\boldsymbol{X}^{(0)}\boldsymbol{R}_2]_{(\text{PRE\_CARRY})j} = \underbrace{\sum_{i \in \{(3\ell+5)-j, (4\ell+6)-j, j\}} 4\boldsymbol{X}^{(0)}_{(\text{NUM})i}[\boldsymbol{R}_1]_{ij}}_{\text{negative}} + \underbrace{\sum_{\substack{2 \le i \le j-1 \\ i \ne (3\ell+5)-j \\ i \ne (4\ell+6)-j}} 4\boldsymbol{X}^{(0)}_{(\text{NUM})i}[\boldsymbol{R}_1]_{ij}}_{\text{positive}},$$

$$\tag{43}$$

where

$$0 \le -\sum_{i \in \{(3\ell+5)-j, (4\ell+6)-j, j\}} 4\boldsymbol{X}^{(0)}_{(\text{NUM})i}[\boldsymbol{R}_1]_{ij} \le \frac{3 \cdot 9(j-4)}{4e^{2M} + (j-4)} \tag{44}$$

and

$$0 \le \sum_{\substack{2 \le i \le j-1 \\ i \ne (3\ell+5)-j \\ i \ne (4\ell+6)-j}} 4\boldsymbol{X}^{(0)}_{(\text{NUM})i}[\boldsymbol{R}_1]_{ij} \le \frac{36(j-4)}{4e^{2M} + (j-4)}. \tag{45}$$

And if $j = 3\ell + 4 = N$,

$$[\boldsymbol{U}_2\boldsymbol{V}_2\boldsymbol{X}^{(0)}\boldsymbol{R}_2]_{(\text{PRE\_CARRY})N} = \underbrace{4\boldsymbol{X}^{(0)}_{(\text{NUM})N}[\boldsymbol{R}_1]_{NN}}_{\text{negative}} + \underbrace{\sum_{2 \le i \le N-1} 4\boldsymbol{X}^{(0)}_{(\text{NUM})i}[\boldsymbol{R}_1]_{iN}}_{\text{positive}}, \tag{46}$$

where

$$0 \le -4\boldsymbol{X}^{(0)}_{(\text{NUM})N}[\boldsymbol{R}_1]_{NN} \le \frac{18(N-2)}{2e^{2M} + N - 2} \tag{47}$$

and

$$0 \le \sum_{2 \le i \le N-1} 4\boldsymbol{X}^{(0)}_{(\text{NUM})i}[\boldsymbol{R}_1]_{iN} \le \frac{36(N-2)}{2e^{2M} + N - 2}. \tag{48}$$

Next, we obtain a bound of the softmax error term at dimension PRE_EOS. Since

$$\sum_{1 \le i \le j} 2\boldsymbol{X}^{(0)}_{(\text{IS\_BOS})i}[\boldsymbol{R}_1]_{ij} = 2\boldsymbol{X}^{(0)}_{(\text{IS\_BOS})1}[\boldsymbol{R}_1]_{1j}, \tag{49}$$

the error term can be bounded as

$$0 \le -[\boldsymbol{U}_2\boldsymbol{V}_2\boldsymbol{X}^{(0)}\boldsymbol{R}_2]_{(\text{PRE\_EOS})j} \le \begin{cases} \dfrac{2(j-3)}{3(3e^{2M} + j - 3)} & \text{if } j = 2\ell + 3 \\[2ex] \dfrac{2(j-4)}{4(4e^{2M} + j - 4)} & \text{if } j \in \{2\ell+4, \ldots, 3\ell+3\}, \\[2ex] \dfrac{(j-2)}{2e^{2M} + j - 2} & \text{if } j = 3\ell + 4. \end{cases} \tag{50}$$

We then can ensure that both $\left|[\boldsymbol{U}_2\boldsymbol{V}_2\boldsymbol{X}^{(0)}\boldsymbol{R}_2]_{(\text{PRE\_SUM})j}\right|$ and $\left|[\boldsymbol{U}_2\boldsymbol{V}_2\boldsymbol{X}^{(0)}\boldsymbol{R}_2]_{(\text{PRE\_EOS})j}\right|$ smaller than 0.1 for each $j \in \{2\ell+3, \dots, 3\ell+4\}$, by letting $M \geq \frac{1}{2}\log(N)+3$. The proof is similar to the one that is presented for head 1.

Table 18: Example of $\boldsymbol{Q}_2\boldsymbol{X}^{(0)}$, continuing from Table 9.

| $\mathcal{I}$ | \$ | 6 | 5 | 3 | + | 0 | 4 | 9 | = | 2 | 0 | 7 | 0 |
|---|---|---|---|---|---|---|---|---|---|---|---|---|---|
| 1–$P$: | $\boldsymbol{0}_P$ | $\sqrt{M}\boldsymbol{v}_3^P$ | $\sqrt{M}\boldsymbol{v}_4^P$ | $\sqrt{M}\boldsymbol{v}_5^P$ | $\sqrt{M}\boldsymbol{v}_6^P$ | $\sqrt{M}\boldsymbol{v}_3^P$ | $\sqrt{M}\boldsymbol{v}_4^P$ | $\sqrt{M}\boldsymbol{v}_5^P$ | $\sqrt{M}\boldsymbol{v}_6^P$ | $\sqrt{M}\boldsymbol{v}_5^P$ | $\sqrt{M}\boldsymbol{v}_4^P$ | $\sqrt{M}\boldsymbol{v}_3^P$ | $\sqrt{M}\boldsymbol{v}_2^P$ |
| $P+1$: | $\sqrt{MP}$ | $\sqrt{MP}$ | $\sqrt{MP}$ | $\sqrt{MP}$ | $\sqrt{MP}$ | $\sqrt{MP}$ | $\sqrt{MP}$ | $\sqrt{MP}$ | $\sqrt{MP}$ | $\sqrt{MP}$ | $\sqrt{MP}$ | $\sqrt{MP}$ | $\sqrt{MP}$ |

Table 19: Example of $\boldsymbol{K}_2\boldsymbol{X}^{(0)}$, continuing from Table 9.

| $\mathcal{I}$ | \$ | 6 | 5 | 3 | + | 0 | 4 | 9 | = | 2 | 0 | 7 | 0 |
|---|---|---|---|---|---|---|---|---|---|---|---|---|---|
| 1–$P$: | $\boldsymbol{0}_P$ | $\sqrt{M}\boldsymbol{v}_3^P$ | $\sqrt{M}\boldsymbol{v}_4^P$ | $\sqrt{M}\boldsymbol{v}_5^P$ | $\sqrt{M}\boldsymbol{v}_6^P$ | $\sqrt{M}\boldsymbol{v}_3^P$ | $\sqrt{M}\boldsymbol{v}_4^P$ | $\sqrt{M}\boldsymbol{v}_5^P$ | $\sqrt{M}\boldsymbol{v}_6^P$ | $\sqrt{M}\boldsymbol{v}_5^P$ | $\sqrt{M}\boldsymbol{v}_4^P$ | $\sqrt{M}\boldsymbol{v}_3^P$ | $\sqrt{M}\boldsymbol{v}_2^P$ |
| $P+1$: | $\sqrt{MP}$ | 0 | 0 | 0 | 0 | 0 | 0 | 0 | 0 | 0 | 0 | 0 | 0 |

Table 20: Example of $\boldsymbol{U}_2\boldsymbol{V}_2\boldsymbol{X}^{(0)}$, continuing from Table 9.

| $\mathcal{I}$ | \$ | 6 | 5 | 3 | + | 0 | 4 | 9 | = | 2 | 0 | 7 | 0 |
|---|---|---|---|---|---|---|---|---|---|---|---|---|---|
| 1: NUM | 0 | 0 | 0 | 0 | 0 | 0 | 0 | 0 | 0 | 0 | 0 | 0 | 0 |
| 2: IS_BOS | 0 | 0 | 0 | 0 | 0 | 0 | 0 | 0 | 0 | 0 | 0 | 0 | 0 |
| 3: FULL_ONES | 0 | 0 | 0 | 0 | 0 | 0 | 0 | 0 | 0 | 0 | 0 | 0 | 0 |
| 4: PRE_SUM | 0 | 0 | 0 | 0 | 0 | 0 | 0 | 0 | 0 | 0 | 0 | 0 | 0 |
| 5: PRE_CARRY | 0 | 24 | 20 | 12 | 0 | 0 | 16 | 36 | 0 | 8 | 0 | 28 | 0 |
| 6: PRE_EOS | 2 | 0 | 0 | 0 | 0 | 0 | 0 | 0 | 0 | 0 | 0 | 0 | 0 |
| 7–16: SUM | $\boldsymbol{0}_{10}$ | $\boldsymbol{0}_{10}$ | $\boldsymbol{0}_{10}$ | $\boldsymbol{0}_{10}$ | $\boldsymbol{0}_{10}$ | $\boldsymbol{0}_{10}$ | $\boldsymbol{0}_{10}$ | $\boldsymbol{0}_{10}$ | $\boldsymbol{0}_{10}$ | $\boldsymbol{0}_{10}$ | $\boldsymbol{0}_{10}$ | $\boldsymbol{0}_{10}$ | $\boldsymbol{0}_{10}$ |
| 17: IS_EOS | 0 | 0 | 0 | 0 | 0 | 0 | 0 | 0 | 0 | 0 | 0 | 0 | 0 |
| 18–end: POS_1 | $\boldsymbol{0}_{2P}$ | $\boldsymbol{0}_{2P}$ | $\boldsymbol{0}_{2P}$ | $\boldsymbol{0}_{2P}$ | $\boldsymbol{0}_{2P}$ | $\boldsymbol{0}_{2P}$ | $\boldsymbol{0}_{2P}$ | $\boldsymbol{0}_{2P}$ | $\boldsymbol{0}_{2P}$ | $\boldsymbol{0}_{2P}$ | $\boldsymbol{0}_{2P}$ | $\boldsymbol{0}_{2P}$ | $\boldsymbol{0}_{2P}$ |

Table 21: Example of $\boldsymbol{U}_2\boldsymbol{V}_2\boldsymbol{X}^{(0)}\boldsymbol{T}_2$, continuing from Table 20. See Table 17 for definition of $\boldsymbol{T}_2$.

| $\mathcal{I}$ | \$ | 6 | 5 | 3 | + | 0 | 4 | 9 | = | 2 | 0 | 7 | 0 |
|---|---|---|---|---|---|---|---|---|---|---|---|---|---|
| 1: NUM | 0 | 0 | 0 | 0 | 0 | 0 | 0 | 0 | 0 | 0 | 0 | 0 | 0 |
| 2: IS_BOS | 0 | 0 | 0 | 0 | 0 | 0 | 0 | 0 | 0 | 0 | 0 | 0 | 0 |
| 3: FULL_ONES | 0 | 0 | 0 | 0 | 0 | 0 | 0 | 0 | 0 | 0 | 0 | 0 | 0 |
| 4: PRE_SUM | 0 | 0 | 0 | 0 | 0 | 0 | 0 | 0 | 0 | 0 | 0 | 0 | 0 |
| 5: PRE_CARRY | 0 | 12 | 10 | 6 | 0 | 8 | 12 | 16 | 0 | 14 | 9 | 13 | 0 |
| 6: PRE_EOS | 2 | 1 | 1 | 1 | 1 | 2/3 | 2/3 | 2/3 | 2/3 | 1/2 | 1/2 | 1/2 | 1 |
| 7–16: SUM | $\boldsymbol{0}_{10}$ | $\boldsymbol{0}_{10}$ | $\boldsymbol{0}_{10}$ | $\boldsymbol{0}_{10}$ | $\boldsymbol{0}_{10}$ | $\boldsymbol{0}_{10}$ | $\boldsymbol{0}_{10}$ | $\boldsymbol{0}_{10}$ | $\boldsymbol{0}_{10}$ | $\boldsymbol{0}_{10}$ | $\boldsymbol{0}_{10}$ | $\boldsymbol{0}_{10}$ | $\boldsymbol{0}_{10}$ |
| 17: IS_EOS | 0 | 0 | 0 | 0 | 0 | 0 | 0 | 0 | 0 | 0 | 0 | 0 | 0 |
| 18–end: POS_1 | $\boldsymbol{0}_{2P}$ | $\boldsymbol{0}_{2P}$ | $\boldsymbol{0}_{2P}$ | $\boldsymbol{0}_{2P}$ | $\boldsymbol{0}_{2P}$ | $\boldsymbol{0}_{2P}$ | $\boldsymbol{0}_{2P}$ | $\boldsymbol{0}_{2P}$ | $\boldsymbol{0}_{2P}$ | $\boldsymbol{0}_{2P}$ | $\boldsymbol{0}_{2P}$ | $\boldsymbol{0}_{2P}$ | $\boldsymbol{0}_{2P}$ |

### E.4.3 Residual Connection

So far we have computed the output of $\texttt{Att}_1$ operation. Passing through the residual connection, the output of the attention layer is the sum of the original input encoding matrix and the output of $\texttt{Att}$ operation:

$$\boldsymbol{Y}^{(1)} = \boldsymbol{X}^{(0)} + \sum_{h \in \{1,2\}} \boldsymbol{U}_h \boldsymbol{V}_h \boldsymbol{X}^{(0)} \boldsymbol{T}_h + \underbrace{\sum_{h \in \{1,2\}} \boldsymbol{U}_h \boldsymbol{V}_h \boldsymbol{X}^{(0)} \boldsymbol{R}_h}_{\text{softmax error term}}. \tag{51}$$

Since the term $\sum_{h\in\{1,2\}} \boldsymbol{U}_h \boldsymbol{V}_h \boldsymbol{X}^{(0)} \boldsymbol{T}_h$ has nonzero entries only at dimensions PRE_SUM, PRE_CARRY, and PRE_EOS, the residual connection plays a role of "filling in some blanks" in the input encoding matrix. A concrete example of the output of residual connection is presented in Table 22, ignoring the softmax error term, whose entries have an absolute value smaller than 0.1.

Table 22: Example output of residual connection, continuing from Tables 9, 15 and 21. Here we ignore the softmax error terms in the orange rows. The gray rows will be filled in later.

| $\mathcal{I}$ | \$ | 6 | 5 | 3 | + | 0 | 4 | 9 | = | 2 | 0 | 7 | 0 |
|---|---|---|---|---|---|---|---|---|---|---|---|---|---|
| 1: NUM | 0 | 6 | 5 | 3 | 0 | 0 | 4 | 9 | 0 | 2 | 0 | 7 | 0 |
| 2: IS_BOS | 1 | 0 | 0 | 0 | 0 | 0 | 0 | 0 | 0 | 0 | 0 | 0 | 0 |
| 3: FULL_ONES | 1 | 1 | 1 | 1 | 1 | 1 | 1 | 1 | 1 | 1 | 1 | 1 | 1 |
| 4: PRE_SUM | 0 | 0 | 9 | 7.5 | 4.5 | 0 | 6 | 9 | 12 | 9 | 6 | 0 | 0 |
| 5: PRE_CARRY | 0 | 12 | 10 | 6 | 0 | 8 | 12 | 16 | 0 | 14 | 9 | 13 | 0 |
| 6: PRE_EOS | 2 | 1 | 1 | 1 | 1 | 2/3 | 2/3 | 2/3 | 2/3 | 1/2 | 1/2 | 1/2 | 1 |
| 7–16: SUM | $\boldsymbol{0}_{10}$ | $\boldsymbol{0}_{10}$ | $\boldsymbol{0}_{10}$ | $\boldsymbol{0}_{10}$ | $\boldsymbol{0}_{10}$ | $\boldsymbol{0}_{10}$ | $\boldsymbol{0}_{10}$ | $\boldsymbol{0}_{10}$ | $\boldsymbol{0}_{10}$ | $\boldsymbol{0}_{10}$ | $\boldsymbol{0}_{10}$ | $\boldsymbol{0}_{10}$ | $\boldsymbol{0}_{10}$ |
| 17: IS_EOS | 0 | 0 | 0 | 0 | 0 | 0 | 0 | 0 | 0 | 0 | 0 | 0 | 0 |
| 18–$(P+17)$: POS_1 | $\boldsymbol{0}_P$ | $\boldsymbol{v}_3^P$ | $\boldsymbol{v}_4^P$ | $\boldsymbol{v}_5^P$ | $\boldsymbol{v}_6^P$ | $\boldsymbol{v}_3^P$ | $\boldsymbol{v}_4^P$ | $\boldsymbol{v}_5^P$ | $\boldsymbol{v}_6^P$ | $\boldsymbol{v}_5^P$ | $\boldsymbol{v}_4^P$ | $\boldsymbol{v}_3^P$ | $\boldsymbol{v}_2^P$ |
| $(P+18)$-$(2P+17)$: POS_2 | $\boldsymbol{0}_P$ | $\boldsymbol{v}_4^P$ | $\boldsymbol{v}_5^P$ | $\boldsymbol{v}_6^P$ | $\boldsymbol{v}_7^P$ | $\boldsymbol{v}_4^P$ | $\boldsymbol{v}_5^P$ | $\boldsymbol{v}_6^P$ | $\boldsymbol{v}_7^P$ | $\boldsymbol{v}_6^P$ | $\boldsymbol{v}_5^P$ | $\boldsymbol{v}_4^P$ | $\boldsymbol{v}_3^P$ |

## E.5 Transformer Block — Token-wise Feed-forward Layer

The goal of the feed-forward layer is to fill in the blanks of the encoding matrix at dimensions SUM and IS_EOS. Be careful that the feed-forward layer can only implement token-wise mappings; a token-wise mapping takes inputs only from the entries in the same column of the encoding matrix. Besides, the architecture of our feed-forward layer (except for the residual connection) is a one-hidden-layer ReLU network.

For a token $\sigma_i$ for $i \in \{2\ell+3, \ldots, 3\ell+3\}$ (from '=' token to the second ), we will put a standard unit vector $\boldsymbol{e}_{k+1}^{10}$ to dimensions SUM if the next token is $k \in \{0, \ldots, 9\}$.

Recall from the discussion in Appendix E.4.2 that we can judge whether a carry 1 is generated at a certain position by exploiting only the digits (of the operands and the sum) in the same significance. Bringing the notation, let $a$ and $b$ be digits of the operands in the same significance, and $c$ be a digit of the sum in the same significance as $a$ and $b$. Then the rule of recognizing that the addition of $a$ and $b$ generates a carry is that

$$\begin{cases} \text{If } a+b-c \in \{9,10\}, & \text{then a carry is generated,} \\ \text{Otherwise: if } a+b-c \in \{-1,0\}, & \text{then the carry is not generated.} \end{cases} \quad (52)$$

A simple case analysis shows that the value of $a+b-c$ must be one of $-1, 0, 9$, and $10$. Let us briefly check this claim in our example:

$$6 + 0 - 7 = -1; \qquad \text{no carry from 6+0} \quad (53)$$
$$5 + 4 - 0 = 9; \qquad \text{there is a carry from 5+4} \quad (54)$$
$$3 + 9 - 2 = 10. \qquad \text{there is a carry from 3+9} \quad (55)$$

Recall that a noisy version of $a+b+c$ is already stored at dimension PRE_CARRY of $\boldsymbol{Y}^{(1)}$, and $c$ is exactly at dimension NUM. Thus, we can (approximately) implement $a+b-c$ for a token $\sigma_j$ by

$$\boldsymbol{Y}^{(0)}_{(\text{PRE\_CARRY})j} - 2\boldsymbol{Y}^{(0)}_{(\text{NUM})j}. \quad (56)$$

This is a kind of token-wise *linear* transform, so we do not need to consume any hidden layer (with ReLU activation $\phi$) to implement it.

Combining with $\boldsymbol{Y}^{(0)}_{(\text{PRE\_SUM})j}$, a noisy version of addition without carry, we can indeed implement the addition. Note that a digit-wise addition should be done as

$$\text{digit-wise addition} = (\text{addition without carry} + \mathbb{1}_{\{\text{carry propagates}\}}) \mod 10. \quad (57)$$

We first describe the formal construction of feed-forward network FF$_1$ for dimensions SUM and IS_EOS and then explain the intuition behind the construction. For the example result of applying the feed-forward network is presented in Table 23.

### E.5.1 Subnetwork 1: Construction for SUM (dimension 7–16).

Given a vector $\boldsymbol{y} = [\boldsymbol{y}_j]_{j=1}^d \in \mathbb{R}^d$, define a linear function $g : \mathbb{R}^d \to \mathbb{R}$ as

$$g(\boldsymbol{y}) := \boldsymbol{y}_{\text{PRE\_SUM}} + \frac{\boldsymbol{y}_{\text{PRE\_CARRY}} - 2\boldsymbol{y}_{\text{NUM}}}{10} + 0.21 = \boldsymbol{y}_3 + \frac{\boldsymbol{y}_4 - 2\boldsymbol{y}_1}{10} + 0.21 \tag{58}$$

and consider a one-hidden-layer ReLU network $f_k : \mathbb{R} \to \mathbb{R}$ $(k = 0, 1, \ldots, 9)$ defined as

$$\begin{aligned} f_k(x) = 2\Big[ &\phi(x - (k - 0.5)) - \phi(x - k) - \phi(x - (k + 0.5)) + \phi(x - (k + 1)) \\ &+ \phi(x - (k + 9.5)) - \phi(x - (k + 10)) - \phi(x - (k + 10.5)) + \phi(x - (k + 11)) \Big]. \end{aligned} \tag{59}$$

Then we construct a subnetwork of our feed-forward network for a token $\sigma_j$ by

$$\left[ \text{FF}_1 \left( \boldsymbol{Y}^{(1)} \right) \right]_{(\text{SUM})j} = \left[ f_0 \left( g \left( \boldsymbol{Y}^{(1)}_{\bullet j} \right) \right) \quad \cdots \quad f_9 \left( g \left( \boldsymbol{Y}^{(1)}_{\bullet j} \right) \right) \right]^\top. \tag{60}$$

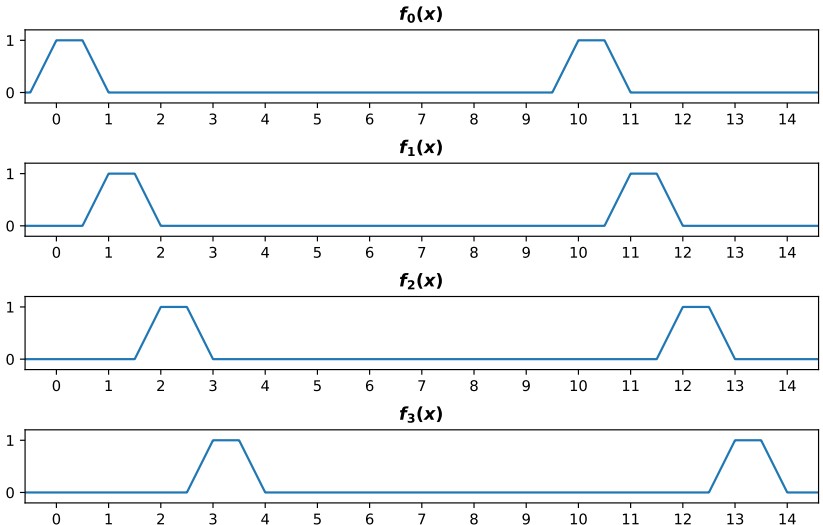

Figure 23: Example plots of $f_k(x)$ defined in Equation (59). ($k = 0, 1, 2, 3$)

**Explanation.** The purpose of the first subnetwork is to generate a 10-dimensional one-hot vector whose position of 1 indicates the next digit: $\boldsymbol{e}_k^{10}$ for the answer of next-token prediction '$k$'. There are two cases where we need to predict the next token as '$k$':

- Case 1: (Addition without carry) $= k \mod 10$ and no carry propagates.
- Case 2: (Addition without carry) $= k - 1 \mod 10$ and there is a propagating carry 1.

In the first case, due to the softmax error (with magnitude at most $0.1$),

$$\boldsymbol{Y}^{(0)}_{(\text{PRE\_SUM})j} \in [k - 0.1, k + 0.1] \cap [k + 9.9, k + 10.1], \tag{61}$$

$$\boldsymbol{Y}^{(0)}_{(\text{PRE\_CARRY})j} - 2\boldsymbol{Y}^{(0)}_{(\text{NUM})j} \in [-1.1, -0.9] \cap [-0.1, 0.1] \subset [-1.1, 0.1]. \tag{62}$$

In the second case, again due to the softmax error (with magnitude at most $0.1$),

$$\boldsymbol{Y}^{(0)}_{(\text{PRE\_SUM})j} + 1 \in [k - 0.1, k + 0.1] \cap [k + 9.9, k + 10.1], \tag{63}$$

$$\boldsymbol{Y}^{(0)}_{(\text{PRE\_CARRY})j} - 2\boldsymbol{Y}^{(0)}_{(\text{NUM})j} - 10 \in [-1.1, -0.9] \cap [-0.1, 0.1] \subset [-1.1, 0.1]. \tag{64}$$

In both cases,

$$Y^{(0)}_{(\text{PRE\_SUM})j} + \frac{Y^{(0)}_{(\text{PRE\_CARRY})j} - 2Y^{(0)}_{(\text{NUM})j}}{10} + 0.21 \in [k, k+0.32] \cap [k+10, k+10.32] \qquad (65)$$

$$\subset [k, k+0.5] \cap [k+10, k+10.5]. \qquad (66)$$

We can map the column $Y^{(0)}_{\bullet j}$ to the set $[k, k+0.5] \cap [k+10, k+10.5]$ if the next token is $\sigma_{j+1} = \text{`}k\text{'}$. This job is done by the function $g$. Note that the resulting sets $[k, k+0.5] \cap [k+10, k+10.5]$ are disjoint for different $k$'s.

Recall that our objective is to output 1 to the dimension $k+6$ (among the dimensions 7, 8, ..., 16 in SUM) and to output 0 to the other dimensions in SUM if we need to predict '$k$' as the next token. To this end, it is enough to map the set $[k, k+0.5] \cap [k+10, k+10.5]$ to 1 and to map the other sets (for different $k$'s) to 0. This can be done by a ReLU network $f_k(x)$ is a ReLU network having two bumps at intervals $[k-0.5, k+1]$ and $[k+9.5, k+11]$. In particular, $f_k(x) = 1$ if $x \in [k, k+0.5] \cup [k+10, k+10.5]$: see Figure 23 for an illustration.

Lastly, we have a desired one-hot vector output for each $j$ by taking a composition between $g$ and $[f_0(\cdot), \ldots, f_9(\cdot)]^\top$ as written in Equation (60).

### E.5.2 Subnetwork 2: Construction for IS_EOS (dimension 17).

We move on to the dimension IS_EOS. For a token $\sigma_j$ for $j \in \{2\ell+3, \ldots, 3\ell+4\}$, if $k$ is the next token, we will put $\mathbb{1}_{\{k=\$\}}$ to dimension IS_EOS: 1 if $k$ is the special token '$\$$' and 0 otherwise. To this end, we define a ReLU network $h : \mathbb{R} \to \mathbb{R}$ as

$$h(x) = 10\phi(x - 0.8) - 10\phi(x - 0.9). \qquad (67)$$

Then, we can construct a subnetwork of our feed-forward network for a token $\sigma_j$ by

$$\left[ \text{FF}_1\left( Y^{(1)} \right) \right]_{(\text{IS\_EOS})j} = h\left( Y^{(1)}_{(\text{PRE\_EOS})j} \right). \qquad (68)$$

**Explanation.** Note that for columns $j \in \{2\ell+3, \ldots, 3\ell+4\}$, if we consider the presence of softmax errors with magnitude at most 0.1, the values that $Y^{(1)}_{(\text{PRE\_EOS})j}$ can have lie in the set $[0.4, 0.6] \cap [2/3 - 0.1, 2/3 + 0.1] \cap [0.9, 1.1] \subset (-\infty, 0.8) \cap [0.9, \infty)$. We want to output 1 if $Y^{(1)}_{(\text{PRE\_EOS})j} \geq 0.9$ and 0 otherwise: this can be done with the ReLU network $h$ with two neurons.

**Remarks:**

- In total, we consume $8 \times 10 + 2 = 82$ ReLU neurons in our feed-forward network $\text{FF}_1$. However, it is possible to construct the addition Transformer with a smaller number of neurons, with a slight modification in the linear readout of the decoding function (Appendix E.6).

- Unlike in the attention layer, now we do not have to worry about softmax errors in the output since the feed-forward ReLU network plays the role of *denoiser*.

Table 23: Example output after applying the feed-forward network.

| $\mathcal{I}$ | $\$$ | 6 | 5 | 3 | + | 0 | 4 | 9 | = | 2 | 0 | 7 | 0 |
|---|---|---|---|---|---|---|---|---|---|---|---|---|---|
| 1: NUM | 0 | 0 | 0 | 0 | 0 | 0 | 0 | 0 | 0 | 0 | 0 | 0 | 0 |
| 2: IS_BOS | 0 | 0 | 0 | 0 | 0 | 0 | 0 | 0 | 0 | 0 | 0 | 0 | 0 |
| 3: FULL_ONES | 0 | 0 | 0 | 0 | 0 | 0 | 0 | 0 | 0 | 0 | 0 | 0 | 0 |
| 4: PRE_SUM | 0 | 0 | 0 | 0 | 0 | 0 | 0 | 0 | 0 | 0 | 0 | 0 | 0 |
| 5: PRE_CARRY | 0 | 0 | 0 | 0 | 0 | 0 | 0 | 0 | 0 | 0 | 0 | 0 | 0 |
| 6: PRE_EOS | 0 | 0 | 0 | 0 | 0 | 0 | 0 | 0 | 0 | 0 | 0 | 0 | 0 |
| 7–16: SUM | $e^{10}_1$ | $e^{10}_1$ | $e^{10}_{10}$ | $e^{10}_8$ | $e^{10}_5$ | $e^{10}_2$ | $e^{10}_7$ | $e^{10}_{10}$ | $e^{10}_3$ | $e^{10}_1$ | $e^{10}_8$ | $e^{10}_1$ | $e^{10}_1$ |
| 17: IS_EOS | 1 | 1 | 1 | 1 | 1 | 0 | 0 | 0 | 0 | 0 | 0 | 0 | 1 |
| 18–($P+17$): POS_1 | $\mathbf{0}_P$ | $v^P_3$ | $v^P_4$ | $v^P_5$ | $v^P_6$ | $v^P_3$ | $v^P_4$ | $v^P_5$ | $v^P_6$ | $v^P_5$ | $v^P_4$ | $v^P_3$ | $v^P_2$ |
| ($P+18$)-($2P+17$): POS_2 | $\mathbf{0}_P$ | $v^P_4$ | $v^P_5$ | $v^P_6$ | $v^P_7$ | $v^P_4$ | $v^P_5$ | $v^P_6$ | $v^P_7$ | $v^P_6$ | $v^P_5$ | $v^P_4$ | $v^P_3$ |

### E.5.3   Residual Connection

The last task of the feed-forward layer is to pass $\mathrm{FF}_1\left(\boldsymbol{Y}^{(1)}\right)$ through the residual connection. As a result, we have

$$\boldsymbol{X}^{(1)} = \boldsymbol{Y}^{(1)} + \mathrm{FF}_1\left(\boldsymbol{Y}^{(1)}\right). \tag{69}$$

A concrete example of the output of the second residual connection is showcased in Table 24.

Table 24: Example output of residual connection, continuing from Table 23. Here we ignore the softmax error terms in the orange rows.

| $\mathcal{I}$ | $ | 6 | 5 | 3 | + | 0 | 4 | 9 | = | 2 | 0 | 7 | 0 |
|---|---|---|---|---|---|---|---|---|---|---|---|---|---|
| 1: NUM | 0 | 6 | 5 | 3 | 0 | 0 | 4 | 9 | 0 | 2 | 0 | 7 | 0 |
| 2: IS_BOS | 1 | 0 | 0 | 0 | 0 | 0 | 0 | 0 | 0 | 0 | 0 | 0 | 0 |
| 3: FULL_ONES | 1 | 1 | 1 | 1 | 1 | 1 | 1 | 1 | 1 | 1 | 1 | 1 | 1 |
| 4: PRE_SUM | 0 | 0 | 9 | 7.5 | 4.5 | 0 | 6 | 9 | 12 | 9 | 6 | 0 | 0 |
| 5: PRE_CARRY | 0 | 12 | 10 | 6 | 0 | 8 | 12 | 16 | 0 | 14 | 9 | 13 | 0 |
| 6: PRE_EOS | 2 | 1 | 1 | 1 | 1 | 2/3 | 2/3 | 2/3 | 2/3 | 1/2 | 1/2 | 1/2 | 1 |
| 7–16: SUM | $\boldsymbol{e}_1^{10}$ | $\boldsymbol{e}_1^{10}$ | $\boldsymbol{e}_{10}^{10}$ | $\boldsymbol{e}_8^{10}$ | $\boldsymbol{e}_5^{10}$ | $\boldsymbol{e}_2^{10}$ | $\boldsymbol{e}_7^{10}$ | $\boldsymbol{e}_{10}^{10}$ | $\boldsymbol{e}_3^{10}$ | $\boldsymbol{e}_1^{10}$ | $\boldsymbol{e}_8^{10}$ | $\boldsymbol{e}_1^{10}$ | $\boldsymbol{e}_1^{10}$ |
| 17: IS_EOS | 1 | 1 | 1 | 1 | 1 | 0 | 0 | 0 | 0 | 0 | 0 | 0 | 1 |
| 18–(P + 17): POS_1 | $\boldsymbol{0}_P$ | $\boldsymbol{v}_3^P$ | $\boldsymbol{v}_4^P$ | $\boldsymbol{v}_5^P$ | $\boldsymbol{v}_6^P$ | $\boldsymbol{v}_3^P$ | $\boldsymbol{v}_4^P$ | $\boldsymbol{v}_5^P$ | $\boldsymbol{v}_6^P$ | $\boldsymbol{v}_5^P$ | $\boldsymbol{v}_4^P$ | $\boldsymbol{v}_3^P$ | $\boldsymbol{v}_2^P$ |
| (P + 18)-(2P + 17): POS_2 | $\boldsymbol{0}_P$ | $\boldsymbol{v}_4^P$ | $\boldsymbol{v}_5^P$ | $\boldsymbol{v}_6^P$ | $\boldsymbol{v}_7^P$ | $\boldsymbol{v}_4^P$ | $\boldsymbol{v}_5^P$ | $\boldsymbol{v}_6^P$ | $\boldsymbol{v}_7^P$ | $\boldsymbol{v}_6^P$ | $\boldsymbol{v}_5^P$ | $\boldsymbol{v}_4^P$ | $\boldsymbol{v}_3^P$ |

### E.6   Decoding Function

As mentioned in Appendix D, the decoding function performs a linear readout (with a weight matrix $\boldsymbol{W}_{\mathrm{out}} \in \mathbb{R}^{|\mathcal{V}| \times d}$) and a (token-wise) arg-max operation. That is,

$$\mathrm{Dec}\left(\boldsymbol{X}^{(1)}\right) := \left(\mathcal{V}_{k_i}\right)_{i=1,\ldots,N} \in \mathcal{V}^N, \tag{70}$$

where $\mathcal{V}_k$ is the $k$-th element of $\mathcal{V}$ and

$$k_i := \arg\max_{k \in [|\mathcal{V}|]} \left\{ o_k : \boldsymbol{W}_{\mathrm{out}} \boldsymbol{X}_{\bullet i}^{(1)} = \begin{bmatrix} o_1 & \cdots & o_{|\mathcal{V}|} \end{bmatrix}^\top \right\}. \tag{71}$$

The objective of the decoding function is to perform a proper next-token prediction for addition, especially utilizing the dimensions SUM and IS_EOS of $\boldsymbol{X}^{(1)}$.

We now construct the weight matrix $\boldsymbol{W}_{\mathrm{out}}$. For a token $\sigma_i$, if the value of dimension IS_EOS of $\boldsymbol{X}^{(1)}$ is 0, then the linear readout output the dimensions SUM as it is to return one of a number token (0-9). On the other hand, if the value of dimension IS_EOS is 1, then the linear readout outputs a large number (like 100 for example) for the token '$' to return EOS ($). This can be implemented by the weight matrix $\boldsymbol{W}_{\mathrm{out}}$ described in Table 25. Also, an example of applying the linear transform is showcased in Table 26.

Table 25: The *transposed* weight matrix $\boldsymbol{W}_{out}^{\top}$ of the linear readout in decoding function.

| $\mathcal{V}$ | 0 | 1 | 2 | 3 | 4 | 5 | 6 | 7 | 8 | 9 | + | = | $ |
|---|---|---|---|---|---|---|---|---|---|---|---|---|---|
| 1-6: NUM-PRE_EOS | $\boldsymbol{0}_6$ | $\boldsymbol{0}_6$ | $\boldsymbol{0}_6$ | $\boldsymbol{0}_6$ | $\boldsymbol{0}_6$ | $\boldsymbol{0}_6$ | $\boldsymbol{0}_6$ | $\boldsymbol{0}_6$ | $\boldsymbol{0}_6$ | $\boldsymbol{0}_6$ | $\boldsymbol{0}_6$ | $\boldsymbol{0}_6$ | $\boldsymbol{0}_6$ |
| 7: SUM$_1$ | 1 | 0 | 0 | 0 | 0 | 0 | 0 | 0 | 0 | 0 | 0 | 0 | 0 |
| 8: SUM$_2$ | 0 | 1 | 0 | 0 | 0 | 0 | 0 | 0 | 0 | 0 | 0 | 0 | 0 |
| 9: SUM$_3$ | 0 | 0 | 1 | 0 | 0 | 0 | 0 | 0 | 0 | 0 | 0 | 0 | 0 |
| 10: SUM$_4$ | 0 | 0 | 0 | 1 | 0 | 0 | 0 | 0 | 0 | 0 | 0 | 0 | 0 |
| 11: SUM$_5$ | 0 | 0 | 0 | 0 | 1 | 0 | 0 | 0 | 0 | 0 | 0 | 0 | 0 |
| 12: SUM$_6$ | 0 | 0 | 0 | 0 | 0 | 1 | 0 | 0 | 0 | 0 | 0 | 0 | 0 |
| 13: SUM$_7$ | 0 | 0 | 0 | 0 | 0 | 0 | 1 | 0 | 0 | 0 | 0 | 0 | 0 |
| 14: SUM$_8$ | 0 | 0 | 0 | 0 | 0 | 0 | 0 | 1 | 0 | 0 | 0 | 0 | 0 |
| 15: SUM$_9$ | 0 | 0 | 0 | 0 | 0 | 0 | 0 | 0 | 1 | 0 | 0 | 0 | 0 |
| 16: SUM$_{10}$ | 0 | 0 | 0 | 0 | 0 | 0 | 0 | 0 | 0 | 1 | 0 | 0 | 0 |
| 17: IS_EOS | 0 | 0 | 0 | 0 | 0 | 0 | 0 | 0 | 0 | 0 | 0 | 0 | 100 |
| 18–end: POS_1, POS_2 | $\boldsymbol{0}_{2P}$ | $\boldsymbol{0}_{2P}$ | $\boldsymbol{0}_{2P}$ | $\boldsymbol{0}_{2P}$ | $\boldsymbol{0}_{2P}$ | $\boldsymbol{0}_{2P}$ | $\boldsymbol{0}_{2P}$ | $\boldsymbol{0}_{2P}$ | $\boldsymbol{0}_{2P}$ | $\boldsymbol{0}_{2P}$ | $\boldsymbol{0}_{2P}$ | $\boldsymbol{0}_{2P}$ | $\boldsymbol{0}_{2P}$ |

Table 26: Example output of linear readout ($\boldsymbol{W}_{\text{out}}\boldsymbol{X}^{(1)}$), continuing from Tables 24 and 25. The yellow cells represent the maximum value of each column, from the '=' token's column to the rightmost column (used for next-token prediction).

| $\mathcal{I}$ | $ | 6 | 5 | 3 | + | 0 | 4 | 9 | = | 2 | 0 | 7 | 0 |
|---|---|---|---|---|---|---|---|---|---|---|---|---|---|
| 0 | 1 | 1 | 0 | 0 | 0 | 0 | 0 | 0 | 0 | 1 | 0 | 1 | 1 |
| 1 | 0 | 0 | 0 | 0 | 0 | 1 | 0 | 0 | 0 | 0 | 0 | 0 | 0 |
| 2 | 0 | 0 | 0 | 0 | 0 | 0 | 0 | 0 | 1 | 0 | 0 | 0 | 0 |
| 3 | 0 | 0 | 0 | 0 | 0 | 0 | 0 | 0 | 0 | 0 | 0 | 0 | 0 |
| 4 | 0 | 0 | 0 | 0 | 1 | 0 | 0 | 0 | 0 | 0 | 0 | 0 | 0 |
| 5 | 0 | 0 | 0 | 0 | 0 | 0 | 0 | 0 | 0 | 0 | 0 | 0 | 0 |
| 6 | 0 | 0 | 0 | 0 | 0 | 0 | 1 | 0 | 0 | 0 | 0 | 0 | 0 |
| 7 | 0 | 0 | 0 | 1 | 0 | 0 | 0 | 0 | 0 | 0 | 1 | 0 | 0 |
| 8 | 0 | 0 | 0 | 0 | 0 | 0 | 0 | 0 | 0 | 0 | 0 | 0 | 0 |
| 9 | 0 | 0 | 1 | 0 | 0 | 0 | 0 | 1 | 0 | 0 | 0 | 0 | 0 |
| + | 0 | 0 | 0 | 0 | 0 | 0 | 0 | 0 | 0 | 0 | 0 | 0 | 0 |
| = | 0 | 0 | 0 | 0 | 0 | 0 | 0 | 0 | 0 | 0 | 0 | 0 | 0 |
| $ | 100 | 100 | 100 | 100 | 100 | 0 | 0 | 0 | 0 | 0 | 0 | 0 | 100 |

Table 27: Example output sequence $\mathcal{O} = \texttt{Dec}\left(\boldsymbol{X}^{(1)}\right)$, continuing from Table 26. The yellow cells in the bottom row exactly predict the next tokens.

| $\mathcal{I}$ | $ | 6 | 5 | 3 | + | 0 | 4 | 9 | = | 2 | 0 | 7 | 0 |
|---|---|---|---|---|---|---|---|---|---|---|---|---|---|
| $\mathcal{O}$ | $ | $ | $ | $ | $ | 1 | 6 | 9 | 2 | 0 | 7 | 0 | $ |

# F   Impossibility of Addition with No Positional Encoding

For the sake of readability, we restate the proposition below.

**Proposition 5.2.** *Consider any depth-1 finite-head decoder-only Transformer model $\mathcal{T}$ without positional encoding (NoPE). Given an input sequence $\mathcal{I}$ and its arbitrary permutation $\mathcal{I}'$, if the last tokens of $\mathcal{I}$ and $\mathcal{I}'$ are identical, then the next tokens predicted by $\mathcal{T}$ will also be identical for both sequences when applying a greedy decoding scheme.*

**Remark.**   We assume the 1-layer ($L = 1$) $H$-head Transformer achitecture specified in Appendix D. Although it omits normalization layers, we remark that Proposition 5.2 remains valid even for the architecture with a standard layer normalization (Ba et al., 2016) or its variants (e.g., Zhang and Sennrich, 2019).

*Proof.*  We keep following the notation about matrices introduced in Appendix E.1. Throughout the proof, we denote the value/vector/matrix related to $\mathcal{I}'$ by appending ‘$'$’ to it.

Let encoding matrices generated from the input sequences $\mathcal{I}, \mathcal{I}' \in \mathcal{V}^N$ as

$$\boldsymbol{X} := \texttt{Enc}(\mathcal{I}) \in \mathbb{R}^{d \times N} \quad \text{and} \quad \boldsymbol{X}' := \texttt{Enc}(\mathcal{I}') \in \mathbb{R}^{d \times N}. \tag{72}$$

Since there is no positional encoding, the encoding function $\texttt{Enc}(\cdot)$ maps the same tokens to the same columns. In particular, $\mathcal{I}_i = \mathcal{I}'_j$ implies $\boldsymbol{X}_{\bullet i} = \boldsymbol{X}'_{\bullet j}$. Since we assume that $\mathcal{I}'$ is a permutation of $\mathcal{I}$ such that $\mathcal{I}_N = \mathcal{I}'_N$, there exists a bijection $\pi : [N] \to [N]$ such that $\mathcal{I}'_i = \mathcal{I}_{\pi(i)}$ for each $i \in [N]$ and $\pi(N) = N$. Then, it follows that $\boldsymbol{X}'_{\bullet i} = \boldsymbol{X}_{\bullet(\pi(i))}$ for each $i$ and, specifically, $\boldsymbol{X}'_{\bullet N} = \boldsymbol{X}_{\bullet N}$.

Recall that the single $H$-head attention layer $\texttt{Att} : \mathbb{R}^{d \times N} \to \mathbb{R}^{d \times N}$ operates as $\texttt{Att}(\boldsymbol{X}) = \sum_{h=1}^{H} \texttt{Head}_h(\boldsymbol{X})$ where the attention head $h$ is defined as

$$\texttt{Head}_h(\boldsymbol{X}) := \boldsymbol{U}_h \boldsymbol{V}_h \boldsymbol{X} \cdot \texttt{softmax}\left((\boldsymbol{K}_h \boldsymbol{X})^\top \boldsymbol{Q}_h \boldsymbol{X}\right) \in \mathbb{R}^{d \times N},$$

where $\boldsymbol{Q}_h, \boldsymbol{K}_h \in \mathbb{R}^{d_{QK} \times d}$, $\boldsymbol{V}_h \in \mathbb{R}^{d_V \times d}$ and $\boldsymbol{U}_h \in \mathbb{R}^{d \times d_V}$.

**Claim:**   $[\texttt{Head}_h(\boldsymbol{X})]_{\bullet N} = [\texttt{Head}_h(\boldsymbol{X}')]_{\bullet N}$ for all $h \in [H]$.

The claim suffices to prove the proposition because of the following: first, the claim implies that the last ($N$-th) columns of the attention layer outputs are the same, i.e., $[\texttt{Att}(\boldsymbol{X})]_{\bullet N} = [\texttt{Att}(\boldsymbol{X}')]_{\bullet N}$. Note that the operations after the attention layer—residual connections, FF, and Dec—all operate in a token-wise (column-by-column) manner: the $j$-th column of the output of a token-wise operation is a function of $j$-th column of the input for the operation. Therefore, the last column of the attention layer output totally determines the next-token prediction at $N$-th input token. As a result, the predicted next-tokens are the same for $\mathcal{I}$ and $\mathcal{I}'$.

The rest of the proof is devoted to proving the aforementioned claim. Fix any $h \in [H]$. Let

$$\left[\texttt{softmax}\left((\boldsymbol{K}_h \boldsymbol{X})^\top \boldsymbol{Q}_h \boldsymbol{X}\right)\right]_{\bullet N} = \begin{bmatrix} s_1 & \dots & s_N \end{bmatrix}^\top, \tag{73}$$

$$\left[\texttt{softmax}\left((\boldsymbol{K}_h \boldsymbol{X}')^\top \boldsymbol{Q}_h \boldsymbol{X}'\right)\right]_{\bullet N} = \begin{bmatrix} s'_1 & \dots & s'_N \end{bmatrix}^\top, \tag{74}$$

which are both stochastic (sum to 1) column vectors. Considering that we are taking the last column of the softmax output, it follows that $s'_i = s_{\pi(i)}$ for each $i \in [N]$: this can be proved by applying the definition of the softmax operation and the fact

$$\left[(\boldsymbol{K}_h \boldsymbol{X}')^\top \boldsymbol{Q}_h \boldsymbol{X}'\right]_{iN} = \boldsymbol{X}'^\top_{\bullet i} \boldsymbol{K}_h^\top \boldsymbol{Q}_h \boldsymbol{X}'_{\bullet N} = \boldsymbol{X}^\top_{\bullet\pi(i)} \boldsymbol{K}_h^\top \boldsymbol{Q}_h \boldsymbol{X}_{\bullet N} = \left[(\boldsymbol{K}_h \boldsymbol{X})^\top \boldsymbol{Q}_h \boldsymbol{X}\right]_{(\pi(i))N}. \tag{75}$$

Consequently, since

$$\sum_{i=1}^{N} s'_i \boldsymbol{X}'_{\bullet i} = \sum_{i=1}^{N} s_{\pi(i)} \boldsymbol{X}_{\bullet(\pi(i))} = \sum_{i=1}^{N} s_i \boldsymbol{X}_{\bullet i}, \tag{76}$$

we have

$$\boldsymbol{X}' \cdot \left[\texttt{softmax}\left((\boldsymbol{K}_h \boldsymbol{X}')^\top \boldsymbol{Q}_h \boldsymbol{X}'\right)\right]_{\bullet N} = \boldsymbol{X} \cdot \left[\texttt{softmax}\left((\boldsymbol{K}_h \boldsymbol{X})^\top \boldsymbol{Q}_h \boldsymbol{X}\right)\right]_{\bullet N}. \tag{77}$$

Therefore, the claim holds. This concludes the proof. $\qquad\square$

Here, we provide the Python code that calculates the maximum possible exact-match accuracy that a 1-layer Transformer with NoPE can achieve for the $m$-digit addition problem.

```python
from itertools import product
from collections import defaultdict

m = 4  # Change here
total = 0
counter_dict = defaultdict(dict)

for a, b in product(product(range(10), repeat=m), product(range(10),
    repeat=m)):
    if a[0] == 0 or b[0] == 0: continue
    total += 1
    c = tuple(sorted(a+b))
    a_num = int(''.join(map(str, a)))
    b_num = int(''.join(map(str, b)))
    ab_sum = a_num + b_num
    if ab_sum in counter_dict[c]:
        counter_dict[c][ab_sum] += 1
    else:
        counter_dict[c][ab_sum] = 1

count = sum(max(d.values()) for _, d in counter_dict.items())

print("m =", m)
print("Permutation Invariant Additions Count:", count)
print("        Total m-digit Additions Count:", total)
print("                             Ratio:", count / total)

"""
[Example Outputs]

m = 1
Permutation Invariant Additions Count: 81
        Total m-digit Additions Count: 81
                             Ratio: 1.0
m = 2
Permutation Invariant Additions Count: 2668
        Total m-digit Additions Count: 8100
                             Ratio: 0.32938271604938274
m = 3
Permutation Invariant Additions Count: 50150
        Total m-digit Additions Count: 810000
                             Ratio: 0.06191358024691358
m = 4
Permutation Invariant Additions Count: 765139
        Total m-digit Additions Count: 81000000
                             Ratio: 0.00944616049382716
m = 5
Permutation Invariant Additions Count: 10033314
        Total m-digit Additions Count: 8100000000
                             Ratio: 0.0012386807407407407
"""
```

# G  (Formal) Construction of $N \times 2$ Multiplication Transformer with Position Coupling

Here we show how to implement the $N \times 2$ multiplication using a depth-2 decoder-only Transformer equipped with position coupling. Our construction involves 3 heads in the first Transformer block and 7 heads in the second Transformer block, requiring a total of 10 heads.

**Theorem 6.1.** *Given an appropriate format of the input sequence, there exists a depth-2 decoder-only Transformer model with coupled positions that can perform the $N \times 2$ multiplication task with next-token prediction. Here, the number of the total heads is 10 and the length of the first operand is at most $2^{\lfloor (d-34)/6 \rfloor} - 3$, where we denote the token embedding dimension by $d \geq 46$.*

We note that our construction for the $N \times 2$ multiplication task permits the use of multiple FFN layers at the second decoder block. However, we believe that there exists a potential improvement in our construction, wherein a single FFN layer could suffice for each decoder block, leveraging the expressivity of the neural network. Additionally, we do not provide a detailed error analysis but assume that the softmax operation with sufficiently large attention weights can reduce small attention scores to zero values, thereby clearly revealing the desired attention patterns.

## G.1  Notation

Consider an ordered vocabulary $\mathcal{V} = (0, 1, 2, 3, 4, 5, 6, 7, 8, 9, \times, =, \$)$. We include a special token '$\$$' that plays the role of both the beginning-of-sequence (BOS) token and the end-of-sequence (EOS) token. We denote $\mathcal{V}_k$ as $k$-th element of $\mathcal{V}$. For instance, $\mathcal{V}_4 = 3$ and $\mathcal{V}_{13} = \$$. Unlike the addition task, our construction for the multiplication involves multiple layers and hence we do not omit the superscripts $(l)$ in the parameter matrices/vectors and the size of dimensions.

## G.2  Input Sequence

Our objective is to use next-token prediction for implementing $a \times b = c$. To this end, we want to transform it into an input sequence $\mathcal{I} = \overline{\$A \times B = C}$ of an appropriate format. Let $\ell_a$ and $\ell_b$ represent the lengths of $a$ and $b$, respectively, and we denote their sum as $\ell = \ell_a + \ell_b$. While our immediate focus is on the case where $\ell_b = 2$, it is worth noting that our approach can be extended to the case where $\ell_b > 2$, as the key insight for the construction does not rely on $\ell_b$. Thus, we present the input sequence and encoding function in a more general form applicable to $\ell_b \geq 2$.

Unlike the addition case, we do not zero-pad both $a$ and $b$. Instead, we only zero-pad the response, as the length of $c$ may either equal the sum of the lengths of $a$ and $b$, or be less than the sum of their lengths by 1. Hence, we zero-pad in front of $c$ for the latter case to fix the length of $c$ by $\ell$. We also reverse the response $c$ to make the part $C$. For instance, if we have $312 \times 24 = 7488$, the input sequence transforms to $\overline{\$312 \times 24 = 88470}$. If we have $589 \times 62 = 36518$, then the input sequence would be $\overline{\$589 \times 62 = 81563}$. The red digit is a zero-padding, and the blue digits are the reversed product.

To recap, the input sequence $\mathcal{I} = \overline{\sigma_1 \sigma_2 \ldots \sigma_N} \in \mathcal{V}^N$ of length $N = 2\ell + 3$ consists of six parts:

1. the BOS token $\sigma_1 = $ '$\$$'
2. the first operand $A = \overline{\sigma_2 \ldots \sigma_{\ell_a+1}}$ where $\sigma_i \in \{0, \ldots, 9\}$;
3. the multiplication symbol $\sigma_{\ell_a+2} = $ '$\times$';
4. the second operand $B = \overline{\sigma_{\ell_a+3} \ldots \sigma_{\ell+2}}$ (note that $\ell = \ell_a + \ell_b$) where $\sigma_i \in \{0, \ldots, 9\}$;
5. the equality symbol $\sigma_{\ell+3} = $ '$=$';
6. the (reversed) product $C = \overline{\sigma_{\ell+4} \ldots \sigma_{2\ell+3}}$ where $\sigma_i \in \{0, \ldots, 9\}$.

Note that the part $C$ might be incomplete (i.e., $N < 2\ell + 3$) at the inference time; we infer the digits of the part $C$ one by one using next-token prediction. Throughout this section on a formal construction, however, we only consider the train time setup in which we infer all the digits of the part $C$ at once using *simultaneous* next-token prediction in a single forward pass. Precisely, we want to use an input sequence $\mathcal{I} = \overline{\sigma_1 \ldots \sigma_N}$ to produce an output sequence $\mathcal{O} = \overline{\sigma'_1 \ldots \sigma'_N}$ where $\overline{\sigma'_{\ell+3} \ldots \sigma'_{N-1}} = C = \overline{\sigma_{\ell+4} \ldots \sigma_N}$ and $\sigma'_N = $ '$\$$' (EOS).

### G.3 Encoding Function

We now explain the input embedding for given an input sequence $\mathcal{I}$ designed as above. The embedding matrix $\boldsymbol{X}^{(0)}$ is of size $d \times N$: each column represents an embedding vector for a token, while each row represents a particular *named* dimension. We concatenate the token embedding and the position embedding, which can be viewed as a *sum* of two different embedding matrices of the same size.

Table 28: Example initial encoding. Here we consider the input sequence $\overline{\$7595 \times 79 = 500006}$ and the starting position ID is chosen as $s = 1$. The vectors $\boldsymbol{v}_\square^P$ are defined in Equation (79). The gray rows will be filled in later.

| $\mathcal{I}$ | $ | 7 | 5 | 9 | 5 | × | 7 | 9 | = | 5 | 0 | 0 | 0 | 0 | 6 |
|---|---|---|---|---|---|---|---|---|---|---|---|---|---|---|---|
| 1: NUM | 0 | 7 | 5 | 9 | 5 | 0 | 7 | 9 | 0 | 5 | 0 | 0 | 0 | 0 | 6 |
| 2: FULL_ONES | 1 | 1 | 1 | 1 | 1 | 1 | 1 | 1 | 1 | 1 | 1 | 1 | 1 | 1 | 1 |
| 3: IS_BOS | 1 | 0 | 0 | 0 | 0 | 0 | 0 | 0 | 0 | 0 | 0 | 0 | 0 | 0 | 0 |
| 4: IS_MUL | 0 | 0 | 0 | 0 | 0 | 1 | 0 | 0 | 0 | 0 | 0 | 0 | 0 | 0 | 0 |
| 5: IS_EQUAL | 0 | 0 | 0 | 0 | 0 | 0 | 0 | 0 | 1 | 0 | 0 | 0 | 0 | 0 | 0 |
| 6: IS_OP2_ONE | 0 | 0 | 0 | 0 | 0 | 0 | 0 | 0 | 0 | 0 | 0 | 0 | 0 | 0 | 0 |
| 7: IS_OP2_TEN | 0 | 0 | 0 | 0 | 0 | 0 | 0 | 0 | 0 | 0 | 0 | 0 | 0 | 0 | 0 |
| 8: OP2_ONE | 0 | 0 | 0 | 0 | 0 | 0 | 0 | 0 | 0 | 0 | 0 | 0 | 0 | 0 | 0 |
| 9: OP2_TEN | 0 | 0 | 0 | 0 | 0 | 0 | 0 | 0 | 0 | 0 | 0 | 0 | 0 | 0 | 0 |
| 10: OP1_SHIFT0 | 0 | 0 | 0 | 0 | 0 | 0 | 0 | 0 | 0 | 0 | 0 | 0 | 0 | 0 | 0 |
| 11: OP1_SHIFT1 | 0 | 0 | 0 | 0 | 0 | 0 | 0 | 0 | 0 | 0 | 0 | 0 | 0 | 0 | 0 |
| 12: OP1_SHIFT2 | 0 | 0 | 0 | 0 | 0 | 0 | 0 | 0 | 0 | 0 | 0 | 0 | 0 | 0 | 0 |
| 13: OP1_SHIFT3 | 0 | 0 | 0 | 0 | 0 | 0 | 0 | 0 | 0 | 0 | 0 | 0 | 0 | 0 | 0 |
| 14: OP1_SHIFT4 | 0 | 0 | 0 | 0 | 0 | 0 | 0 | 0 | 0 | 0 | 0 | 0 | 0 | 0 | 0 |
| 15: RESULT1 | 0 | 0 | 0 | 0 | 0 | 0 | 0 | 0 | 0 | 0 | 0 | 0 | 0 | 0 | 0 |
| 16: RESULT2 | 0 | 0 | 0 | 0 | 0 | 0 | 0 | 0 | 0 | 0 | 0 | 0 | 0 | 0 | 0 |
| 17: RESULT3 | 0 | 0 | 0 | 0 | 0 | 0 | 0 | 0 | 0 | 0 | 0 | 0 | 0 | 0 | 0 |
| 18: RESULT4 | 0 | 0 | 0 | 0 | 0 | 0 | 0 | 0 | 0 | 0 | 0 | 0 | 0 | 0 | 0 |
| 19: PRE_PROD | 0 | 0 | 0 | 0 | 0 | 0 | 0 | 0 | 0 | 0 | 0 | 0 | 0 | 0 | 0 |
| 20: PRE_CARRY | 0 | 0 | 0 | 0 | 0 | 0 | 0 | 0 | 0 | 0 | 0 | 0 | 0 | 0 | 0 |
| 21: PRE_EOS1 | 0 | 0 | 0 | 0 | 0 | 0 | 0 | 0 | 0 | 0 | 0 | 0 | 0 | 0 | 0 |
| 22: PRE_EOS2 | 0 | 0 | 0 | 0 | 0 | 0 | 0 | 0 | 0 | 0 | 0 | 0 | 0 | 0 | 0 |
| 23-32: PROD | $\mathbf{0}_{10}$ | $\mathbf{0}_{10}$ | $\mathbf{0}_{10}$ | $\mathbf{0}_{10}$ | $\mathbf{0}_{10}$ | $\mathbf{0}_{10}$ | $\mathbf{0}_{10}$ | $\mathbf{0}_{10}$ | $\mathbf{0}_{10}$ | $\mathbf{0}_{10}$ | $\mathbf{0}_{10}$ | $\mathbf{0}_{10}$ | $\mathbf{0}_{10}$ | $\mathbf{0}_{10}$ | $\mathbf{0}_{10}$ |
| 33: IS_EOS | 0 | 0 | 0 | 0 | 0 | 0 | 0 | 0 | 0 | 0 | 0 | 0 | 0 | 0 | 0 |
| 34: MASK | 0 | 0 | 0 | 0 | 0 | 0 | 0 | 0 | 0 | 0 | 0 | 0 | 0 | 0 | 0 |
| 35–($P$+34): POS_2_MASK | $\mathbf{0}_P$ | $\mathbf{0}_P$ | $\mathbf{0}_P$ | $\mathbf{0}_P$ | $\mathbf{0}_P$ | $\mathbf{0}_P$ | $\mathbf{0}_P$ | $\mathbf{0}_P$ | $\mathbf{0}_P$ | $\mathbf{0}_P$ | $\mathbf{0}_P$ | $\mathbf{0}_P$ | $\mathbf{0}_P$ | $\mathbf{0}_P$ | $\mathbf{0}_P$ |
| ($P$+35)–($2P$+34): POS_1 | $\mathbf{0}_P$ | $v_3^P$ | $v_4^P$ | $v_5^P$ | $v_6^P$ | $v_7^P$ | $v_5^P$ | $v_6^P$ | $v_7^P$ | $v_6^P$ | $v_5^P$ | $v_4^P$ | $v_3^P$ | $v_2^P$ | $v_1^P$ |
| ($2P$+35)–($3P$+34): POS_2 | $\mathbf{0}_P$ | $v_4^P$ | $v_5^P$ | $v_6^P$ | $v_7^P$ | $v_8^P$ | $v_6^P$ | $v_7^P$ | $v_8^P$ | $v_7^P$ | $v_6^P$ | $v_5^P$ | $v_4^P$ | $v_3^P$ | $v_2^P$ |
| ($3P$+35)–($4P$+34): POS_3 | $\mathbf{0}_P$ | $v_5^P$ | $v_6^P$ | $v_7^P$ | $v_8^P$ | $v_9^P$ | $v_7^P$ | $v_8^P$ | $v_9^P$ | $v_8^P$ | $v_7^P$ | $v_6^P$ | $v_5^P$ | $v_4^P$ | $v_3^P$ |
| ($4P$+35)–($5P$+34): POS_4 | $\mathbf{0}_P$ | $v_6^P$ | $v_7^P$ | $v_8^P$ | $v_9^P$ | $v_{10}^P$ | $v_8^P$ | $v_9^P$ | $v_{10}^P$ | $v_9^P$ | $v_8^P$ | $v_7^P$ | $v_6^P$ | $v_5^P$ | $v_4^P$ |
| ($5P$+35)–($6P$+34): POS_5 | $\mathbf{0}_P$ | $v_7^P$ | $v_8^P$ | $v_9^P$ | $v_{10}^P$ | $v_{11}^P$ | $v_9^P$ | $v_{10}^P$ | $v_{11}^P$ | $v_{10}^P$ | $v_9^P$ | $v_8^P$ | $v_7^P$ | $v_6^P$ | $v_5^P$ |

### G.3.1 Token Embedding

The token embedding consists of $(34 + P)$ dimensions, where $P$ represents the dimension for the position embedding which will be described in the very next section. While the token embedding dimension for the addition task was independent of $P$, our construction strategy for the multiplication task involves copying the position embedding into the token embedding. This is why we have the $P$ term in our token embedding dimension. For the first $34$ dimensions, we label them as:

$$1=\text{NUM}, 2=\text{FULL\_ONES}, 3=\text{IS\_BOS}, 4=\text{IS\_MUL}, 5=\text{IS\_EQUAL},$$

$$6=\text{IS\_OP2\_ONE}, 7=\text{IS\_OP2\_TEN}, 8=\text{OP2\_ONE}, 9=\text{OP2\_TEN},$$

$$10=\text{OP1\_SHIFT0}, 11=\text{OP1\_SHIFT1}, 12=\text{OP1\_SHIFT2}, 13=\text{OP1\_SHIFT3}, 14=\text{OP1\_SHIFT4},$$

$$15=\text{RESULT1}, 16=\text{RESULT2}, 17=\text{RESULT3}, 18=\text{RESULT4},$$

$$19=\text{PRE\_PROD}, 20=\text{PRE\_CARRY}, 21=\text{PRE\_EOS1}, 22=\text{PRE\_EOS2}$$

$$\{23,...,32\}=\text{PROD}, 33=\text{IS\_EOS}, 34=\text{MASK},$$

and for the last $P$ dimensions ($\{35, ..., 34 + P\}$), we named them as POS_2_MASK.

The initial token embedding fills only NUM, FULL_ONES, IS_BOS, IS_MUL, and IS_EQUAL, leaving the other $(29 + P)$ dimensions empty (i.e., all zeros). These $(29 + P)$ dimensions will be filled by passing through the layers. Here we describe how we fill the first 5 dimensions.

**Dimension 1 (NUM).**   For a number token $(0, \ldots, 9)$, we put itself into the dimension NUM. For the other tokens $(\times, =, \$)$, we put 0.

**Dimension 2 (FULL_ONES).**   We put 1 everywhere in this dimension.

**Dimension 3 (IS_BOS).**   For a special token '$\$$', we put 1 into the dimension IS_BOS. Otherwise, we put 0.

**Dimension 4 (IS_MUL).**   For a special token '$\times$', we put 1 into the dimension IS_MUL. Otherwise, we put 0.

**Dimension 5 (IS_EQUAL).**   For a special token '$=$', we put 1 into the dimension IS_EQUAL. Otherwise, we put 0.

### G.3.2   Coupled Position IDs and Position Embedding

We now specify the allocation of coupled position IDs for the $N \times M$ multiplication task as the following: given an input sequence $\mathcal{I} = \overline{\sigma_1 \ldots \sigma_N}$,

$$
p(i) = \begin{cases} 0, & i = 1, \\ s + i - 2 + \ell_b, & i = 2, \ldots, \ell_a + 2, \\ s + i - 3, & i = \ell_a + 3, \ldots, \ell + 3, \\ s - i + 3 + 2\ell & i = \ell + 4, \ldots, 2\ell + 3. \end{cases} \tag{78}
$$

Compared to the addition case, the position allocating function $p$ becomes more complicated since the length of two operands can be different, but the core remains simple: coupling the position IDs for the least significant digit in the first operand $(A)$, the second operand $(B)$, and the result $(C)$, and then decreasing the IDs as the digit position increases for each $A$, $B$, and $C$.

Now we explain the position embedding. We utilize the same $\boldsymbol{v}_k^D$ ($k \in [2^D]$) defined for the addition task, specifically

$$
\boldsymbol{v}_k^D = \left[ (-1)^{b_i^{(D,k)}} \right]_{i=1}^D \in \mathbb{R}^D \tag{79}
$$

where $b_i^{(D,k)}$ is defined as the $i$-th (from left) digit of $D$-digit binary representation of $k - 1$. Using $\boldsymbol{v}_k^D$, we design the position embedding for each position ID $p(i)$ by

$$
\begin{bmatrix} \boldsymbol{v}_{p(i)}^P \\ \boldsymbol{v}_{p(i)+1}^P \\ \boldsymbol{v}_{p(i)+2}^P \\ \boldsymbol{v}_{p(i)+3}^P \\ \boldsymbol{v}_{p(i)+4}^P \end{bmatrix}. \tag{80}
$$

The first $P$ dimensions of the position embedding are named as POS_1, and subsequent sets of $P$ dimensions are named as POS_2, POS_3, POS_4, and POS_5, respectively. Thus, the position embedding is a $5P$-dimensional vector. In case of $p(i) + j$ ($j \in [4]$) exceeding $2^P$, we use $\boldsymbol{v}_{p(i)+j-2^P}^P$ instead of $\boldsymbol{v}_{p(i)+j}^P$. If $p(i) = 0$, we let $\boldsymbol{0}_{5P}$ as a position embedding vector.

By concatenating the token embedding and the position embedding, we get the input embedding $\boldsymbol{X}^{(0)}$. Specifically, the position embedding is placed under the token embedding $((P + 35)$-th to $(6P + 34)$-th dimension). See Table 9 for an example. As a result, the total embedding dimension is $d = 6P + 34$. Note the maximum possible position ID that can be represented with $\boldsymbol{v}_k^P$'s is $\texttt{max\_pos} = 2^P = 2^{\lfloor (d-34)/6 \rfloor}$. Therefore, the length of the first operand must be $\ell_a \leq \texttt{max\_pos} - \ell_b - 1 = 2^{\lfloor (d-34)/6 \rfloor} - \ell_b - 1$. For the case when $\ell_b = 2$, this inequality becomes $\ell_a \leq 2^{\lfloor (d-34)/6 \rfloor} - 3$.

### G.4 Construction Idea

Here, we provide an example that demonstrates how we construct the $N \times 2$ multiplication. Consider the calculation $7595 \times 79 = 600005$. While a typical method for computing such a multiplication is illustrated in Table 29, we consider an alternative approach, as shown in Table 30. In this method, we pair the digits from the first and second operands at each step where the sum of their digit positions is the same, and then calculate the sum of the pairwise products. For example, the number 116 in Table 30 is generated by $9 \times 9 + 5 \times 7$, and the number 108 is generated by $5 \times 9 + 9 \times 7$, where blue indicates numbers from the first operand and red indicates numbers from the second operand. The main reason for considering such a method is to provide a clearer intuition for determining which numbers from each operand we should attend to when predicting the next token.

Table 29: Multiplication I

|   |   | 7 | 5 | 9 | 5 |
|---|---|---|---|---|---|
| × |   |   |   | 7 | 9 |
|   | 6 | 8 | 3 | 5 | 5 |
| 5 | 3 | 1 | 6 | 5 |   |
| 6 | 0 | 0 | 0 | 0 | 5 |

Table 30: Multiplication II

|   |   | 7 | 5 | 9 | 5 |
|---|---|---|---|---|---|
| × |   |   |   | 7 | 9 |
|   |   |   | 4 | 5 |   |
|   |   | 1 | 1 | 6 |   |
|   | 1 | 0 | 8 |   |   |
|   | 9 | 8 |   |   |   |
| 4 | 9 |   |   |   |   |
| 6 | 0 | 0 | 0 | 0 | 5 |

Suppose the current input sequence is $\overline{\$7595 \times 79 = 5000}$. During this step, the model is tasked with predicting 0 (the 0 just before 6) for the next token. As illustrated in Table 30, this 0 is computed from the sum of 9, 9, 1, and an additional 1, representing the carry from the previous step. Similar to the explanation in E.4.2, we highlight that the carry 1 can be detected by computing $8$ (ones digit of 98) $+ 0$ (tens digit of 108) $+ 1$ (hundreds digit of 116) $- 0$ (current token): yielding a result of 9, indicating the occurrence of a carry 1.

In summary, the correct prediction of the next token 0 (the 0 just before 6) can be achieved by summing the main summation part and the carry part, where the main summation part is computed using 49, 98, 108, and the carry part is calculated using 98, 108, and 116. Additionally, it's noteworthy to detail the breakdowns:

- $49 = 0 \times 9 + 7 \times 7$,
- $98 = 7 \times 9 + 5 + 7$,
- $108 = 5 \times 9 + 9 + 7$,
- $116 = 9 \times 9 + 5 + 7$.

Thus, for predicting the next token, we need $0, 7, 5, 9, 5, 9, 7$. Here, we highlight that this structure, requiring 5 consecutive tokens from the first operand and every token from the second operand for the next-token prediction, remains unchanged for any prediction time and any query length.

As we will see in the later subsection, a depth-2 decoder-only Transformer model can be constructed to fill OP2_ONE by 9, OP2_TEN by 7, and OP1_SHIFT0 to OP1_SHIFT4 by 0, 7, 5, 9, and 5, respectively. One may be concerned that 0 is not given in the first operand at the input sequence. This requirement of 0 beyond the most significant digit arises in the later stage of the prediction, i.e., predicting the token that is near the most significant digit of the response. Although 0 is not explicitly given in the first operand, our construction can automatically manage as if the 0 were originally at the start of the first operand. A similar situation occurs in the early stage of the prediction that 0 is required before the least significant digit of the first operand, and our construction is also capable of handling this issue.

Consequently, the embedding vector of the current token 0 (the 0 preceding 60) will be structured as the left-most table in Table 31, with some irrelevant dimensions omitted for readability. We then utilize a feed-forward layer to fill

- RESULT1 with OP1_SHIFT0 $\times$ OP2_ONE $+$ OP1_SHIFT1 $\times$ OP2_TEN,

- RESULT2 with OP1_SHIFT1 × OP2_ONE + OP1_SHIFT2 × OP2_TEN,

- RESULT3 with OP1_SHIFT2 × OP2_ONE + OP1_SHIFT3 × OP2_TEN,

- RESULT4 with OP1_SHIFT3 × OP2_ONE + OP1_SHIFT4 × OP2_TEN.

The result is illustrated in the center table of Table 31. Next, we employ an additional feed-forward layer to fill

- PRE_PROD with ones digit of RESULT1 + tens digit of RESULT2 + hundreds digit of RESULT3,

- PRE_CARRY with ones digit of RESULT2 + tens digit of RESULT3 + hundreds digit of RESULT4.

These computations yield the result illustrated in the right-most table of Table 31. Once this process is done, we can finally predict the next token by the following two steps:

- $\text{CARRY} = \begin{cases} 0, & \text{if PRE\_CARRY} - \text{NUM} \in \{-2, -1, 0\}, \\ 1, & \text{if PRE\_CARRY} - \text{NUM} \in \{8, 9, 10\}, \\ 2, & \text{if PRE\_CARRY} - \text{NUM} \in \{18, 19, 20\}, \end{cases}$

- $\text{NEXT\_TOKEN} = \text{PRE\_PROD} + \text{CARRY} \pmod{10}$.

Table 31: Illustration of the construction idea.

| $\mathcal{I}$ | 0 | | $\mathcal{I}$ | 0 | | $\mathcal{I}$ | 0 |
|---|---|---|---|---|---|---|---|
| 1: NUM | 0 | | 1: NUM | 0 | | 1: NUM | 0 |
| 2: FULL_ONES | 1 | | 2: FULL_ONES | 1 | | 2: FULL_ONES | 1 |
| 3: IS_BOS | 0 | | 3: IS_BOS | 0 | | 3: IS_BOS | 0 |
| 4: IS_MUL | 0 | | 4: IS_MUL | 0 | | 4: IS_MUL | 0 |
| 5: IS_EQUAL | 0 | | 5: IS_EQUAL | 0 | | 5: IS_EQUAL | 0 |
| 8: OP2_ONE | 9 | | 8: OP2_ONE | 9 | | 8: OP2_ONE | 9 |
| 9: OP2_TEN | 7 | | 9: OP2_TEN | 7 | | 9: OP2_TEN | 7 |
| 10: OP1_SHIFT0 | 0 | | 10: OP1_SHIFT0 | 0 | | 10: OP1_SHIFT0 | 0 |
| 11: OP1_SHIFT1 | 7 | | 11: OP1_SHIFT1 | 7 | | 11: OP1_SHIFT1 | 7 |
| 12: OP1_SHIFT2 | 5 | | 12: OP1_SHIFT2 | 5 | | 12: OP1_SHIFT2 | 5 |
| 13: OP1_SHIFT3 | 9 | $\rightarrow$ | 13: OP1_SHIFT3 | 9 | $\rightarrow$ | 13: OP1_SHIFT3 | 9 |
| 14: OP1_SHIFT4 | 5 | | 14: OP1_SHIFT4 | 5 | | 14: OP1_SHIFT4 | 5 |
| 15: RESULT1 | 0 | | 15: RESULT1 | 49 | | 15: RESULT1 | 49 |
| 16: RESULT2 | 0 | | 16: RESULT2 | 98 | | 16: RESULT2 | 98 |
| 17: RESULT3 | 0 | | 17: RESULT3 | 108 | | 17: RESULT3 | 108 |
| 18: RESULT4 | 0 | | 18: RESULT4 | 116 | | 18: RESULT4 | 116 |
| 19: PRE_PROD | 0 | | 19: PRE_PROD | 0 | | 19: PRE_PROD | 19 |
| 20: PRE_CARRY | 0 | | 20: PRE_CARRY | 0 | | 20: PRE_CARRY | 9 |
| $(P{+}35)$–$(2P{+}34)$: POS_1 | $\boldsymbol{v}_3^P$ | | $(P{+}35)$–$(2P{+}34)$: POS_1 | $\boldsymbol{v}_3^P$ | | $(P{+}35)$–$(2P{+}34)$: POS_1 | $\boldsymbol{v}_3^P$ |
| $(2P{+}35)$–$(3P{+}34)$: POS_2 | $\boldsymbol{v}_4^P$ | | $(2P{+}35)$–$(3P{+}34)$: POS_2 | $\boldsymbol{v}_4^P$ | | $(2P{+}35)$–$(3P{+}34)$: POS_2 | $\boldsymbol{v}_4^P$ |
| $(3P{+}35)$–$(4P{+}34)$: POS_3 | $\boldsymbol{v}_5^P$ | | $(3P{+}35)$–$(4P{+}34)$: POS_3 | $\boldsymbol{v}_5^P$ | | $(3P{+}35)$–$(4P{+}34)$: POS_3 | $\boldsymbol{v}_5^P$ |
| $(4P{+}35)$–$(5P{+}34)$: POS_4 | $\boldsymbol{v}_6^P$ | | $(4P{+}35)$–$(5P{+}34)$: POS_4 | $\boldsymbol{v}_6^P$ | | $(4P{+}35)$–$(5P{+}34)$: POS_4 | $\boldsymbol{v}_6^P$ |
| $(5P{+}35)$–$(6P{+}34)$: POS_5 | $\boldsymbol{v}_7^P$ | | $(5P{+}35)$–$(6P{+}34)$: POS_5 | $\boldsymbol{v}_7^P$ | | $(5P{+}35)$–$(6P{+}34)$: POS_5 | $\boldsymbol{v}_7^P$ |

## G.5 Transformer Block 1 — Causal Attention Layer

To implement the concept introduced in Appendix G.4, it is essential to design a Transformer block capable of generating an embedding matrix depicted in the left-most table of Table 31. The goal of the first Transformer block is to fill IS_OP2_ONE (6-th dimension) and IS_OP2_TEN (7-th dimension) by 1 if the token corresponds to the ones or tens digit of the second operand, respectively, and 0 otherwise. These two dimensions enable the filling of OP2_ONE (8-th dimension) and OP2_TEN (9-th dimension) at the second Transformer block. Furthermore, we will fill MASK (34-th dimension) in the first block, which will serve as a base for filling OP1_SHIFT0 to OP1_SHIFT4 in the second block. Thus, we currently have 3 objectives(IS_OP2_ONE, IS_OP2_TEN, MASK), each of which will be addressed by an individual head.

### G.5.1 Attention Head 1: Detecting the Ones Digit of the Second Operand

The goal of the first head is to make the dimension IS_OP2_ONE as a one-hot row vector, where $1$ is placed only at the token corresponding to the ones digit of the second operand.

Recall that $d = 6P + 34$ and let $d_{QK,11} = P + 1$. Let $M > 0$ be a sufficiently large positive real number. Let

$$\boldsymbol{Q}_1^{(1)} = \begin{pmatrix} \mathbf{0}_{P \times (P+34)} & \mathbf{0}_{P \times P} & \sqrt{M} \boldsymbol{I}_P & \mathbf{0}_{P \times P} & \mathbf{0}_{P \times P} & \mathbf{0}_{P \times P} \\ \sqrt{MP} \left( \boldsymbol{e}_{\text{FULL\_ONES}}^{P+34} \right)^\top & \mathbf{0}_{1 \times P} & \mathbf{0}_{1 \times P} & \mathbf{0}_{1 \times P} & \mathbf{0}_{1 \times P} & \mathbf{0}_{1 \times P} \end{pmatrix} \in \mathbb{R}^{d_{QK,11} \times d}, \tag{81}$$

$$\boldsymbol{K}_1^{(1)} = \begin{pmatrix} \mathbf{0}_{P \times (P+34)} & \sqrt{M} \boldsymbol{I}_P & \mathbf{0}_{P \times P} & \mathbf{0}_{P \times P} & \mathbf{0}_{P \times P} & \mathbf{0}_{P \times P} \\ \sqrt{MP} \left( \boldsymbol{e}_{\text{IS\_BOS}}^{P+34} \right)^\top & \mathbf{0}_{1 \times P} & \mathbf{0}_{1 \times P} & \mathbf{0}_{1 \times P} & \mathbf{0}_{1 \times P} & \mathbf{0}_{1 \times P} \end{pmatrix} \in \mathbb{R}^{d_{QK,11} \times d}. \tag{82}$$

Unlike the construction for the addition task, we do not provide the table for the exact matrix and detailed error analysis due to their complex characterization. Instead, we provide an illustrative example for each step. We will also simply regard $M$ as a sufficiently large real scalar and thus the attention values can be clearly separated after going through the softmax operation.

The matrix $\boldsymbol{Q}_1^{(1)}$ maps the embedding matrix $\boldsymbol{X}^{(0)}$ into a query matrix $\boldsymbol{Q}_1^{(1)} \boldsymbol{X}^{(0)} \in \mathbb{R}^{(P+1) \times N}$, where the first $P$ rows are obtained by copying from the dimensions POS_2 and scaling by $\sqrt{M}$, while the last row is the copy of the dimension FULL_ONES scaled by $\sqrt{MP}$. Similarly, the matrix $\boldsymbol{K}_1^{(1)}$ maps the embedding matrix to a key matrix $\boldsymbol{K}_1^{(1)} \boldsymbol{X}^{(0)} \in \mathbb{R}^{(P+1) \times N}$. In this case, the first $P$ rows are obtained by copying from the dimensions POS_1 and scaled by $\sqrt{M}$, with the last row being the dimension IS_BOS, scaled by $\sqrt{MP}$. For concrete examples, refer to Tables 32 and 33.

Table 32: Example of $\boldsymbol{Q}_1^{(1)} \boldsymbol{X}^{(0)}$, continuing from Table 28.

| $\mathcal{I}$ | \$ | 7 | 5 | 9 | 5 | $\times$ | 7 | 9 | = | 5 | 0 | 0 | 0 | 0 | 6 |
|---|---|---|---|---|---|---|---|---|---|---|---|---|---|---|---|
| $1$–$P$: | $\mathbf{0}_P$ | $\sqrt{M}\boldsymbol{v}_4^P$ | $\sqrt{M}\boldsymbol{v}_5^P$ | $\sqrt{M}\boldsymbol{v}_6^P$ | $\sqrt{M}\boldsymbol{v}_7^P$ | $\sqrt{M}\boldsymbol{v}_8^P$ | $\sqrt{M}\boldsymbol{v}_6^P$ | $\sqrt{M}\boldsymbol{v}_7^P$ | $\sqrt{M}\boldsymbol{v}_8^P$ | $\sqrt{M}\boldsymbol{v}_7^P$ | $\sqrt{M}\boldsymbol{v}_6^P$ | $\sqrt{M}\boldsymbol{v}_5^P$ | $\sqrt{M}\boldsymbol{v}_4^P$ | $\sqrt{M}\boldsymbol{v}_3^P$ | $\sqrt{M}\boldsymbol{v}_2^P$ |
| $P+1$: | $\sqrt{MP}$ | $\sqrt{MP}$ | $\sqrt{MP}$ | $\sqrt{MP}$ | $\sqrt{MP}$ | $\sqrt{MP}$ | $\sqrt{MP}$ | $\sqrt{MP}$ | $\sqrt{MP}$ | $\sqrt{MP}$ | $\sqrt{MP}$ | $\sqrt{MP}$ | $\sqrt{MP}$ | $\sqrt{MP}$ | $\sqrt{MP}$ |

Table 33: Example of $\boldsymbol{K}_1^{(1)} \boldsymbol{X}^{(0)}$, continuing from Table 28.

| $\mathcal{I}$ | \$ | 7 | 5 | 9 | 5 | $\times$ | 7 | 9 | = | 5 | 0 | 0 | 0 | 0 | 6 |
|---|---|---|---|---|---|---|---|---|---|---|---|---|---|---|---|
| $1$–$P$: | $\mathbf{0}_P$ | $\sqrt{M}\boldsymbol{v}_3^P$ | $\sqrt{M}\boldsymbol{v}_4^P$ | $\sqrt{M}\boldsymbol{v}_5^P$ | $\sqrt{M}\boldsymbol{v}_6^P$ | $\sqrt{M}\boldsymbol{v}_7^P$ | $\sqrt{M}\boldsymbol{v}_5^P$ | $\sqrt{M}\boldsymbol{v}_6^P$ | $\sqrt{M}\boldsymbol{v}_7^P$ | $\sqrt{M}\boldsymbol{v}_6^P$ | $\sqrt{M}\boldsymbol{v}_5^P$ | $\sqrt{M}\boldsymbol{v}_4^P$ | $\sqrt{M}\boldsymbol{v}_3^P$ | $\sqrt{M}\boldsymbol{v}_2^P$ | $\sqrt{M}\boldsymbol{v}_1^P$ |
| $P+1$: | $\sqrt{MP}$ | 0 | 0 | 0 | 0 | 0 | 0 | 0 | 0 | 0 | 0 | 0 | 0 | 0 | 0 |

By these, the attention score matrix $\boldsymbol{C}_1^{(1)} := (\boldsymbol{K}_1^{(1)} \boldsymbol{X}^{(0)})^\top \boldsymbol{Q}_1^{(1)} \boldsymbol{X}^{(0)}$ and the attention matrix $\boldsymbol{A}_1^{(1)} := \mathtt{softmax}(\boldsymbol{C}_1^{(1)}) \in \mathbb{R}^{N \times N}$ can be obtained. We provide the example of $\boldsymbol{A}_1^{(1)}$ in Table 34. Specifically, an entry in $\boldsymbol{A}_1^{(1)}$ is non-zero if and only if the inner product between the query and key vectors equals $MP$.

Now let $d_{V,11} = 1$ and define

$$\boldsymbol{V}_1^{(1)} = 2 (\boldsymbol{e}_{\text{IS\_MUL}}^d - \boldsymbol{e}_{\text{IS\_EQUAL}}^d)^\top \in \mathbb{R}^{d_{V,11} \times d}, \tag{83}$$

$$\boldsymbol{U}_1^{(1)} = \boldsymbol{e}_{\text{IS\_OP2\_ONE}}^d \in \mathbb{R}^{d \times d_{V,11}}. \tag{84}$$

The matrix $\boldsymbol{U}_1^{(1)} \boldsymbol{V}_1^{(1)} \boldsymbol{X}^{(0)}$ takes the dimension IS_MUL and IS_EQUAL from the embedding matrix $\boldsymbol{X}^{(0)}$, subtracts one from the other, scales the result by 2, and puts it to the dimension IS_OP2_SUM. Consequently, the matrix $\boldsymbol{U}_1^{(1)} \boldsymbol{V}_1^{(1)} \boldsymbol{X}^{(0)} \boldsymbol{A}_1^{(1)}$ is a matrix that matches the size of the input embedding matrix $\boldsymbol{X}^{(0)}$ and is filled with zeroes, except for a unique 1 located at the ones place of the second operand in the input sequence, in the dimension IS_OP2_ONE (6-th). A concrete example is provided in Tables 35 and 36.

Table 34: Example of $\boldsymbol{A}_1^{(1)}$ (with explicit row/column indices and sufficiently large $M$), continuing from Tables 32 and 33.

| row \ col | $j=1$ | 2 | 3 | 4 | 5 | 6 | 7 | 8 | 9 | 10 | 11 | 12 | 13 | 14 | 15 |
|---|---|---|---|---|---|---|---|---|---|---|---|---|---|---|---|
| $i=1$ | 1 | 1 | 1 | 1 | 1 | 1 | 1/2 | 1/2 | 1 | 1/3 | 1/4 | 1/4 | 1/3 | 1/3 | 1/2 |
| 2 | 0 | 0 | 0 | 0 | 0 | 0 | 0 | 0 | 0 | 0 | 0 | 0 | 0 | 1/3 | 0 |
| 3 | 0 | 0 | 0 | 0 | 0 | 0 | 0 | 0 | 0 | 0 | 0 | 0 | 1/3 | 0 | 0 |
| 4 | 0 | 0 | 0 | 0 | 0 | 0 | 0 | 0 | 0 | 0 | 0 | 1/4 | 0 | 0 | 0 |
| 5 | 0 | 0 | 0 | 0 | 0 | 0 | 1/2 | 0 | 0 | 0 | 1/4 | 0 | 0 | 0 | 0 |
| 6 | 0 | 0 | 0 | 0 | 0 | 0 | 0 | 1/2 | 0 | 1/3 | 0 | 0 | 0 | 0 | 0 |
| 7 | 0 | 0 | 0 | 0 | 0 | 0 | 0 | 0 | 0 | 0 | 0 | 1/4 | 0 | 0 | 0 |
| 8 | 0 | 0 | 0 | 0 | 0 | 0 | 0 | 0 | 0 | 0 | 1/4 | 0 | 0 | 0 | 0 |
| 9 | 0 | 0 | 0 | 0 | 0 | 0 | 0 | 0 | 0 | 1/3 | 0 | 0 | 0 | 0 | 0 |
| 10 | 0 | 0 | 0 | 0 | 0 | 0 | 0 | 0 | 0 | 0 | 1/4 | 0 | 0 | 0 | 0 |
| 11 | 0 | 0 | 0 | 0 | 0 | 0 | 0 | 0 | 0 | 0 | 0 | 1/4 | 0 | 0 | 0 |
| 12 | 0 | 0 | 0 | 0 | 0 | 0 | 0 | 0 | 0 | 0 | 0 | 0 | 1/3 | 0 | 0 |
| 13 | 0 | 0 | 0 | 0 | 0 | 0 | 0 | 0 | 0 | 0 | 0 | 0 | 0 | 1/3 | 0 |
| 14 | 0 | 0 | 0 | 0 | 0 | 0 | 0 | 0 | 0 | 0 | 0 | 0 | 0 | 0 | 1/2 |
| 15 | 0 | 0 | 0 | 0 | 0 | 0 | 0 | 0 | 0 | 0 | 0 | 0 | 0 | 0 | 0 |

Table 35: Example of $\boldsymbol{U}_1^{(1)}\boldsymbol{V}_1^{(1)}\boldsymbol{X}^{(0)}$, continuing from Table 28. (Irrelevant dimensions are omitted for readability)

| $\mathcal{I}$ | \$ | 7 | 5 | 9 | 5 | × | 7 | 9 | = | 5 | 0 | 0 | 0 | 0 | 6 |
|---|---|---|---|---|---|---|---|---|---|---|---|---|---|---|---|
| 1: NUM | 0 | 0 | 0 | 0 | 0 | 0 | 0 | 0 | 0 | 0 | 0 | 0 | 0 | 0 | 0 |
| 2: FULL_ONES | 0 | 0 | 0 | 0 | 0 | 0 | 0 | 0 | 0 | 0 | 0 | 0 | 0 | 0 | 0 |
| 3: IS_BOS | 0 | 0 | 0 | 0 | 0 | 0 | 0 | 0 | 0 | 0 | 0 | 0 | 0 | 0 | 0 |
| 4: IS_MUL | 0 | 0 | 0 | 0 | 0 | 0 | 0 | 0 | 0 | 0 | 0 | 0 | 0 | 0 | 0 |
| 5: IS_EQUAL | 0 | 0 | 0 | 0 | 0 | 0 | 0 | 0 | 0 | 0 | 0 | 0 | 0 | 0 | 0 |
| 6: IS_OP2_ONE | 0 | 0 | 0 | 0 | 0 | 2 | 0 | 0 | -2 | 0 | 0 | 0 | 0 | 0 | 0 |

Table 36: Example of $\boldsymbol{U}_1^{(1)}\boldsymbol{V}_1^{(1)}\boldsymbol{X}^{(0)}\boldsymbol{A}_1^{(1)}$, continuing from Tables 34 and 35. (Irrelevant dimensions are omitted for readability)

| $\mathcal{I}$ | \$ | 7 | 5 | 9 | 5 | × | 7 | 9 | = | 5 | 0 | 0 | 0 | 0 | 6 |
|---|---|---|---|---|---|---|---|---|---|---|---|---|---|---|---|
| 1: NUM | 0 | 0 | 0 | 0 | 0 | 0 | 0 | 0 | 0 | 0 | 0 | 0 | 0 | 0 | 0 |
| 2: FULL_ONES | 0 | 0 | 0 | 0 | 0 | 0 | 0 | 0 | 0 | 0 | 0 | 0 | 0 | 0 | 0 |
| 3: IS_BOS | 0 | 0 | 0 | 0 | 0 | 0 | 0 | 0 | 0 | 0 | 0 | 0 | 0 | 0 | 0 |
| 4: IS_MUL | 0 | 0 | 0 | 0 | 0 | 0 | 0 | 0 | 0 | 0 | 0 | 0 | 0 | 0 | 0 |
| 5: IS_EQUAL | 0 | 0 | 0 | 0 | 0 | 0 | 0 | 0 | 0 | 0 | 0 | 0 | 0 | 0 | 0 |
| 6: IS_OP2_ONE | 0 | 0 | 0 | 0 | 0 | 0 | 0 | 1 | 0 | 0 | 0 | 0 | 0 | 0 | 0 |

### G.5.2 Attention Head 2: Detecting the Tens Digit of the Second Operand

In the previous head, we set the dimension IS_OP2_ONE (6-th dimension) to a one-hot row vector, where 1 is placed only in the token corresponding to the ones digit of the second operand. The objective of Attention head 2 is to fill the dimension IS_OP2_TEN (7-th dimension) similarly to IS_OP2_ONE, but with 1 placed only in the tens digit of the second operand.

The design of the query, key, and value weight is not significantly different from the previous head. Compared to the construction of attention head 1, we only push $\sqrt{\overline{M}}\boldsymbol{I}_P$ to the next block for designing

$\boldsymbol{Q}_2^{(1)}$. Specifically, $\boldsymbol{Q}_2^{(1)}$ and $\boldsymbol{K}_2^{(1)}$ are defined as

$$\boldsymbol{Q}_2^{(1)} = \begin{pmatrix} \mathbf{0}_{P \times (P+34)} & \mathbf{0}_{P \times P} & \mathbf{0}_{P \times P} & \sqrt{M}\boldsymbol{I}_P & \mathbf{0}_{P \times P} & \mathbf{0}_{P \times P} \\ \sqrt{MP}\left(\boldsymbol{e}_{\text{FULL\_ONES}}^{P+34}\right)^\top & \mathbf{0}_{1 \times P} & \mathbf{0}_{1 \times P} & \mathbf{0}_{1 \times P} & \mathbf{0}_{1 \times P} & \mathbf{0}_{1 \times P} \end{pmatrix} \in \mathbb{R}^{d_{QK,12} \times d},$$

(85)

$$\boldsymbol{K}_2^{(1)} = \begin{pmatrix} \mathbf{0}_{P \times (P+34)} & \sqrt{M}\boldsymbol{I}_P & \mathbf{0}_{P \times P} & \mathbf{0}_{P \times P} & \mathbf{0}_{P \times P} & \mathbf{0}_{P \times P} \\ \sqrt{MP}\left(\boldsymbol{e}_{\text{IS\_BOS}}^{P+34}\right)^\top & \mathbf{0}_{1 \times P} & \mathbf{0}_{1 \times P} & \mathbf{0}_{1 \times P} & \mathbf{0}_{1 \times P} & \mathbf{0}_{1 \times P} \end{pmatrix} \in \mathbb{R}^{d_{QK,12} \times d},$$

(86)

where $d_{QK,12}$ is set to $P + 1$. We refer to Tables 37 and 38 for specific examples.

Table 37: Example of $\boldsymbol{Q}_2^{(1)} \boldsymbol{X}^{(0)}$, continuing from Table 28.

| $\mathcal{I}$ | \$ | 7 | 5 | 9 | 5 | $\times$ | 7 | 9 | = | 5 | 0 | 0 | 0 | 0 | 6 |
|---|---|---|---|---|---|---|---|---|---|---|---|---|---|---|---|
| $1-P$: | $\mathbf{0}_P$ | $\sqrt{M}\boldsymbol{v}_5^P$ | $\sqrt{M}\boldsymbol{v}_6^P$ | $\sqrt{M}\boldsymbol{v}_7^P$ | $\sqrt{M}\boldsymbol{v}_8^P$ | $\sqrt{M}\boldsymbol{v}_9^P$ | $\sqrt{M}\boldsymbol{v}_7^P$ | $\sqrt{M}\boldsymbol{v}_8^P$ | $\sqrt{M}\boldsymbol{v}_9^P$ | $\sqrt{M}\boldsymbol{v}_8^P$ | $\sqrt{M}\boldsymbol{v}_7^P$ | $\sqrt{M}\boldsymbol{v}_6^P$ | $\sqrt{M}\boldsymbol{v}_5^P$ | $\sqrt{M}\boldsymbol{v}_4^P$ | $\sqrt{M}\boldsymbol{v}_3^P$ |
| $P+1$: | $\sqrt{MP}$ | $\sqrt{MP}$ | $\sqrt{MP}$ | $\sqrt{MP}$ | $\sqrt{MP}$ | $\sqrt{MP}$ | $\sqrt{MP}$ | $\sqrt{MP}$ | $\sqrt{MP}$ | $\sqrt{MP}$ | $\sqrt{MP}$ | $\sqrt{MP}$ | $\sqrt{MP}$ | $\sqrt{MP}$ | $\sqrt{MP}$ |

Table 38: Example of $\boldsymbol{K}_2^{(1)} \boldsymbol{X}^{(0)}$, continuing from Table 28.

| $\mathcal{I}$ | \$ | 7 | 5 | 9 | 5 | $\times$ | 7 | 9 | = | 5 | 0 | 0 | 0 | 0 | 6 |
|---|---|---|---|---|---|---|---|---|---|---|---|---|---|---|---|
| $1-P$: | $\mathbf{0}_P$ | $\sqrt{M}\boldsymbol{v}_3^P$ | $\sqrt{M}\boldsymbol{v}_4^P$ | $\sqrt{M}\boldsymbol{v}_5^P$ | $\sqrt{M}\boldsymbol{v}_6^P$ | $\sqrt{M}\boldsymbol{v}_7^P$ | $\sqrt{M}\boldsymbol{v}_5^P$ | $\sqrt{M}\boldsymbol{v}_6^P$ | $\sqrt{M}\boldsymbol{v}_7^P$ | $\sqrt{M}\boldsymbol{v}_6^P$ | $\sqrt{M}\boldsymbol{v}_5^P$ | $\sqrt{M}\boldsymbol{v}_4^P$ | $\sqrt{M}\boldsymbol{v}_3^P$ | $\sqrt{M}\boldsymbol{v}_2^P$ | $\sqrt{M}\boldsymbol{v}_1^P$ |
| $P+1$: | $\sqrt{MP}$ | 0 | 0 | 0 | 0 | 0 | 0 | 0 | 0 | 0 | 0 | 0 | 0 | 0 | 0 |

By these, the attention score matrix $\boldsymbol{C}_2^{(1)} := (\boldsymbol{K}_2^{(1)} \boldsymbol{X}^{(0)})^\top \boldsymbol{Q}_2^{(1)} \boldsymbol{X}^{(0)}$ and the attention matrix $\boldsymbol{A}_2^{(1)} := \texttt{softmax}(\boldsymbol{C}_2^{(1)}) \in \mathbb{R}^{N \times N}$ can be obtained, and the example of $\boldsymbol{A}_2^{(1)}$ is provided in Table 39.

Table 39: Example of $\boldsymbol{A}_2^{(1)}$ (with explicit row/column indices and sufficiently large $M$), continuing from Tables 37 and 38.

| row \ col | $j = 1$ | 2 | 3 | 4 | 5 | 6 | 7 | 8 | 9 | 10 | 11 | 12 | 13 | 14 | 15 |
|---|---|---|---|---|---|---|---|---|---|---|---|---|---|---|---|
| $i = 1$ | 1 | 1 | 1 | 1 | 1 | 1 | 1/2 | 1 | 1 | 1 | 1/3 | 1/4 | 1/4 | 1/3 | 1/3 |
| 2 | 0 | 0 | 0 | 0 | 0 | 0 | 0 | 0 | 0 | 0 | 0 | 0 | 0 | 0 | 1/3 |
| 3 | 0 | 0 | 0 | 0 | 0 | 0 | 0 | 0 | 0 | 0 | 0 | 0 | 0 | 1/3 | 0 |
| 4 | 0 | 0 | 0 | 0 | 0 | 0 | 0 | 0 | 0 | 0 | 0 | 0 | 1/4 | 0 | 0 |
| 5 | 0 | 0 | 0 | 0 | 0 | 0 | 0 | 0 | 0 | 0 | 0 | 1/4 | 0 | 0 | 0 |
| 6 | 0 | 0 | 0 | 0 | 0 | 0 | 1/2 | 0 | 0 | 0 | 1/3 | 0 | 0 | 0 | 0 |
| 7 | 0 | 0 | 0 | 0 | 0 | 0 | 0 | 0 | 0 | 0 | 0 | 0 | 1/4 | 0 | 0 |
| 8 | 0 | 0 | 0 | 0 | 0 | 0 | 0 | 0 | 0 | 0 | 0 | 1/4 | 0 | 0 | 0 |
| 9 | 0 | 0 | 0 | 0 | 0 | 0 | 0 | 0 | 0 | 0 | 1/3 | 0 | 0 | 0 | 0 |
| 10 | 0 | 0 | 0 | 0 | 0 | 0 | 0 | 0 | 0 | 0 | 0 | 1/4 | 0 | 0 | 0 |
| 11 | 0 | 0 | 0 | 0 | 0 | 0 | 0 | 0 | 0 | 0 | 0 | 0 | 1/4 | 0 | 0 |
| 12 | 0 | 0 | 0 | 0 | 0 | 0 | 0 | 0 | 0 | 0 | 0 | 0 | 0 | 1/3 | 0 |
| 13 | 0 | 0 | 0 | 0 | 0 | 0 | 0 | 0 | 0 | 0 | 0 | 0 | 0 | 0 | 1/3 |
| 14 | 0 | 0 | 0 | 0 | 0 | 0 | 0 | 0 | 0 | 0 | 0 | 0 | 0 | 0 | 0 |
| 15 | 0 | 0 | 0 | 0 | 0 | 0 | 0 | 0 | 0 | 0 | 0 | 0 | 0 | 0 | 0 |

Finally, we set $\boldsymbol{V}_2^{(1)}$ and $\boldsymbol{U}_2^{(1)}$ the same to that of the previous head. That is, with $d_{V,12} = 1$,

$$\boldsymbol{V}_2^{(1)} = 2(\boldsymbol{e}_{\text{IS\_MUL}}^d - \boldsymbol{e}_{\text{IS\_EQUAL}}^d)^\top \in \mathbb{R}^{d_{V,12} \times d}, \tag{87}$$

$$\boldsymbol{U}_2^{(1)} = \boldsymbol{e}_{\text{IS\_OP2\_TEN}}^d \in \mathbb{R}^{d \times d_{V,12}}, \tag{88}$$

and the example of $\boldsymbol{U}_2^{(1)} \boldsymbol{V}_2^{(1)} \boldsymbol{X}^{(0)}$ and $\boldsymbol{U}_2^{(1)} \boldsymbol{V}_2^{(1)} \boldsymbol{X}^{(0)} \boldsymbol{A}_2^{(1)}$ is provided in Tables 40 and 41. Consequently, the matrix $\boldsymbol{U}_2^{(1)} \boldsymbol{V}_2^{(1)} \boldsymbol{X}^{(0)} \boldsymbol{A}_2^{(1)}$ is a matrix that matches the size of the input embedding

matrix and is filled with zeroes, except for a unique 1 located at the tens place of the second operand in the input sequence, with the dimension IS_OP2_TEN (7-th dimension).

Table 40: Example of $\boldsymbol{U}_2^{(1)}\boldsymbol{V}_2^{(1)}\boldsymbol{X}^{(0)}$, continuing from Table 28. (Irrelevant dimensions are omitted for readability)

| $\mathcal{I}$ | \$ | 7 | 5 | 9 | 5 | × | 7 | 9 | = | 5 | 0 | 0 | 0 | 0 | 6 |
|---|---|---|---|---|---|---|---|---|---|---|---|---|---|---|---|
| 1: NUM | 0 | 0 | 0 | 0 | 0 | 0 | 0 | 0 | 0 | 0 | 0 | 0 | 0 | 0 | 0 |
| 2: FULL_ONES | 0 | 0 | 0 | 0 | 0 | 0 | 0 | 0 | 0 | 0 | 0 | 0 | 0 | 0 | 0 |
| 3: IS_BOS | 0 | 0 | 0 | 0 | 0 | 0 | 0 | 0 | 0 | 0 | 0 | 0 | 0 | 0 | 0 |
| 4: IS_MUL | 0 | 0 | 0 | 0 | 0 | 0 | 0 | 0 | 0 | 0 | 0 | 0 | 0 | 0 | 0 |
| 5: IS_EQUAL | 0 | 0 | 0 | 0 | 0 | 0 | 0 | 0 | 0 | 0 | 0 | 0 | 0 | 0 | 0 |
| 7: IS_OP2_TEN | 0 | 0 | 0 | 0 | 0 | 2 | 0 | 0 | -2 | 0 | 0 | 0 | 0 | 0 | 0 |

Table 41: Example of $\boldsymbol{U}_2^{(1)}\boldsymbol{V}_2^{(1)}\boldsymbol{X}^{(0)}\boldsymbol{A}_2^{(1)}$, continuing from Tables 39 and 40. (Irrelevant dimensions are omitted for readability)

| $\mathcal{I}$ | \$ | 7 | 5 | 9 | 5 | × | 7 | 9 | = | 5 | 0 | 0 | 0 | 0 | 6 |
|---|---|---|---|---|---|---|---|---|---|---|---|---|---|---|---|
| 1: NUM | 0 | 0 | 0 | 0 | 0 | 0 | 0 | 0 | 0 | 0 | 0 | 0 | 0 | 0 | 0 |
| 2: FULL_ONES | 0 | 0 | 0 | 0 | 0 | 0 | 0 | 0 | 0 | 0 | 0 | 0 | 0 | 0 | 0 |
| 3: IS_BOS | 0 | 0 | 0 | 0 | 0 | 0 | 0 | 0 | 0 | 0 | 0 | 0 | 0 | 0 | 0 |
| 4: IS_MUL | 0 | 0 | 0 | 0 | 0 | 0 | 0 | 0 | 0 | 0 | 0 | 0 | 0 | 0 | 0 |
| 5: IS_EQUAL | 0 | 0 | 0 | 0 | 0 | 0 | 0 | 0 | 0 | 0 | 0 | 0 | 0 | 0 | 0 |
| 7: IS_OP2_TEN | 0 | 0 | 0 | 0 | 0 | 0 | 1 | 0 | 0 | 0 | 0 | 0 | 0 | 0 | 0 |

### G.5.3 Attention Head 3: Position Masking

The goal of Attention head 3 is to generate a binary mask at the dimension MASK (34-th dimension), with '0' placed before the multiplication symbol (×) and '1' placed starting from the multiplication symbol to the end.

To this end, we set $d_{QK,13} = 1$ and design query and key weights by

$$\boldsymbol{Q}_3^{(1)} = \left(\boldsymbol{e}_{\text{FULL\_ONES}}^d\right)^\top \in \mathbb{R}^{d_{QK,13}\times d}, \tag{89}$$

$$\boldsymbol{K}_3^{(1)} = \left(\boldsymbol{e}_{\text{IS\_MUL}}^d\right)^\top \in \mathbb{R}^{d_{QK,13}\times d}. \tag{90}$$

The matrices $\boldsymbol{Q}_3^{(1)}\boldsymbol{X}^{(0)}$ and $\boldsymbol{K}_3^{(1)}\boldsymbol{X}^{(0)}$ take the dimension FULL_ONES and IS_MUL, respectively, from the input embedding matrix. For concrete examples, please refer to Tables 42 and 43.

Table 42: Example of $\boldsymbol{Q}_3^{(1)}\boldsymbol{X}^{(0)}$, continuing from Table 28.

| $\mathcal{I}$ | \$ | 7 | 5 | 9 | 5 | × | 7 | 9 | = | 5 | 0 | 0 | 0 | 0 | 6 |
|---|---|---|---|---|---|---|---|---|---|---|---|---|---|---|---|
| 1: | 1 | 1 | 1 | 1 | 1 | 1 | 1 | 1 | 1 | 1 | 1 | 1 | 1 | 1 | 1 |

Table 43: Example of $\boldsymbol{K}_3^{(1)}\boldsymbol{X}^{(0)}$, continuing from Table 28.

| $\mathcal{I}$ | \$ | 7 | 5 | 9 | 5 | × | 7 | 9 | = | 5 | 0 | 0 | 0 | 0 | 6 |
|---|---|---|---|---|---|---|---|---|---|---|---|---|---|---|---|
| 1: | 0 | 0 | 0 | 0 | 0 | 1 | 0 | 0 | 0 | 0 | 0 | 0 | 0 | 0 | 0 |

By these, the attention score matrix $\boldsymbol{C}_3^{(1)} := (\boldsymbol{K}_3^{(1)}\boldsymbol{X}^{(0)})^\top\boldsymbol{Q}_3^{(1)}\boldsymbol{X}^{(0)}$ and the attention matrix $\boldsymbol{A}_3^{(1)} := \text{softmax}(\boldsymbol{C}_3^{(1)}) \in \mathbb{R}^{N\times N}$ can be obtained and the example of $\boldsymbol{A}_3^{(1)}$ is provided in Table 44.

Table 44: Example of $\boldsymbol{A}_3^{(1)}$ (with explicit row/column indices), continuing from Tables 42 and 43.

| row \ col | $j = 1$ | 2 | 3 | 4 | 5 | 6 | 7 | 8 | 9 | 10 | 11 | 12 | 13 | 14 | 15 |
|---|---|---|---|---|---|---|---|---|---|---|---|---|---|---|---|
|  | 1 | 1 | 1 | 1 | 1 | 1 | 1 | 1 | 1 | 1 | 1 | 1 | 1 | 1 | 1 |
| $i = 1$  0 | 1 | 1/2 | 1/3 | 1/4 | 1/5 | 0 | 0 | 0 | 0 | 0 | 0 | 0 | 0 | 0 | 0 |
| 2  0 | 0 | 1/2 | 1/3 | 1/4 | 1/5 | 0 | 0 | 0 | 0 | 0 | 0 | 0 | 0 | 0 | 0 |
| 3  0 | 0 | 0 | 1/3 | 1/4 | 1/5 | 0 | 0 | 0 | 0 | 0 | 0 | 0 | 0 | 0 | 0 |
| 4  0 | 0 | 0 | 0 | 1/4 | 1/5 | 0 | 0 | 0 | 0 | 0 | 0 | 0 | 0 | 0 | 0 |
| 5  0 | 0 | 0 | 0 | 0 | 1/5 | 0 | 0 | 0 | 0 | 0 | 0 | 0 | 0 | 0 | 0 |
| 6  1 | 0 | 0 | 0 | 0 | 0 | 1 | 1 | 1 | 1 | 1 | 1 | 1 | 1 | 1 | 1 |
| 7  0 | 0 | 0 | 0 | 0 | 0 | 0 | 0 | 0 | 0 | 0 | 0 | 0 | 0 | 0 | 0 |
| 8  0 | 0 | 0 | 0 | 0 | 0 | 0 | 0 | 0 | 0 | 0 | 0 | 0 | 0 | 0 | 0 |
| 9  0 | 0 | 0 | 0 | 0 | 0 | 0 | 0 | 0 | 0 | 0 | 0 | 0 | 0 | 0 | 0 |
| 10  0 | 0 | 0 | 0 | 0 | 0 | 0 | 0 | 0 | 0 | 0 | 0 | 0 | 0 | 0 | 0 |
| 11  0 | 0 | 0 | 0 | 0 | 0 | 0 | 0 | 0 | 0 | 0 | 0 | 0 | 0 | 0 | 0 |
| 12  0 | 0 | 0 | 0 | 0 | 0 | 0 | 0 | 0 | 0 | 0 | 0 | 0 | 0 | 0 | 0 |
| 13  0 | 0 | 0 | 0 | 0 | 0 | 0 | 0 | 0 | 0 | 0 | 0 | 0 | 0 | 0 | 0 |
| 14  0 | 0 | 0 | 0 | 0 | 0 | 0 | 0 | 0 | 0 | 0 | 0 | 0 | 0 | 0 | 0 |
| 15  0 | 0 | 0 | 0 | 0 | 0 | 0 | 0 | 0 | 0 | 0 | 0 | 0 | 0 | 0 | 0 |

Finally, we set $\boldsymbol{V}_3^{(1)}$ and $\boldsymbol{U}_3^{(1)}$ by $d_{V,13} = 1$ and

$$\boldsymbol{V}_3^{(1)} = (\boldsymbol{e}_{\text{IS\_MUL}}^d)^\top \in \mathbb{R}^{d_{V,13} \times d}, \tag{91}$$

$$\boldsymbol{U}_3^{(1)} = \boldsymbol{e}_{\text{MASK}}^d \in \mathbb{R}^{d \times d_{V,13}}. \tag{92}$$

The example of $\boldsymbol{U}_3^{(1)} \boldsymbol{V}_3^{(1)} \boldsymbol{X}^{(0)}$ and $\boldsymbol{U}_3^{(1)} \boldsymbol{V}_3^{(1)} \boldsymbol{X}^{(0)} \boldsymbol{A}_3^{(1)}$ is provided in Tables 45 and 46. Consequently, the matrix $\boldsymbol{U}_3^{(1)} \boldsymbol{V}_3^{(1)} \boldsymbol{X}^{(0)} \boldsymbol{A}_3^{(1)}$ is a matrix that matches the size of the input embedding matrix and is filled with 1 only at the dimension MASK (34-th dimension) starting from the $\times$ token to the end of sequence, and 0 otherwise.

At this point, the objective of attention head 3 may seem somewhat unclear. We note that the output of Attention head 3 will be utilized to fill the dimensions POS_2_MASK in the subsequent FFN layer, and this POS_2_MASK plays a crucial role in designing the key matrices in the Attention heads 3 to 7 at the second Transformer block.

Table 45: Example of $\boldsymbol{U}_3^{(1)} \boldsymbol{V}_3^{(1)} \boldsymbol{X}^{(0)}$, continuing from Table 28. (Irrelevant dimensions are omitted for readability)

| $\mathcal{I}$ | $ | 7 | 5 | 9 | 5 | $\times$ | 7 | 9 | = | 5 | 0 | 0 | 0 | 0 | 6 |
|---|---|---|---|---|---|---|---|---|---|---|---|---|---|---|---|
| 1: NUM | 0 | 0 | 0 | 0 | 0 | 0 | 0 | 0 | 0 | 0 | 0 | 0 | 0 | 0 | 0 |
| 2: FULL_ONES | 0 | 0 | 0 | 0 | 0 | 0 | 0 | 0 | 0 | 0 | 0 | 0 | 0 | 0 | 0 |
| 3: IS_BOS | 0 | 0 | 0 | 0 | 0 | 0 | 0 | 0 | 0 | 0 | 0 | 0 | 0 | 0 | 0 |
| 4: IS_MUL | 0 | 0 | 0 | 0 | 0 | 0 | 0 | 0 | 0 | 0 | 0 | 0 | 0 | 0 | 0 |
| 5: IS_EQUAL | 0 | 0 | 0 | 0 | 0 | 0 | 0 | 0 | 0 | 0 | 0 | 0 | 0 | 0 | 0 |
| 34: MASK | 0 | 0 | 0 | 0 | 0 | 1 | 0 | 0 | 0 | 0 | 0 | 0 | 0 | 0 | 0 |

Table 46: Example of $\boldsymbol{U}_3^{(1)} \boldsymbol{V}_3^{(1)} \boldsymbol{X}^{(0)} \boldsymbol{A}_3^{(1)}$, continuing from Tables 44 and 45. (Irrelevant dimensions are omitted for readability)

| $\mathcal{I}$ | $ | 7 | 5 | 9 | 5 | $\times$ | 7 | 9 | = | 5 | 0 | 0 | 0 | 0 | 6 |
|---|---|---|---|---|---|---|---|---|---|---|---|---|---|---|---|
| 1: NUM | 0 | 0 | 0 | 0 | 0 | 0 | 0 | 0 | 0 | 0 | 0 | 0 | 0 | 0 | 0 |
| 2: FULL_ONES | 0 | 0 | 0 | 0 | 0 | 0 | 0 | 0 | 0 | 0 | 0 | 0 | 0 | 0 | 0 |
| 3: IS_BOS | 0 | 0 | 0 | 0 | 0 | 0 | 0 | 0 | 0 | 0 | 0 | 0 | 0 | 0 | 0 |
| 4: IS_MUL | 0 | 0 | 0 | 0 | 0 | 0 | 0 | 0 | 0 | 0 | 0 | 0 | 0 | 0 | 0 |
| 5: IS_EQUAL | 0 | 0 | 0 | 0 | 0 | 0 | 0 | 0 | 0 | 0 | 0 | 0 | 0 | 0 | 0 |
| 34: MASK | 0 | 0 | 0 | 0 | 0 | 1 | 1 | 1 | 1 | 1 | 1 | 1 | 1 | 1 | 1 |

## G.5.4 Residual Connection

So far we have computed the output of $\texttt{Att}_1$ operation. Passing through the residual connection, the output of the attention layer becomes the sum of the original input embedding matrix and the output of $\texttt{Att}_1$ operation:

$$\boldsymbol{Y}^{(1)} = \boldsymbol{X}^{(0)} + \sum_{h \in \{1,2,3\}} \boldsymbol{U}_h^{(1)} \boldsymbol{V}_h^{(1)} \boldsymbol{X}^{(0)} \boldsymbol{A}_h^{(1)}. \tag{93}$$

An example of the output of residual connection is presented in Table 47.

Table 47: Example output of residual connection, continuing from Tables 28, 36, 41 and 46. Uncolored rows represent the initial embedding. Yellow rows indicate the rows filled by the attention heads in the first Transformer block. A pink row indicates the row that will be filled by the subsequent FFN layer.

| $\mathcal{I}$ | $ | 7 | 5 | 9 | 5 | × | 7 | 9 | = | 5 | 0 | 0 | 0 | 0 | 6 |
|---|---|---|---|---|---|---|---|---|---|---|---|---|---|---|---|
| 1: NUM | 0 | 7 | 5 | 9 | 5 | 0 | 7 | 9 | 0 | 5 | 0 | 0 | 0 | 0 | 6 |
| 2: FULL_ONES | 1 | 1 | 1 | 1 | 1 | 1 | 1 | 1 | 1 | 1 | 1 | 1 | 1 | 1 | 1 |
| 3: IS_BOS | 1 | 0 | 0 | 0 | 0 | 0 | 0 | 0 | 0 | 0 | 0 | 0 | 0 | 0 | 0 |
| 4: IS_MUL | 0 | 0 | 0 | 0 | 0 | 1 | 0 | 0 | 0 | 0 | 0 | 0 | 0 | 0 | 0 |
| 5: IS_EQUAL | 0 | 0 | 0 | 0 | 0 | 0 | 0 | 0 | 1 | 0 | 0 | 0 | 0 | 0 | 0 |
| 6: IS_OP2_ONE | 0 | 0 | 0 | 0 | 0 | 0 | 0 | 1 | 0 | 0 | 0 | 0 | 0 | 0 | 0 |
| 7: IS_OP2_TEN | 0 | 0 | 0 | 0 | 0 | 0 | 1 | 0 | 0 | 0 | 0 | 0 | 0 | 0 | 0 |
| 34: MASK | 0 | 0 | 0 | 0 | 0 | 1 | 1 | 1 | 1 | 1 | 1 | 1 | 1 | 1 | 1 |
| 35–$(P{+}34)$: POS_2_MASK | $\boldsymbol{0}_P$ | $\boldsymbol{0}_P$ | $\boldsymbol{0}_P$ | $\boldsymbol{0}_P$ | $\boldsymbol{0}_P$ | $\boldsymbol{0}_P$ | $\boldsymbol{0}_P$ | $\boldsymbol{0}_P$ | $\boldsymbol{0}_P$ | $\boldsymbol{0}_P$ | $\boldsymbol{0}_P$ | $\boldsymbol{0}_P$ | $\boldsymbol{0}_P$ | $\boldsymbol{0}_P$ | $\boldsymbol{0}_P$ |
| $(P{+}35)$–$(2P{+}34)$: POS_1 | $\boldsymbol{0}_P$ | $v_3^P$ | $v_4^P$ | $v_5^P$ | $v_6^P$ | $v_7^P$ | $v_5^P$ | $v_6^P$ | $v_7^P$ | $v_6^P$ | $v_5^P$ | $v_4^P$ | $v_3^P$ | $v_2^P$ | $v_1^P$ |
| $(2P{+}35)$–$(3P{+}34)$: POS_2 | $\boldsymbol{0}_P$ | $v_4^P$ | $v_5^P$ | $v_6^P$ | $v_7^P$ | $v_8^P$ | $v_6^P$ | $v_7^P$ | $v_8^P$ | $v_7^P$ | $v_6^P$ | $v_5^P$ | $v_4^P$ | $v_3^P$ | $v_2^P$ |
| $(3P{+}35)$–$(4P{+}34)$: POS_3 | $\boldsymbol{0}_P$ | $v_5^P$ | $v_6^P$ | $v_7^P$ | $v_8^P$ | $v_9^P$ | $v_7^P$ | $v_8^P$ | $v_9^P$ | $v_8^P$ | $v_7^P$ | $v_6^P$ | $v_5^P$ | $v_4^P$ | $v_3^P$ |
| $(4P{+}35)$–$(5P{+}34)$: POS_4 | $\boldsymbol{0}_P$ | $v_6^P$ | $v_7^P$ | $v_8^P$ | $v_9^P$ | $v_{10}^P$ | $v_8^P$ | $v_9^P$ | $v_{10}^P$ | $v_9^P$ | $v_8^P$ | $v_7^P$ | $v_6^P$ | $v_5^P$ | $v_4^P$ |
| $(5P{+}35)$–$(6P{+}34)$: POS_5 | $\boldsymbol{0}_P$ | $v_7^P$ | $v_8^P$ | $v_9^P$ | $v_{10}^P$ | $v_{11}^P$ | $v_9^P$ | $v_{10}^P$ | $v_{11}^P$ | $v_{10}^P$ | $v_9^P$ | $v_8^P$ | $v_7^P$ | $v_6^P$ | $v_5^P$ |

## G.6 Transformer Block 1 — Token-wise Feed-forward Layer

The goal of the feed-forward layer involves filling the dimensions POS_2_MASK. Specifically, for each token $\sigma_i$, if the dimension MASK is 1 (i.e., $\boldsymbol{Y}^{(1)}_{(\text{MASK})i} = 1$), we want to fill the dimensions POS_2_MASK by copying the the corresponding token's POS_2; otherwise, we want to fill with $\boldsymbol{0}_P$. Be careful that the feed-forward operation is restricted to a token-wise mapping, meaning it only takes inputs from entries within the same column of the encoding matrix.

**Construction for POS_2_MASK.** Given a vector $\boldsymbol{y} = [\boldsymbol{y}_j]_{j=1}^d \in \mathbb{R}^d$, define functions $g_l, h_l : \mathbb{R}^d \to \mathbb{R}$ for every $j \in [P]$ as

$$g_l(\boldsymbol{y}) := \boldsymbol{y}_{\text{POS\_2},l} - 2\boldsymbol{y}_{\text{MASK}} \tag{94}$$
$$h_l(\boldsymbol{y}) := -\boldsymbol{y}_{\text{POS\_2},l} - 2\boldsymbol{y}_{\text{MASK}} \tag{95}$$

where $\boldsymbol{y}_{\text{POS\_2},l} \in \mathbb{R}$ is the $l$-th dimension of $\boldsymbol{y}_{\text{POS\_2}} \in \mathbb{R}^P$ ($l \in 1, 2, \ldots, P$).

Consider a simple one-hidden-layer ReLU networks $f_l : \mathbb{R}^d \to \mathbb{R}$ defined as

$$f_l(\boldsymbol{y}) = \phi(g_l(\boldsymbol{y})) - \phi(h_l(\boldsymbol{y})).$$

Using the fact that $\boldsymbol{y}_{\text{POS\_2},l}$ is either $-1$ or $1$, we can easily check that $f_l(\boldsymbol{y}) = \boldsymbol{y}_{\text{POS\_2},l}$ if $\boldsymbol{y}_{\text{MASK}}$ is 0, and $f_l(\boldsymbol{y}) = 0$ if $\boldsymbol{y}_{\text{MASK}}$ is 1.

Now, we can construct the width-$2P$ feed-forward network that outputs the desired value at the dimension POS_2_MASK by

$$\left[ \text{FF}_1\left( \boldsymbol{Y}^{(1)} \right) \right]_{(\text{POS\_2\_MASK})i} = \left[ f_1\left( \boldsymbol{Y}^{(1)}_{\bullet i} \right) \quad \cdots \quad f_P\left( \boldsymbol{Y}^{(1)}_{\bullet i} \right) \right]^\top \in \mathbb{R}^{P \times 1}, \tag{96}$$

and 0 for any other dimensions. The example output for this layer is presented in Table 48.

Table 48: Example output of FFN layer at the first Transformer block, continuing from Table 47.

| $\mathcal{I}$ | $ | 7 | 5 | 9 | 5 | × | 7 | 9 | = | 5 | 0 | 0 | 0 | 0 | 6 |
|---|---|---|---|---|---|---|---|---|---|---|---|---|---|---|---|
| 35–$(P+34)$: POS_2_MASK | $\mathbf{0}_P$ | $v_4^P$ | $v_5^P$ | $v_6^P$ | $v_7^P$ | $\mathbf{0}_P$ | $\mathbf{0}_P$ | $\mathbf{0}_P$ | $\mathbf{0}_P$ | $\mathbf{0}_P$ | $\mathbf{0}_P$ | $\mathbf{0}_P$ | $\mathbf{0}_P$ | $\mathbf{0}_P$ | $\mathbf{0}_P$ |

### G.6.1 Residual Connection

The last task of the feed-forward layer is to pass $\mathrm{FF}_1\left(\boldsymbol{Y}^{(1)}\right)$ through the residual connection. As a result, we have

$$\boldsymbol{X}^{(1)} = \boldsymbol{Y}^{(1)} + \mathrm{FF}_1\left(\boldsymbol{Y}^{(1)}\right). \tag{97}$$

This is the end of the first Transformer block, and a concrete example of $\boldsymbol{X}^{(1)}$ is illustrated in Table 49.

Table 49: Example embedding matrix after the first Transformer block. The yellow rows represent the results introduced during the first block, while the gray rows will be filled in the second block.

| $\mathcal{I}$ | $ | 7 | 5 | 9 | 5 | × | 7 | 9 | = | 5 | 0 | 0 | 0 | 0 | 6 |
|---|---|---|---|---|---|---|---|---|---|---|---|---|---|---|---|
| 1: NUM | 0 | 7 | 5 | 9 | 5 | 0 | 7 | 9 | 0 | 5 | 0 | 0 | 0 | 0 | 6 |
| 2: FULL_ONES | 1 | 1 | 1 | 1 | 1 | 1 | 1 | 1 | 1 | 1 | 1 | 1 | 1 | 1 | 1 |
| 3: IS_BOS | 1 | 0 | 0 | 0 | 0 | 0 | 0 | 0 | 0 | 0 | 0 | 0 | 0 | 0 | 0 |
| 4: IS_MUL | 0 | 0 | 0 | 0 | 0 | 1 | 0 | 0 | 0 | 0 | 0 | 0 | 0 | 0 | 0 |
| 5: IS_EQUAL | 0 | 0 | 0 | 0 | 0 | 0 | 0 | 0 | 1 | 0 | 0 | 0 | 0 | 0 | 0 |
| 6: IS_OP2_ONE | 0 | 0 | 0 | 0 | 0 | 0 | 0 | 1 | 0 | 0 | 0 | 0 | 0 | 0 | 0 |
| 7: IS_OP2_TEN | 0 | 0 | 0 | 0 | 0 | 0 | 1 | 0 | 0 | 0 | 0 | 0 | 0 | 0 | 0 |
| 8: OP2_ONE | 0 | 0 | 0 | 0 | 0 | 0 | 0 | 0 | 0 | 0 | 0 | 0 | 0 | 0 | 0 |
| 9: OP2_TEN | 0 | 0 | 0 | 0 | 0 | 0 | 0 | 0 | 0 | 0 | 0 | 0 | 0 | 0 | 0 |
| 10: OP1_SHIFT0 | 0 | 0 | 0 | 0 | 0 | 0 | 0 | 0 | 0 | 0 | 0 | 0 | 0 | 0 | 0 |
| 11: OP1_SHIFT1 | 0 | 0 | 0 | 0 | 0 | 0 | 0 | 0 | 0 | 0 | 0 | 0 | 0 | 0 | 0 |
| 12: OP1_SHIFT2 | 0 | 0 | 0 | 0 | 0 | 0 | 0 | 0 | 0 | 0 | 0 | 0 | 0 | 0 | 0 |
| 13: OP1_SHIFT3 | 0 | 0 | 0 | 0 | 0 | 0 | 0 | 0 | 0 | 0 | 0 | 0 | 0 | 0 | 0 |
| 14: OP1_SHIFT4 | 0 | 0 | 0 | 0 | 0 | 0 | 0 | 0 | 0 | 0 | 0 | 0 | 0 | 0 | 0 |
| 15: RESULT1 | 0 | 0 | 0 | 0 | 0 | 0 | 0 | 0 | 0 | 0 | 0 | 0 | 0 | 0 | 0 |
| 16: RESULT2 | 0 | 0 | 0 | 0 | 0 | 0 | 0 | 0 | 0 | 0 | 0 | 0 | 0 | 0 | 0 |
| 17: RESULT3 | 0 | 0 | 0 | 0 | 0 | 0 | 0 | 0 | 0 | 0 | 0 | 0 | 0 | 0 | 0 |
| 18: RESULT4 | 0 | 0 | 0 | 0 | 0 | 0 | 0 | 0 | 0 | 0 | 0 | 0 | 0 | 0 | 0 |
| 19: PRE_PROD | 0 | 0 | 0 | 0 | 0 | 0 | 0 | 0 | 0 | 0 | 0 | 0 | 0 | 0 | 0 |
| 20: PRE_CARRY | 0 | 0 | 0 | 0 | 0 | 0 | 0 | 0 | 0 | 0 | 0 | 0 | 0 | 0 | 0 |
| 21: PRE_EOS1 | 0 | 0 | 0 | 0 | 0 | 0 | 0 | 0 | 0 | 0 | 0 | 0 | 0 | 0 | 0 |
| 22: PRE_EOS2 | 0 | 0 | 0 | 0 | 0 | 0 | 0 | 0 | 0 | 0 | 0 | 0 | 0 | 0 | 0 |
| 23-32: PROD | $\mathbf{0}_{10}$ | $\mathbf{0}_{10}$ | $\mathbf{0}_{10}$ | $\mathbf{0}_{10}$ | $\mathbf{0}_{10}$ | $\mathbf{0}_{10}$ | $\mathbf{0}_{10}$ | $\mathbf{0}_{10}$ | $\mathbf{0}_{10}$ | $\mathbf{0}_{10}$ | $\mathbf{0}_{10}$ | $\mathbf{0}_{10}$ | $\mathbf{0}_{10}$ | $\mathbf{0}_{10}$ | $\mathbf{0}_{10}$ |
| 33: IS_EOS | 0 | 0 | 0 | 0 | 0 | 0 | 0 | 0 | 0 | 0 | 0 | 0 | 0 | 0 | 0 |
| 34: MASK | 0 | 0 | 0 | 0 | 0 | 1 | 1 | 1 | 1 | 1 | 1 | 1 | 1 | 1 | 1 |
| 35–$(P+34)$: POS_2_MASK | $\mathbf{0}_P$ | $v_4^P$ | $v_5^P$ | $v_6^P$ | $v_7^P$ | $\mathbf{0}_P$ | $\mathbf{0}_P$ | $\mathbf{0}_P$ | $\mathbf{0}_P$ | $\mathbf{0}_P$ | $\mathbf{0}_P$ | $\mathbf{0}_P$ | $\mathbf{0}_P$ | $\mathbf{0}_P$ | $\mathbf{0}_P$ |
| $(P+35)$–$(2P+34)$: POS_1 | $\mathbf{0}_P$ | $v_3^P$ | $v_4^P$ | $v_5^P$ | $v_6^P$ | $v_7^P$ | $v_5^P$ | $v_6^P$ | $v_7^P$ | $v_6^P$ | $v_5^P$ | $v_4^P$ | $v_3^P$ | $v_2^P$ | $v_1^P$ |
| $(2P+35)$–$(3P+34)$: POS_2 | $\mathbf{0}_P$ | $v_4^P$ | $v_5^P$ | $v_6^P$ | $v_7^P$ | $v_8^P$ | $v_6^P$ | $v_7^P$ | $v_8^P$ | $v_7^P$ | $v_6^P$ | $v_5^P$ | $v_4^P$ | $v_3^P$ | $v_2^P$ |
| $(3P+35)$–$(4P+34)$: POS_3 | $\mathbf{0}_P$ | $v_5^P$ | $v_6^P$ | $v_7^P$ | $v_8^P$ | $v_9^P$ | $v_7^P$ | $v_8^P$ | $v_9^P$ | $v_8^P$ | $v_7^P$ | $v_6^P$ | $v_5^P$ | $v_4^P$ | $v_3^P$ |
| $(4P+35)$–$(5P+34)$: POS_4 | $\mathbf{0}_P$ | $v_6^P$ | $v_7^P$ | $v_8^P$ | $v_9^P$ | $v_{10}^P$ | $v_8^P$ | $v_9^P$ | $v_{10}^P$ | $v_9^P$ | $v_8^P$ | $v_7^P$ | $v_6^P$ | $v_5^P$ | $v_4^P$ |
| $(5P+35)$–$(6P+34)$: POS_5 | $\mathbf{0}_P$ | $v_7^P$ | $v_8^P$ | $v_9^P$ | $v_{10}^P$ | $v_{11}^P$ | $v_9^P$ | $v_{10}^P$ | $v_{11}^P$ | $v_{10}^P$ | $v_9^P$ | $v_8^P$ | $v_7^P$ | $v_6^P$ | $v_5^P$ |

### G.7 Transformer Block 2 — Causal Attention Layer

Consider a scenario where the model is at the step of predicting the $i$-th least significant digit of the multiplication result. There are two goals for the causal attention layer at the second Transformer block. The first goal is to generate the embedding matrix as the left-most figure in Table 31, that is,

fill OP2_ONE, OP2_TEN, and OP1_SHIFT0 to OP1_SHIFT4 with the ones digit of the second operand, the tens digit of the second operand, and the $i, (i-1), (i-2), (i-3), (i-4)$-th least significant digit of the first operand, respectively. Our construction assigns each head to each dimension. The second goal is to fill PRE_EOS1 and PRE_EOS2 with appropriate values. These 2 dimensions will be utilized in the subsequent FFN layer to predict whether we should predict the next token as EOS or not. Also, we note that filling these 2 dimensions can be implemented within the same head for OP1_SHIFT0 and OP1_SHIFT2 respectively, thus requiring a total of seven heads.

### G.7.1 Attention Head 1: Copying the Ones Digit of the Second Operand

The objective of Attention head 1 is to fill the dimension OP2_ONE with the ones digit of the second operand. To do so, we design the weights by defining $d_{QK,21} = 1$ and

$$\mathbf{Q}_1^{(2)} = \left(\mathbf{e}_{\text{FULL\_ONES}}^d\right)^\top \in \mathbb{R}^{d_{QK,21} \times d}, \tag{98}$$

$$\mathbf{K}_1^{(2)} = \left(\mathbf{e}_{\text{IS\_OP2\_ONE}}^d\right)^\top \in \mathbb{R}^{d_{QK,21} \times d}. \tag{99}$$

We also define $d_{V,21} = 1$ and

$$\mathbf{V}_1^{(2)} = \left(\mathbf{e}_{\text{NUM}}^d\right)^\top \in \mathbb{R}^{d_{V,21} \times d}, \tag{100}$$

$$\mathbf{U}_1^{(2)} = \mathbf{e}_{\text{OP2\_ONE}}^d \in \mathbb{R}^{d \times d_{V,21}}. \tag{101}$$

A concrete example of $\mathbf{Q}_1^{(2)} X^{(1)}$, $\mathbf{K}_1^{(2)} X^{(1)}$, $\mathbf{A}_1^2$, $\mathbf{U}_1^{(2)} \mathbf{V}_1^{(2)} X^{(1)}$, and $\mathbf{U}_1^{(2)} \mathbf{V}_1^{(2)} X^{(1)} \mathbf{A}_1^{(2)}$ is provided in Tables 50 to 54. One might be concerned that in Table 54, the dimension OP2_ONE is not completely filled with '9', but only the latter part. However, we note that given our focus on next-token prediction, it suffices to accurately fill values starting from the = token, and filling the preceding tokens with placeholder values does not cause any issues.

Table 50: Example of $\mathbf{Q}_1^{(2)} X^{(1)}$, continuing from Table 49.

| $\mathcal{I}$ | \$ | 7 | 5 | 9 | 5 | × | 7 | 9 | = | 5 | 0 | 0 | 0 | 0 | 6 |
|---|---|---|---|---|---|---|---|---|---|---|---|---|---|---|---|
| 1: | 1 | 1 | 1 | 1 | 1 | 1 | 1 | 1 | 1 | 1 | 1 | 1 | 1 | 1 | 1 |

Table 51: Example of $\mathbf{K}_1^{(2)} X^{(1)}$, continuing from Table 49.

| $\mathcal{I}$ | \$ | 7 | 5 | 9 | 5 | × | 7 | 9 | = | 5 | 0 | 0 | 0 | 0 | 6 |
|---|---|---|---|---|---|---|---|---|---|---|---|---|---|---|---|
| 1: | 0 | 0 | 0 | 0 | 0 | 0 | 0 | 1 | 0 | 0 | 0 | 0 | 0 | 0 | 0 |

Table 53: Example of $\mathbf{U}_1^{(2)} \mathbf{V}_1^{(2)} X^{(1)}$, continuing from Table 49. (Irrelevant dimensions are omitted for readability)

| $\mathcal{I}$ | \$ | 7 | 5 | 9 | 5 | × | 7 | 9 | = | 5 | 0 | 0 | 0 | 0 | 6 |
|---|---|---|---|---|---|---|---|---|---|---|---|---|---|---|---|
| 1: NUM | 0 | 0 | 0 | 0 | 0 | 0 | 0 | 0 | 0 | 0 | 0 | 0 | 0 | 0 | 0 |
| 2: FULL_ONES | 0 | 0 | 0 | 0 | 0 | 0 | 0 | 0 | 0 | 0 | 0 | 0 | 0 | 0 | 0 |
| 3: IS_BOS | 0 | 0 | 0 | 0 | 0 | 0 | 0 | 0 | 0 | 0 | 0 | 0 | 0 | 0 | 0 |
| 4: IS_MUL | 0 | 0 | 0 | 0 | 0 | 0 | 0 | 0 | 0 | 0 | 0 | 0 | 0 | 0 | 0 |
| 5: IS_EQUAL | 0 | 0 | 0 | 0 | 0 | 0 | 0 | 0 | 0 | 0 | 0 | 0 | 0 | 0 | 0 |
| 8: OP2_ONE | 0 | 7 | 5 | 9 | 5 | 0 | 7 | 9 | 0 | 5 | 0 | 0 | 0 | 0 | 6 |

Table 52: Example of $\boldsymbol{A}_1^{(2)}$ (with explicit row/column indices), continuing from Tables 50 and 51.

| row \ col | $j=1$ | 2 | 3 | 4 | 5 | 6 | 7 | 8 | 9 | 10 | 11 | 12 | 13 | 14 | 15 |
|---|---|---|---|---|---|---|---|---|---|---|---|---|---|---|---|
| | 1 | 1 | 1 | 1 | 1 | 1 | 1 | 1 | 1 | 1 | 1 | 1 | 1 | 1 | 1 |
| $i=1$  0 | 1 | 1/2 | 1/3 | 1/4 | 1/5 | 1/6 | 1/7 | 0 | 0 | 0 | 0 | 0 | 0 | 0 | 0 |
| 2  0 | 0 | 1/2 | 1/3 | 1/4 | 1/5 | 1/6 | 1/7 | 0 | 0 | 0 | 0 | 0 | 0 | 0 | 0 |
| 3  0 | 0 | 0 | 1/3 | 1/4 | 1/5 | 1/6 | 1/7 | 0 | 0 | 0 | 0 | 0 | 0 | 0 | 0 |
| 4  0 | 0 | 0 | 0 | 1/4 | 1/5 | 1/6 | 1/7 | 0 | 0 | 0 | 0 | 0 | 0 | 0 | 0 |
| 5  0 | 0 | 0 | 0 | 0 | 1/5 | 1/6 | 1/7 | 0 | 0 | 0 | 0 | 0 | 0 | 0 | 0 |
| 6  0 | 0 | 0 | 0 | 0 | 0 | 1/6 | 1/7 | 0 | 0 | 0 | 0 | 0 | 0 | 0 | 0 |
| 7  0 | 0 | 0 | 0 | 0 | 0 | 0 | 1/7 | 0 | 0 | 0 | 0 | 0 | 0 | 0 | 0 |
| 8  1 | 0 | 0 | 0 | 0 | 0 | 0 | 0 | 1 | 1 | 1 | 1 | 1 | 1 | 1 | 1 |
| 9  0 | 0 | 0 | 0 | 0 | 0 | 0 | 0 | 0 | 0 | 0 | 0 | 0 | 0 | 0 | 0 |
| 10  0 | 0 | 0 | 0 | 0 | 0 | 0 | 0 | 0 | 0 | 0 | 0 | 0 | 0 | 0 | 0 |
| 11  0 | 0 | 0 | 0 | 0 | 0 | 0 | 0 | 0 | 0 | 0 | 0 | 0 | 0 | 0 | 0 |
| 12  0 | 0 | 0 | 0 | 0 | 0 | 0 | 0 | 0 | 0 | 0 | 0 | 0 | 0 | 0 | 0 |
| 13  0 | 0 | 0 | 0 | 0 | 0 | 0 | 0 | 0 | 0 | 0 | 0 | 0 | 0 | 0 | 0 |
| 14  0 | 0 | 0 | 0 | 0 | 0 | 0 | 0 | 0 | 0 | 0 | 0 | 0 | 0 | 0 | 0 |
| 15  0 | 0 | 0 | 0 | 0 | 0 | 0 | 0 | 0 | 0 | 0 | 0 | 0 | 0 | 0 | 0 |

Table 54: Example of $\boldsymbol{U}_1^{(2)}\boldsymbol{V}_1^{(2)}\boldsymbol{X}^{(1)}\boldsymbol{A}_1^{(2)}$, continuing from Tables 52 and 53. (Irrelevant dimensions are omitted for readability)

| $\mathcal{I}$ | \$ | 7 | 5 | 9 | 5 | $\times$ | 7 | 9 | = | 5 | 0 | 0 | 0 | 0 | 6 |
|---|---|---|---|---|---|---|---|---|---|---|---|---|---|---|---|
| 1: NUM | 0 | 0 | 0 | 0 | 0 | 0 | 0 | 0 | 0 | 0 | 0 | 0 | 0 | 0 | 0 |
| 2: FULL_ONES | 0 | 0 | 0 | 0 | 0 | 0 | 0 | 0 | 0 | 0 | 0 | 0 | 0 | 0 | 0 |
| 3: IS_BOS | 0 | 0 | 0 | 0 | 0 | 0 | 0 | 0 | 0 | 0 | 0 | 0 | 0 | 0 | 0 |
| 4: IS_MUL | 0 | 0 | 0 | 0 | 0 | 0 | 0 | 0 | 0 | 0 | 0 | 0 | 0 | 0 | 0 |
| 5: IS_EQUAL | 0 | 0 | 0 | 0 | 0 | 0 | 0 | 0 | 0 | 0 | 0 | 0 | 0 | 0 | 0 |
| 8: OP2_ONE | 0 | 7/2 | 4 | 21/4 | 26/5 | 13/3 | 33/7 | 9 | 9 | 9 | 9 | 9 | 9 | 9 | 9 |

### G.7.2   Attention Head 2: Copying the Tens Digit of the Second Operand

The objective of Attention head 2 is to fill the dimension OP2_TEN with the tens digit of the second operand. We take a similar approach to Attention head 1, but the main difference is that we utilize the dimension IS_OP2_TEN instead of IS_OP2_ONE for generating the key weight. We design the weights by defining $d_{QK,22} = 1$ and

$$\boldsymbol{Q}_2^{(2)} = \left(\boldsymbol{e}_{\text{FULL\_ONES}}^d\right)^\top \in \mathbb{R}^{d_{QK,22} \times d}, \tag{102}$$

$$\boldsymbol{K}_2^{(2)} = \left(\boldsymbol{e}_{\text{IS\_OP2\_TEN}}^d\right)^\top \in \mathbb{R}^{d_{QK,22} \times d}. \tag{103}$$

We also define $d_{V,22} = 1$ and

$$\boldsymbol{V}_2^{(2)} = \left(\boldsymbol{e}_{\text{NUM}}^d\right)^\top \in \mathbb{R}^{d_{V,22} \times d}, \tag{104}$$

$$\boldsymbol{U}_2^{(2)} = \boldsymbol{e}_{\text{OP2\_TEN}}^d \in \mathbb{R}^{d \times d_{V,22}}. \tag{105}$$

A concrete example of $\boldsymbol{Q}_2^{(2)}X^{(1)}$, $\boldsymbol{K}_2^{(2)}X^{(1)}$, $\boldsymbol{A}_2^2$, $\boldsymbol{U}_2^{(2)}\boldsymbol{V}_2^{(2)}\boldsymbol{X}^{(1)}$, and $\boldsymbol{U}_2^{(2)}\boldsymbol{V}_2^{(2)}\boldsymbol{X}^{(1)}\boldsymbol{A}_2^{(2)}$ is provided in Tables 55 to 59. Once again, the dimension OP2_TEN is not entirely filled with '7' in Table 59. As mentioned in the previous head, this does not cause any issues because the front part (before =) does not affect the final prediction unless additional attention blocks are introduced after the second Transformer block.

Table 55: Example of $\boldsymbol{Q}_2^{(2)}\boldsymbol{X}^{(1)}$, continuing from Table 49.

| $\mathcal{I}$ | \$ | 7 | 5 | 9 | 5 | $\times$ | 7 | 9 | = | 5 | 0 | 0 | 0 | 0 | 6 |
|---|---|---|---|---|---|---|---|---|---|---|---|---|---|---|---|
| 1: | 1 | 1 | 1 | 1 | 1 | 1 | 1 | 1 | 1 | 1 | 1 | 1 | 1 | 1 | 1 |

Table 56: Example of $\boldsymbol{K}_2^{(2)}\boldsymbol{X}^{(1)}$, continuing from Table 49.

| $\mathcal{I}$ | $ | 7 | 5 | 9 | 5 | × | 7 | 9 | = | 5 | 0 | 0 | 0 | 0 | 6 |
|---|---|---|---|---|---|---|---|---|---|---|---|---|---|---|---|
| 1: | 0 | 0 | 0 | 0 | 0 | 0 | 1 | 0 | 0 | 0 | 0 | 0 | 0 | 0 | 0 |

Table 57: Example of $\boldsymbol{A}_2^{(2)}$ (with explicit row/column indices), continuing from Tables 55 and 56.

| row \ col | $j=1$ | 2 | 3 | 4 | 5 | 6 | 7 | 8 | 9 | 10 | 11 | 12 | 13 | 14 | 15 |
|---|---|---|---|---|---|---|---|---|---|---|---|---|---|---|---|
| | 1 | 1 | 1 | 1 | 1 | 1 | 1 | 1 | 1 | 1 | 1 | 1 | 1 | 1 | 1 |
| $i=1$ ₀ | 1 | 1/2 | 1/3 | 1/4 | 1/5 | 1/6 | 0 | 0 | 0 | 0 | 0 | 0 | 0 | 0 | 0 |
| 2 ₀ | 0 | 1/2 | 1/3 | 1/4 | 1/5 | 1/6 | 0 | 0 | 0 | 0 | 0 | 0 | 0 | 0 | 0 |
| 3 ₀ | 0 | 0 | 1/3 | 1/4 | 1/5 | 1/6 | 0 | 0 | 0 | 0 | 0 | 0 | 0 | 0 | 0 |
| 4 ₀ | 0 | 0 | 0 | 1/4 | 1/5 | 1/6 | 0 | 0 | 0 | 0 | 0 | 0 | 0 | 0 | 0 |
| 5 ₀ | 0 | 0 | 0 | 0 | 1/5 | 1/6 | 0 | 0 | 0 | 0 | 0 | 0 | 0 | 0 | 0 |
| 6 ₀ | 0 | 0 | 0 | 0 | 0 | 1/6 | 0 | 0 | 0 | 0 | 0 | 0 | 0 | 0 | 0 |
| 7 ₁ | 0 | 0 | 0 | 0 | 0 | 0 | 1 | 1 | 1 | 1 | 1 | 1 | 1 | 1 | 1 |
| 8 ₀ | 0 | 0 | 0 | 0 | 0 | 0 | 0 | 0 | 0 | 0 | 0 | 0 | 0 | 0 | 0 |
| 9 ₀ | 0 | 0 | 0 | 0 | 0 | 0 | 0 | 0 | 0 | 0 | 0 | 0 | 0 | 0 | 0 |
| 10 ₀ | 0 | 0 | 0 | 0 | 0 | 0 | 0 | 0 | 0 | 0 | 0 | 0 | 0 | 0 | 0 |
| 11 ₀ | 0 | 0 | 0 | 0 | 0 | 0 | 0 | 0 | 0 | 0 | 0 | 0 | 0 | 0 | 0 |
| 12 ₀ | 0 | 0 | 0 | 0 | 0 | 0 | 0 | 0 | 0 | 0 | 0 | 0 | 0 | 0 | 0 |
| 13 ₀ | 0 | 0 | 0 | 0 | 0 | 0 | 0 | 0 | 0 | 0 | 0 | 0 | 0 | 0 | 0 |
| 14 ₀ | 0 | 0 | 0 | 0 | 0 | 0 | 0 | 0 | 0 | 0 | 0 | 0 | 0 | 0 | 0 |
| 15 ₀ | 0 | 0 | 0 | 0 | 0 | 0 | 0 | 0 | 0 | 0 | 0 | 0 | 0 | 0 | 0 |

Table 58: Example of $\boldsymbol{U}_2^{(2)}\boldsymbol{V}_2^{(2)}\boldsymbol{X}^{(1)}$, continuing from Table 49. (Irrelevant dimensions are omitted for readability)

| $\mathcal{I}$ | $ | 7 | 5 | 9 | 5 | × | 7 | 9 | = | 5 | 0 | 0 | 0 | 0 | 6 |
|---|---|---|---|---|---|---|---|---|---|---|---|---|---|---|---|
| 1: NUM | 0 | 0 | 0 | 0 | 0 | 0 | 0 | 0 | 0 | 0 | 0 | 0 | 0 | 0 | 0 |
| 2: FULL_ONES | 0 | 0 | 0 | 0 | 0 | 0 | 0 | 0 | 0 | 0 | 0 | 0 | 0 | 0 | 0 |
| 3: IS_BOS | 0 | 0 | 0 | 0 | 0 | 0 | 0 | 0 | 0 | 0 | 0 | 0 | 0 | 0 | 0 |
| 4: IS_MUL | 0 | 0 | 0 | 0 | 0 | 0 | 0 | 0 | 0 | 0 | 0 | 0 | 0 | 0 | 0 |
| 5: IS_EQUAL | 0 | 0 | 0 | 0 | 0 | 0 | 0 | 0 | 0 | 0 | 0 | 0 | 0 | 0 | 0 |
| 9: OP2_TEN | 0 | 7 | 5 | 9 | 5 | 0 | 7 | 9 | 0 | 5 | 0 | 0 | 0 | 0 | 6 |

Table 59: Example of $\boldsymbol{U}_2^{(2)}\boldsymbol{V}_2^{(2)}\boldsymbol{X}^{(1)}\boldsymbol{A}_2^{(2)}$, continuing from Tables 57 and 58. (Irrelevant dimensions are omitted for readability)

| $\mathcal{I}$ | $ | 7 | 5 | 9 | 5 | × | 7 | 9 | = | 5 | 0 | 0 | 0 | 0 | 6 |
|---|---|---|---|---|---|---|---|---|---|---|---|---|---|---|---|
| 1: NUM | 0 | 0 | 0 | 0 | 0 | 0 | 0 | 0 | 0 | 0 | 0 | 0 | 0 | 0 | 0 |
| 2: FULL_ONES | 0 | 0 | 0 | 0 | 0 | 0 | 0 | 0 | 0 | 0 | 0 | 0 | 0 | 0 | 0 |
| 3: IS_BOS | 0 | 0 | 0 | 0 | 0 | 0 | 0 | 0 | 0 | 0 | 0 | 0 | 0 | 0 | 0 |
| 4: IS_MUL | 0 | 0 | 0 | 0 | 0 | 0 | 0 | 0 | 0 | 0 | 0 | 0 | 0 | 0 | 0 |
| 5: IS_EQUAL | 0 | 0 | 0 | 0 | 0 | 0 | 0 | 0 | 0 | 0 | 0 | 0 | 0 | 0 | 0 |
| 9: OP2_TEN | 0 | 7/2 | 4 | 21/4 | 26/5 | 13/3 | 7 | 7 | 7 | 7 | 7 | 7 | 7 | 7 | 7 |

### G.7.3 Attention Head 3: Copying the Appropriate Digit from the First Operand I

The objectives of the first and the second Attention heads were to extract the ones and tens digits of the second operand and display them in the dimensions OP2_ONE and OP2_TEN, respectively. For Attention head 3 to 7, we mainly focus on the first operand. Specifically, in Attention head 3, the goal is to fill the dimension OP1_SHIFT0 at the $i$-th least significant digit of the response (when predicting

the $(i+1)$-th least significant digit of the response) with the $(i+1)$-th least significant digit of the first operand. For our example, we want to fill OP1_SHIFT0 of the token $=$ by 5. Here, $i$ ranges from 0 to $\ell_a + 2$, where the 0-th least significant digit of the response denotes the equal token. In cases where $i \geq \ell_a$, we fill by 0.

Additionally, the third head has an extra objective: filling the dimension PRE_EOS1. This dimension is utilized for EOS prediction in the subsequent FFN layer along with PRE_EOS2, which is filled by the fifth head of the same layer. We observed that both objectives can be achieved by utilizing the same attention map. Thus, instead of implementing these objectives in separate heads, we can achieve them by utilizing the matrices $\boldsymbol{V}_3^{(2)}$ and $\boldsymbol{U}_3^{(2)}$ described below. Unlike previous heads, $\boldsymbol{V}_3^{(2)}$ and $\boldsymbol{U}_3^{(2)}$ each have two elements, with each element contributing to one of the objectives.

Our specific construction is as follows. With $d_{QK,23} = P + 1$,

$$
\boldsymbol{Q}_3^{(2)} = \begin{pmatrix} \boldsymbol{0}_{P \times 34} & \boldsymbol{0}_{P \times P} & \sqrt{M}\boldsymbol{I}_P & \boldsymbol{0}_{P \times P} & \boldsymbol{0}_{P \times P} & \boldsymbol{0}_{P \times P} & \boldsymbol{0}_{P \times P} \\ \sqrt{MP}\left(\boldsymbol{e}_{\text{FULL\_ONES}}^{34}\right)^{\top} & \boldsymbol{0}_{1 \times P} & \boldsymbol{0}_{1 \times P} & \boldsymbol{0}_{1 \times P} & \boldsymbol{0}_{1 \times P} & \boldsymbol{0}_{1 \times P} & \boldsymbol{0}_{1 \times P} \end{pmatrix} \in \mathbb{R}^{d_{QK,23} \times d},
\tag{106}
$$

$$
\boldsymbol{K}_3^{(2)} = \begin{pmatrix} \boldsymbol{0}_{P \times 34} & \sqrt{M}\boldsymbol{I}_P & \boldsymbol{0}_{P \times P} & \boldsymbol{0}_{P \times P} & \boldsymbol{0}_{P \times P} & \boldsymbol{0}_{P \times P} & \boldsymbol{0}_{P \times P} \\ \sqrt{MP}\left(\boldsymbol{e}_{\text{IS\_BOS}}^{34}\right)^{\top} & \boldsymbol{0}_{1 \times P} & \boldsymbol{0}_{1 \times P} & \boldsymbol{0}_{1 \times P} & \boldsymbol{0}_{1 \times P} & \boldsymbol{0}_{1 \times P} & \boldsymbol{0}_{1 \times P} \end{pmatrix} \in \mathbb{R}^{d_{QK,23} \times d}.
\tag{107}
$$

and with $d_{V,23} = 2$,

$$
\boldsymbol{V}_3^{(2)} = \begin{pmatrix} 2\left(\boldsymbol{e}_{\text{NUM}}^{d}\right)^{\top} \\ \left(\boldsymbol{e}_{\text{IS\_BOS}}^{d}\right)^{\top} \end{pmatrix} \in \mathbb{R}^{d_{V,23} \times d},
\tag{108}
$$

$$
\boldsymbol{U}_3^{(2)} = \begin{pmatrix} \boldsymbol{e}_{\text{OP1\_SHIFT0}}^{d} & \boldsymbol{e}_{\text{PRE\_EOS1}}^{d} \end{pmatrix} \in \mathbb{R}^{d \times d_{V,23}}.
\tag{109}
$$

We provide the examples in Tables 60 to 64. We note that within the dimension PRE_EOS1 of the matrix $\boldsymbol{U}_3^{(2)}\boldsymbol{V}_3^{(2)}\boldsymbol{X}^{(1)}\boldsymbol{A}_3^{(2)}$, if we restrict our view to the equal symbol $=$ and the response sequence, 1 is only assigned to the first, second, and third most significant digits of the response (regardless of the query length).

Table 60: Example of $\boldsymbol{Q}_3^{(2)}\boldsymbol{X}^{(1)}$, continuing from Table 49.

| $\mathcal{I}$ | \$ | 7 | 5 | 9 | 5 | $\times$ | 7 | 9 | $=$ | 5 | 0 | 0 | 0 | 0 | 6 |
|---|---|---|---|---|---|---|---|---|---|---|---|---|---|---|---|
| $1$–$P$: | $\boldsymbol{0}_P$ | $\sqrt{M}\boldsymbol{v}_3^P$ | $\sqrt{M}\boldsymbol{v}_4^P$ | $\sqrt{M}\boldsymbol{v}_5^P$ | $\sqrt{M}\boldsymbol{v}_6^P$ | $\sqrt{M}\boldsymbol{v}_7^P$ | $\sqrt{M}\boldsymbol{v}_5^P$ | $\sqrt{M}\boldsymbol{v}_6^P$ | $\sqrt{M}\boldsymbol{v}_7^P$ | $\sqrt{M}\boldsymbol{v}_6^P$ | $\sqrt{M}\boldsymbol{v}_5^P$ | $\sqrt{M}\boldsymbol{v}_4^P$ | $\sqrt{M}\boldsymbol{v}_3^P$ | $\sqrt{M}\boldsymbol{v}_2^P$ | $\sqrt{M}\boldsymbol{v}_1^P$ |
| $P+1$: | $\sqrt{MP}$ | $\sqrt{MP}$ | $\sqrt{MP}$ | $\sqrt{MP}$ | $\sqrt{MP}$ | $\sqrt{MP}$ | $\sqrt{MP}$ | $\sqrt{MP}$ | $\sqrt{MP}$ | $\sqrt{MP}$ | $\sqrt{MP}$ | $\sqrt{MP}$ | $\sqrt{MP}$ | $\sqrt{MP}$ | $\sqrt{MP}$ |

Table 61: Example of $\boldsymbol{K}_3^{(2)}\boldsymbol{X}^{(1)}$, continuing from Table 49.

| $\mathcal{I}$ | \$ | 7 | 5 | 9 | 5 | $\times$ | 7 | 9 | $=$ | 5 | 0 | 0 | 0 | 0 | 6 |
|---|---|---|---|---|---|---|---|---|---|---|---|---|---|---|---|
| $1$–$P$: | $\boldsymbol{0}_P$ | $\sqrt{M}\boldsymbol{v}_4^P$ | $\sqrt{M}\boldsymbol{v}_5^P$ | $\sqrt{M}\boldsymbol{v}_6^P$ | $\sqrt{M}\boldsymbol{v}_7^P$ | $\boldsymbol{0}_P$ | $\boldsymbol{0}_P$ | $\boldsymbol{0}_P$ | $\boldsymbol{0}_P$ | $\boldsymbol{0}_P$ | $\boldsymbol{0}_P$ | $\boldsymbol{0}_P$ | $\boldsymbol{0}_P$ | $\boldsymbol{0}_P$ | $\boldsymbol{0}_P$ |
| $P+1$: | $\sqrt{MP}$ | 0 | 0 | 0 | 0 | 0 | 0 | 0 | 0 | 0 | 0 | 0 | 0 | 0 | 0 |

Table 62: Example of $\boldsymbol{A}_3^{(2)}$ (with explicit row/column indices and sufficiently large $M$), continuing from Tables 60 and 61.

| row \ col | $j=1$ | 2 | 3 | 4 | 5 | 6 | 7 | 8 | 9 | 10 | 11 | 12 | 13 | 14 | 15 |
|---|---|---|---|---|---|---|---|---|---|---|---|---|---|---|---|
| $i=1$ | 1 | 1 | 1/2 | 1/2 | 1/2 | 1/2 | 1/2 | 1/2 | 1/2 | 1/2 | 1/2 | 1/2 | 1 | 1 | 1 |
| 2 | 0 | 0 | 1/2 | 0 | 0 | 0 | 0 | 0 | 0 | 0 | 0 | 1/2 | 0 | 0 | 0 |
| 3 | 0 | 0 | 0 | 1/2 | 0 | 0 | 1/2 | 0 | 0 | 0 | 1/2 | 0 | 0 | 0 | 0 |
| 4 | 0 | 0 | 0 | 0 | 1/2 | 0 | 0 | 1/2 | 0 | 1/2 | 0 | 0 | 0 | 0 | 0 |
| 5 | 0 | 0 | 0 | 0 | 0 | 1/2 | 0 | 0 | 1/2 | 0 | 0 | 0 | 0 | 0 | 0 |
| 6 | 0 | 0 | 0 | 0 | 0 | 0 | 0 | 0 | 0 | 0 | 0 | 0 | 0 | 0 | 0 |
| 7 | 0 | 0 | 0 | 0 | 0 | 0 | 0 | 0 | 0 | 0 | 0 | 0 | 0 | 0 | 0 |
| 8 | 0 | 0 | 0 | 0 | 0 | 0 | 0 | 0 | 0 | 0 | 0 | 0 | 0 | 0 | 0 |
| 9 | 0 | 0 | 0 | 0 | 0 | 0 | 0 | 0 | 0 | 0 | 0 | 0 | 0 | 0 | 0 |
| 10 | 0 | 0 | 0 | 0 | 0 | 0 | 0 | 0 | 0 | 0 | 0 | 0 | 0 | 0 | 0 |
| 11 | 0 | 0 | 0 | 0 | 0 | 0 | 0 | 0 | 0 | 0 | 0 | 0 | 0 | 0 | 0 |
| 12 | 0 | 0 | 0 | 0 | 0 | 0 | 0 | 0 | 0 | 0 | 0 | 0 | 0 | 0 | 0 |
| 13 | 0 | 0 | 0 | 0 | 0 | 0 | 0 | 0 | 0 | 0 | 0 | 0 | 0 | 0 | 0 |
| 14 | 0 | 0 | 0 | 0 | 0 | 0 | 0 | 0 | 0 | 0 | 0 | 0 | 0 | 0 | 0 |
| 15 | 0 | 0 | 0 | 0 | 0 | 0 | 0 | 0 | 0 | 0 | 0 | 0 | 0 | 0 | 0 |

Table 63: Example of $\boldsymbol{U}_3^{(2)}\boldsymbol{V}_3^{(2)}\boldsymbol{X}^{(1)}$, continuing from Table 49. (Irrelevant dimensions are omitted for readability)

| $\mathcal{I}$ | $ | 7 | 5 | 9 | 5 | $\times$ | 7 | 9 | = | 5 | 0 | 0 | 0 | 0 | 6 |
|---|---|---|---|---|---|---|---|---|---|---|---|---|---|---|---|
| 1: NUM | 0 | 0 | 0 | 0 | 0 | 0 | 0 | 0 | 0 | 0 | 0 | 0 | 0 | 0 | 0 |
| 2: FULL_ONES | 0 | 0 | 0 | 0 | 0 | 0 | 0 | 0 | 0 | 0 | 0 | 0 | 0 | 0 | 0 |
| 3: IS_BOS | 0 | 0 | 0 | 0 | 0 | 0 | 0 | 0 | 0 | 0 | 0 | 0 | 0 | 0 | 0 |
| 4: IS_MUL | 0 | 0 | 0 | 0 | 0 | 0 | 0 | 0 | 0 | 0 | 0 | 0 | 0 | 0 | 0 |
| 5: IS_EQUAL | 0 | 0 | 0 | 0 | 0 | 0 | 0 | 0 | 0 | 0 | 0 | 0 | 0 | 0 | 0 |
| 10: OP1_SHIFT0 | 0 | 14 | 10 | 18 | 10 | 0 | 14 | 18 | 0 | 10 | 0 | 0 | 0 | 0 | 12 |
| 21: PRE_EOS1 | 1 | 0 | 0 | 0 | 0 | 0 | 0 | 0 | 0 | 0 | 0 | 0 | 0 | 0 | 0 |

Table 64: Example of $\boldsymbol{U}_3^{(2)}\boldsymbol{V}_3^{(2)}\boldsymbol{X}^{(1)}\boldsymbol{A}_3^{(2)}$, continuing from Tables 62 and 63. (Irrelevant dimensions are omitted for readability)

| $\mathcal{I}$ | $ | 7 | 5 | 9 | 5 | $\times$ | 7 | 9 | = | 5 | 0 | 0 | 0 | 0 | 6 |
|---|---|---|---|---|---|---|---|---|---|---|---|---|---|---|---|
| 1: NUM | 0 | 0 | 0 | 0 | 0 | 0 | 0 | 0 | 0 | 0 | 0 | 0 | 0 | 0 | 0 |
| 2: FULL_ONES | 0 | 0 | 0 | 0 | 0 | 0 | 0 | 0 | 0 | 0 | 0 | 0 | 0 | 0 | 0 |
| 3: IS_BOS | 0 | 0 | 0 | 0 | 0 | 0 | 0 | 0 | 0 | 0 | 0 | 0 | 0 | 0 | 0 |
| 4: IS_MUL | 0 | 0 | 0 | 0 | 0 | 0 | 0 | 0 | 0 | 0 | 0 | 0 | 0 | 0 | 0 |
| 5: IS_EQUAL | 0 | 0 | 0 | 0 | 0 | 0 | 0 | 0 | 0 | 0 | 0 | 0 | 0 | 0 | 0 |
| 10: OP1_SHIFT0 | 0 | 0 | 7 | 5 | 9 | 5 | 5 | 9 | 5 | 9 | 5 | 7 | 0 | 0 | 0 |
| 21: PRE_EOS1 | 1 | 1 | 1/2 | 1/2 | 1/2 | 1/2 | 1/2 | 1/2 | 1/2 | 1/2 | 1/2 | 1/2 | 1 | 1 | 1 |

### G.7.4 Attention Head 4: Copying the Appropriate Digit from the First Operand II

The objective of Attention head 4 is to fill the dimension OP1_SHIFT1 at the $i$-th least significant digit of the response (when predicting the $(i+1)$-th least significant digit of the response) with the $i$-th least significant digit of the first operand. Similarly to the previous head, $i$ ranges from $0$ to $\ell_a + 2$. In cases where the $i$-th least significant digit of the first operand is not well-defined (i.e., $i \in \{0, \ell_a + 1, \ell_a + 2\}$), we assign $0$.

The design of Attention head 4 is as follows. With $d_{QK,24} = P + 1$,

$$Q_4^{(2)} = \begin{pmatrix} \mathbf{0}_{P\times 34} & \mathbf{0}_{P\times P} & \mathbf{0}_{P\times P} & \sqrt{M}\mathbf{I}_P & \mathbf{0}_{P\times P} & \mathbf{0}_{P\times P} & \mathbf{0}_{P\times P} \\ \sqrt{MP}\left(e_{\text{FULL\_ONES}}^{34}\right)^\top & \mathbf{0}_{1\times P} & \mathbf{0}_{1\times P} & \mathbf{0}_{1\times P} & \mathbf{0}_{1\times P} & \mathbf{0}_{1\times P} & \mathbf{0}_{1\times P} \end{pmatrix} \in \mathbb{R}^{d_{QK,24}\times d},$$

(110)

$$K_4^{(2)} = \begin{pmatrix} \mathbf{0}_{P\times 34} & \sqrt{M}\mathbf{I}_P & \mathbf{0}_{P\times P} & \mathbf{0}_{P\times P} & \mathbf{0}_{P\times P} & \mathbf{0}_{P\times P} & \mathbf{0}_{P\times P} \\ \sqrt{MP}\left(e_{\text{IS\_BOS}}^{34}\right)^\top & \mathbf{0}_{1\times P} & \mathbf{0}_{1\times P} & \mathbf{0}_{1\times P} & \mathbf{0}_{1\times P} & \mathbf{0}_{1\times P} & \mathbf{0}_{1\times P} \end{pmatrix} \in \mathbb{R}^{d_{QK,24}\times d},$$

(111)

and with $d_{V,24} = 1$,

$$V_4^{(2)} = 2(e_{\text{NUM}}^d)^\top \in \mathbb{R}^{d_{V,24}\times d},$$

(112)

$$U_4^{(2)} = e_{\text{OP1\_SHIFT1}}^d \in \mathbb{R}^{d\times d_{V,24}}.$$

(113)

We provide the examples in Tables 65 to 69.

Table 65: Example of $Q_4^{(2)}X^{(1)}$, continuing from Table 49.

| $\mathcal{I}$ | \$ | 7 | 5 | 9 | 5 | $\times$ | 7 | 9 | = | 5 | 0 | 0 | 0 | 0 | 6 |
|---|---|---|---|---|---|---|---|---|---|---|---|---|---|---|---|
| 1–$P$: | $\mathbf{0}_P$ | $\sqrt{M}v_4^P$ | $\sqrt{M}v_5^P$ | $\sqrt{M}v_6^P$ | $\sqrt{M}v_7^P$ | $\sqrt{M}v_8^P$ | $\sqrt{M}v_6^P$ | $\sqrt{M}v_7^P$ | $\sqrt{M}v_8^P$ | $\sqrt{M}v_7^P$ | $\sqrt{M}v_6^P$ | $\sqrt{M}v_5^P$ | $\sqrt{M}v_4^P$ | $\sqrt{M}v_3^P$ | $\sqrt{M}v_2^P$ |
| $P+1$: | $\sqrt{MP}$ | $\sqrt{MP}$ | $\sqrt{MP}$ | $\sqrt{MP}$ | $\sqrt{MP}$ | $\sqrt{MP}$ | $\sqrt{MP}$ | $\sqrt{MP}$ | $\sqrt{MP}$ | $\sqrt{MP}$ | $\sqrt{MP}$ | $\sqrt{MP}$ | $\sqrt{MP}$ | $\sqrt{MP}$ | $\sqrt{MP}$ |

Table 66: Example of $K_4^{(2)}X^{(1)}$, continuing from Table 49.

| $\mathcal{I}$ | \$ | 7 | 5 | 9 | 5 | $\times$ | 7 | 9 | = | 5 | 0 | 0 | 0 | 0 | 6 |
|---|---|---|---|---|---|---|---|---|---|---|---|---|---|---|---|
| 1–$P$: | $\mathbf{0}_P$ | $\sqrt{M}v_4^P$ | $\sqrt{M}v_5^P$ | $\sqrt{M}v_6^P$ | $\sqrt{M}v_7^P$ | $\mathbf{0}_P$ | $\mathbf{0}_P$ | $\mathbf{0}_P$ | $\mathbf{0}_P$ | $\mathbf{0}_P$ | $\mathbf{0}_P$ | $\mathbf{0}_P$ | $\mathbf{0}_P$ | $\mathbf{0}_P$ | $\mathbf{0}_P$ |
| $P+1$: | $\sqrt{MP}$ | 0 | 0 | 0 | 0 | 0 | 0 | 0 | 0 | 0 | 0 | 0 | 0 | 0 | 0 |

Table 67: Example of $A_4^{(2)}$ (with explicit row/column indices and sufficiently large $M$), continuing from Tables 65 and 66.

| row \ col | $j=1$ | 2 | 3 | 4 | 5 | 6 | 7 | 8 | 9 | 10 | 11 | 12 | 13 | 14 | 15 |
|---|---|---|---|---|---|---|---|---|---|---|---|---|---|---|---|
| $i=1$ | 1 | 1/2 | 1/2 | 1/2 | 1/2 | 1 | 1/2 | 1/2 | 1 | 1/2 | 1/2 | 1/2 | 1/2 | 1 | 1 |
| 2 | 0 | 1/2 | 0 | 0 | 0 | 0 | 0 | 0 | 0 | 0 | 0 | 0 | 1/2 | 0 | 0 |
| 3 | 0 | 0 | 1/2 | 0 | 0 | 0 | 0 | 0 | 0 | 0 | 0 | 1/2 | 0 | 0 | 0 |
| 4 | 0 | 0 | 0 | 1/2 | 0 | 0 | 1/2 | 0 | 0 | 0 | 1/2 | 0 | 0 | 0 | 0 |
| 5 | 0 | 0 | 0 | 0 | 1/2 | 0 | 0 | 1/2 | 0 | 1/2 | 0 | 0 | 0 | 0 | 0 |
| 6 | 0 | 0 | 0 | 0 | 0 | 0 | 0 | 0 | 0 | 0 | 0 | 0 | 0 | 0 | 0 |
| 7 | 0 | 0 | 0 | 0 | 0 | 0 | 0 | 0 | 0 | 0 | 0 | 0 | 0 | 0 | 0 |
| 8 | 0 | 0 | 0 | 0 | 0 | 0 | 0 | 0 | 0 | 0 | 0 | 0 | 0 | 0 | 0 |
| 9 | 0 | 0 | 0 | 0 | 0 | 0 | 0 | 0 | 0 | 0 | 0 | 0 | 0 | 0 | 0 |
| 10 | 0 | 0 | 0 | 0 | 0 | 0 | 0 | 0 | 0 | 0 | 0 | 0 | 0 | 0 | 0 |
| 11 | 0 | 0 | 0 | 0 | 0 | 0 | 0 | 0 | 0 | 0 | 0 | 0 | 0 | 0 | 0 |
| 12 | 0 | 0 | 0 | 0 | 0 | 0 | 0 | 0 | 0 | 0 | 0 | 0 | 0 | 0 | 0 |
| 13 | 0 | 0 | 0 | 0 | 0 | 0 | 0 | 0 | 0 | 0 | 0 | 0 | 0 | 0 | 0 |
| 14 | 0 | 0 | 0 | 0 | 0 | 0 | 0 | 0 | 0 | 0 | 0 | 0 | 0 | 0 | 0 |
| 15 | 0 | 0 | 0 | 0 | 0 | 0 | 0 | 0 | 0 | 0 | 0 | 0 | 0 | 0 | 0 |

Table 68: Example of $\boldsymbol{U}_4^{(2)}\boldsymbol{V}_4^{(2)}\boldsymbol{X}^{(1)}$, continuing from Table 49. (Irrelevant dimensions are omitted for readability)

| $\mathcal{I}$ | $ | 7 | 5 | 9 | 5 | × | 7 | 9 | = | 5 | 0 | 0 | 0 | 0 | 6 |
|---|---|---|---|---|---|---|---|---|---|---|---|---|---|---|---|
| 1: NUM | 0 | 0 | 0 | 0 | 0 | 0 | 0 | 0 | 0 | 0 | 0 | 0 | 0 | 0 | 0 |
| 2: FULL_ONES | 0 | 0 | 0 | 0 | 0 | 0 | 0 | 0 | 0 | 0 | 0 | 0 | 0 | 0 | 0 |
| 3: IS_MUL | 0 | 0 | 0 | 0 | 0 | 0 | 0 | 0 | 0 | 0 | 0 | 0 | 0 | 0 | 0 |
| 4: IS_EQUAL | 0 | 0 | 0 | 0 | 0 | 0 | 0 | 0 | 0 | 0 | 0 | 0 | 0 | 0 | 0 |
| 11: OP1_SHIFT1 | 0 | 14 | 10 | 18 | 10 | 0 | 14 | 18 | 0 | 10 | 0 | 0 | 0 | 0 | 12 |

Table 69: Example of $\boldsymbol{U}_4^{(2)}\boldsymbol{V}_4^{(2)}\boldsymbol{X}^{(1)}\boldsymbol{A}_4^{(2)}$, continuing from Tables 67 and 68. (Irrelevant dimensions are omitted for readability)

| $\mathcal{I}$ | $ | 7 | 5 | 9 | 5 | × | 7 | 9 | = | 5 | 0 | 0 | 0 | 0 | 6 |
|---|---|---|---|---|---|---|---|---|---|---|---|---|---|---|---|
| 1: NUM | 0 | 0 | 0 | 0 | 0 | 0 | 0 | 0 | 0 | 0 | 0 | 0 | 0 | 0 | 0 |
| 2: FULL_ONES | 0 | 0 | 0 | 0 | 0 | 0 | 0 | 0 | 0 | 0 | 0 | 0 | 0 | 0 | 0 |
| 3: IS_MUL | 0 | 0 | 0 | 0 | 0 | 0 | 0 | 0 | 0 | 0 | 0 | 0 | 0 | 0 | 0 |
| 4: IS_EQUAL | 0 | 0 | 0 | 0 | 0 | 0 | 0 | 0 | 0 | 0 | 0 | 0 | 0 | 0 | 0 |
| 11: OP1_SHIFT1 | 0 | 7 | 5 | 9 | 5 | 0 | 9 | 5 | 0 | 5 | 9 | 5 | 7 | 0 | 0 |

### G.7.5 Attention Head 5: Copying the Appropriate Digit from the First Operand III

The main objective of Attention head 5 is to fill the dimension OP1_SHIFT2 at the $i$-th least significant digit of the response (when predicting the $(i+1)$-th least significant digit of the response) with the $(i-1)$-th least significant digit of the first operand. Similarly to the previous head, $i$ ranges from 0 to $\ell_a + 2$, and in cases where the $i$-th least significant digit of the first operand is not well-defined (i.e., $i \in \{0, 1, \ell_a + 2\}$), we assign 0.

As mentioned in Attention head 3, we assign an extra goal to Attention head 5, which is to fill the dimension PRE_EOS2.

The design of the fifth head is as follows. With $d_{QK,25} = P + 1$,

$$\boldsymbol{Q}_5^{(2)} = \begin{pmatrix} \boldsymbol{0}_{P \times 34} & \boldsymbol{0}_{P \times P} & \boldsymbol{0}_{P \times P} & \boldsymbol{0}_{P \times P} & \sqrt{M}\boldsymbol{I}_P & \boldsymbol{0}_{P \times P} & \boldsymbol{0}_{P \times P} \\ \sqrt{MP}\left(\boldsymbol{e}_{\text{FULL\_ONES}}^{34}\right)^\top & \boldsymbol{0}_{1 \times P} & \boldsymbol{0}_{1 \times P} & \boldsymbol{0}_{1 \times P} & \boldsymbol{0}_{1 \times P} & \boldsymbol{0}_{1 \times P} & \boldsymbol{0}_{1 \times P} \end{pmatrix} \in \mathbb{R}^{d_{QK,25} \times d}, \tag{114}$$

$$\boldsymbol{K}_5^{(2)} = \begin{pmatrix} \boldsymbol{0}_{P \times 34} & \sqrt{M}\boldsymbol{I}_P & \boldsymbol{0}_{P \times P} & \boldsymbol{0}_{P \times P} & \boldsymbol{0}_{P \times P} & \boldsymbol{0}_{P \times P} & \boldsymbol{0}_{P \times P} \\ \sqrt{MP}\left(\boldsymbol{e}_{\text{IS\_BOS}}^{34}\right)^\top & \boldsymbol{0}_{1 \times P} & \boldsymbol{0}_{1 \times P} & \boldsymbol{0}_{1 \times P} & \boldsymbol{0}_{1 \times P} & \boldsymbol{0}_{1 \times P} & \boldsymbol{0}_{1 \times P} \end{pmatrix} \in \mathbb{R}^{d_{QK,25} \times d}, \tag{115}$$

and with $d_{V,25} = 2$,

$$\boldsymbol{V}_5^{(2)} = \begin{pmatrix} 2(\boldsymbol{e}_{\text{NUM}}^d)^\top \\ (\boldsymbol{e}_{\text{IS\_BOS}}^d)^\top \end{pmatrix} \in \mathbb{R}^{d_{V,25} \times d}, \tag{116}$$

$$\boldsymbol{U}_5^{(2)} = \begin{pmatrix} \boldsymbol{e}_{\text{OP1\_SHIFT2}}^d & \boldsymbol{e}_{\text{PRE\_EOS2}}^d \end{pmatrix} \in \mathbb{R}^{d \times d_{V,25}}. \tag{117}$$

We provide the examples in Tables 70 to 74. Note that within the dimension PRE_EOS2 of the matrix $\boldsymbol{U}_5^{(2)}\boldsymbol{V}_5^{(2)}\boldsymbol{X}^{(1)}\boldsymbol{A}_5^{(2)}$, if we restrict our view to the equal symbol $=$ and the response sequence, 1 is only assigned to the most and the least significant digit of the response, and the equal token. An important observation is that upon comparing PRE_EOS1 and PRE_EOS2, the most significant digit of the response is the only token that has a value of 1 in both dimensions. This observation plays a crucial role in predicting EOS for the next token, and we will elaborate further in the later section discussing the FFN layer.

#### Table 70: Example of $\boldsymbol{Q}_5^{(2)}\boldsymbol{X}^{(1)}$, continuing from Table 49.

| $\mathcal{I}$ | \$ | 7 | 5 | 9 | 5 | $\times$ | 7 | 9 | = | 5 | 0 | 0 | 0 | 0 | 6 |
|---|---|---|---|---|---|---|---|---|---|---|---|---|---|---|---|
| 1–P: | $\mathbf{0}_P$ | $\sqrt{M}\boldsymbol{v}_5^P$ | $\sqrt{M}\boldsymbol{v}_6^P$ | $\sqrt{M}\boldsymbol{v}_7^P$ | $\sqrt{M}\boldsymbol{v}_8^P$ | $\sqrt{M}\boldsymbol{v}_9^P$ | $\sqrt{M}\boldsymbol{v}_7^P$ | $\sqrt{M}\boldsymbol{v}_8^P$ | $\sqrt{M}\boldsymbol{v}_9^P$ | $\sqrt{M}\boldsymbol{v}_8^P$ | $\sqrt{M}\boldsymbol{v}_7^P$ | $\sqrt{M}\boldsymbol{v}_6^P$ | $\sqrt{M}\boldsymbol{v}_5^P$ | $\sqrt{M}\boldsymbol{v}_4^P$ | $\sqrt{M}\boldsymbol{v}_3^P$ |
| $P+1$: | $\sqrt{MP}$ | $\sqrt{MP}$ | $\sqrt{MP}$ | $\sqrt{MP}$ | $\sqrt{MP}$ | $\sqrt{MP}$ | $\sqrt{MP}$ | $\sqrt{MP}$ | $\sqrt{MP}$ | $\sqrt{MP}$ | $\sqrt{MP}$ | $\sqrt{MP}$ | $\sqrt{MP}$ | $\sqrt{MP}$ | $\sqrt{MP}$ |

#### Table 71: Example of $\boldsymbol{K}_5^{(2)}\boldsymbol{X}^{(1)}$, continuing from Table 49.

| $\mathcal{I}$ | \$ | 7 | 5 | 9 | 5 | $\times$ | 7 | 9 | = | 5 | 0 | 0 | 0 | 0 | 6 |
|---|---|---|---|---|---|---|---|---|---|---|---|---|---|---|---|
| 1–P: | $\mathbf{0}_P$ | $\sqrt{M}\boldsymbol{v}_4^P$ | $\sqrt{M}\boldsymbol{v}_5^P$ | $\sqrt{M}\boldsymbol{v}_6^P$ | $\sqrt{M}\boldsymbol{v}_7^P$ | $\mathbf{0}_P$ | $\mathbf{0}_P$ | $\mathbf{0}_P$ | $\mathbf{0}_P$ | $\mathbf{0}_P$ | $\mathbf{0}_P$ | $\mathbf{0}_P$ | $\mathbf{0}_P$ | $\mathbf{0}_P$ | $\mathbf{0}_P$ |
| $P+1$: | $\sqrt{MP}$ | 0 | 0 | 0 | 0 | 0 | 0 | 0 | 0 | 0 | 0 | 0 | 0 | 0 | 0 |

#### Table 72: Example of $\boldsymbol{A}_5^{(2)}$ (with explicit row/column indices and sufficiently large $M$), continuing from Tables 70 and 71.

| row \ col | $j=1$ | 2 | 3 | 4 | 5 | 6 | 7 | 8 | 9 | 10 | 11 | 12 | 13 | 14 | 15 |
|---|---|---|---|---|---|---|---|---|---|---|---|---|---|---|---|
| $i=1$ | 1 | 1 | 1 | 1 | 1 | 1 | 1/2 | 1 | 1 | 1 | 1/2 | 1/2 | 1/2 | 1/2 | 1 |
| 2 | 0 | 0 | 0 | 0 | 0 | 0 | 0 | 0 | 0 | 0 | 0 | 0 | 0 | 1/2 | 0 |
| 3 | 0 | 0 | 0 | 0 | 0 | 0 | 0 | 0 | 0 | 0 | 0 | 0 | 1/2 | 0 | 0 |
| 4 | 0 | 0 | 0 | 0 | 0 | 0 | 0 | 0 | 0 | 0 | 0 | 1/2 | 0 | 0 | 0 |
| 5 | 0 | 0 | 0 | 0 | 0 | 0 | 1/2 | 0 | 0 | 0 | 1/2 | 0 | 0 | 0 | 0 |
| 6 | 0 | 0 | 0 | 0 | 0 | 0 | 0 | 0 | 0 | 0 | 0 | 0 | 0 | 0 | 0 |
| 7 | 0 | 0 | 0 | 0 | 0 | 0 | 0 | 0 | 0 | 0 | 0 | 0 | 0 | 0 | 0 |
| 8 | 0 | 0 | 0 | 0 | 0 | 0 | 0 | 0 | 0 | 0 | 0 | 0 | 0 | 0 | 0 |
| 9 | 0 | 0 | 0 | 0 | 0 | 0 | 0 | 0 | 0 | 0 | 0 | 0 | 0 | 0 | 0 |
| 10 | 0 | 0 | 0 | 0 | 0 | 0 | 0 | 0 | 0 | 0 | 0 | 0 | 0 | 0 | 0 |
| 11 | 0 | 0 | 0 | 0 | 0 | 0 | 0 | 0 | 0 | 0 | 0 | 0 | 0 | 0 | 0 |
| 12 | 0 | 0 | 0 | 0 | 0 | 0 | 0 | 0 | 0 | 0 | 0 | 0 | 0 | 0 | 0 |
| 13 | 0 | 0 | 0 | 0 | 0 | 0 | 0 | 0 | 0 | 0 | 0 | 0 | 0 | 0 | 0 |
| 14 | 0 | 0 | 0 | 0 | 0 | 0 | 0 | 0 | 0 | 0 | 0 | 0 | 0 | 0 | 0 |
| 15 | 0 | 0 | 0 | 0 | 0 | 0 | 0 | 0 | 0 | 0 | 0 | 0 | 0 | 0 | 0 |

#### Table 73: Example of $\boldsymbol{U}_5^{(2)}\boldsymbol{V}_5^{(2)}\boldsymbol{X}^{(1)}$, continuing from Table 49. (Irrelevant dimensions are omitted for readability)

| $\mathcal{I}$ | \$ | 7 | 5 | 9 | 5 | $\times$ | 7 | 9 | = | 5 | 0 | 0 | 0 | 0 | 6 |
|---|---|---|---|---|---|---|---|---|---|---|---|---|---|---|---|
| 1: NUM | 0 | 0 | 0 | 0 | 0 | 0 | 0 | 0 | 0 | 0 | 0 | 0 | 0 | 0 | 0 |
| 2: FULL_ONES | 0 | 0 | 0 | 0 | 0 | 0 | 0 | 0 | 0 | 0 | 0 | 0 | 0 | 0 | 0 |
| 3: IS_MUL | 0 | 0 | 0 | 0 | 0 | 0 | 0 | 0 | 0 | 0 | 0 | 0 | 0 | 0 | 0 |
| 4: IS_EQUAL | 0 | 0 | 0 | 0 | 0 | 0 | 0 | 0 | 0 | 0 | 0 | 0 | 0 | 0 | 0 |
| 12: OP1_SHIFT2 | 0 | 14 | 10 | 18 | 10 | 0 | 14 | 18 | 0 | 10 | 0 | 0 | 0 | 0 | 12 |
| 22: PRE_EOS2 | 1 | 0 | 0 | 0 | 0 | 0 | 0 | 0 | 0 | 0 | 0 | 0 | 0 | 0 | 0 |

Table 74: Example of $\boldsymbol{U}_5^{(2)}\boldsymbol{V}_5^{(2)}\boldsymbol{X}^{(1)}\boldsymbol{A}_5^{(2)}$, continuing from Tables 72 and 73. (Irrelevant dimensions are omitted for readability)

| $\mathcal{I}$ | \$ | 7 | 5 | 9 | 5 | $\times$ | 7 | 9 | = | 5 | 0 | 0 | 0 | 0 | 6 |
|---|---|---|---|---|---|---|---|---|---|---|---|---|---|---|---|
| 1: NUM | 0 | 0 | 0 | 0 | 0 | 0 | 0 | 0 | 0 | 0 | 0 | 0 | 0 | 0 | 0 |
| 2: FULL_ONES | 0 | 0 | 0 | 0 | 0 | 0 | 0 | 0 | 0 | 0 | 0 | 0 | 0 | 0 | 0 |
| 3: IS_MUL | 0 | 0 | 0 | 0 | 0 | 0 | 0 | 0 | 0 | 0 | 0 | 0 | 0 | 0 | 0 |
| 4: IS_EQUAL | 0 | 0 | 0 | 0 | 0 | 0 | 0 | 0 | 0 | 0 | 0 | 0 | 0 | 0 | 0 |
| 12: OP1_SHIFT2 | 0 | 0 | 0 | 0 | 0 | 0 | 5 | 0 | 0 | 0 | 5 | 9 | 5 | 7 | 0 |
| 22: PRE_EOS2 | 1 | 1 | 1 | 1 | 1 | 1 | 1/2 | 1 | 1 | 1 | 1/2 | 1/2 | 1/2 | 1/2 | 1 |

### G.7.6 Attention Head 6: Copying the Appropriate Digit from the First Operand IV

The objective of Attention head 6 is to fill the dimension OP1_SHIFT3 at the $i$-th least significant digit of the response (when predicting the $(i+1)$-th least significant digit of the response) with the $(i-2)$-th least significant digit of the first operand. Similarly to the previous head, $i$ ranges from $0$ to $\ell_a + 2$. In cases where the $i$-th least significant digit of the first operand is not well-defined (i.e., $i \in \{0, 1, 2\}$), we assign $0$.

The design of Attention head 6 is as follows. With $d_{QK,26} = P + 1$,

$$\boldsymbol{Q}_6^{(2)} = \begin{pmatrix} \boldsymbol{0}_{P\times 34} & \boldsymbol{0}_{P\times P} & \boldsymbol{0}_{P\times P} & \boldsymbol{0}_{P\times P} & \boldsymbol{0}_{P\times P} & \sqrt{M}\boldsymbol{I}_P & \boldsymbol{0}_{P\times P} \\ \sqrt{MP}\left(\boldsymbol{e}_{\text{FULL\_ONES}}^{34}\right)^\top & \boldsymbol{0}_{P\times P} & \boldsymbol{0}_{1\times P} & \boldsymbol{0}_{1\times P} & \boldsymbol{0}_{1\times P} & \boldsymbol{0}_{1\times P} & \boldsymbol{0}_{1\times P} \end{pmatrix} \in \mathbb{R}^{d_{QK,26}\times d},$$
(118)

$$\boldsymbol{K}_6^{(2)} = \begin{pmatrix} \boldsymbol{0}_{P\times 34} & \sqrt{M}\boldsymbol{I}_P & \boldsymbol{0}_{P\times P} & \boldsymbol{0}_{P\times P} & \boldsymbol{0}_{P\times P} & \boldsymbol{0}_{P\times P} & \boldsymbol{0}_{P\times P} \\ \sqrt{MP}\left(\boldsymbol{e}_{\text{IS\_BOS}}^{34}\right)^\top & \boldsymbol{0}_{1\times P} & \boldsymbol{0}_{1\times P} & \boldsymbol{0}_{1\times P} & \boldsymbol{0}_{1\times P} & \boldsymbol{0}_{1\times P} & \boldsymbol{0}_{1\times P} \end{pmatrix} \in \mathbb{R}^{d_{QK,26}\times d}.$$
(119)

With $d_{V,26} = 1$,

$$\boldsymbol{V}_6^{(2)} = 2(\boldsymbol{e}_{\text{NUM}}^d)^\top \in \mathbb{R}^{d_{V,26}\times d},$$
(120)

$$\boldsymbol{U}_6^{(2)} = \boldsymbol{e}_{\text{OP1\_SHIFT3}}^d \in \mathbb{R}^{d\times d_{V,26}}.$$
(121)

We provide the examples in Tables 75 to 79.

Table 75: Example of $\boldsymbol{Q}_6^{(2)}\boldsymbol{X}^{(1)}$, continuing from Table 49.

| $\mathcal{I}$ | \$ | 7 | 5 | 9 | 5 | $\times$ | 7 | 9 | = | 5 | 0 | 0 | 0 | 0 | 6 |
|---|---|---|---|---|---|---|---|---|---|---|---|---|---|---|---|
| $1$–$P$: | $\boldsymbol{0}_P$ | $\sqrt{M}\boldsymbol{v}_6^P$ | $\sqrt{M}\boldsymbol{v}_7^P$ | $\sqrt{M}\boldsymbol{v}_8^P$ | $\sqrt{M}\boldsymbol{v}_9^P$ | $\sqrt{M}\boldsymbol{v}_{10}^P$ | $\sqrt{M}\boldsymbol{v}_8^P$ | $\sqrt{M}\boldsymbol{v}_9^P$ | $\sqrt{M}\boldsymbol{v}_{10}^P$ | $\sqrt{M}\boldsymbol{v}_9^P$ | $\sqrt{M}\boldsymbol{v}_8^P$ | $\sqrt{M}\boldsymbol{v}_7^P$ | $\sqrt{M}\boldsymbol{v}_6^P$ | $\sqrt{M}\boldsymbol{v}_5^P$ | $\sqrt{M}\boldsymbol{v}_4^P$ |
| $P+1$: | $\sqrt{MP}$ | $\sqrt{MP}$ | $\sqrt{MP}$ | $\sqrt{MP}$ | $\sqrt{MP}$ | $\sqrt{MP}$ | $\sqrt{MP}$ | $\sqrt{MP}$ | $\sqrt{MP}$ | $\sqrt{MP}$ | $\sqrt{MP}$ | $\sqrt{MP}$ | $\sqrt{MP}$ | $\sqrt{MP}$ | $\sqrt{MP}$ |

Table 76: Example of $\boldsymbol{K}_6^{(2)}\boldsymbol{X}^{(1)}$, continuing from Table 49.

| $\mathcal{I}$ | \$ | 7 | 5 | 9 | 5 | $\times$ | 7 | 9 | = | 5 | 0 | 0 | 0 | 0 | 6 |
|---|---|---|---|---|---|---|---|---|---|---|---|---|---|---|---|
| $1$–$P$: | $\boldsymbol{0}_P$ | $\sqrt{M}\boldsymbol{v}_4^P$ | $\sqrt{M}\boldsymbol{v}_5^P$ | $\sqrt{M}\boldsymbol{v}_6^P$ | $\sqrt{M}\boldsymbol{v}_7^P$ | $\boldsymbol{0}_P$ | $\boldsymbol{0}_P$ | $\boldsymbol{0}_P$ | $\boldsymbol{0}_P$ | $\boldsymbol{0}_P$ | $\boldsymbol{0}_P$ | $\boldsymbol{0}_P$ | $\boldsymbol{0}_P$ | $\boldsymbol{0}_P$ | $\boldsymbol{0}_P$ |
| $P+1$: | $\sqrt{MP}$ | 0 | 0 | 0 | 0 | 0 | 0 | 0 | 0 | 0 | 0 | 0 | 0 | 0 | 0 |

Table 77: Example of $\boldsymbol{A}_6^{(2)}$ (with explicit row/column indices and sufficiently large $M$), continuing from Tables 75 and 76.

| row \ col | $j=1$ | 2 | 3 | 4 | 5 | 6 | 7 | 8 | 9 | 10 | 11 | 12 | 13 | 14 | 15 |
|---|---|---|---|---|---|---|---|---|---|---|---|---|---|---|---|
| $i=1$ | 1 | 1 | 1 | 1 | 1 | 1 | 1 | 1 | 1 | 1 | 1 | 1/2 | 1/2 | 1/2 | 1/2 |
| 2 | 0 | 0 | 0 | 0 | 0 | 0 | 0 | 0 | 0 | 0 | 0 | 0 | 0 | 0 | 1/2 |
| 3 | 0 | 0 | 0 | 0 | 0 | 0 | 0 | 0 | 0 | 0 | 0 | 0 | 0 | 1/2 | 0 |
| 4 | 0 | 0 | 0 | 0 | 0 | 0 | 0 | 0 | 0 | 0 | 0 | 0 | 1/2 | 0 | 0 |
| 5 | 0 | 0 | 0 | 0 | 0 | 0 | 0 | 0 | 0 | 0 | 0 | 1/2 | 0 | 0 | 0 |
| 6 | 0 | 0 | 0 | 0 | 0 | 0 | 0 | 0 | 0 | 0 | 0 | 0 | 0 | 0 | 0 |
| 7 | 0 | 0 | 0 | 0 | 0 | 0 | 0 | 0 | 0 | 0 | 0 | 0 | 0 | 0 | 0 |
| 8 | 0 | 0 | 0 | 0 | 0 | 0 | 0 | 0 | 0 | 0 | 0 | 0 | 0 | 0 | 0 |
| 9 | 0 | 0 | 0 | 0 | 0 | 0 | 0 | 0 | 0 | 0 | 0 | 0 | 0 | 0 | 0 |
| 10 | 0 | 0 | 0 | 0 | 0 | 0 | 0 | 0 | 0 | 0 | 0 | 0 | 0 | 0 | 0 |
| 11 | 0 | 0 | 0 | 0 | 0 | 0 | 0 | 0 | 0 | 0 | 0 | 0 | 0 | 0 | 0 |
| 12 | 0 | 0 | 0 | 0 | 0 | 0 | 0 | 0 | 0 | 0 | 0 | 0 | 0 | 0 | 0 |
| 13 | 0 | 0 | 0 | 0 | 0 | 0 | 0 | 0 | 0 | 0 | 0 | 0 | 0 | 0 | 0 |
| 14 | 0 | 0 | 0 | 0 | 0 | 0 | 0 | 0 | 0 | 0 | 0 | 0 | 0 | 0 | 0 |
| 15 | 0 | 0 | 0 | 0 | 0 | 0 | 0 | 0 | 0 | 0 | 0 | 0 | 0 | 0 | 0 |

Table 78: Example of $\boldsymbol{U}_6^{(2)}\boldsymbol{V}_6^{(2)}\boldsymbol{X}^{(1)}$, continuing from Table 49. (Irrelevant dimensions are omitted for readability)

| $\mathcal{I}$ | \$ | 7 | 5 | 9 | 5 | $\times$ | 7 | 9 | $=$ | 5 | 0 | 0 | 0 | 0 | 6 |
|---|---|---|---|---|---|---|---|---|---|---|---|---|---|---|---|
| 1: NUM | 0 | 0 | 0 | 0 | 0 | 0 | 0 | 0 | 0 | 0 | 0 | 0 | 0 | 0 | 0 |
| 2: FULL_ONES | 0 | 0 | 0 | 0 | 0 | 0 | 0 | 0 | 0 | 0 | 0 | 0 | 0 | 0 | 0 |
| 3: IS_MUL | 0 | 0 | 0 | 0 | 0 | 0 | 0 | 0 | 0 | 0 | 0 | 0 | 0 | 0 | 0 |
| 4: IS_EQUAL | 0 | 0 | 0 | 0 | 0 | 0 | 0 | 0 | 0 | 0 | 0 | 0 | 0 | 0 | 0 |
| 13: OP1_SHIFT3 | 0 | 14 | 10 | 18 | 10 | 0 | 14 | 18 | 0 | 10 | 0 | 0 | 0 | 0 | 12 |

Table 79: Example of $\boldsymbol{U}_6^{(2)}\boldsymbol{V}_6^{(2)}\boldsymbol{X}^{(1)}\boldsymbol{A}_6^{(2)}$, continuing from Tables 77 and 78. (Irrelevant dimensions are omitted for readability)

| $\mathcal{I}$ | \$ | 7 | 5 | 9 | 5 | $\times$ | 7 | 9 | $=$ | 5 | 0 | 0 | 0 | 0 | 6 |
|---|---|---|---|---|---|---|---|---|---|---|---|---|---|---|---|
| 1: NUM | 0 | 0 | 0 | 0 | 0 | 0 | 0 | 0 | 0 | 0 | 0 | 0 | 0 | 0 | 0 |
| 2: FULL_ONES | 0 | 0 | 0 | 0 | 0 | 0 | 0 | 0 | 0 | 0 | 0 | 0 | 0 | 0 | 0 |
| 3: IS_MUL | 0 | 0 | 0 | 0 | 0 | 0 | 0 | 0 | 0 | 0 | 0 | 0 | 0 | 0 | 0 |
| 4: IS_EQUAL | 0 | 0 | 0 | 0 | 0 | 0 | 0 | 0 | 0 | 0 | 0 | 0 | 0 | 0 | 0 |
| 13: OP1_SHIFT3 | 0 | 0 | 0 | 0 | 0 | 0 | 0 | 0 | 0 | 0 | 0 | 5 | 9 | 5 | 7 |

### G.7.7 Attention Head 7: Copying the Appropriate Digit from the First Operand V

The objective of Attention head 7 is to fill the dimension OP1_SHIFT4 at the $i$-th least significant digit of the response (when predicting the $(i+1)$-th least significant digit of the response) with the $(i-3)$-th least significant digit of the first operand. Similarly to the previous head, $i$ ranges from $0$ to $\ell_a + 2$. In cases where the $i$-th least significant digit of the first operand is not well-defined (i.e., $i \in \{0, 1, 2, 3\}$), we assign $0$.

The design of Attention head 7 is as follows. With $d_{QK,27} = P + 1$,

$$Q_7^{(2)} = \begin{pmatrix} \mathbf{0}_{P \times 34} & \mathbf{0}_{P \times P} & \mathbf{0}_{P \times P} & \mathbf{0}_{P \times P} & \mathbf{0}_{P \times P} & \mathbf{0}_{P \times P} & \sqrt{M}\boldsymbol{I}_P \\ \sqrt{MP}\left(\boldsymbol{e}_{\text{FULL\_ONES}}^{34}\right)^\top & \mathbf{0}_{1 \times P} & \mathbf{0}_{1 \times P} & \mathbf{0}_{1 \times P} & \mathbf{0}_{1 \times P} & \mathbf{0}_{1 \times P} & \mathbf{0}_{1 \times P} \end{pmatrix} \in \mathbb{R}^{d_{QK,27} \times d},$$

$$(122)$$

$$K_7^{(2)} = \begin{pmatrix} \mathbf{0}_{P \times 34} & \sqrt{M}\boldsymbol{I}_P & \mathbf{0}_{P \times P} & \mathbf{0}_{P \times P} & \mathbf{0}_{P \times P} & \mathbf{0}_{P \times P} & \mathbf{0}_{P \times P} \\ \sqrt{MP}\left(\boldsymbol{e}_{\text{IS\_BOS}}^{34}\right)^\top & \mathbf{0}_{1 \times P} & \mathbf{0}_{1 \times P} & \mathbf{0}_{1 \times P} & \mathbf{0}_{1 \times P} & \mathbf{0}_{1 \times P} & \mathbf{0}_{1 \times P} \end{pmatrix} \in \mathbb{R}^{d_{QK,27} \times d}.$$

$$(123)$$

With $d_{V,27} = 1$,

$$\boldsymbol{V}_7^{(2)} = 2\left(\boldsymbol{e}_{\text{NUM}}^d\right)^\top \in \mathbb{R}^{d_{V,27} \times d}, \tag{124}$$

$$\boldsymbol{U}_7^{(2)} = \boldsymbol{e}_{\text{OP1\_SHIFT4}}^d \in \mathbb{R}^{d \times d_{V,27}}. \tag{125}$$

We provide the examples in Tables 80 to 84.

Table 80: Example of $\boldsymbol{Q}_7^{(2)}\boldsymbol{X}^{(1)}$, continuing from Table 49.

| $\mathcal{I}$ | \$ | 7 | 5 | 9 | 5 | $\times$ | 7 | 9 | = | 5 | 0 | 0 | 0 | 0 | 6 |
|---|---|---|---|---|---|---|---|---|---|---|---|---|---|---|---|
| $1$–$P$: | $\mathbf{0}_P$ | $\sqrt{M}\boldsymbol{v}_7^P$ | $\sqrt{M}\boldsymbol{v}_8^P$ | $\sqrt{M}\boldsymbol{v}_9^P$ | $\sqrt{M}\boldsymbol{v}_{10}^P$ | $\sqrt{M}\boldsymbol{v}_{11}^P$ | $\sqrt{M}\boldsymbol{v}_9^P$ | $\sqrt{M}\boldsymbol{v}_{10}^P$ | $\sqrt{M}\boldsymbol{v}_{11}^P$ | $\sqrt{M}\boldsymbol{v}_{10}^P$ | $\sqrt{M}\boldsymbol{v}_9^P$ | $\sqrt{M}\boldsymbol{v}_8^P$ | $\sqrt{M}\boldsymbol{v}_7^P$ | $\sqrt{M}\boldsymbol{v}_6^P$ | $\sqrt{M}\boldsymbol{v}_5^P$ |
| $P+1$: | $\sqrt{MP}$ | $\sqrt{MP}$ | $\sqrt{MP}$ | $\sqrt{MP}$ | $\sqrt{MP}$ | $\sqrt{MP}$ | $\sqrt{MP}$ | $\sqrt{MP}$ | $\sqrt{MP}$ | $\sqrt{MP}$ | $\sqrt{MP}$ | $\sqrt{MP}$ | $\sqrt{MP}$ | $\sqrt{MP}$ | $\sqrt{MP}$ |

Table 81: Example of $\boldsymbol{K}_7^{(2)}\boldsymbol{X}^{(1)}$, continuing from Table 49.

| $\mathcal{I}$ | \$ | 7 | 5 | 9 | 5 | $\times$ | 7 | 9 | = | 5 | 0 | 0 | 0 | 0 | 6 |
|---|---|---|---|---|---|---|---|---|---|---|---|---|---|---|---|
| $1$–$P$: | $\mathbf{0}_P$ | $\sqrt{M}\boldsymbol{v}_4^P$ | $\sqrt{M}\boldsymbol{v}_5^P$ | $\sqrt{M}\boldsymbol{v}_6^P$ | $\sqrt{M}\boldsymbol{v}_7^P$ | $\mathbf{0}_P$ | $\mathbf{0}_P$ | $\mathbf{0}_P$ | $\mathbf{0}_P$ | $\mathbf{0}_P$ | $\mathbf{0}_P$ | $\mathbf{0}_P$ | $\mathbf{0}_P$ | $\mathbf{0}_P$ | $\mathbf{0}_P$ |
| $P+1$: | $\sqrt{MP}$ | 0 | 0 | 0 | 0 | 0 | 0 | 0 | 0 | 0 | 0 | 0 | 0 | 0 | 0 |

Table 82: Example of $\boldsymbol{A}_7^{(2)}$ (with explicit row/column indices and sufficiently large $M$), continuing from Tables 80 and 81.

| row \ col | $j=1$ | 2 | 3 | 4 | 5 | 6 | 7 | 8 | 9 | 10 | 11 | 12 | 13 | 14 | 15 |
|---|---|---|---|---|---|---|---|---|---|---|---|---|---|---|---|
| $i=1$ | 1 | 1 | 1 | 1 | 1 | 1 | 1 | 1 | 1 | 1 | 1 | 1 | 1/2 | 1/2 | 1/2 |
| 2 | 0 | 0 | 0 | 0 | 0 | 0 | 0 | 0 | 0 | 0 | 0 | 0 | 0 | 0 | 0 |
| 3 | 0 | 0 | 0 | 0 | 0 | 0 | 0 | 0 | 0 | 0 | 0 | 0 | 0 | 0 | 1/2 |
| 4 | 0 | 0 | 0 | 0 | 0 | 0 | 0 | 0 | 0 | 0 | 0 | 0 | 0 | 1/2 | 0 |
| 5 | 0 | 0 | 0 | 0 | 0 | 0 | 0 | 0 | 0 | 0 | 0 | 0 | 1/2 | 0 | 0 |
| 6 | 0 | 0 | 0 | 0 | 0 | 0 | 0 | 0 | 0 | 0 | 0 | 0 | 0 | 0 | 0 |
| 7 | 0 | 0 | 0 | 0 | 0 | 0 | 0 | 0 | 0 | 0 | 0 | 0 | 0 | 0 | 0 |
| 8 | 0 | 0 | 0 | 0 | 0 | 0 | 0 | 0 | 0 | 0 | 0 | 0 | 0 | 0 | 0 |
| 9 | 0 | 0 | 0 | 0 | 0 | 0 | 0 | 0 | 0 | 0 | 0 | 0 | 0 | 0 | 0 |
| 10 | 0 | 0 | 0 | 0 | 0 | 0 | 0 | 0 | 0 | 0 | 0 | 0 | 0 | 0 | 0 |
| 11 | 0 | 0 | 0 | 0 | 0 | 0 | 0 | 0 | 0 | 0 | 0 | 0 | 0 | 0 | 0 |
| 12 | 0 | 0 | 0 | 0 | 0 | 0 | 0 | 0 | 0 | 0 | 0 | 0 | 0 | 0 | 0 |
| 13 | 0 | 0 | 0 | 0 | 0 | 0 | 0 | 0 | 0 | 0 | 0 | 0 | 0 | 0 | 0 |
| 14 | 0 | 0 | 0 | 0 | 0 | 0 | 0 | 0 | 0 | 0 | 0 | 0 | 0 | 0 | 0 |
| 15 | 0 | 0 | 0 | 0 | 0 | 0 | 0 | 0 | 0 | 0 | 0 | 0 | 0 | 0 | 0 |

Table 83: Example of $\boldsymbol{U}_7^{(2)}\boldsymbol{V}_7^{(2)}\boldsymbol{X}^{(1)}$, continuing from Table 49. (Irrelevant dimensions are omitted for readability)

| $\mathcal{I}$ | $ | 7 | 5 | 9 | 5 | × | 7 | 9 | = | 5 | 0 | 0 | 0 | 0 | 6 |
|---|---|---|---|---|---|---|---|---|---|---|---|---|---|---|---|
| 1: NUM | 0 | 0 | 0 | 0 | 0 | 0 | 0 | 0 | 0 | 0 | 0 | 0 | 0 | 0 | 0 |
| 2: FULL_ONES | 0 | 0 | 0 | 0 | 0 | 0 | 0 | 0 | 0 | 0 | 0 | 0 | 0 | 0 | 0 |
| 3: IS_MUL | 0 | 0 | 0 | 0 | 0 | 0 | 0 | 0 | 0 | 0 | 0 | 0 | 0 | 0 | 0 |
| 4: IS_EQUAL | 0 | 0 | 0 | 0 | 0 | 0 | 0 | 0 | 0 | 0 | 0 | 0 | 0 | 0 | 0 |
| 14: OP1_SHIFT4 | 0 | 14 | 10 | 18 | 10 | 0 | 14 | 18 | 0 | 10 | 0 | 0 | 0 | 0 | 12 |

Table 84: Example of $\boldsymbol{U}_7^{(2)}\boldsymbol{V}_7^{(2)}\boldsymbol{X}^{(1)}\boldsymbol{A}_7^{(2)}$, continuing from Tables 82 and 83. (Irrelevant dimensions are omitted for readability)

| $\mathcal{I}$ | $ | 7 | 5 | 9 | 5 | × | 7 | 9 | = | 5 | 0 | 0 | 0 | 0 | 6 |
|---|---|---|---|---|---|---|---|---|---|---|---|---|---|---|---|
| 1: NUM | 0 | 0 | 0 | 0 | 0 | 0 | 0 | 0 | 0 | 0 | 0 | 0 | 0 | 0 | 0 |
| 2: FULL_ONES | 0 | 0 | 0 | 0 | 0 | 0 | 0 | 0 | 0 | 0 | 0 | 0 | 0 | 0 | 0 |
| 3: IS_MUL | 0 | 0 | 0 | 0 | 0 | 0 | 0 | 0 | 0 | 0 | 0 | 0 | 0 | 0 | 0 |
| 4: IS_EQUAL | 0 | 0 | 0 | 0 | 0 | 0 | 0 | 0 | 0 | 0 | 0 | 0 | 0 | 0 | 0 |
| 14: OP1_SHIFT4 | 0 | 0 | 0 | 0 | 0 | 0 | 0 | 0 | 0 | 0 | 0 | 0 | 5 | 9 | 5 |

### G.7.8 Residual Connection

So far we have computed the output of $\texttt{Att}_2$ operation. Passing through the residual connection, the output of the attention layer becomes the sum of $\boldsymbol{X}^{(1)}$ (the input to the second Transformer block) and the output of $\texttt{Att}_2$ operation:

$$\boldsymbol{Y}^{(2)} = \boldsymbol{X}^{(1)} + \sum_{h\in[7]} \boldsymbol{U}_h^{(2)}\boldsymbol{V}_h^{(2)}\boldsymbol{X}^{(1)}\boldsymbol{A}_h^{(2)}. \tag{126}$$

A concrete example of the output of residual connection is presented in Table 85.

Table 85: Example output of residual connection, continuing from Tables 49, 54, 59, 64, 69, 74, 79 and 84. Uncolored rows represent the initial embedding. Gray rows indicate the rows filled by the first Transformer block. Yellow rows indicate the rows filled by the attention layers at the second Transformer block. Pink rows indicate the rows that will be filled by the subsequent FFN layer.

| $\mathcal{I}$ | $ | 7 | 5 | 9 | 5 | × | 7 | 9 | = | 5 | 0 | 0 | 0 | 0 | 6 |
|---|---|---|---|---|---|---|---|---|---|---|---|---|---|---|---|
| 1: NUM | 0 | 7 | 5 | 9 | 5 | 0 | 7 | 9 | 0 | 5 | 0 | 0 | 0 | 0 | 6 |
| 2: FULL_ONES | 1 | 1 | 1 | 1 | 1 | 1 | 1 | 1 | 1 | 1 | 1 | 1 | 1 | 1 | 1 |
| 3: IS_BOS | 1 | 0 | 0 | 0 | 0 | 0 | 0 | 0 | 0 | 0 | 0 | 0 | 0 | 0 | 0 |
| 4: IS_MUL | 0 | 0 | 0 | 0 | 0 | 1 | 0 | 0 | 0 | 0 | 0 | 0 | 0 | 0 | 0 |
| 5: IS_EQUAL | 0 | 0 | 0 | 0 | 0 | 0 | 0 | 0 | 1 | 0 | 0 | 0 | 0 | 0 | 0 |
| 6: IS_OP2_ONE | 0 | 0 | 0 | 0 | 0 | 0 | 0 | 1 | 0 | 0 | 0 | 0 | 0 | 0 | 0 |
| 7: IS_OP2_TEN | 0 | 0 | 0 | 0 | 0 | 0 | 1 | 0 | 0 | 0 | 0 | 0 | 0 | 0 | 0 |
| 8: OP2_ONE | 0 | 7/2 | 4 | 21/4 | 26/5 | 13/3 | 33/7 | 9 | 9 | 9 | 9 | 9 | 9 | 9 | 9 |
| 9: OP2_TEN | 0 | 7/2 | 4 | 21/4 | 26/5 | 13/3 | 7 | 7 | 7 | 7 | 7 | 7 | 7 | 7 | 7 |
| 10: OP1_SHIFT0 | 0 | 0 | 7 | 5 | 9 | 5 | 5 | 9 | 5 | 9 | 5 | 7 | 0 | 0 | 0 |
| 11: OP1_SHIFT1 | 0 | 7 | 5 | 9 | 5 | 0 | 9 | 5 | 0 | 5 | 9 | 5 | 7 | 0 | 0 |
| 12: OP1_SHIFT2 | 0 | 0 | 0 | 0 | 0 | 0 | 5 | 0 | 0 | 0 | 5 | 9 | 5 | 7 | 0 |
| 13: OP1_SHIFT3 | 0 | 0 | 0 | 0 | 0 | 0 | 0 | 0 | 0 | 0 | 0 | 5 | 9 | 5 | 7 |
| 14: OP1_SHIFT4 | 0 | 0 | 0 | 0 | 0 | 0 | 0 | 0 | 0 | 0 | 0 | 0 | 5 | 9 | 5 |
| 15: RESULT1 | 0 | 0 | 0 | 0 | 0 | 0 | 0 | 0 | 0 | 0 | 0 | 0 | 0 | 0 | 0 |
| 16: RESULT2 | 0 | 0 | 0 | 0 | 0 | 0 | 0 | 0 | 0 | 0 | 0 | 0 | 0 | 0 | 0 |
| 17: RESULT3 | 0 | 0 | 0 | 0 | 0 | 0 | 0 | 0 | 0 | 0 | 0 | 0 | 0 | 0 | 0 |
| 18: RESULT4 | 0 | 0 | 0 | 0 | 0 | 0 | 0 | 0 | 0 | 0 | 0 | 0 | 0 | 0 | 0 |
| 19: PRE_PROD | 0 | 0 | 0 | 0 | 0 | 0 | 0 | 0 | 0 | 0 | 0 | 0 | 0 | 0 | 0 |
| 20: PRE_CARRY | 0 | 0 | 0 | 0 | 0 | 0 | 0 | 0 | 0 | 0 | 0 | 0 | 0 | 0 | 0 |
| 21: PRE_EOS1 | 1 | 1 | 1/2 | 1/2 | 1/2 | 1/2 | 1/2 | 1/2 | 1/2 | 1/2 | 1/2 | 1/2 | 1 | 1 | 1 |
| 22: PRE_EOS2 | 1 | 1 | 1 | 1 | 1 | 1 | 1/2 | 1 | 1 | 1 | 1/2 | 1/2 | 1/2 | 1/2 | 1 |
| 23–32: PROD | $\mathbf{0}_{10}$ | $\mathbf{0}_{10}$ | $\mathbf{0}_{10}$ | $\mathbf{0}_{10}$ | $\mathbf{0}_{10}$ | $\mathbf{0}_{10}$ | $\mathbf{0}_{10}$ | $\mathbf{0}_{10}$ | $\mathbf{0}_{10}$ | $\mathbf{0}_{10}$ | $\mathbf{0}_{10}$ | $\mathbf{0}_{10}$ | $\mathbf{0}_{10}$ | $\mathbf{0}_{10}$ | $\mathbf{0}_{10}$ |
| 33: IS_EOS | 0 | 0 | 0 | 0 | 0 | 0 | 0 | 0 | 0 | 0 | 0 | 0 | 0 | 0 | 0 |
| 34: MASK | 0 | 0 | 0 | 0 | 0 | 1 | 1 | 1 | 1 | 1 | 1 | 1 | 1 | 1 | 1 |
| 35–(P+34): POS_2_MASK | $\mathbf{0}_P$ | $\mathbf{0}_P$ | $\mathbf{0}_P$ | $\mathbf{0}_P$ | $\mathbf{0}_P$ | $\mathbf{0}_P$ | $\mathbf{0}_P$ | $\mathbf{0}_P$ | $\mathbf{0}_P$ | $\mathbf{0}_P$ | $\mathbf{0}_P$ | $\mathbf{0}_P$ | $\mathbf{0}_P$ | $\mathbf{0}_P$ | $\mathbf{0}_P$ |
| (P+35)–(2P+34): POS_1 | $\mathbf{0}_P$ | $v_3^P$ | $v_4^P$ | $v_5^P$ | $v_6^P$ | $v_7^P$ | $v_5^P$ | $v_6^P$ | $v_7^P$ | $v_6^P$ | $v_5^P$ | $v_4^P$ | $v_3^P$ | $v_2^P$ | $v_1^P$ |
| (2P+35)–(3P+34): POS_2 | $\mathbf{0}_P$ | $v_4^P$ | $v_5^P$ | $v_6^P$ | $v_7^P$ | $v_8^P$ | $v_6^P$ | $v_7^P$ | $v_8^P$ | $v_7^P$ | $v_6^P$ | $v_5^P$ | $v_4^P$ | $v_3^P$ | $v_2^P$ |
| (3P+35)–(4P+34): POS_3 | $\mathbf{0}_P$ | $v_5^P$ | $v_6^P$ | $v_7^P$ | $v_8^P$ | $v_9^P$ | $v_7^P$ | $v_8^P$ | $v_9^P$ | $v_8^P$ | $v_7^P$ | $v_6^P$ | $v_5^P$ | $v_4^P$ | $v_3^P$ |
| (4P+35)–(5P+34): POS_4 | $\mathbf{0}_P$ | $v_6^P$ | $v_7^P$ | $v_8^P$ | $v_9^P$ | $v_{10}^P$ | $v_8^P$ | $v_9^P$ | $v_{10}^P$ | $v_9^P$ | $v_8^P$ | $v_7^P$ | $v_6^P$ | $v_5^P$ | $v_4^P$ |
| (5P+35)–(6P+34): POS_5 | $\mathbf{0}_P$ | $v_7^P$ | $v_8^P$ | $v_9^P$ | $v_{10}^P$ | $v_{11}^P$ | $v_9^P$ | $v_{10}^P$ | $v_{11}^P$ | $v_{10}^P$ | $v_9^P$ | $v_8^P$ | $v_7^P$ | $v_6^P$ | $v_5^P$ |

## G.8 Transformer Block 2 — Token-wise Feed-forward Layer

Our ultimate goal is to fill the dimensions PROD and IS_EOS with appropriate values. The dimensions RESULT1 to RESULT4, PRE_PROD, and PRE_CARRY serve as temporary memories for storing intermediate values, which will help us achieve our ultimate goal. Our construction involves sequentially stacking the MLP networks step-by-step to generate each of these temporary values. (As mentioned in the theorem statement below, we allow $FF_2$ to be a multi-layer MLP.)

While our current construction for $FF_2$ involves multiple hidden layers, we believe that our construction can be improved to employ a single hidden layer. If employing multiple hidden layers in the FFN is not feasible, this issue can be addressed by introducing additional Transformer blocks. Specifically, we can bypass the attention layer in these extra blocks by residual connection and only utilize their FFNs.

**Step 1. Filling RESULT_1 to RESULT_4** Here, we first assume the existence of a single-hidden-layer MLP network, denoted as $f : \mathbb{R}^2 \to \mathbb{R}$, such that given any integers $a, b \in \{0, 1, \ldots, 9\}$, $f(a, b)$ equals to their multiplication, i.e., $ab$. Such a network can be implemented with 100 hidden nodes (Zhang et al., 2021).

Recalling Appendix G.4, we construct the first MLP network by utilizing eight instances of the function $f$ in parallel as follows:

1. RESULT1 $= f(\text{OP1\_SHIFT0}, \text{OP2\_ONE}) + f(\text{OP1\_SHIFT1}, \text{OP2\_TEN}) \in \{0, 1, \ldots, 162\}$,

2. $\text{RESULT2} = f(\text{OP1\_SHIFT1}, \text{OP2\_ONE}) + f(\text{OP1\_SHIFT2}, \text{OP2\_TEN}) \in \{0, 1, \ldots, 162\}$,

3. $\text{RESULT3} = f(\text{OP1\_SHIFT2}, \text{OP2\_ONE}) + f(\text{OP1\_SHIFT3}, \text{OP2\_TEN}) \in \{0, 1, \ldots, 162\}$,

4. $\text{RESULT4} = f(\text{OP1\_SHIFT3}, \text{OP2\_ONE}) + f(\text{OP1\_SHIFT4}, \text{OP2\_TEN}) \in \{0, 1, \ldots, 162\}$.

### Step 2. Filling PRE_PROD and PRE_CARRY

Here, we assume the existence of the following three single-hidden-layer MLP networks, denoted as $g_1, g_2, g_3 : \mathbb{R} \to \mathbb{R}$, such that given any at most 3-digit integer $a \in \{0, 1, \ldots, 162\}$, $g_1(a)$, $g_2(a)$ and $g_3(a)$ output the ones, tens, and hundreds digit of $a$, respectively. Similarly to the previous step, each network can be implemented with 163 hidden nodes (Zhang et al., 2021).

Recalling Appendix G.4, we construct the second MLP network on top of the first MLP network, by utilizing 2 instances of each of the function $g_1$, $g_2$, and $g_3$ in parallel as follows:

- $\text{PRE\_PROD} = g_1(\text{RESULT1}) + g_2(\text{RESULT2}) + g_3(\text{RESULT3}) \in \{0, 1, \ldots, 27\}$,
- $\text{PRE\_CARRY} = g_1(\text{RESULT2}) + g_2(\text{RESULT3}) + g_3(\text{RESULT4}) \in \{0, 1, \ldots, 27\}$.

**Step 3. Filling PROD**  Here, we assume the existence of a single-hidden-layer MLP network, denoted as $h : \mathbb{R}^2 \to \mathbb{R}$, such that given any integers $a \in \{0, 1, \ldots, 27\}$, $b \in \{0, 1, \ldots, 9\}$ satisfying $a - b \in \{-2, -1, 0, 8, 9, 10, 18, 19, 20\}$, $h$ satisfies

$$h(a, b) = \begin{cases} 0, & \text{if } a - b \in \{-2, -1, 0\}, \\ 1, & \text{if } a - b \in \{8, 9, 10\}, \\ 2, & \text{if } a - b \in \{18, 19, 20\}. \end{cases}$$

We also assume the existence of a single-hidden-layer MLP network, denoted as $h' : \mathbb{R} \to \mathbb{R}$, such that given any integer $a \in \{0, 1, \ldots, 19\}$, $h'(a)$ equals to $a \pmod{10}$.

We finally assume the existence of a single-hidden-layer MLP network $q_i : \mathbb{R} \to \mathbb{R}$ for each $i \in \{0, 1, \ldots, 9\}$, such that given any integers $a \in \{0, 1, \ldots, 9\}$, $q_i$ satisfies

$$q_i(a) = \mathbb{1}(i = a).$$

Similarly to the previous step, each network can be implemented with 280, 20, and 10 hidden nodes. Recalling Appendix G.4, we construct the third MLP network, on top of the second MLP network, by

- $\text{PROD} = \begin{pmatrix} q_0(h'(\text{PRE\_PROD} + h(\text{PRE\_CARRY}, \text{NUM}))) \\ q_1(h'(\text{PRE\_PROD} + h(\text{PRE\_CARRY}, \text{NUM}))) \\ \vdots \\ q_9(h'(\text{PRE\_PROD} + h(\text{PRE\_CARRY}, \text{NUM}))) \end{pmatrix} \in \mathbb{R}^{10}$.

One can easily check that $h'(\text{PRE\_PROD} + h(\text{PRE\_CARRY}, \text{NUM}))$ yields an element of $0, 1, \ldots, 9$, and thus PROD is an one-hot column vector. Specifically, if $h'(\text{PRE\_PROD} + h(\text{PRE\_CARRY}, \text{NUM})) = i$, then PROD becomes $e_{i+1}^{10}$.

**Step 4. Filling IS_EOS**  We construct a single-hidden-layer MLP network $r : \mathbb{R}^2 \to \mathbb{R}$ by

$$r(a, b) = 2\phi(a + b - 1.5).$$

We then can fill the dimension IS_EOS by

- $\text{IS\_EOS} = r(\text{PRE\_EOS1}, \text{PRE\_EOS2})$.

Since PRE_EOS1 and PRE_EOS2 can have either $1/2$ or 1, IS_EOS equals 1 only when both PRE_EOS1 and PRE_EOS2 are 1. Additionally, we note that PRE_EOS1 and PRE_EOS2 are the direct outputs from the attention layer. Therefore, the network $r$ can be deployed in parallel with the first MLP network and does not require an additional FFN layer.

The example output resulting from passing through all these steps is presented in Table 86.

Table 86: Example output of FFN layer in the second Transformer block, continuing from Table 85. Here, we mark $-$ for the entries before the equal token, as these entries do not affect the next-token prediction in our construction and are thus not important.

| $\mathcal{I}$ | $ | 7 | 5 | 9 | 5 | $\times$ | 7 | 9 | = | 5 | 0 | 0 | 0 | 0 | 6 |
|---|---|---|---|---|---|---|---|---|---|---|---|---|---|---|---|
| 15: RESULT1 | - | - | - | - | - | - | - | - | 45 | 116 | 108 | 98 | 49 | 0 | 0 |
| 16: RESULT2 | - | - | - | - | - | - | - | - | 0 | 45 | 116 | 108 | 98 | 49 | 0 |
| 17: RESULT3 | - | - | - | - | - | - | - | - | 0 | 0 | 45 | 116 | 108 | 98 | 49 |
| 18: RESULT4 | - | - | - | - | - | - | - | - | 0 | 0 | 0 | 45 | 116 | 108 | 98 |
| 19: PRE_PROD | - | - | - | - | - | - | - | - | 5 | 10 | 9 | 9 | 19 | 4 | 0 |
| 20: PRE_CARRY | - | - | - | - | - | - | - | - | 0 | 5 | 10 | 9 | 9 | 19 | 4 |
| 23-32: PROD | - | - | - | - | - | - | - | - | $e_6^{10}$ | $e_1^{10}$ | $e_1^{10}$ | $e_1^{10}$ | $e_1^{10}$ | $e_7^{10}$ | $e_1^{10}$ |
| 33: IS_EOS | - | - | - | - | - | - | - | - | 0 | 0 | 0 | 0 | 0 | 0 | 1 |

### G.8.1 Residual Connection

The last task of the feed-forward layer is to pass $\text{FF}_2\left(\boldsymbol{Y}^{(2)}\right)$ through the residual connection. As a result, we have

$$\boldsymbol{X}^{(2)} = \boldsymbol{Y}^{(2)} + \text{FF}_2\left(\boldsymbol{Y}^{(2)}\right). \tag{127}$$

This is the end of the second Transformer block, and an example of $\boldsymbol{X}^{(2)}$ is illustrated in Table 87.

### G.9 Decoding Function

As mentioned in Appendix D, the decoding function performs a linear readout (with a weight matrix $\boldsymbol{W}_{\text{out}} \in \mathbb{R}^{|\mathcal{V}| \times d}$) and a (token-wise) arg-max operation. That is,

$$\text{Dec}\left(\boldsymbol{X}^{(1)}\right) := \left(\mathcal{V}_{k_i}\right)_{i=1,\dots,N} \in \mathcal{V}^N, \tag{128}$$

where $\mathcal{V}_k$ is the $k$-th element of $\mathcal{V}$ and

$$k_i := \arg\max_{k \in [|\mathcal{V}|]} \left\{ o_k : \boldsymbol{W}_{\text{out}} \boldsymbol{X}_{\bullet i}^{(1)} = \begin{bmatrix} o_1 & \cdots & o_{|\mathcal{V}|} \end{bmatrix}^\top \right\}. \tag{129}$$

The objective of the decoding function is to perform a proper next-token prediction for $N \times 2$ multiplication, especially utilizing the dimensions PROD and IS_EOS of $\boldsymbol{X}^{(2)}$.

We now construct the weight matrix $\boldsymbol{W}_{\text{out}}$. For a token $\sigma_i$, if the value of dimension IS_EOS of $\boldsymbol{X}^{(2)}$ is 0, then the linear readout output the dimensions PROD as it is to return one of a number token (0-9). On the other hand, if the value of dimension IS_EOS is 1, then the linear readout outputs a large number (like 9 for example) for the token '\$' to return EOS (\$). This can be implemented by the weight matrix $\boldsymbol{W}_{\text{out}}$ described in Table 88. Also, an example of applying the linear transform is showcased in Tables 89 and 90.

Table 87: Example embedding matrix after the second Transformer block. The yellow rows represent the results introduced during the second block, while the gray rows indicate the results from the first block. Similarly to Table 85, we mark $-$ for the entries before the equal token, as these entries do not affect the next-token prediction in our construction and are thus not important.

| $\mathcal{I}$ | $ | 7 | 5 | 9 | 5 | × | 7 | 9 | = | 5 | 0 | 0 | 0 | 0 | 6 |
|---|---|---|---|---|---|---|---|---|---|---|---|---|---|---|---|
| 1: NUM | 0 | 7 | 5 | 9 | 5 | 0 | 7 | 9 | 0 | 5 | 0 | 0 | 0 | 0 | 6 |
| 2: FULL_ONES | 1 | 1 | 1 | 1 | 1 | 1 | 1 | 1 | 1 | 1 | 1 | 1 | 1 | 1 | 1 |
| 3: IS_BOS | 1 | 0 | 0 | 0 | 0 | 0 | 0 | 0 | 0 | 0 | 0 | 0 | 0 | 0 | 0 |
| 4: IS_MUL | 0 | 0 | 0 | 0 | 0 | 1 | 0 | 0 | 0 | 0 | 0 | 0 | 0 | 0 | 0 |
| 5: IS_EQUAL | 0 | 0 | 0 | 0 | 0 | 0 | 0 | 0 | 1 | 0 | 0 | 0 | 0 | 0 | 0 |
| 6: IS_OP2_ONE | 0 | 0 | 0 | 0 | 0 | 0 | 0 | 1 | 0 | 0 | 0 | 0 | 0 | 0 | 0 |
| 7: IS_OP2_TEN | 0 | 0 | 0 | 0 | 0 | 0 | 1 | 0 | 0 | 0 | 0 | 0 | 0 | 0 | 0 |
| 8: OP2_ONE | - | - | - | - | - | - | - | - | 9 | 9 | 9 | 9 | 9 | 9 | 9 |
| 9: OP2_TEN | - | - | - | - | - | - | - | - | 7 | 7 | 7 | 7 | 7 | 7 | 7 |
| 10: OP1_SHIFT0 | - | - | - | - | - | - | - | - | 5 | 9 | 5 | 7 | 0 | 0 | 0 |
| 11: OP1_SHIFT1 | - | - | - | - | - | - | - | - | 0 | 5 | 9 | 5 | 7 | 0 | 0 |
| 12: OP1_SHIFT2 | - | - | - | - | - | - | - | - | 0 | 0 | 5 | 9 | 5 | 7 | 0 |
| 13: OP1_SHIFT3 | - | - | - | - | - | - | - | - | 0 | 0 | 0 | 5 | 9 | 5 | 7 |
| 14: OP1_SHIFT4 | - | - | - | - | - | - | - | - | 0 | 0 | 0 | 0 | 5 | 9 | 5 |
| 15: RESULT1 | - | - | - | - | - | - | - | - | 45 | 116 | 108 | 98 | 49 | 0 | 0 |
| 16: RESULT2 | - | - | - | - | - | - | - | - | 0 | 45 | 116 | 108 | 98 | 49 | 0 |
| 17: RESULT3 | - | - | - | - | - | - | - | - | 0 | 0 | 45 | 116 | 108 | 98 | 49 |
| 18: RESULT4 | - | - | - | - | - | - | - | - | 0 | 0 | 0 | 45 | 116 | 108 | 98 |
| 19: PRE_PROD | - | - | - | - | - | - | - | - | 5 | 10 | 9 | 9 | 19 | 4 | 0 |
| 20: PRE_CARRY | - | - | - | - | - | - | - | - | 0 | 5 | 10 | 9 | 9 | 19 | 4 |
| 21: PRE_EOS1 | - | - | - | - | - | - | - | - | 1/2 | 1/2 | 1/2 | 1/2 | 1 | 1 | 1 |
| 22: PRE_EOS2 | - | - | - | - | - | - | - | - | 1 | 1 | 1/2 | 1/2 | 1/2 | 1/2 | 1 |
| 23-32: PROD | - | - | - | - | - | - | - | - | $e_6^{10}$ | $e_1^{10}$ | $e_1^{10}$ | $e_1^{10}$ | $e_1^{10}$ | $e_7^{10}$ | $e_1^{10}$ |
| 33: IS_EOS | - | - | - | - | - | - | - | - | 0 | 0 | 0 | 0 | 0 | 0 | 1 |
| 34: MASK | 0 | 0 | 0 | 0 | 0 | 1 | 1 | 1 | 1 | 1 | 1 | 1 | 1 | 1 | 1 |
| 35–(P+34): POS_2_MASK | $\mathbf{0}_P$ | $\mathbf{0}_P$ | $\mathbf{0}_P$ | $\mathbf{0}_P$ | $\mathbf{0}_P$ | $\mathbf{0}_P$ | $\mathbf{0}_P$ | $\mathbf{0}_P$ | $\mathbf{0}_P$ | $\mathbf{0}_P$ | $\mathbf{0}_P$ | $\mathbf{0}_P$ | $\mathbf{0}_P$ | $\mathbf{0}_P$ | $\mathbf{0}_P$ |
| (P+35)–(2P+34): POS_1 | $\mathbf{0}_P$ | $v_3^P$ | $v_4^P$ | $v_5^P$ | $v_6^P$ | $v_7^P$ | $v_5^P$ | $v_6^P$ | $v_7^P$ | $v_6^P$ | $v_5^P$ | $v_4^P$ | $v_3^P$ | $v_2^P$ | $v_1^P$ |
| (2P+35)–(3P+34): POS_2 | $\mathbf{0}_P$ | $v_4^P$ | $v_5^P$ | $v_6^P$ | $v_7^P$ | $v_8^P$ | $v_6^P$ | $v_7^P$ | $v_8^P$ | $v_7^P$ | $v_6^P$ | $v_5^P$ | $v_4^P$ | $v_3^P$ | $v_2^P$ |
| (3P+35)–(4P+34): POS_3 | $\mathbf{0}_P$ | $v_5^P$ | $v_6^P$ | $v_7^P$ | $v_8^P$ | $v_9^P$ | $v_7^P$ | $v_8^P$ | $v_9^P$ | $v_8^P$ | $v_7^P$ | $v_6^P$ | $v_5^P$ | $v_4^P$ | $v_3^P$ |
| (4P+35)–(5P+34): POS_4 | $\mathbf{0}_P$ | $v_6^P$ | $v_7^P$ | $v_8^P$ | $v_9^P$ | $v_{10}^P$ | $v_8^P$ | $v_9^P$ | $v_{10}^P$ | $v_9^P$ | $v_8^P$ | $v_7^P$ | $v_6^P$ | $v_5^P$ | $v_4^P$ |
| (5P+35)–(6P+34): POS_5 | $\mathbf{0}_P$ | $v_7^P$ | $v_8^P$ | $v_9^P$ | $v_{10}^P$ | $v_{11}^P$ | $v_9^P$ | $v_{10}^P$ | $v_{11}^P$ | $v_{10}^P$ | $v_9^P$ | $v_8^P$ | $v_7^P$ | $v_6^P$ | $v_5^P$ |

Table 88: The *transposed* weight matrix $\boldsymbol{W}_{out}^{\top}$ of the linear readout in decoding function. $P'$ represents $6P + 1$.

| $\mathcal{V}$ | 0 | 1 | 2 | 3 | 4 | 5 | 6 | 7 | 8 | 9 | × | = | $ |
|---|---|---|---|---|---|---|---|---|---|---|---|---|---|
| 1-22: NUM-PRE_EOS_2 | $\mathbf{0}_{22}$ | $\mathbf{0}_{22}$ | $\mathbf{0}_{22}$ | $\mathbf{0}_{22}$ | $\mathbf{0}_{22}$ | $\mathbf{0}_{22}$ | $\mathbf{0}_{22}$ | $\mathbf{0}_{22}$ | $\mathbf{0}_{22}$ | $\mathbf{0}_{22}$ | $\mathbf{0}_{22}$ | $\mathbf{0}_{22}$ | $\mathbf{0}_{22}$ |
| 23: PROD$_1$ | 1 | 0 | 0 | 0 | 0 | 0 | 0 | 0 | 0 | 0 | 0 | 0 | 0 |
| 24: PROD$_2$ | 0 | 1 | 0 | 0 | 0 | 0 | 0 | 0 | 0 | 0 | 0 | 0 | 0 |
| 25: PROD$_3$ | 0 | 0 | 1 | 0 | 0 | 0 | 0 | 0 | 0 | 0 | 0 | 0 | 0 |
| 26: PROD$_4$ | 0 | 0 | 0 | 1 | 0 | 0 | 0 | 0 | 0 | 0 | 0 | 0 | 0 |
| 27: PROD$_5$ | 0 | 0 | 0 | 0 | 1 | 0 | 0 | 0 | 0 | 0 | 0 | 0 | 0 |
| 28: PROD$_6$ | 0 | 0 | 0 | 0 | 0 | 1 | 0 | 0 | 0 | 0 | 0 | 0 | 0 |
| 29: PROD$_7$ | 0 | 0 | 0 | 0 | 0 | 0 | 1 | 0 | 0 | 0 | 0 | 0 | 0 |
| 30: PROD$_8$ | 0 | 0 | 0 | 0 | 0 | 0 | 0 | 1 | 0 | 0 | 0 | 0 | 0 |
| 31: PROD$_9$ | 0 | 0 | 0 | 0 | 0 | 0 | 0 | 0 | 1 | 0 | 0 | 0 | 0 |
| 32: PROD$_{10}$ | 0 | 0 | 0 | 0 | 0 | 0 | 0 | 0 | 0 | 1 | 0 | 0 | 0 |
| 33: IS_EOS | 0 | 0 | 0 | 0 | 0 | 0 | 0 | 0 | 0 | 0 | 0 | 0 | 100 |
| 34-end | $\mathbf{0}_{P'}$ | $\mathbf{0}_{P'}$ | $\mathbf{0}_{P'}$ | $\mathbf{0}_{P'}$ | $\mathbf{0}_{P'}$ | $\mathbf{0}_{P'}$ | $\mathbf{0}_{P'}$ | $\mathbf{0}_{P'}$ | $\mathbf{0}_{P'}$ | $\mathbf{0}_{P'}$ | $\mathbf{0}_{P'}$ | $\mathbf{0}_{P'}$ | $\mathbf{0}_{P'}$ |

Table 89: Example output of linear readout ($\boldsymbol{W}_{\text{out}}\boldsymbol{X}^{(2)}$), continuing from Tables 87 and 88. The yellow cells represent the maximum value of each column, from the '=' token's column to the rightmost column (which are used for next-token prediction).

| $\mathcal{I}$ | $ | 7 | 5 | 9 | 5 | × | 7 | 9 | = | 5 | 0 | 0 | 0 | 0 | 6 |
|---|---|---|---|---|---|---|---|---|---|---|---|---|---|---|---|
| 0 | - | - | - | - | - | - | - | - | 0 | 1 | 1 | 1 | 1 | 0 | 1 |
| 1 | - | - | - | - | - | - | - | - | 0 | 0 | 0 | 0 | 0 | 0 | 0 |
| 2 | - | - | - | - | - | - | - | - | 0 | 0 | 0 | 0 | 0 | 0 | 0 |
| 3 | - | - | - | - | - | - | - | - | 0 | 0 | 0 | 0 | 0 | 0 | 0 |
| 4 | - | - | - | - | - | - | - | - | 0 | 0 | 0 | 0 | 0 | 0 | 0 |
| 5 | - | - | - | - | - | - | - | - | 1 | 0 | 0 | 0 | 0 | 0 | 0 |
| 6 | - | - | - | - | - | - | - | - | 0 | 0 | 0 | 0 | 0 | 1 | 0 |
| 7 | - | - | - | - | - | - | - | - | 0 | 0 | 0 | 0 | 0 | 0 | 0 |
| 8 | - | - | - | - | - | - | - | - | 0 | 0 | 0 | 0 | 0 | 0 | 0 |
| 9 | - | - | - | - | - | - | - | - | 0 | 0 | 0 | 0 | 0 | 0 | 0 |
| × | - | - | - | - | - | - | - | - | 0 | 0 | 0 | 0 | 0 | 0 | 0 |
| = | - | - | - | - | - | - | - | - | 0 | 0 | 0 | 0 | 0 | 0 | 0 |
| $ | - | - | - | - | - | - | - | - | 0 | 0 | 0 | 0 | 0 | 0 | 100 |

Table 90: Example output sequence $\mathcal{O} = \text{Dec}\left(\boldsymbol{X}^{(2)}\right)$, continuing from Table 89. The yellow cells in the bottom row exactly predict the next tokens.

| $\mathcal{I}$ | $ | 7 | 5 | 9 | 5 | × | 7 | 9 | = | 5 | 0 | 0 | 0 | 0 | 6 |
|---|---|---|---|---|---|---|---|---|---|---|---|---|---|---|---|
| $\mathcal{O}$ | - | - | - | - | - | - | - | - | - | 5 | 0 | 0 | 0 | 0 | 6 | $ |

