# OpenReview forum: "Position Coupling: Improving Length Generalization of Arithmetic Transformers Using Task Structure"
_NeurIPS.cc/2024/Conference — NeurIPS 2024 poster_

### Official Review · Reviewer_U8WH · 2024-07-09

**Soundness:** 3
**Presentation:** 3
**Contribution:** 2
**Rating:** 6
**Confidence:** 4

**Summary:**

This paper proposes a method called "position coupling" to enhance the length generalization of Transformer models, specifically targeting arithmetic tasks such as integer addition. The authors claim both empirical success and theoretical guarantees for their approach. The method involves assigning the same position IDs to semantically related tokens to better embed the task structure into the Transformer’s positional encoding.

**Strengths:**

1. **Novelty**: The idea of coupling positional embeddings to reflect semantic relationships is interesting and novel.
2. **Empirical Results**: The paper provides detailed empirical results showing improved performance on length generalization for integer addition tasks.
3. **Theoretical Analysis**: The authors offer theoretical insights into why position coupling should work, which is a positive aspect of the paper.

**Weaknesses:**

1. **Scope of Application**: While the paper demonstrates the effectiveness of position coupling on integer addition tasks, the generalizability to broader and more complex tasks is not convincingly shown. The examples provided are limited and do not cover a wide range of real-world applications.
2. **Complexity and Practicality**: The proposed method introduces additional complexity in the positional encoding process. This complexity may limit the practical applicability of the method, especially for larger and more diverse datasets.
3. **Proper Citation of Related Work**: In section 3, lines 113-115, the randpos method for length extrapolation, introduced by "Randomized Positional Encodings Boost Length Generalization of Transformers" (ACL 2023), is not properly cited.

**Questions:**

1. Can you provide insights on whether "position coupling" can be generalized to other positional encoding schemes beyond Absolute Positional Encoding? It would be beneficial to understand if and how this method can be adapted or integrated with other popular positional encodings, such as Rotary Positional Embedding (RoPE).

**Limitations:**

As mentioned above.

---

> ### Author Rebuttal · Authors · 2024-08-07
>
> We are grateful for the reviewer’s effort and constructive feedback. Below we summarize your feedback/questions and address these one by one.
>
> > **W1. Scope of Application: the generalizability of position coupling to broader and more complex tasks is not convincingly shown.**
>
> - We agree that the applicability of position coupling to broader tasks has not yet been explored in this paper. However, we want to note that enhancing the length generalization ability of models, even for the addition tasks, is considered an important problem as addressing and improving the arithmetic abilities of models can lead to a better understanding of the model’s capabilities.
> - Definitely, the next step of our work is to extend the application of position coupling to complex, real-world tasks. To do so, we plan to develop a method called “automatic coupling”. Unlike position coupling which requires manual assignment of position ID for each token, it will automatically learn the task structure and assign appropriate position IDs without human intervention. This advancement will enable broader applications beyond simple arithmetic/algorithmic tasks. We believe our findings can serve as a stepping stone towards this future development.
>
> > **W2. Complexity and Practicality: Position coupling introduces additional complexities in PE, which may limit its practical applicability.**
>
> - The introduction of position coupling does add complexity, particularly in designing task-specific couplings. However, once the design is determined, the additional overhead due to its implementation is negligible, as explained in lines 659-662.
> - Furthermore, in contrast to approaches such as [1] and [2] that employ index hinting which requires doubling the input sequence length, position coupling does not increase the sequence length. As a result, we believe that there are no significant additional complexities (e.g., time, memory) introduced during the training phase.
> - Additionally, we believe that the development of “automatic coupling”, as mentioned earlier, has the potential to fully address the complexity issue.
> - If there are remaining concerns regarding additional complexity introduced by our method, please let us know without hesitation.
>
> > **W3. Proper Citation of Related Work: “Randomized Position Encodings Boost Length Generalization of Transformers” (ACL 2023) is not properly cited.**
>
> - We agree that there are some similarities in the underlying concepts, and therefore we will add the citation and provide a more detailed comparison.
> - We note that our method (described in lines 113-115) differs from that of Ruoss et al., 2023. Our method randomly selects the start position ID and assigns a sequence of consecutive numbers. In contrast, Ruoss et al., 2023 assigns a sequence of increasing integers, which are generally not consecutive.
>
> > **Q1. Can position coupling be generalized to other PE schemes than APE?**
>
> - Thank you for an insightful question. During the rebuttal period, we realized that our proposed coupling method could be extended to Relative PEs such as RoPE and T5’s relative bias. The relative position between the query and key is determined by the difference in position IDs that were assigned by our position coupling method. Specifically for RoPE, we conducted experiments and observed that position coupling enhances the length generalization capability of RoPE (See Fig. 4 of the PDF file in our Global Response). We believe that this approach has significant potential for the research of length generalization and we will add experiments in our next revision.
> - Instead of adapting our method to other PE methods than APE, it is also possible to integrate position coupling with existing RPE methods such as standard RPEs, RoPE, or FIRE [3]. In this approach, RPE methods are used independently alongside position coupling, which may provide hope for application to general LLM models. Thus, we think this is another promising direction to further improve the applicability of our proposed methodology, so we will conduct some experiments and consider adding the results to our final manuscript.
>
> We hope our response has adequately addressed the reviewer's concerns, and we would appreciate it if you could reconsider your assessment.
>
> ---
> **References**
>
> [1] Zhou et al., What algorithms can transformers learn? a study in length generalization. ICLR 2024.
> [2] Zhou et al., Transformers can achieve length generalization but not robustly. arXiv preprint, 2024.
> [3] Li et al., Functional interpolation for relative positions improves long context transformers. arXiv preprint, 2023.

---

> > ### Comment · Reviewer_U8WH · 2024-08-08
> >
> > Thank you for your feedback. I will raise my score from 5 to 6.

---

> > > ### Author Response · Authors · 2024-08-11
> > >
> > > Thank you for your feedback and for reconsidering the score. We will be sure to incorporate your suggestions in our next revision. If you have any further questions or comments, please feel free to share them.

---

### Official Review · Reviewer_ksWg · 2024-07-10

**Soundness:** 3
**Presentation:** 3
**Contribution:** 3
**Rating:** 7
**Confidence:** 4

**Summary:**

This paper proposes a new way to bake in the positional structure of problems for transformers. The authors also analyze the potential for models with and without their proposed positional coupling to solve problems of arbitrary size. They also show empirically that their method helps a small transformer learn addition.

**Strengths:**

1. Originality: To the best of my knowledge the theoretical analyses are novel and the methods are novel (up to concurrent works).
1. Quality: The work is detailed and thorough.
1. Clarity: The paper is well written and clear.

**Weaknesses:**

1. The significance is limited.
  i. I think often algorithm learning papers can come across as limited in impact and I don't mean to bring this up. Rather, this paper specifically makes a portion of its contributions around 1-layer transformers where some of the claims seem particularly limited. For example, the proof of impossibility of a 1-layer decoder only transformer is interesting, but I don't think it applies to even a 2-layer model weakening the motivation for the use of this positional coupling in practical settings.
  ii. The authors state the limitation that this method requires a priori understanding of the problem structure. I appreciate that this is acknowledged as it feels important to me. With *enough* a priori knowledge of problem structure, one can often solve the problem without ML. This paper is a cool demonstration of learning addition from data, but it is limited in this sense.
  iii. The fact that operands are zero padded to match in length is also a limitation here. This is another instance of requiring problem-aware data processing. The test set is only made up of operands of the same size (clarifying questions below).

**Questions:**

1. Is my understanding of the test set construction accurate? All addition problems are of the form A + B where A and B have the same number of digits?  What happens when we test these models on things like "20345 + 34 = " for example?
1. Why is only the answer reversed? In related work, reversal is often discussed, but this particular scheme requires even more structure-aware processing. I think blindly reversing all numbers that a model encounters would be more compelling than knowing to only reverse the answer. Do the methods in this paper perform better/worse with more consistent data processing?
1. In Section 4, deeper models are shown to perform worse than the shallower models, which the authors attribute to the difficulty of training deep networks. Do the deeper models fit the training data as well as the shallow ones? It seems this way from the plot and if it is the case, then I think another explanation is needed here as the optimization does not seem to be the problem. Perhaps there is an argument to be made about over-fitting, but the rate of degradation seems to vary somewhat nicely with added depth raising more questions about what could be happening. Can the authors offer any other insights here? Maybe some understanding of what is happening at operand lengths where accuracy is ~80%, i.e. which examples are correct/wrong would lend some clarity here.

I'm excited to discuss my review with the authors during the rebuttal period and remain open to improving my score with answers to my questions.

**Limitations:**

The authors adequately addressed all limitations in the limitations section. The significance is limited in ways I addressed above, but these are relatively minor.

---

> ### Author Rebuttal · Authors · 2024-08-07
>
> We deeply appreciate the reviewer's valuable questions and rich comments, and we hope our response relevantly addresses all points raised in the review.
>
> > **W1. Some of the contributions are limited to 1-layer Transformers.**
>
> - We first highlight that our theoretical construction (Thm 5.1) is not limited to 1-layer models because it naturally extends to multi-layer cases.
> - As you pointed out, our impossibility result on NoPE may not hold for a multi-layer model. However, the main purpose of this result is to show a provable gap in the expressive power between position coupling and NoPE. Therefore, we believe that the inapplicability of the impossibility result in multi-layer models does not weaken the motivation for using position coupling.
>
> > **W2. With enough knowledge of the problem, we don’t need ML.**
>
> - We highlight that length generalization is a crucial issue in LLM. Plenty of existing research [1–4] (including us) regards arithmetic/algorithmic tasks as manageable but interesting test beds for studying length generalization because LLMs fail to length-generalize even on such simple tasks.
> - The main message from our work is that incorporating task structure when designing positional encodings can effectively improve the length generalization capabilities. We believe our findings can serve as a stepping stone for future research in this area.
>
> > **W3. Zero-padding is a problem-aware data processing.**
>
> - Note that zero-padding to match the operand lengths is prevalent in this research area ([3–5]).
> - One can implement our method without relying on zero-padding. We present experimental results using a no-padding scheme in the attached PDF file (Fig. 1) in our General Response. While the no-padding position coupling approach fails in in-distribution generalization when trained on a 1-layer model (due to the increased complexity of the algorithm that the model should learn), combined with proper reversing of the number(s), it functionally extrapolates the performance of deeper models.
>
> > **Q1. Should the operands in every testing example have the same length?**
>
> - You’re correct: we sampled the operands to be the same length while testing. To test the model (trained on zero-padding format) with the sample “20345+34=”, we could input it as “20345+00034=”.
> - To address the concern, we tested on operands sampled with different lengths. See Fig. 3 of our PDF attached to the General Response. Each axis corresponds to the lengths of each operand. The results show that the model is capable of solving tasks even when the two operands have different lengths, although zero-padding is applied to ensure consistency in the input format. We will add this result to our revision.
>
> > **Q2. Reversing the answer solely is a problem-aware data processing.**
>
> - Note that solely reversing the answer is also a common practice in this research area [3–5]. From now on, let us compare two different formattings: (a) solely reversing the answer and (b) reversing all the numbers.
> - We first empirically compare (a) and (b) while applying position coupling (see Fig. 1 of the PDF in our General Response). For 1-layer models, both formats exhibit near-perfect length extrapolation and show little difference. Conversely, for 6-layer models, a noticeable performance gap emerges. With zero-paddings, (b) performs better than (a); but if there’s no padding, (a) performs much better than (b). There seems to be no clear winner between the two.
> - The similarity between (a) and (b) for 1-layer models is expected. If we assign the position IDs based on the significance of the digits as usual, there is NO effective difference between (a) and (b) for a 1-layer model in terms of its prediction, which can be deduced from Prop 5.2. Accordingly, our Theorem 5.1 based on (a) can also be applied to (b) without any modification.
> - The difference between (a) and (b) for deeper models is also expected. In multi-layer models, the causal attention mask causes each token embedding to depend on the embeddings of preceding tokens after the first layer. Thus, unlike in the previous case, the predictions of the model may differ depending on the input format.
>
> > **Q3. The optimization might not be the reason why deeper models perform worse.**
>
> - There are two key aspects of optimization: convergence and implicit bias. Our explanation primarily concerns implicit bias. To answer your first question, deeper models do fit the training samples just as well as the shallower models. Thus, convergence is not the issue. We also think that overfitting is not the case, as the trained models only struggle with longer sequences while achieving perfect accuracy on in-distribution samples.
> - Therefore, we believe that the issue lies in the implicit bias of the models. Among the infinite number of solutions that achieve zero training loss, shallower models seem to possess a better implicit bias, which allows them to find solutions that generalize better for longer sequences.
> - Specifically, we conjecture that the outstanding performance of the 1-layer model is due to its relatively restricted expressivity and that the model has no way to fit the training samples other than learning the true algorithm. However, for deeper models, the model can still fit the training samples without necessarily learning the true algorithm due to greater expressivity.
> - For a detailed discussion on this, please refer to our General Response.
>
> ---
> **References**
>
> [1] Jelassi et al. Length generalization in arithmetic transformers. arXiv preprint, 2023.
> [2] Kazemnejad et al.. The impact of positional encoding on length generalization in transformers. NeurIPS 2023.
> [3] Zhou et al., What algorithms can transformers learn? a study in length generalization. ICLR 2024.
> [4] Zhou et al. Transformers can achieve length generalization but not robustly. arXiv preprint, 2024.
> [5] Lee et al. Teaching arithmetic to small transformers. ICLR 2024.

---

> > ### Comment · Reviewer_ksWg · 2024-08-09
> > **Reviewer Response**
> >
> > I have read the detailed reply from the authors. The additional clarity and the details in the general response and the PDF have addressed my concerns. I have changed my score from a 6 to a 7. At this point, I think the paper is clearly above the acceptance threshold.

---

> > > ### Author Response · Authors · 2024-08-11
> > >
> > > Thank you for your feedback and for reconsidering the score. We are glad to hear that our response addressed your concerns and that you view the paper as above the acceptance threshold. We would also be happy to hear if you have any additional thoughts or suggestions.

---

### Official Review · Reviewer_CVfs · 2024-07-13

**Soundness:** 2
**Presentation:** 3
**Contribution:** 3
**Rating:** 5
**Confidence:** 3

**Summary:**

This paper proposes "position coupling", a novel technique to improve the length generalization ability of decoder-only Transformers. Unlike standard positional embeddings, position coupling assigns the same position ID to semantically related tokens across the input sequence, directly embedding task structure within the model. This approach achieves near-perfect length generalization on integer addition, extrapolating from training on up to 30-digit sums to successfully solving 200-digit additions. The authors theoretically prove the capability of a 1-layer Transformer with coupled positions to perform additions with exponentially long operands. They further demonstrate the effectiveness of position coupling on tasks like addition with multiple summands, N×2 multiplication, and copy/reverse operations, providing a theoretical construction for the multiplication task as well. The paper explores the application of position coupling in 2D tasks, showcasing its potential beyond 1D sequences.

**Strengths:**

- While inspired by index hinting, position coupling offers a novel and more elegant solution for incorporating task structure into Transformers. It directly embeds this information within the positional encoding, eliminating the need for augmenting the input sequence and simplifying model training.
- The paper demonstrates a high level of technical rigor. The proposed method is well-motivated and thoroughly evaluated on a variety of tasks, including both empirical analyses and theoretical constructions. The experimental design is comprehensive, with thorough comparisons against relevant baselines and ablations on various architectural choices.
- The paper is well-written and easy to follow. The authors clearly articulate the problem, their proposed solution, and the key contributions. The use of figures and examples effectively illustrates the concepts and makes the theoretical constructions more accessible.
- This work addresses a crucial challenge in Transformer-based learning: length generalization. The impressive results on arithmetic tasks, particularly the significant extrapolation achieved in addition, highlight the potential of position coupling for enabling Transformers to learn algorithms and generalize far beyond their training data. The theoretical analyses provide valuable insights into the mechanism of position coupling and its role in achieving length generalization. The extension to 2D tasks further broadens the applicability and impact of this work.

**Weaknesses:**

- While Theorem 5.1 provides a strong theoretical foundation for the capabilities of 1-layer Transformers with position coupling, the paper lacks a theoretical understanding of why deeper models might perform worse despite their greater expressivity. Further theoretical analysis on the interaction between position coupling and depth, especially on tasks like N×2 multiplication where deeper models are necessary, would significantly strengthen the work.
- The success of position coupling heavily relies on the specific input format (reversed response, zero-padding, etc.). It remains unclear how robust the method is to variations in input format and whether it can be applied to tasks where such specific formatting is not possible or desirable. Exploring alternative position coupling schemes that are less sensitive to the input format or evaluating the method on tasks with diverse input structures would strengthen the claims of generalizability.
- The paper primarily compares position coupling against basic positional embedding techniques. Including a broader range of recent length generalization techniques in the experimental comparison, such as those based on relative positional encodings would provide a more comprehensive understanding of the method's effectiveness and potential advantages.
- The minesweeper generator task serves as a preliminary investigation into the potential of position coupling for multi-dimensional tasks. However, exploring the applicability and effectiveness of position coupling on a wider range of 2D or even higher-dimensional tasks, potentially with more complex structures, would further highlight the significance and generalizability of the proposed method.

**Questions:**

- The paper acknowledges the reliance on a specific input format for optimal performance. Could you elaborate on the sensitivity of position coupling to variations in input format? Have you experimented with alternative formats and if so, what were the outcomes?
- While the paper explores several arithmetic and algorithmic tasks, it would be beneficial to understand the limitations of position coupling. Are there specific task characteristics or structures that might render position coupling less effective or even inapplicable?
- The design of the position coupling scheme relies on an intuitive understanding of the task structure. Could you formalize the notion of task structure and provide guidelines for designing appropriate position coupling schemes for different tasks?
- For the 2D task, why does using the same embedding layer for both position coupling modules perform better than separate layers? Is this specific to the task or a general observation?
- Would it be possible to extend the evaluation of position coupling to more complex 2D tasks, such as image-related tasks or tasks involving graphs or other non-sequential structures?

---

> ### Author Rebuttal · Authors · 2024-08-07
>
> We thank the reviewer for the insightful comments. We provide our response to the reviewer's concerns.
>
> > **W1. Theoretically, why do deeper models perform worse despite their expressivity?**
>
> - For a broader answer to the question, see our General Response.
> - We hypothesize that the performance degradation is due to the bad implicit bias of deep models (learning shortcuts to only achieve in-distribution generalization) when learning a simple algorithm to solve the task. We believe exploring a theoretical explanation for the bad implicit bias of large models on low-complexity tasks is a promising research direction.
>
> > **W2+Q1. How robust is position coupling to input formats? Can it be applied to tasks when specific formatting is undesirable?**
>
> - See our General Response for details on the robustness to input formats.
> - As we emphasized there, proper use of the input format is crucial for solving the task even with position coupling. Also, the model’s performance depends on the choice of input format. Thus, if we cannot apply any formatting, we should not expect significant success in solving the given task, not just in terms of length generalization.
>
> > **W3. Compare with recent techniques for length generalization.**
>
> - Thank you for your suggestion. One notable length generalization result for the addition task before us appears in [1], combining FIRE (a relative PE method) and index hinting. They achieve near-perfect length generalization up to operand length 100 by training on up to 40-digit additions. (Recall that we achieve 30-to-200 generalization with a 1-layer model.) Despite their great performance, they require doubling the input length because of index hinting. In contrast, our method doesn't require doubling the input sequence, so we believe our method is more efficient to run.
> - We also conducted experiments combining RoPE and our method and achieved an improved length generalization compared to vanilla RoPE: see Fig. 4 of PDF in our General Response.
> - We will add these comparisons in our final manuscript.
> - [1] Zhou et al., Transformers can achieve length generalization but not robustly. arXiv, 2024.
>
> > **W4. Can position coupling be applied to a wider range of 2D or higher-dim tasks?**
>
> - As demonstrated in the paper, our method significantly helps Transformers to length-generalize on the tasks with a clear structure between token positions. Extending this to various multi-dim tasks is an interesting future work.
> - A challenge in multi-dim tasks (not only for length generalization and for applying our method) is the exponential growth (in task dimension) of the number of tokens, which makes it difficult for the model to analyze queries and generate responses. Overcoming this dimensionality problem is an interesting future direction.
>
> > **Q2. Are there specific task structures that render position coupling less effective or inapplicable?**
>
> - Let us give you some examples. Our preliminary experiments weren’t very successful for some tasks including (1) addition with a varying number of operands and (2) multiplication with varying lengths of both operands. We tried couplings similar to the ones applied to simple addition and Nx2 multiplication, respectively, but they weren’t effective (although we did not invest much effort in making it work).
>   - The algorithm for solving (1) needs to attend to a varying (over problem instances) number of positions for generating tokens.
>   - The algorithm for solving (2) needs to attend to positions in varying relative distances in terms of the “coupled” position IDs.
> On the contrary, our theoretical construction for simple addition and Nx2 tasks does not suffer from these difficulties, which is the key reason for successful length generalization; thus, we do not expect length generalization on tasks (1) and (2) without any advance in input formats.
> - Some tasks don’t have any structure between specific positions (e.g., sorting and mode), thereby we cannot directly apply position coupling. (See our response to Reviewer ow6J.)
>
> > **Q3. Can you formalize the notion of task structure and provide guidelines for designing proper coupling for different tasks?**
>
> - Defining task structure involves the relationship between the query and response, focusing on which tokens in the query influence the determination of each token in the response. Designing position coupling relies on human intuition, making it challenging to provide concrete guidelines. However, for tasks with unclear coupling structures, assigning position IDs (piece-wise) consecutively may be worth trying. It is shown to be empirically effective in the Nx2 multiplication task (which is later theoretically backed by Thm 6.1), although it's not intuitive how the coupled positions capture the task structure.
>
> > **Q4. Why does using the same embedding layer for both position coupling modules perform better than separate layers?**
>
> - As mentioned in lines 655–657, we do not have a clear explanation for why sharing the embedding layer performs better. We conjecture it is due to the row-column symmetry of the Minesweeper generator task, meaning that transposing the board still results in a valid problem. We believe that tasks where rows and columns have different semantic structures might not benefit from using the same embedding layer.
>
> > **Q5. Can we extend the evaluation of position coupling to more complex 2D tasks?**
>
> - Although our study focuses on simple arithmetic/algorithmic tasks, we are eager to explore more complex multi-dimensional tasks.
> - For image-related tasks (e.g., involving ViT), we could apply position coupling similar to what we did for the Minesweeper generator task. For graphs and other non-sequential tasks, the position coupling scheme may need refinement, but extending it to these complex tasks is promising future work.
>
> We hope our response has adequately addressed the reviewer's concerns, and we would appreciate it if you could reconsider your assessment.

---

> > ### Author Response · Authors · 2024-08-11
> >
> > Dear Reviewer CVfs,
> >
> > Thank you for taking the time to review our work and for providing such insightful and constructive feedback. We understand that you may have a busy schedule, but we wanted to follow up to ensure that our responses have sufficiently addressed your concerns. If you have any further questions or comments, we would be glad to hear them.

---

### Official Review · Reviewer_ow6J · 2024-07-24

**Soundness:** 4
**Presentation:** 4
**Contribution:** 3
**Rating:** 7
**Confidence:** 3

**Summary:**

This work considers the problem of length generalization of Transformers and proposes injecting the task structure through positional embeddings for improving length generalization. Task structures are known and therefore, the authors come up with a (relatively) general heuristic to leverage this structure.

The paper relies on the observation that for tasks like addition, there are “groups” of tokens that should be treated similarly as they carry the same semantics. I.e., the digits of the summands from least significant to most significant should be embedded with the same positional embedding (called *position coupling*) so that the model can take advantage of this structure and carry out the sum correctly.

Position coupling is used in conjunction with a number of other tricks such as reversing the sum, zero padding, and using BOS/EOS.

The proposed method is then evaluated comprehensively on the addition task (for which it was proposed), and strong length generalization is observed. Ablations on the number of layers, and different positional embeddings are carried out to further emphasize the importance of position coupling.

The results are also backed by theory, and interestingly the attention patterns predicted by theory are observed in the experiments.

Other than the addition task, Multiplication ($N\times2$) and a 2D task are considered in the experiments as well.

**Strengths:**

- The empirical results are quite strong and proper ablations and baselines are considered.
- The definitions, presentation, and method are quite clear.
- The method is backed by theory.
- The method is relatively general, however, it is only applicable to a class of tasks where there is a clear structure to be exploited.
- The predictions of the theory are further supported in the experiments (the attention patterns for carry detection)
- The experiments go beyond addition to multiplication and a 2D game (where more explanation is required)

**Weaknesses:**

The proposed method, though general for some tasks, seems to be very much geared towards a specific class of tasks of interest in length generalization. In particular, tasks like sorting, mode, and parity seem to be automatically out of reach for positional coupling and limit the generality of its applicability.

**Questions:**

None.

**Limitations:**

See weaknesses.

---

> ### Author Rebuttal · Authors · 2024-08-07
>
> We thank the reviewer for the positive review and valuable comments. Below, we address the reviewer’s concern.
>
> > **W1. Position coupling is geared towards specific tasks in length generalization.**
>
> First, as mentioned in the conclusion of our paper, we focus on certain tasks with a handy structure between specific positions of tokens. Since vanilla Transformers often fail to learn the true structure of the tasks—even for simple algorithmic tasks—as observed by plenty of previous works [1–10], we aim to guide the model to properly learn the structure by coupling the positions, thereby improving length generalization. The structure between positions is important for employing our method, so we admit that it is not easy to apply our method to some tasks. **However**, we address the three tasks raised by the reviewer.
>
> **Parity**.
> - Given a binary query sequence, the objective is to output 1 if there is an odd number of 1s in the query and to output 0 otherwise.
> - Without any data processing, it is a difficult task for Transformers to achieve length generalization [1,4,5,7,9]. However, with the help of a scratchpad generated by leveraging the idea of unrolling the query, Transformers can achieve length generalization [9]. To utilize the power of position coupling, we opt for a different scratchpad from that used in [9].
> - By unrolling the query, we can generate a sequence of partial results. Let’s put the partial result from the 1st to n-th token of the query into the n-th token of the response sequence. For example, if the query is given as 010011, the response with scratchpad would be 011101. Then, the last token of the response immediately becomes the answer for the original task. We train the model to generate the whole response (including scratchpad) for each query to solve the task step-by-step with next-token prediction. Now, we can naturally couple the n-th positions of the query and the response when we apply our method. For example, given an input sequence “010011=011101”, we can assign (3,4,5,6,7,8,2,3,4,5,6,7,8) as position IDs (when the starting ID is randomly chosen as 3).
> - To showcase the efficacy of position coupling on the parity task with a proper input format, we compare 4 different settings: position coupling with/without scratchpad and NoPE with/without scratchpad. For the “position coupling without scratchpad” setting, we naively couple all the positions (except for the ‘=’ token) with the same position ID (e.g., “010011=1” can get (3,3,3,3,3,3,4,3)). We train the models on the queries of lengths 1–20 and test the lengths up to 100. We measure exact-match accuracy (including scratchpad if applicable) as well as the accuracy only for the single token at the position of the last token of the response except for the EOS token (called “parity accuracy”). The result of the experiments is shown in the PDF file attached in our General Response. Without scratchpads, both NoPE and position coupling cannot even achieve good in-distribution performances. Even with our scratchpads, without any position embeddings (NoPE), a 6-layer 8-head model showcases a very restricted length generalization capability up to the length of ~30. The model performs worse than random from the length of 40 because the model sometimes outputs tokens other than 0 or 1. Most importantly, our 1-layer 4-head model with position coupling and scratchpad achieves perfect length generalization up to length 100! We strongly believe that thanks to the combination of coupled position IDs and scratchpad enabling Transformers to learn a simple algorithm for solving the task with next-token prediction, we could achieve another remarkable length generalization result.
> - We will add this result to our final manuscript with more ablations.
>
> **Sorting** and **Mode**.
> - The sorting task aims to generate a response equal to the sorted sequence of the given query; the mode task aims to find the most frequent token appearing in the given query.
> - In both tasks, there is no exploitable structure between the positions of tokens. Thus, it is not straightforward to apply position coupling to solve these tasks, thereby we did not test our method on these tasks.
> - Not only is it unnatural to couple the positions, but it is also unnecessary to do so. Vanilla Transformers already length-generalize well on these tasks [9].
>
> In short, position coupling is an effective method if we can create a clear structure between positions with proper usage of input format; however, it is inapplicable if there is no such structure. Nonetheless, there are a lot of real-world tasks whose underlying structure between positions is vague or unavailable. We leave the research direction of extending our idea to such tasks by automatically discovering appropriate couplings of the positions as interesting future work.
>
>  Please let us know without hesitation if you have further questions or comments.
>
> ---
> **References**
>
> [1] Bhattamishra et al., On the ability and limitations of transformers to recognize formal languages. arXiv preprint, 2020.
> [2] Kim et al., Have you seen that number? investigating extrapolation in question answering models. NeurIPS 2021.
> [3] Nye et al., Show your work: Scratchpads for intermediate computation with language models. arXiv preprint, 2021.
> [4] Chiang and Cholak., Overcoming a theoretical limitation of self-attention, arXiv preprint, 2022.
> [5] Delétang et al., Neural networks and the Chomsky hierarchy, ICLR 2023.
> [6] Kazemnejad et al.. The impact of positional encoding on length generalization in transformers. NeurIPS 2023.
> [7] Ruoss et al., Randomized positional encodings boost length generalization of transformers, ACL 2023.
> [8] Lee et al., Teaching arithmetic to small transformers. ICLR 2024.
> [9] Zhou et al., What algorithms can transformers learn? a study in length generalization. ICLR 2024.
> [10] Zhou et al., Transformers can achieve length generalization but not robustly. arXiv preprint, 2024.

---

> > ### Comment · Reviewer_ow6J · 2024-08-09
> >
> > Thank you for your efforts and your detailed rebuttal, especially for coming up with how to apply position coupling to the parity task and showing successful results on it (and thanks for acknowledging the limitation of the method w.r.t. tasks like sorting and mode).
> >
> > I'd add that the 2D experiment could benefit from a bit more explanation and how position coupling is applied to it. I am happy with the rebuttal and like the paper, thus I'll maintain my score, and would just encourage the authors to see if they can extend their image experiment to other tasks with more structure, but regardless I find the work solid.
> >
> > As a side note, the discussion on length generalization getting worse with deeper models seems to be somewhat connected to a concurrent work [1], it might be worth seeing if there's indeed any relation.
> >
> > [1] https://arxiv.org/pdf/2402.04875

---

> > > ### Author Response · Authors · 2024-08-11
> > >
> > > Thank you for taking the time to provide valuable feedback. We appreciate your suggestions on clarifying the 2D experiment, and will certainly consider these points in our future work. We also appreciate the reference you provided and will look into the potential connection.

---

### Author Rebuttal · Authors · 2024-08-07

We deeply appreciate all reviewers for their insightful and detailed reviews, questions, and comments on our work. We assure the reviewers that all the answers and discussions will be incorporated into our final manuscript.

We are encouraged to see that the reviewers recognized that our method is novel and significant (CVfs, ksWg, U8WH), our experiments are thorough and detailed (ow6J, CVfs, ksWg, U8WH), and our theoretical analysis is insightful and well-motivates our approach (ow6J, CVfs, ksWg, U8WH).

Please check out our PDF file containing:
* Fig. 1: ablations on the input formats and the model size.
* Fig. 2: Parity task solved with position coupling + scratchpad.
* Fig. 3: Testing on separate operand lengths.
* Fig. 4: RoPE + Position Coupling.

Now, we will provide our response to two commonly raised questions.

## **1. Ablations on Input Formats**
* Reviewers raised a question on the robustness of our method to input formats (CVfs) and a concern that input formatting is problem-aware processing (ksWg).
* We would like to clarify that our input format is primarily selected to simplify the algorithm of solving the addition task, not through extensive ablation studies. Thus, we are not arguing that our choice of input format is empirically optimal for training Transformers.
* However, we note that applying proper input formatting is crucial and natural in general ML. Even in a simple image classification task, appropriate standardization and augmentation are often useful. When applying these techniques, we often utilize the fact that typical image pixels range in [0, 255] (to apply standardization) and that appropriate augmentation methods may differ by the image type. Hence, enough understanding of the task leads to proper input processing that helps the model to effectively solve a given task.
* Our additional experiments on position coupling with various input formats show that the model’s performance varies with different input formats (refer to the attachment). It is expected, as the complexity of the algorithm that the model should learn changes according to the input format.
  * Small models (1-layer 4-head) achieve near-perfect generalization when the numbers are zero-padded and the answer or all numbers are reversed. We believe this is because the combination of zero-padding and reversing enabled a small Transformer to learn a simple length-generalizing algorithm. If we flip the answer or all the numbers without zero-padding, small models exhibit a bit worse in-distribution performance and a restricted length generalization capability. Without reversing, the models perform poorly.
  * Larger models (6-layer 16-head) perform better than the small model when the numbers are no longer zero-padded (especially when the answer is reversed). We believe this is because the task-solving algorithm with reversing and without zero-padding that the model should learn is more sophisticated, which larger models can learn more easily. Contrarily, we observe a degradation in performance when we add zero-padding in the larger model, which suggests that the model may have learned a "shortcut" due to its (overly) strong expressive power relative to the problem's complexity. (Refer to the next question for more details on this matter.)

## **2. Deeper models seem to perform worse. Why?**
* Some reviewers expressed concerns that position coupling performs worse when applied to deeper networks. Here, we provide our thoughts on this phenomenon.
* The first thing to note is that position coupling enables the model to solve the addition problem using a much simpler algorithm. As proven in Theorem 5.1, even a 1-layer model is sufficiently expressive. We hypothesize that the reason for the outstanding performance of a 1-layer model is that the architecture is simple so the model has no way to fit the training samples other than by learning the true (or length-generalizable) function.
* In contrast, we believe that deeper models perform poorly because their larger expressive capacity allows them to learn shortcuts to fit the training distribution that may not generalize well across different lengths. This phenomenon differs from classical overfitting in that these models still generalize well to in-distribution samples.
* A similar observation was made in [1]: there exist non-extrapolating models that generalize well to in-distribution samples but struggle with longer ones. The authors interpreted this phenomenon as indicating that there might be unexpected ways to solve the problem and the model may rely on shortcut-like solutions that work for in-distribution samples but fail on longer samples.
* The superior performance of the position coupling scheme without zero-padding in deeper models also aligns with our interpretation. In this scenario, the query sequence becomes less consistent: the lengths of two operands may differ and some position IDs appear only once. This makes the functions the model needs to learn more complex. This complexity is evident from 1-layer experiments that the no-padding scheme fails for even in-distribution generalization. We believe that such a complex structure of the target function compels the model to learn the true function rather than a shortcut-like solution, resulting in strong length generalization performance.
* Combining the answers for the first two questions, we do not believe that position coupling itself is bad for deep models. The performance degradation on deep models can also be attributed to the input formats, which may make the task-solving algorithm way simpler, and other training details such as small dataset size, which can make the model easily overfit to training distribution.

Again, we deeply thank all reviewers for their time and effort in reviewing our work. We are excited to hear more feedback.

Warm regards,
Authors

---
**References**

[1] Zhang et al., Unveiling transformers with lego: a synthetic reasoning task. arXiv preprint, 2022.

---

### Decision · Program_Chairs · 2024-09-25

**Decision:**

Accept (poster)

**Comment:**

The paper studies the issue that Transformers are not able to generalize algorithms learned on sequences of length $n$ to longer sequences $n+k$ for even moderate values of $k$. This is demonstrated on many algorithmic toy tasks like addition where the algorithm is known, relatively simple, and yet Transformers still fail. Theoretical results for the 1-layer cases are presented to show that the proposed solution of "Position Coupling" theoretically (and empirically) improves the representational power of the Transformers.

All reviewers are generally positive on the paper. The work is significantly improved in rebuttal by showing the parity problem can also be tackled with their approach as can other base positional embeddings. This gives the work a broader potential scope and applicability.

There are still weaknesses that could be better incorporated into the manuscript to set appropriate expectations. In particular, it is not made obvious that the position coupling strategy is single-purpose in that it must be specified for each unique problem, and does not act as a more generic "plug-and-play" later/encoding that can be used for arbitrary problems. This reduces the scope to a theoretical inspection of the limits of Transformers and potential paths forward rather than a generally useful tool. Also noted by the reviewers is the drop in performance as more layers are added.

However, immediate practical utility is not a prerequisite for publication. The work is overall interesting, sheds light on a new approach to talking a real limitation, and is relatively easy to implement. As such, I recommend the paper for acceptance but note that the reviewers should temper the reader's expectation to the manual work needed to apply position coupling to new tasks.